



# The biophysics, ecology, and biogeochemistry of functionally diverse, vertically- and horizontally-heterogeneous ecosystems: the Ecosystem Demography Model, version 2.2 — Part 1: Model description

Marcos Longo[1,2,3], Ryan G. Knox[4,5], David M. Medvigy[6], Naomi M. Levine[7], Michael C. Dietze[8], Yeonjoo Kim[9], Abigail L. S. Swann[10], Ke Zhang[11], Christine R. Rollinson[12], Rafael L. Bras[13], Steven C. Wofsy[1], and Paul R. Moorcroft[1]

[1]Harvard University, Cambridge, MA, United States
[2]Embrapa Agricultural Informatics, Campinas, SP, Brazil
[3]Jet Propulsion Laboratory, California Institute of Technology, Pasadena, CA, United States
[4]Massachusetts Institute of Technology, Cambridge, MA, United States
[5]Lawrence Berkeley National Laboratory, Berkeley, CA, United States
[6]University of Notre Dame, Notre Dame, IN, United States
[7]University of Southern California, Los Angeles, CA, United States
[8]Boston University, Boston, MA, United States
[9]Department of Civil and Environmental Engineering, Yonsei University, Seoul 03722, Republic of Korea
[10]University of Washington, Seattle, WA, United States
[11]Hohai University, Nanjing, Jiangsu, China
[12]The Morton Arboretum, Lisle, IL, United States
[13]Georgia Institute of Technology, Atlanta, GA, United States

**Correspondence:** M. Longo
(mlongo@post.harvard.edu)

**Abstract.** Earth System Models (ESMs) have been developed to represent the role of terrestrial ecosystems on the energy, water, and carbon cycles. However, many ESMs still lack representation of within-ecosystem heterogeneity and diversity. In this manuscript, we present the Ecosystem Demography Model version 2.2 (ED-2.2). In ED-2.2, the biophysical and physiological cycles account for the horizontal and vertical heterogeneity of the ecosystem: the energy, water, and carbon cycles are solved

5  separately for each group of individual trees of similar size and functional group (cohorts) living in a micro-environment with similar disturbance history (patches). We define the equations that describe the energy, water, and carbon cycles in terms of total energy, water, and carbon, which simplifies the ordinary differential equations and guarantees excellent conservation of these quantities in long-term simulation ($< 0.1\%$ error over 50 years). We also show examples of ED-2.2 simulation results at single sites and across tropical South America. These results demonstrate the model's ability to characterize the variability of

10  ecosystem structure, composition and functioning both at stand- and continental-scales. In addition, a detailed model evaluation was carried out and presented in a companion paper. Finally, we highlight some of the ongoing developments in ED-2.2 that aim at reducing the uncertainties identified in this study and the inclusion of processes hitherto not represented in the model.



# 1 Introduction

The dynamics of the terrestrial biosphere play an integral role in the earth's carbon, water and energy cycles (Betts and Silva Dias, 2010; Santanello Jr et al., 2018; Le Quéré et al., 2018), and consequently, how the earth's climate system is expected to change over the coming decades due to the increasing levels of atmospheric carbon dioxide arising from anthropogenic

activities (IPCC, 2014; Le Quéré et al., 2018). Models for the dynamics of the terrestrial biosphere and its bi-directional interaction with the atmosphere have evolved considerably over the past decades (Levis, 2010; Fisher et al., 2014, 2018). Initially developed to provide global surface boundary conditions in climate model simulations, the first generation of terrestrial biosphere models consisted of relatively simple formulations describing the dynamics of radiation, roughness, and transpiration (e.g., NCAR/BATS, Dickinson et al., 1986) and photosynthesis (e.g. SiB, Sellers et al., 1986). However, increasing recognition

of the role of vegetation in mediating the exchanges of carbon, water and energy between the land and the atmosphere led to terrestrial biosphere models being expanded to incorporate explicit representations of plant photosynthesis, and resulting dynamics of terrestrial carbon uptake, turnover and release within terrestrial ecosystems (e.g. LSM (Bonan, 1995) and SiB2 (Sellers et al., 1996)). While the fluxes of carbon, water and energy predicted by these models would change in response to changes in their climate forcing, the biophysical and biogeochemical properties of the ecosystem within each climatological

grid cell was prescribed, and thus did not change over time.

Subsequently, building upon previous work (Prentice et al., 1992; Neilson, 1995; Haxeltine and Prentice, 1996), Foley et al. (1996) adopted an approach to calculate the productivity of a series of plant functional types (PFTs), based on a leaf-level model of photosynthesis. The abundance of each PFT within each grid cell was dynamic, with the abundance changes being determined by the relative productivity of the PFTs. This allowed the fast-timescale exchanges of carbon, water, and energy

within the plant canopy to be explictly linked with the long-term dynamics of the ecosystem. This approach followed the concept of dynamic global vegetation model (DGVM), originally coined by Prentice et al. (1989) to describe this kind of terrestrial biosphere model in which changes in climate could drive changes in ecosystem composition, structure and functioning, and which when run coupled to atmospheric models would then feedback onto climate. The subsequent generation of terrestrial biosphere-based DGVMs (i.e. DGVMs incorporating couple carbon, water, and energy fluxes) such as LPJ (Sitch et al., 2003),

CLM-DGVM (Levis et al., 2004) and TRIFFID/JULES (Hughes et al., 2004; Clark et al., 2011; Mangeon et al., 2016) have included additional mechanisms such as disturbance through fires and multiple types of mortality.

Analyses have shown that most terrestrial biosphere models are capable of reproducing the current distribution of global biomes (e.g. Sitch et al., 2003; Blyth et al., 2011) and their carbon stocks and fluxes (Piao et al., 2013). However, they diverge markedly in their predictions of how terrestrial ecosystems will respond to future climate change (Friedlingstein et al., 2014).

In fully-coupled Earth System Model simulations, some of these differing predictions arise from divergent predictions about the direction and magnitude of regional climate change. However, off-line analyses, in which the models are forced with prescribed climatological forcing, have shown that there is also substantial disagreement between the models about how terrestrial ecosystems will respond to any shift in climate (e.g. Sitch et al., 2008; Zhang et al., 2015). In addition, the transitions between biome types, for example, the transition that occurs between closed-canopy tropical forests and grass- and shrub-dominated



savannahs in South America, are generally far more abrupt in typical DGVM results than in observations (Good et al., 2011; Levine et al., 2016).

One important limitation of most DGVMs is that they cannot represent within-ecosystem diversity and heterogeneity. The representation of plant functional diversity within terrestrial biosphere models is normally coarse, with broadly-defined PFTs defined from a combination of morphological and leaf physiological attributes (Purves and Pacala, 2008). In addition, there is limited variation in the resource conditions (light, water, nutrient levels) experienced by individual plants within the climatological grid cells of traditional DGVMs. Some models — e.g. SiB (Sellers et al., 1996) and CLM (Oleson et al., 2013) — have vertical above-ground heterogeneity in the form of a multi-layer plant canopy that allows for sun and shade leaves, and/or differences in rooting depth between PFTs; however, resource conditions are assumed to be horizontally homogeneous, meaning that there is no horizontal spatial variation in resource conditions experienced by individuals. The lack of significant variability in resource conditions limits the range of environmental niches within the climatological grid cells of terrestrial biosphere and makes the coexistence between PFTs difficult; consequently model ecosystems become comprised of single homogeneous vegetation types (Moorcroft, 2003, 2006).

Field- and laboratory-based studies conducted over the past thirty years indicate that plant functional diversity significantly affects ecosystem functioning (Loreau and Hector, 2001; Tilman et al., 2014, and references therein), and variations in trait expression are strongly driven by disturbances and local heterogeneity of abiotic factors such as soil characteristics (Bruelheide et al., 2018; Both et al., 2019). In many cases, biodiversity increases ecosystem productivity and ecosystem stability (e.g., Tilman and Downing, 1994; Naeem and Li, 1997; Cardinale et al., 2007; García-Palacios et al., 2018), and biodiversity has also been shown to contribute to enhanced ecosystem functionality in highly stressed environments (e.g. Jucker and Coomes, 2012). Other studies have also established correlations between tropical forest diversity and carbon storage and primary productivity (Cavanaugh et al., 2014; Poorter et al., 2015; Liang et al., 2016; Huang et al., 2018).

In addition to the absence of within-ecosystem diversity in conventional terrestrial biosphere models, plants of each PFT are also assumed to be homogeneous in size while, in contrast, most terrestrial ecosystems, particularly forests and woodlands, exhibit marked size-structure of individuals within plant canopies (Hutchings, 1997). This size-related heterogeneity is important because plant size strongly affects the amount of light, water, and nutrients individual plants within the canopy can access, which, in turn, affects their performance, dynamics and responses to climatological stress. It also allows representing the dynamics of pervasive human-driven degradation of forest ecosystems (Lewis et al., 2015; Haddad et al., 2015), which affects carbon stocks, structure and composition of forests that cannot be easily represented in highly aggregated models (Longo and Keller, 2019).

An alternative approach to simulating the dynamics of terrestrial ecosystems has been individual based vegetation models (Friend et al., 1997; Bugmann, 2001; Sato et al., 2007; Fischer et al., 2016). Also known as forest gap models, due to the importance of canopy gaps for the dynamics of closed canopy forests, these models simulate the birth, growth, and death of individual plants, thereby incorporating diversity and heterogeneity of the plant canopy mechanistically. In forest gap models, the ecosystem properties such as total carbon stocks, and net ecosystem productivity are emergent properties resulting from competition of limiting resources and the differential ability of plants to survive and be productive under a variety of micro-



environments (e.g. gaps or the understory of a densely populated patch of old-growth forest). This approach has two main advantages. First, gap models represent the dynamic changes in the ecosystem structure caused by disturbances such as tree fall, selective logging, and fires. These disturbances create new micro-environments that are significantly different from old-growth vegetation areas, and allow plants with different life strategies (for example, shade-intolerants) to co-exist in the landscape.

Second, because individual trees are represented in the model, the results can be directly compared with field measurements. Gap models have various degrees of complexity, with some models being able to represent the interactions between climate variability and gross primary productivity (Friend et al., 1997; Sato et al., 2007), as well as the impact of climate change in the ecosystem carbon balance (Fischer et al., 2016, and references therein). However, because the birth and death of individuals within a plant canopy are stochastic processes, multiple realizations of given model formulation are required to determine the

long-term, large-scale dynamics of these models, which limits their applicability over large regions or global scales, and has precluded their use in Earth System Modeling studies.

The Ecosystem Demography Model (ED Moorcroft et al., 2001) is a size- and age-structured approximation of an individual-based vegetation model. Through this approach, it addresses the need to incorporate heterogeneity into models of the long-term, large scale response of terrestrial ecosystems to changes in climate and other environmental forcings within a deterministic

modeling framework. The size and age-structured partial differential equations that describe the plant community are derived from individual-level properties, but are properly scaled to account for the spatially-localized nature of interactions within plant canopies. The model was later extended by Hurtt et al. (2002) and Albani et al. (2006) to incorporate multiple forms of disturbance including land-clearing, land-abandonment, and forest harvesting. An important difference between ED and most DGVMs is that in ED, PFTs are defined not simply based on their biogeographic ranges, but also represent diversity in plant

life-history strategies within any given ecosystem. These different PFTs represent a suite of physiological, morphological, and life-history traits that mechanistically represent the ways different kinds of plants utilize resources (Fisher et al., 2010). Because these models also represent functional diversity and heterogeneity of micro-environments, the ecosystem's structure, diversity and functioning also emerge from the interactions between plants with different life strategies under different resource availability, albeit at a lesser extent than individual-based models (Fisher et al., 2018).

The original ED model formulation was an off-line ecosystem model describing the coupled carbon and water fluxes of a heterogeneous tropical forest ecosystem (Moorcroft et al., 2001). Subsequently Medvigy et al. (2009) applied a similar approach to develop the Ecosystem Demography model version 2 (ED-2) that describes coupled carbon, water and energy fluxes of the land surface, and is capable of being run both offline (e.g. Medvigy et al., 2009; Antonarakis et al., 2011; Zhang et al., 2015; Castanho et al., 2016; Levine et al., 2016), or interactively with a regional atmospheric model (e.g. Knox et al.,

2015; Swann et al., 2015).

In this paper, we describe in detail the biophysical, physiological, ecological and biogeochemical formulation of the most recent version of the ED-2 model (ED-2.2), focusing in particular on the model's formulation of the fast time-scale dynamics of the heterogeneous plant canopy that occur at sub-daily timescales. While many parameterizations and sub-models in ED-2.2 are based on approaches that are also used in other DVGMs, their implementation in ED-2.2 has some critical differences from

other ecosystem models and also previous versions of ED: (1) In ED-2.2, the fundamental budget equations use energy and



total mass as the main prognostic variables; because we use equations that directly track the time changes of the properties we seek to conserve, we can assess the model conservation of such properties with fewer assumptions . (2) In ED-2.2, all thermodynamic properties are scalable with mass, and the model is constructed such that when individual biomass changes due to growth and turnover, the thermodynamic properties are also updated to reflect changes in heat and water holding capacity.

(3) The water and energy budget equations for vegetation are solved at the individual level and the corresponding equations for environments shared by plants such as soils and canopy air space are solved for each micro-environment in the landscape, and thus ecosystem-scale fluxes are emerging properties of the plant community. This approach allows the model to represent both the horizontal and vertical heterogeneity of environments of the plant communities. It also links the individual's ability to access resources such as light and water and accumulate carbon under a variety of micro-environments, which ultimately

drives the long-term dynamics of growth, reproduction, and survivorship.

## 2   Model overview

### 2.1   The Representation of Ecosystem Heterogeneity in ED-2.2

In ED-2.2 the terrestrial ecosystem within a given region of interest is represented through a hierarchy of structures to capture the physical and biological heterogeneity in the ecosystem's properties (Fig. 1).

*Physical Heterogeneity:* The domain of interest is geographically divided into *polygons*. Within each polygon, the time-varying meteorological forcing above the plant canopy is assumed to be uniform. For example, a single polygon may be used to simulate the dynamics of an ecosystem in the neighborhood of an eddy flux tower, or alternatively, a polygon may represent the lower boundary condition within one horizontal grid-cell in an atmospheric model. Each polygon is sub-divided into one or more *sites* that are designed to represent landscape-scale variation in other abiotic properties, such as soil texture,

depth, elevation, slope, aspect, and topographic moisture index. Each site is defined as a fractional area within the polygon and represents all regions within the polygon that share similar time-invariant physical (abiotic) properties. Both polygons and sites are defined at the beginning of the simulation and are fixed in time, and no geographic information exists below the level of the polygon.

*Biotic Heterogeneity:* Within each site, horizontal, disturbance-related heterogeneity in the ecosystem at any given time $t$

is characterized through a series of *patches* that are defined by the time elapsed since last disturbance (i.e. *age*, $a$) and the type of disturbance that generated them. Like sites, patches are not physically contiguous: each patch represents the collection of canopy gap-sized ($\sim 10\,\mathrm{m}$) areas within the site that have a similar disturbance history, defined in terms of the type of disturbance experienced (represented by subscript $q$, $q \in 1, 2, \ldots, N_Q$) and time since the disturbance event occurred. The disturbance types accounted for in ED-2.2, and the possible transitions between different disturbance types, are shown in

Fig. S1. The collection of gaps within each given site belonging to a polygon follows a probability distribution function $\alpha$,



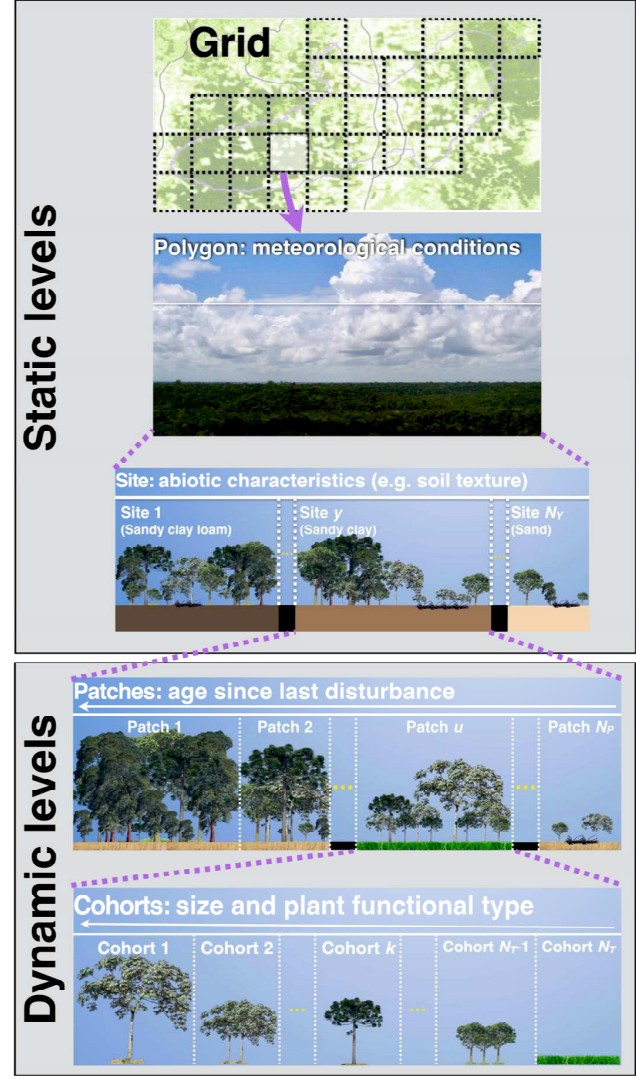

**Figure 1.** Schematic representation of the multiple hierarchical levels in ED-2.2, organized by increasing level of detail from top to bottom. Static levels (grid, polygons, and sites) are assigned during the model initialization and remain constant throughout the simulation. Dynamic levels (patches and cohorts) may change during the simulation according to the dynamics of the ecosystem.

which can be also thought of as the relative area within a site, that satisfies:

$$\sum_{q=1}^{N_Q} \left[ \int_0^\infty \alpha_q\left(a,t\right) \mathrm{d}a \right] = 1. \tag{1}$$

Similarly, the plant community population is characterized by the number of plants per unit area (hereafter number density, $n$), and is further classified according to their plant functional type (PFT), represented by subscript $f$ ($f \in 1, 2, \ldots, N_F$) and the





type of gap ($q$). The number density distribution depends on the individuals' biomass characteristics (size, $\mathbf{C}$), the age since last disturbance ($a$) and the time ($t$), and is expressed as $n_{fq}(\mathbf{C}, a, t)$. Size is defined as a vector $\mathbf{C} = n_{fq}^{-1} (C_l; C_r; C_\sigma; C_h; C_n)$ (units: $\mathrm{kg_C\,plant^{-1}}$) corresponding to biomass of leaves, fine roots, sapwood, heartwood, and non-structural storage (starch and sugars), respectively.

Following Moorcroft et al. (2001), Albani et al. (2006), and Medvigy and Moorcroft (2012), the fundamental partial differential equations that describe the dynamics of demographic density and probability distribution within each site in the size-and-age structured model are defined as (dependencies omitted in the equations for clarity):

$$\underbrace{\frac{\partial n_{fq}}{\partial t}}_{\text{Change rate}} = \underbrace{-\frac{\partial n_{fq}}{\partial a}}_{\text{Aging}} \underbrace{-\boldsymbol{\nabla}_{\mathbf{C}} \cdot (\mathbf{g}_f \, n_{fq})}_{\text{Growth}} \underbrace{-m_f \, n_{fq}}_{\text{Mortality}}, \tag{2}$$

$$\underbrace{\frac{\partial \alpha_q}{\partial t}}_{\text{Change rate}} = \underbrace{\frac{\partial \alpha_q}{\partial a}}_{\text{Aging}} \underbrace{- \sum_{q'=1}^{N_Q} (\lambda_{q'q} \, \alpha_q)}_{\text{Disturbance}}, \tag{3}$$

where $m_f$ is mortality rate, which may depend on the PFT, size, and the individual carbon balance; $\mathbf{g}_f$ is the vector of the net growth rates for each carbon pool, which also may depend on the PFT, size, and carbon balance; $\boldsymbol{\nabla}_{\mathbf{C}}\cdot$ is the divergent operator for the size vector; and $\lambda_{q'q}$ is the transition matrix from gaps generated by previous disturbance $q'$ affected by new disturbance of type $q$, which may depend on environmental conditions. Boundary conditions are shown in Supplement S1.

    Equation (2) and Eq. (3) cannot be solved analytically except for the most trivial cases; therefore the age distribution is

discretized into $N_P$ patches of similar age and same disturbance type, and the population size structure living in any given patch is discretized into $N_T$ cohorts of similar size and same PFT (Fig. 1). Unlike polygons and sites, patches and cohorts are dynamic levels: changes in distribution (fractional area) of patches are driven by aging and disturbance rates, whereas changes in the distribution of cohorts in each patch are driven by growth, mortality, and recruitment (Fig. S2).

    The environment perceived by each plant (e.g. incident light, temperature, vapor pressure deficit) varies across large scales

as a consequence of changes in climate (macro-environment), but also varies at small scales (within the landscape; micro-environment) because of the horizontal and vertical position of each individual relative to other individuals in the plant community (e.g. Bazzaz, 1979) and the position of the local community in landscapes with complex terrains. Both macro- and micro-environmental conditions drive the net primary productivity of each individual, and ultimately determine growth, mortality, and recruitment rates for each individual. Likewise, they can also affect the disturbance rates: for example, during drought

conditions (macro-environment) open canopy patches (micro-environment) may experience faster ground desiccation and consequently increase local fire risk. To account for the variability in micro-environments within the landscape and within local plant communities, in ED-2.2 the energy, water, and carbon dioxide cycles are solved separately for each patch, and within each patch, fluxes and storage associated with individual plants are solved for each cohort.

    The ED-2.2 model represents processes that have inherently different time scales, therefore the model also has a hierarchy of

time steps, in order to attain maximum computational efficiency (Table 1). Processes associated with the short-term dynamics

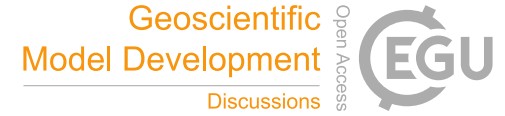



**Table 1.** Time steps associated with processes resolved by ED-2.2. The thermodynamic sub-step is dynamic and it depends on the error evaluation of the integrator, but it cannot be longer than the biophysics step, which is defined by the user. Other steps are fixed as of ED-2.2. Processes marked with ⋆ are presented in the main paper. Other processes are briefly described in Supplements S2 and S3.

| Time Step | Time scale | Processes |
|---|---|---|
| Thermodynamics ($\Delta t_{\text{Thermo}}$) | $1\,\text{s}$ - $\Delta t_{\text{Bio}}$ | ⋆ Energy and water fluxes<br>⋆ Eddy fluxes (including $CO_2$ flux)<br>⋆ Most thermodynamic state functions |
| Biophysics ($\Delta t_{\text{Bio}}$) | $2 - 15\,\text{min}$ | ⋆ Meteorological and $CO_2$ Forcing<br>⋆ Radiation model<br>⋆ Photosynthesis model<br>⋆ Respiration fluxes (autotrophic and heterotrophic)<br>⋆ Evaluation of energy, water, and $CO_2$ budgets |
| Phenology ($\Delta t_{\text{Phen}}$) | $1\,\text{day}$ | Maintenance of active tissues<br>Update of the storage pool<br>Leaf phenology<br>Plant carbon balance<br>Integration of mortality rate due to cold<br>Soil litter pools |
| Cohort dynamics ($\Delta t_{\text{CD}}$) | $1\,\text{month}$ | Growth of structural tissues<br>Mortality rate<br>Reproduction – Cohort creation<br>Integration of fire disturbance rate<br>Cohort fusion, fission, and extinction |
| Patch dynamics ($\Delta t_{\text{PD}}$) | $1\,\text{yr}$ | Annual disturbance rates and patch creation<br>Patch fusion and termination |

are presented in this manuscript. A summary of the phenological processes and those associated with longer term dynamics is presented in Supplement S2 (see also Moorcroft et al., 2001; Albani et al., 2006; Medvigy et al., 2009).

## 2.2 Software requirements and model architecture

*Software requirements*. The ED-2.2 source code is mainly written in Fortran 90, with a few file management routines written
5 in C. Most input and output files use the Hierarchical Data Format 5 (HDF5) format and libraries (The HDF Group, 2016). In addition, the Message Passing Interface (MPI) is highly recommended for regional simulations and is required for simulations



coupled with the *Brazilian Improvements on Regional Atmospheric Modeling System* (BRAMS) atmospheric model (Knox et al., 2015; Swann et al., 2015; Freitas et al., 2017). The source code can be also compiled with Shared Memory Processing (SMP) libraries, which enable parallel processing of thermodynamics and biophysics steps at the patch level, and thus allowing shorter simulation time.

*Code design and parallel structure*. ED-2.2 has been designed to be run in three different configurations; (1) As a stand-alone land-surface model over a small list of specified locations (sites), or (2) as a stand-alone land-surface model distributed over a regional grid, or (3) as coupled with the BRAMS atmospheric model as distributed over a regional grid (ED-BRAMS, Knox et al., 2015; Swann et al., 2015). For regional stand-alone grids, the model partitions the grid into spatially contiguous tiles of polygons, which access the initial and boundary conditions and are integrated independently of each other, but write the results
to a unified output file using collective input/output functions from HDF5. In the case of simulations dynamically coupled with the BRAMS model, polygons are defined to match each atmospheric grid cell.

*Memory allocation*. The code uses dynamic allocation of variables and extensive use of pointers to efficiently reduce the amount of data transferred between routines. To reduce the output file size, polygon-, site-, patch-, and cohort-level variables are always written as long vectors, and auxiliary index vectors are used to map variables from higher hierarchical levels to
lower hierarchical levels (for example, to which patch a cohort-level variable belongs).

## 2.3   Model inputs

Every ED-2.2 simulation requires an initial state for forest structure and composition (initial state), a description of soil characteristics (edaphic conditions), and a time-varying list of meteorological drivers (atmospheric conditions).

*Initial state*. To initialize a plant community from inventory data, one must have either the diameter at breast height of every
individual or the stem density of different diameter size classes, along with plant functional type identification and location; in addition necromass from the litter layer, woody debris and soil organic carbon are needed. Alternatively, initial conditions can be obtained from airborne LiDAR measurements (Antonarakis et al., 2011, 2014) or a prescribed near bare ground condition may be used for long-term spin up simulations. Previous simulations can be used as initial conditions as well.

*Edaphic conditions*. The user must also provide soil characteristics such as total soil depth, total number of soil layers, the
thickness of each layer, as well as soil texture, color and the bottom soil boundary condition (bedrock, reduced drainage, free drainage, or permanent water table). This flexibility allows the user to easily adjust the soil characteristics according to their regions of interest. Soil texture can be read from standard data sets (e.g. Tempel et al., 1996; Hengl et al., 2017) or provided directly by the user. Soil layers, soil color and bottom boundary condition must be provided directly by the user as of ED-2.2. In addition, simulations with multiple sites per polygon also need to provide the fractional areas of each site and the mean soil
texture class, slope, aspect, elevation, and topographic moisture index of each site.

*Atmospheric conditions*. Meteorological conditions needed to drive ED-2.2 include temperature, specific humidity, $CO_2$ molar fraction, pressure of the air above the canopy, precipitation rate, incoming solar (shortwave) irradiance (radiation flux) and incoming thermal (longwave) irradiance (Table 2), at a reference height that is at least a few meters above the canopy. Sub-daily measurements (0.5-6 hours) are highly recommended so the model can properly simulate the diurnal cycle and





**Table 2.** Atmospheric boundary conditions driving the ED-2.2 model. Variable names and subscript follow a standard notation throughout the manuscript (Tables S1 and S2). Flux variables between two thermodynamic systems are defined by a dot and two indices separated by a comma, and they are positive when the net flux goes from the thermodynamic system represented by the first index to the one represented by the second index.

| Variable | Description | Units |
|---|---|---|
| $u_x$ | Zonal wind speed | $\mathrm{m\,s^{-1}}$ |
| $u_y$ | Meridional wind speed | $\mathrm{m\,s^{-1}}$ |
| $p_a$ | Free air pressure | Pa |
| $T_a$ | Free air temperature | K |
| $w_a$ | Free air specific humidity | $\mathrm{kg_W\,kg^{-1}}$ |
| $c_a$ | Free air CO$_2$ mixing ratio | $\mathrm{\mu mol_C\,mol^{-1}}$ |
| $z_a$ | Height of the reference point above canopy | m |
| $\dot{W}_{\infty,a}$ | Precipitation mass rate | $\mathrm{kg_W\,m^{-2}\,s^{-1}}$ |
| $\dot{Q}_{\mathrm{TIR}(\infty,a)}^{\Downarrow}$ | Downward thermal infrared irradiance | $\mathrm{W\,m^{-2}}$ |
| $\dot{Q}_{\mathrm{PAR}(\infty,a)}^{\odot}$ | Downward photosynthetically active irradiance, direct | $\mathrm{W\,m^{-2}}$ |
| $\dot{Q}_{\mathrm{PAR}(\infty,a)}^{\Downarrow}$ | Downward photosynthetically active irradiance, diffuse | $\mathrm{W\,m^{-2}}$ |
| $\dot{Q}_{\mathrm{NIR}(\infty,a)}^{\odot}$ | Downward near infrared irradiance, direct | $\mathrm{W\,m^{-2}}$ |
| $\dot{Q}_{\mathrm{NIR}(\infty,a)}^{\Downarrow}$ | Downward near infrared irradiance, diffuse | $\mathrm{W\,m^{-2}}$ |

interdiurnal variability. Alternatively, the meteorological forcing may be provided directly by BRAMS (Knox et al., 2015; Swann et al., 2015).

## 3 Overview of enthalpy, water, and carbon dioxide cycles

Here we present the fundamental equations that describe the biogeophysical and biogeochemical cycles. Because the environmental conditions are a function of the local plant community and resources are shared by the individuals, these cycles must be described at the patch level, and the response of the plant community can be aggregated to the polygon level once the cycles are resolved for each patch. In ED-2.2, patches do not exchange enthalpy, water, and carbon dioxide with other patches; thus patches are treated as independent systems. Throughout this section, we will only refer to the patch- and cohort-levels, and indices associated with patches, sites and polygons will be omitted for clarity.

### 3.1 Definition of the thermodynamic state

Each patch is defined by a *thermodynamic envelope* (Fig. 2), comprised of multiple thermodynamic systems: each soil layer (total number of layers $N_G$), each temporary surface water or snow layer (total number of layers $N_S$), aboveground part each cohort (total number of cohorts $N_T$), and the canopy air space. For simplicity, roots are assumed to be in thermal equilibrium





with the soil layers and have negligible heat capacity compared to the soil layers. Although patches do not exchange heat and mass with other patches, they are allowed to exchange heat and mass with the free air (i.e. the atmosphere above and outside of the air-space control-volume we deem as within canopy) and lose water and associated energy through surface and sub-surface runoff. We also assume that intensive variables such as pressure and temperature are uniform within each thermodynamic

system. Note that free air is not considered a thermodynamic system in ED-2 because the thermodynamic state is determined directly from the boundary conditions, and thus external to the model.

The fundamental equations that describe the system thermodynamics are the first law of thermodynamics in terms of enthalpy $H$ ($\mathrm{J\,m^{-2}}$), and mass continuity for incompressible fluids for total water mass $W$ ($\mathrm{kg_W\,m^{-2}}$):

$$\underbrace{\frac{\mathrm{d}H}{\mathrm{d}t}}_{\text{Change in enthalpy}} = \underbrace{\dot{Q}}_{\text{Net heat flux}} + \underbrace{\dot{H}}_{\text{Enthalpy flux due to mass flux}} - \underbrace{\mathcal{V}\frac{\mathrm{d}p}{\mathrm{d}t}}_{\text{Pressure change}} , \qquad (4)$$

$$\underbrace{\frac{\mathrm{d}W}{\mathrm{d}t}}_{\text{Change in water mass}} = \underbrace{\dot{W}}_{\text{Net water mass flux}} , \qquad (5)$$

where $\mathcal{V}$ is the volume of the thermodynamic system and $p$ is the ambient pressure. The components in the right-hand side of Eq. (4) and Eq. (5) depend on the thermodynamic system, and will be presented in detail in the following sections. Net heat fluxes ($\dot{Q}$) represent changes in enthalpy that are not associated with mass exchange (radiative and sensible heat fluxes), whereas the remaining enthalpy fluxes ($\dot{H}$) correspond to changes in heat capacity due to addition or removal of mass from

each thermodynamic system.

The merit of solving the changes in enthalpy over internal energy is that changes in enthalpy are equivalent to the net energy flux when pressure is constant (Eq. 4). Pressure is commonly included in atmospheric measurements, making it easy to track changes in enthalpy not related to energy fluxes. In reality, the only thermodynamic system where the distinction between internal energy and enthalpy matters is the canopy air space. Work associated with thermal expansion of solids and

liquids is several orders of magnitude smaller than heat (Dufour and van Mieghem, 1975), and changes in pressure contribute significantly less to enthalpy because the specific volume of solids and liquids are comparatively small. Likewise, enthalpy fluxes that do not involve gas phase (e.g. canopy dripping and runoff) are nearly indistinguishable from internal energy flux, whereas differences between enthalpy and internal energy fluxes are significant when gas phase is involved (e.g. transpiration and eddy flux). For simplicity, from this point on we will use the term *enthalpy* whenever internal energy is indistinguishable

from enthalpy. The complete list of state variables in ED-2.2 is shown in Table 3.

Variations in enthalpy are more important than their actual values, but they must be consistently defined relative to a pre-determined and known thermodynamic state, at which we define enthalpy to be zero. For any material other than water, enthalpy is defined as zero when the material temperature is $0\,\mathrm{K}$; for water, enthalpy is defined as zero when water is at $0\,\mathrm{K}$ and completely frozen. The general definitions of enthalpy and internal energy states used in all thermodynamic systems in

ED-2.2 are described in Supplement S4. In ED-2.2, enthalpy is used as the prognostic variable because these are directly and linearly related to the governing ordinary differential equation (Eq. 4). Temperature is diagnostically obtained based on





**Table 3.** List of state variables solved in ED-2.2. Unless otherwise noted, the reference equation is the ordinary differential equation that defines the rate of change of the thermodynamic state. The list of fluxes that describe the thermodynamic state is presented in Table 4. For a complete list of subscripts and variables used in this manuscript, refer to Tables S1-S2.

| State variable | Description | Units | Budget equation |
|---|---|---|---|
| $c_c$ | $CO_2$ mixing ratio — canopy air space | $\mu mol_C\,mol^{-1}$ | (23) |
| $h_c$ | Specific enthalpy — canopy air space | $J\,kg^{-1}$ | (18) |
| $\mathcal{H}_{g_j}$ | Volumetric enthalpy — soil layer $j$ | $J\,m^{-3}$ | (4)[a] |
| $H_{s_j}$ | Enthalpy — temporary surface water layer $j$ | $J\,m^{-2}$ | (4) |
| $H_{t_k}$ | Enthalpy — cohort $k$ | $J\,m^{-2}$ | (4) |
| $p_c$ | Atmospheric pressure — canopy air space | Pa | (S66)[b] |
| $w_c$ | Specific humidity — canopy air space | $kg_W\,kg^{-1}$ | (19) |
| $W_{s_j}$ | Water mass — TSW layer $j$ | $kg_W\,m^{-2}$ | (5) |
| $W_{t_k}$ | Intercepted/dew/frost water mass — cohort $k$ | $kg_W\,m^{-2}$ | (5) |
| $z_c$ | Depth (specific volume) — canopy air space | m | (17)[c] |
| $\vartheta_{g_j}$ | Volumetric soil moisture — soil layer $j$ | $m_W^3\,m^{-3}$ | (5)[a] |

[a] Budget fluxes are in units of area, and the state variable is updated following the conversion described in Section 3.2.1.

[b] Canopy air space pressure is not solved using ordinary differential equations, but based on the atmospheric pressure from the meteorological forcing.

[c] Canopy air space depth is determined from vegetation characteristics, not from an ordinary differential equation.

the heat capacity of each thermodynamic system, and the heat capacities of different thermodynamic systems are defined in Supplement S5.

## 3.2   Heat ($\dot{Q}$), water ($\dot{W}$), and enthalpy ($\dot{H}$) fluxes

The enthalpy and water cycles for each patch in ED-2.2 are summarized in Fig. 2, and these cycles are solved every thermody-
5   namic sub-step ($\Delta t_{\text{Thermo}}$), using a fourth-order Runge-Kutta integrator with dynamic time steps to maintain the error within prescribed tolerance. For all fluxes and variables, we follow the subscript notation described in Table S1, and denote flux variables with a dot and two indices separated by a comma, denoting the systems impacted by the flux. Fluxes are positive when they go from the system represented by the first subscript towards the second subscript; arrows in Fig. 2 represent allowed directions. The list of fluxes solved in ED-2.2 is provided in Table 4, and a complete list of variables is provided in Table S2.
10   In addition, the values of global constants and global parameters are listed in Tables S3 and S4, respectively, and the default parameters specific for each tropical plant functional type are presented in Table S5; similar parameters for temperate plant functional types are found in Medvigy et al. (2009).





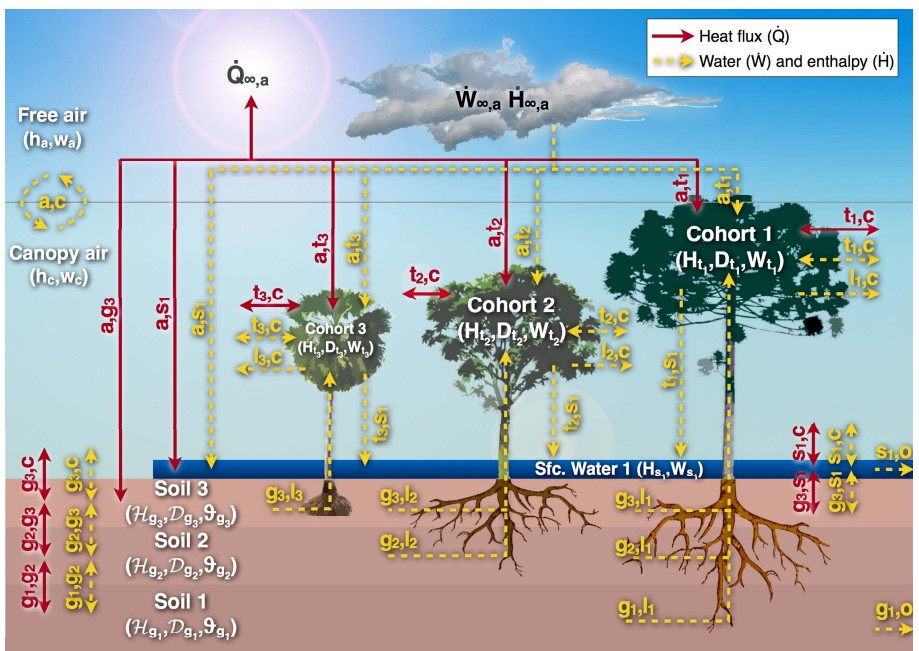

**Figure 2.** Schematic of the fluxes that are solved in ED-2.2 for a single patch (thermodynamic envelope). In this example, the patch has three cohorts, a single surface water layer, and three soil layers. Letters near the arrows are the subscripts associated with fluxes, although the flux variable has been omitted here for clarity. Solid red arrows represent heat flux with no exchange of mass, and dashed blue arrows represent exchange of mass and associated enthalpy. Arrows that point to a single direction represent fluxes that can only go in one (non-negative) direction, and arrows pointing to both directions represent fluxes that can be positive, negative, or zero.

### 3.2.1 Soil

In ED-2.2, the soil characteristics (depth, number of layers, texture and color) are defined during initialization and assumed fixed in time. Within each patch, each soil layer (comprised by soil matrix and soil water in each layer) is considered a separate thermodynamic system, with the main size dimension being the layer thickness $\Delta z_{g_j}$, with $j = 1$ being the deepest soil layer, and $j = N_G$ being the topmost soil layer. Typically, the top layer thickness is set to $\Delta z_{g_{N_G}} = 0.02\,\mathrm{m}$, which is a compromise between computational efficiency and ability to represent the stronger gradients near the surface, and layers with increasing thickness ($\Delta z_{g_j}$) are added for the entire rooting zone.

The thermodynamic state is defined in terms of the soil volume: the bulk specific enthalpy $\mathcal{H}_{g_j}$ ($\mathrm{J\,m^{-3}}$) and volumetric soil water content $\vartheta_{g_j}$ ($\mathrm{m^3_W\,m^{-3}}$), which can be related to Eq. (4)-(5) by defining $H_{g_j} = \mathcal{H}_{g_j}\Delta z_{g_j}$ and $W_{g_j} = \rho_\ell \cdot \vartheta_{g_j} \cdot \Delta z_{g_j}$, where $\rho_\ell$ is the density of liquid water (Table S3). Soil net fluxes for any layer $j$ are defined as:

$$\underbrace{\dot{Q}_{g_j}}_{\substack{\text{Net heat flux}}} = \underbrace{\dot{Q}_{g_{j-1},g_j} - \dot{Q}_{g_j,g_{j+1}}}_{\substack{\text{Net sensible heat flux} \\ \text{between consecutive layers} \\ (4.1)}} + \underbrace{\delta_{g_j g_{N_G}}\,\dot{Q}_{a,g_{N_G}}}_{\substack{\text{Absorbed irradiance} \\ (4.3.2)}} - \underbrace{\delta_{g_j g_{N_G}}\,\dot{Q}_{g_{N_G},c}}_{\substack{\text{Ground–CAS sensible heat} \\ (4.5.2 \text{ and } 4.5.3)}} \ , \tag{6}$$

**Table 4.** List of energy, water, and carbon dioxide fluxes that define the thermodynamic state in ED-2.2, along with sections and equation that define them. Fluxes are denoted by a dotted letter, and two subscripts separated with a comma: $\dot{X}_{m,n}$. Positive fluxes go from thermodynamic system $m$ to thermodynamic system $n$; negative fluxes go in the opposite direction. Acronyms in the description column: canopy air space (CAS); temporary surface water (TSW). The complete list of subscripts and variables used in this manuscript is available in Tables S1-S2, and the list of state variables is shown in Table 3.

| Variable | Description | Section | Equation | Units |
|---|---|---|---|---|
| $\dot{C}_{a,c}$ | $CO_2$ flux from turbulent mixing | 4.4 | 55 | |
| $\dot{C}_{e_j,c}$ | Heterotrophic respiration flux (soil carbon pool $e_j$) | 4.8 | 100 | |
| $\dot{C}_{l_k,c}$ | Net leaf (cohort $k$)–CAS $CO_2$ flux[a] | 4.6 | 91 | $mol_C\ m^{-2}\ s^{-1}$ |
| $\dot{C}_{n_k,c}$ | Storage turnover (cohort $k$) respiration flux | 4.7 | 98 | |
| $\dot{C}_{r_k,c}$ | Fine-root (cohort $k$) metabolic respiration flux | 4.7 | 97 | |
| $\dot{C}_{\Delta_k,c}$ | Growth and maintenance (cohort $k$) respiration flux | 4.7 | 99 | |
| $\dot{Q}_{a,g_{NG}}$ | Net absorbed irradiance (topmost soil layer) | 4.3.2 | 52 | |
| $\dot{Q}_{a,s_j}$ | Net absorbed irradiance (TSW layer $j$) | 4.3.2 | 51 | |
| $\dot{Q}_{a,t_k}$ | Net absorbed irradiance (cohort $k$) | 4.3.1 | 49 | |
| $\dot{Q}_{g_{NG},c}$ | Ground–CAS net sensible heat flux | 4.5.2 | 66 | $W\ m^{-2}$ |
| $\dot{Q}_{g_{j-1},g_j}$ | Net sensible heat flux between two soil layers | 4.1 | 26 | |
| $\dot{Q}_{s_{NS},c}$ | TSW–CAS net sensible heat flux | 4.5.2 | 64 | |
| $\dot{Q}_{s_{j-1},s_j}$ | Net sensible heat flux between two TSW layers | 4.1 | 27 | |
| $\dot{Q}_{t_k,c}$ | Cohort $k$–CAS net sensible heat flux | 4.5.1 | 58 | |
| $\dot{H}_{a,c}$ | Enthalpy flux from turbulent mixing at the top of CAS. | 4.4 | 54 | |
| $\dot{H}_{a,s_{NS}}$ | Enthalpy flux to the top TSW layer associated with throughfall precipitation | 4.2 | 42 | |
| $\dot{H}_{a,t_k}$ | Enthaply flux associated with rainfall interception by cohort $k$ | 4.2 | 41 | |
| $\dot{H}_{g_{NG},c}$ | Enthalpy flux associated with ground–CAS evaporation[b] | 4.5.3 | 73 | |
| $\dot{H}_{g_{j-1},g_j}$ | Enthalpy flux associated with water percolation between two soil layers | 4.1 | 35 | |
| $\dot{H}_{g_j,l_k}$ | Enthalpy flux associated with soil water extraction from soil layer $j$ by cohort $k$ | 4.6 | 95 | $W\ m^{-2}$ |
| $\dot{H}_{g_1,o}$ | Enthalpy flux associated with sub-surface runoff from the bottom soil layer | 4.1 | 29 | |
| $\dot{H}_{l_k,c}$ | Enthalpy flux associated with transpiration by cohort $k$ | 4.6 | 96 | |
| $\dot{H}_{s_{NS},c}$ | Enthalpy flux associated with TSW–CAS evaporation[b] | 4.5.3 | 73 | |
| $\dot{H}_{s_{NS},o}$ | Enthalpy flux associated with surface runoff from the top TSW layer | 4.1 | 34 | |
| $\dot{H}_{s_{j-1},s_j}$ | Enthalpy flux associated with water percolation between two TSW layers | 4.1 | 35 | |
| $\dot{H}_{t_k,c}$ | Enthalpy flux associated with evaporation[b] of intercepted water (cohort $k$) | 4.5.3 | 73 | |
| $\dot{H}_{t_k,s_{NS}}$ | Enthalpy flux associated with canopy dripping from cohort $k$ to the top TSW layer | 4.2 | 44 | |
| $\dot{W}_{a,c}$ | Water flux from turbulent mixing at the top of CAS. | 4.4 | 53 | |
| $\dot{W}_{a,s_{NS}}$ | Precipitation throughfall flux to the top TSW layer | 4.2 | 37 | |
| $\dot{W}_{a,t_k}$ | Water flux from rainfall interception (cohort $k$) | 4.2 | 36 | |
| $\dot{W}_{g_{NG},c}$ | Ground–CAS evaporation[b] flux | 4.5.2 | 67 | |
| $\dot{W}_{g_{j-1},g_j}$ | Water percolation flux between two soil layers | 4.1 | 28 | |
| $\dot{W}_{g_j,l_k}$ | Water flux associated with soil water extraction by plants | 4.6 | 94 | $kg_W\ m^{-2}\ s^{-1}$ |
| $\dot{W}_{g_1,o}$ | Water flux associated with sub-surface runoff from the bottom soil layer | 4.1 | 33 | |
| $\dot{W}_{l_k,c}$ | Transpiration flux (cohort $k$) | 4.6 | 92 | |
| $\dot{W}_{s_{j-1},s_j}$ | Water percolation flux between two TSW layers | 4.1 | 30–31 | |
| $\dot{W}_{s_{NS},o}$ | Surface runoff water flux from the top TSW layer | 4.1 | 34 | |
| $\dot{W}_{s_{NS},c}$ | TSW–CAS evaporation[b] flux | 4.5.2 | 65 | |
| $\dot{W}_{t_k,c}$ | Evaporation[b] flux from intercepted water (cohort $k$) | 4.5.1 | 59 | |
| $\dot{W}_{t_k,s_{NS}}$ | Canopy dripping flux from cohort $k$ to the top TSW layer | 4.2 | 43 | |

[a] Net flux between leaf respiration (positive) and gross primary productivity (negative).
[b] When negative, this flux corresponds to dew or frost formation.

$$\underbrace{\dot{H}_{g_j}}_{\substack{\text{Net enthalpy flux} \\ \text{due to water flux}}} = \underbrace{\dot{H}_{g_{j-1},g_j} - \dot{H}_{g_j,g_{j+1}}}_{\substack{\text{Water percolation} \\ \text{between consecutive layers} \\ (4.1)}} - \underbrace{\delta_{g_jg_1}\dot{H}_{g_1,o}}_{\substack{\text{Sub-surface runoff} \\ (4.1)}} - \underbrace{\delta_{g_jg_{N_G}}\dot{H}_{g_{NG},c}}_{\substack{\text{Gnd. Evaporation} \\ (4.5.2 \text{ and } 4.5.3)}} - \underbrace{\sum_{k=1}^{N_T}\dot{H}_{g_j,l_k}}_{\substack{\text{Water uptake} \\ \text{by cohorts} \\ (4.6)}}, \tag{7}$$





$$\underbrace{\dot{W}_{g_j}}_{\text{Net water flux}} = \underbrace{\dot{W}_{g_{j-1},g_j} - \dot{W}_{g_j,g_{j+1}}}_{\substack{\text{Water percolation} \\ \text{between consecutive layers} \\ (4.1)}} - \underbrace{\delta_{g_j g_1} \dot{W}_{g_1,o}}_{\substack{\text{Sub-surface runoff} \\ (4.1)}} - \underbrace{\delta_{g_j g_{N_G}} \dot{W}_{g_{N_G},c}}_{\substack{\text{Gnd. Evaporation} \\ (4.5.2 \text{ and } 4.5.3)}} - \underbrace{\sum_{k=1}^{N_T} \dot{W}_{g_j,l_k}}_{\substack{\text{water uptake} \\ \text{by cohorts} \\ (4.6)}}, \tag{8}$$

where $\delta_{g_j g_{j'}}$ is the Kronecker delta (1 if $j = j'$, 0 otherwise), CAS is the canopy air space, and subscript $o$ denotes the loss through runoff. References in parentheses underneath the terms correspond to the sections in which each term is presented in detail. The equations above assume $(\dot{Q}_{g_0,g_1}; \dot{H}_{g_0,g_1}; \dot{W}_{g_0,g_1})$ to be zero, and $(\dot{Q}_{g_{N_G},g_{N_G+1}}; \dot{H}_{g_{N_G},g_{N_G+1}}; \dot{W}_{g_{N_G},g_{N_G+1}})$ to be equivalent to $(\dot{Q}_{g_{N_G},s_1}; \dot{H}_{g_{N_G},s_1}; \dot{W}_{g_{N_G},s_1})$, which are the fluxes between topmost soil layer and bottommost temporary surface

water layer (see also section 3.2.2).

### 3.2.2 Temporary surface water (TSW)

Temporary surface water (TSW) exists whenever water falls to the ground, or dew or frost develops on the ground. The layer will be maintained only when the amount of water that reaches the ground exceeds the water holding capacity of the top soil layer (a function of the soil porosity), or when precipitation falls as snow. The maximum number of temporary surface water

layers $N_S^{\max}$ is defined by the user, but the actual number of layers $N_S$ and the thickness of each layer depends on the total mass and the water phase, following Walko et al. (2000). When the layer is in liquid phase, only one layer ($N_S = 1$) is maintained. If a snowpack develops, the temporary surface water can be divided into several layers (subscript $j$, with $j = 1$ being the deepest soil layer, and $j = N_S$ being the topmost TSW layer), and the thickness of each layer ($\Delta z_{s_j}$) is defined using the same algorithm as LEAF-2 (Walko et al., 2000). Net TSW fluxes are defined as:

$$\underbrace{\dot{Q}_{s_j}}_{\text{Net heat flux}} = \underbrace{\dot{Q}_{s_{j-1},s_j} - \dot{Q}_{s_j,s_{j+1}}}_{\substack{\text{Net sensible heat flux} \\ \text{between consecutive layers} \\ (4.1)}} + \underbrace{\dot{Q}_{a,s_j}}_{\substack{\text{Absorbed irradiance} \\ (4.3.2)}} - \underbrace{\delta_{s_j s_{N_S}} \dot{Q}_{s_{N_S},c}}_{\substack{\text{Ground–CAS sensible heat} \\ (4.5.2 \text{ and } 4.5.3)}} , \tag{9}$$

$$\underbrace{\dot{H}_{s_j}}_{\substack{\text{Net enthalpy flux} \\ \text{due to water flux}}} = \underbrace{\dot{H}_{s_{j-1},s_j} - \dot{H}_{s_j,s_{j+1}}}_{\substack{\text{Water percolation} \\ \text{between consecutive layers} \\ (4.1)}} + \underbrace{\delta_{s_j s_{N_S}} \dot{H}_{a,s_{N_S}}}_{\substack{\text{Throughfall} \\ \text{precipitation} \\ (4.2)}} + \underbrace{\delta_{s_j s_{N_S}} \left( \sum_{k=1}^{N_T} \dot{H}_{t_k,s_{N_S}} \right)}_{\substack{\text{Canopy dripping} \\ \text{from cohorts} \\ (4.2)}} - \underbrace{\delta_{s_j s_{N_S}} \dot{H}_{s_{N_S},o}}_{\substack{\text{Surface runoff} \\ (4.1)}} - \underbrace{\delta_{s_j s_{N_S}} \dot{H}_{s_{N_S},c}}_{\substack{\text{Surface water} \\ \text{evaporation} \\ (4.5.2 \text{ and } 4.5.3)}}, \tag{10}$$

$$\underbrace{\dot{W}_{s_j}}_{\text{Water flux}} = \underbrace{\dot{W}_{s_{j-1},s_j} - \dot{W}_{s_j,s_{j+1}}}_{\substack{\text{Water percolation} \\ \text{between consecutive layers} \\ (4.1)}} + \underbrace{\delta_{s_j s_{N_S}} \dot{W}_{a,s_{N_S}}}_{\substack{\text{Throughfall} \\ \text{precipitation} \\ (4.2)}} + \underbrace{\delta_{s_j s_{N_S}} \left( \sum_{k=1}^{N_T} \dot{W}_{t_k,s_{N_S}} \right)}_{\substack{\text{Canopy dripping} \\ \text{from cohorts} \\ (4.2)}} - \underbrace{\delta_{s_j s_{N_S}} \dot{W}_{s_{N_S},o}}_{\substack{\text{Surface runoff} \\ (4.1)}} - \underbrace{\delta_{s_j s_{N_S}} \dot{W}_{s_{N_S},c}}_{\substack{\text{Surface water} \\ \text{evaporation} \\ (4.5.2 \text{ and } 4.5.3)}}, \tag{11}$$

where $\delta_{s_j s_{j'}}$ is the Kronecker delta (1 if $j = j'$, 0 otherwise), CAS is the canopy air space, and subscript $o$ denotes loss from the thermodynamic envelope through runoff. Terms are described in detail in the sections shown underneath each term. Similarly

to the soil fluxes (Section 3.2.1), we assume that $(\dot{Q}_{s_0,s_1}; \dot{H}_{s_0,s_1}; \dot{W}_{s_0,s_1})$ is equivalent to $(\dot{Q}_{g_{N_G},s_1}; \dot{H}_{g_{N_G},s_1}; \dot{W}_{g_{N_G},s_1})$, the





fluxes between the topmost soil layer and the bottommost TSW layer; and that $(\dot{Q}_{s_{N_S},s_{N_S+1}}; \dot{H}_{s_{N_S},s_{N_S+1}}; \dot{W}_{s_{N_S},s_{N_S+1}})$ are all zero, as layer $N_S + 1$ does not exist.

### 3.2.3 Vegetation

In ED-2.2, vegetation is solved as an independent thermodynamic system only if the cohort is sufficiently large. The minimum size is an adjustable parameter and the typical minimum heat capacity solved by ED-2.2 is on the order of $10 \, \mathrm{J \, m^{-2} \, K^{-1}}$ and total area index of $0.005 \mathrm{m^2_{leaf+wood} \, m^{-2}}$. Cohorts smaller than this are excluded from all energy and water cycle calculations and assumed to be in thermal equilibrium with canopy air space. The net fluxes of heat, enthalpy and water each cohort $k$ that can be resolved are:

$$\underbrace{\dot{Q}_{t_k}}_{\substack{\text{Net heat flux}}} = \underbrace{\dot{Q}_{a,t_k}}_{\substack{\text{Cohort's net} \\ \text{absorbed irradiance} \\ (4.3.1)}} - \underbrace{\dot{Q}_{t_k,c}}_{\substack{\text{Cohort–CAS} \\ \text{sensible heat} \\ (4.5.1)}}, \tag{12}$$

$$\underbrace{\dot{H}_{t_k}}_{\substack{\text{Net enthalpy flux} \\ \text{due to water flux}}} = \underbrace{\dot{H}_{a,t_k}}_{\substack{\text{Rainfall} \\ \text{interception} \\ (4.2)}} - \underbrace{\dot{H}_{t_k,s_{N_S}}}_{\substack{\text{Canopy dripping} \\ (4.2)}} + \underbrace{\left(\sum_{j=1}^{N_G} \dot{H}_{g_j,l_k}\right)}_{\substack{\text{Ground water} \\ \text{uptake (transpiration)} \\ (4.6)}} - \underbrace{\dot{H}_{l_k,c}}_{\substack{\text{Transpiration} \\ (4.6)}} - \underbrace{\dot{H}_{t_k,c}}_{\substack{\text{Evaporation of} \\ \text{intercepted water} \\ (4.5.3)}}, \tag{13}$$

$$\underbrace{\dot{W}_{t_k}}_{\substack{\text{Water flux}}} = \underbrace{\dot{W}_{a,t_k}}_{\substack{\text{Rainfall} \\ \text{interception} \\ (4.2)}} - \underbrace{\dot{W}_{t_k,s_{N_S}}}_{\substack{\text{Canopy dripping} \\ (4.2)}} + \underbrace{\left(\sum_{j=1}^{N_G} \dot{W}_{g_j,l_k}\right)}_{\substack{\text{Ground water} \\ \text{uptake (transpiration)} \\ (4.6)}} - \underbrace{\dot{W}_{l_k,c}}_{\substack{\text{Transpiration} \\ (4.6)}} - \underbrace{\dot{W}_{t_k,c}}_{\substack{\text{Evaporation of} \\ \text{intercepted water} \\ (4.5.3)}}. \tag{14}$$

Each term is described in detail in the sections shown underneath each term on the right-hand side of Eq. (12)-(14).

### 3.2.4 Canopy air space (CAS)

The canopy air space is a gas, therefore extensive properties akin to the other thermodynamic systems are not intuitive because total mass and total volume cannot be directly compared to observations. Therefore, all prognostic and diagnostic variables are solved in the intensive form. Total enthalpy $H_c$ and total water mass $W_c$ of the canopy air space can be written in terms of air density $\rho_c$ and the equivalent depth of the canopy air space $\overline{z}_c$ as:

$$H_c = \rho_c \overline{z}_c h_c, \tag{15}$$

$$W_c = \rho_c \overline{z}_c w_c, \tag{16}$$

$$\overline{z}_c = \max\left(5.0, \frac{\sum_{k=1}^{N_{T(\text{canopy})}} n_{t_k} \mathrm{BA}_{t_k} z_{t_k}}{\sum_{k=1}^{N_{T(\text{canopy})}} n_{t_k} \mathrm{BA}_{t_k}}\right), \tag{17}$$

where $\mathrm{BA}_{t_k}$ (cm$^2$) and $z_{t_k}$ (m) are the basal area and the height of cohort $k$, respectively; and $N_{T(\text{canopy})}$ is the number of cohorts that are in the canopy, and we assume that cohorts are ordered from tallest to shortest. In case the canopy is open,



$N_{T(\text{canopy})}$ is the total number of cohorts, and a minimum value of 5m is imposed when vegetation is absent or too short, to prevent numerical instabilities. Because the equivalent canopy depth depends only on the cohort size, $\overline{z}_c$ is updated at the cohort dynamics step ($\Delta t_{\text{CD}}$, Table 1). If we substitute Eq. (15) and Eq. (16) into Eq. (4) and Eq. (5), respectively, and assume that changes in density over short time steps are much smaller than changes in enthalpy or humidity, and that changes we obtain the following equations for the canopy air space budget:

$$\frac{\mathrm{d}h_c}{\mathrm{d}t} = \frac{1}{\rho_c \overline{z}_c} \left( \dot{Q}_c + \dot{H}_c + \overline{z}_c \frac{\mathrm{d}p_c}{\mathrm{d}t} \right), \tag{18}$$

$$\frac{\mathrm{d}w_c}{\mathrm{d}t} = \frac{1}{\rho_c \overline{z}_c} \dot{W}_c, \tag{19}$$

where

$$\underbrace{\dot{Q}_c}_{\text{Net heat flux}} = \underbrace{\left( \sum_{k=1}^{N_T} \dot{Q}_{t_k,c} \right)}_{\substack{\text{Cohort–CAS} \\ \text{sensible heat (4.5.1 and 4.5.3)}}} + \underbrace{\dot{Q}_{s_{N_S},c}}_{\substack{\text{Surface water–CAS} \\ \text{sensible heat (4.5.2 and 4.5.3)}}} + \underbrace{\dot{Q}_{g_{N_G},c}}_{\substack{\text{Ground–CAS} \\ \text{sensible heat (4.5.2 and 4.5.3)}}}, \tag{20}$$

$$\underbrace{\dot{H}_c}_{\text{Net enthalpy flux}} = \underbrace{\dot{H}_{a,c}}_{\substack{\text{Enthalpy flux from} \\ \text{Turbulent mixing (4.4)}}} + \underbrace{\left( \sum_{k=1}^{N_T} \dot{H}_{t_k,c} \right)}_{\substack{\text{Evaporation of} \\ \text{intercepted water} \\ \text{(4.5.1 and 4.5.3)}}} + \underbrace{\left( \sum_{k=1}^{N_T} \dot{H}_{l_k,c} \right)}_{\substack{\text{Transpiration} \\ \text{(4.6)}}} + \underbrace{\dot{H}_{s_{N_S},c}}_{\substack{\text{Surface water} \\ \text{evaporation} \\ \text{(4.5.2 and 4.5.3)}}} + \underbrace{\dot{H}_{g_{N_G},c}}_{\substack{\text{Ground evaporation} \\ \text{(4.5.2 and 4.5.3)}}}, \tag{21}$$

$$\underbrace{\dot{W}_c}_{\text{Water flux}} = \underbrace{\dot{W}_{a,c}}_{\substack{\text{Water flux from} \\ \text{Turbulent mixing (4.4)}}} + \underbrace{\left( \sum_{k=1}^{N_T} \dot{W}_{t_k,c} \right)}_{\substack{\text{Evaporation of} \\ \text{intercepted water} \\ \text{(4.5.1 and 4.5.3)}}} + \underbrace{\left( \sum_{k=1}^{N_T} \dot{W}_{l_k,c} \right)}_{\substack{\text{Transpiration} \\ \text{(4.6)}}} + \underbrace{\dot{W}_{s_{N_S},c}}_{\substack{\text{Surface water} \\ \text{evaporation} \\ \text{(4.5.2 and 4.5.3)}}} + \underbrace{\dot{W}_{g_{N_G},c}}_{\substack{\text{Ground evaporation} \\ \text{(4.5.2 and 4.5.3)}}}. \tag{22}$$

Unlike in the other thermodynamic systems (soil, temporary surface water, and vegetation), the net enthalpy flux of the canopy air space is not exclusively due to associated water flux: the eddy flux between the free air and the canopy air space ($\dot{H}_{a,c}$) includes both water transport and flux associated with mixing of air with different temperatures, and thus enthalpy, between canopy air space and free air.

In addition, we must also track the canopy-air-space pressure $p_c$. In ED-2.2, CAS pressure is not solved through a differential equation: instead $p_c$ is updated whenever the meteorological forcing is updated, using the ideal gas law and hydrostatic equilibrium following the method described in Supplement S6. The rate of change of canopy air pressure is then applied in Eq. (18). Likewise, CAS density ($\rho_c$) is updated at the end of each thermodynamic step to ensure that the CAS conforms to the ideal gas law.

### 3.3 Carbon dioxide cycle

In ED-2.2, the carbon dioxide cycle is a subset of the full carbon cycle, which is shown in Fig. 3. The canopy air space is the only thermodynamic system with $CO_2$ storage that is solved by ED-2.2; nonetheless, we assume that the contribution of $CO_2$





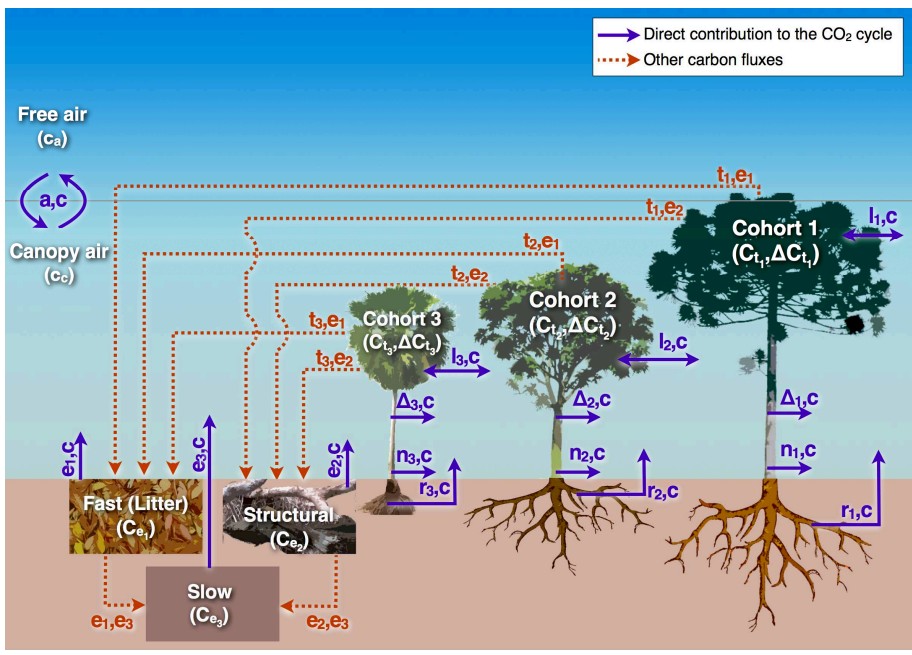

**Figure 3.** Schematic of the patch-level carbon cycle solved in ED-2.2 for a patch containing two cohorts. Like Figure 2, letters near the arrows are the subscripts associated with fluxes. Fluxes shown in black lines are part of the $CO_2$ cycle discussed in this manuscript, and grey lines are part of the carbon cycle but do not directly affect the $CO_2$ flux; these fluxes are summarized in Supplements S2 and S3.

to density and heat capacity of the canopy air space is negligible, hence only the molar $CO_2$ mixing ratio $c_c$ ($\mathrm{mol_C\,mol^{-1}}$) is traced.

The change in $CO_2$ storage in the canopy air space is determined by the following differential equation:

$$\frac{\mathrm{d}c_c}{\mathrm{d}t} = \frac{\mathcal{M}_d}{\mathcal{M}_C}\frac{1}{\rho_c\,\overline{z}_c}\dot{C}_c, \tag{23}$$

$$\underbrace{\dot{C}_c}_{\substack{\text{Net carbon flux}}} = \underbrace{\dot{C}_{a,c}}_{\substack{\text{Carbon flux from}\\\text{Turbulent mixing}\\(4.4)}} + \underbrace{\sum_{k=1}^{N_T}\dot{C}_{l_k,c}}_{\substack{\text{Net Leaf–CAS flux}\\\text{(Respiration-GPP)}\\(4.6)}} + \underbrace{\sum_{k=1}^{N_T}\dot{C}_{r_k,c}}_{\substack{\text{Fine-root}\\\text{Respiration}\\(4.7)}} + \underbrace{\sum_{k=1}^{N_T}\dot{C}_{n_k,c}}_{\substack{\text{Storage turnover}\\\text{Respiration}\\(4.7)}} + \underbrace{\sum_{k=1}^{N_T}\dot{C}_{\Delta_k,c}}_{\substack{\text{Growth and maintenance}\\\text{Respiration}\\(4.7)}} + \underbrace{\sum_{j=1}^{3}\dot{C}_{e_j,c}}_{\substack{\text{Heterotrophic}\\\text{Respiration}\\(4.8)}}, \tag{24}$$

where $\mathcal{M}_d$ and $\mathcal{M}_C$ are the molar masses of dry air and carbon, respectively, used to convert mass to molar fraction ($1\,\mathrm{mol_C} =$

5   $1\,\mathrm{mol_{CO_2}}$). The terms on the right-hand side of Eq. (24) are described in detail in the sections displayed underneath each term. The net leaf–CAS flux ($\dot{C}_{l_k,c}$) for any cohort $k$ is positive when leaf respiration exceeds photosynthetic assimilation. The heterotrophic respiration is based on a simplified implementation of the CENTURY model (Bolker et al., 1998) that combines the decomposition rates from three soil carbon pools, defined by their characteristic life time: fast (metabolic litter and microbial; $e_1$), intermediate (structural debris; $e_2$), and slow (humified and passive soil carbon; $e_3$). Note that the soil

10   carbon pools are not directly related to the soil layers used to describe the thermodynamic state (Section 3.2.1).





In addition to canopy air space, we also define a virtual cohort pool of carbon corresponding to the accumulated carbon balance ($C_{\Delta_k}$). The accumulated carbon balance links short-term carbon cycle components such as photosynthesis and respiration with long-term dynamics that depend on carbon balance such as such as carbon allocation to growth and reproduction, and mortality (Long-term dynamics described in Supplement S2). The accumulated carbon balance is defined by the following equation:

$$\underbrace{\frac{\mathrm{d}C_{\Delta_k}}{\mathrm{d}t}}_{\substack{\text{Change in}\\\text{carbon balance}}} = -\underbrace{\dot{C}_{l_k,c}}_{\substack{\text{Net Leaf–CAS flux}\\\text{(Respiration-GPP)}\\\text{(4.6)}}} - \underbrace{\dot{C}_{r_k,c}}_{\substack{\text{Fine-root}\\\text{Respiration}\\\text{(4.7)}}} - \underbrace{\dot{C}_{n_k,c}}_{\substack{\text{Storage turnover}\\\text{Respiration}\\\text{(4.7)}}} - \underbrace{\dot{C}_{\Delta_k,c}}_{\substack{\text{Growth and maintenance}\\\text{Respiration}\\\text{(4.7)}}} - \underbrace{\dot{C}_{t_k,e_1}}_{\substack{\text{Turnover of}\\\text{non-lignified litter}\\\text{(S3)}}} - \underbrace{\dot{C}_{t_k,e_2}}_{\substack{\text{Turnover of}\\\text{lignified litter}\\\text{(S3)}}} , \qquad (25)$$

where $\dot{C}_{t_k,e_1}$ and $\dot{C}_{t_k,e_2}$ are the individual carbon losses caused by leaf shedding and turnover of living tissues that become part of the litter ($\dot{C}_{t_k,e_1}$) and structural debris ($\dot{C}_{t_k,e_2}$). The transfer of carbon from plants to the soil carbon pools and between the soil carbon pools do not directly impact the carbon dioxide budget, but contribute to the long-term ecosystem carbon stock distribution and carbon balance. These components have been discussed in previous ED and ED2 publications (Moorcroft et al., 2001; Albani et al., 2006; Medvigy, 2006; Medvigy et al., 2009) and are summarized in Supplement S3.

## 4 Sub-models and parameterizations of terms of the general equations

### 4.1 Hydrology sub-model and ground energy exchange

The ground model encompasses heat, enthalpy, and water fluxes between adjacent layers of soil and temporary surface water, as well as losses of water and enthalpy due to surface runoff and drainage. Fluxes between adjacent layers are positive when they are upwards, and runoff and drainage fluxes are positive or zero.

Sensible heat flux between two adjacent soil or temporary surface water layers $j-1$ and $j$ are determined based on thermal conductivity $\Upsilon_Q$ and temperature gradient (Bonan, 2008), with an additional term for temporary surface water to scale the flux when the temporary surface water covers only a fraction $f_{\mathrm{TSW}}$ of the ground:

$$\dot{Q}_{g_{j-1},g_j} = -\langle \Upsilon_Q \rangle_{g_{j-1},g_j} \left( \frac{\partial T_g}{\partial z} \right)_{g_{j-1},g_j} , \qquad (26)$$

$$\dot{Q}_{s_{j-1},s_j} = -f_{\mathrm{TSW}} \langle \Upsilon_Q \rangle_{s_{j-1},s_j} \left( \frac{\partial T_s}{\partial z} \right)_{s_{j-1},s_j} , \qquad (27)$$

where the operator $\langle \ \rangle$ is the log-linear interpolation from the mid-point height of layers $j-1$ and $j$ to the height at the interface. The bottom boundary condition of Eq. (26) is $\left( \frac{\partial T}{\partial z} \right)_{g_0,g_1} \equiv 0$. The interface between the top soil layer and the first temporary surface water ($\dot{Q}_{g_{N_G},s_1}$) is found by applying Eq. (27) with $\left( T_{s_0}; \Upsilon_{Q_{s_0}}; \Delta z_{s_0} \right) = \left( T_{g_{N_G}}; \Upsilon_{Q_{g_{N_G}}}; \Delta z_{g_{N_G}} \right)$. Soil thermal conductivity depends on soil moisture and texture properties, and the parameterization is described in Supplement S7. Both the fraction of ground covered by the temporary surface water and the thermal conductivity of the temporary surface

water are described in Supplement S8.





Ground water exchange between layers occurs only if water is in liquid phase. The water flux between soil layers $g_{j-1}$ and $g_j, j \in \{2, 3, \ldots, N_G\}$ is determined from Darcy's law (Bonan, 2008):

$$\dot{W}_{g_{j-1},g_j} = -\rho_\ell \left\langle \Upsilon_\Psi \right\rangle_{g_{j-1},g_j} \left[ \frac{\partial \Psi}{\partial z} + \frac{\mathrm{d}z_g}{\mathrm{d}z} \right]_{g_{j-1},g_j}, \tag{28}$$

where $\Psi$ is the soil matric potential and $\Upsilon_\Psi$ is the hydraulic conductivity, both defined after Brooks and Corey (1964), with an additional correction term applied to hydraulic conductivity to reduce conductivity in case the soil is partially or completely frozen (Supplement S7). The bottom boundary condition for soil matric potential gradient is $\left( \frac{\partial \Psi}{\partial z} \right)_{g_0,g_1} \equiv 0$.

The term $\frac{\mathrm{d}z_g}{\mathrm{d}z}$ in Eq. (28) is the flux due to gravity, and it is 1 for all layers except the bottom boundary condition, which
depends on the sub-surface drainage. Sub-surface drainage at the bottom boundary depends on the type of drainage, and is determined using a slight modification of Eq. (28). Let $\eth$ be an angle-like parameter that controls the drainage beneath the lowest level. Because we assume zero gradient in soil matric potential between the lowest layer and the boundary condition, the sub-surface drainage flux ($\dot{W}_{g_1,o}$) becomes:

$$\dot{W}_{g_1,o} = -\dot{W}_{g_0,g_1} = \rho_\ell \, \Upsilon_{\Psi_{g_1}} \, \sin \eth. \tag{29}$$

Special cases of Eq. (29) are the zero-flow conditions ($\eth = 0$) and free drainage ($\eth = \frac{\pi}{2}$).

For the temporary surface water, water flux between layers through percolation is calculated similarly to LEAF-2 (Walko et al., 2000). Liquid water in excess of $10\,\%$ is in principle free to percolate to the layer below, although the maximum percolation of the first surface water layer is limited by the amount of pore space available at the top ground layer:

$$\dot{W}_{g_{N_G},s_1} = -\frac{1}{\Delta t_{\text{Thermo}}} \max \left[ 0, W_{s_1} \left( \frac{\ell_{s_1} - 0.1}{0.9} \right), \rho_\ell \left( \vartheta_{\text{Po}} - \vartheta_{g_{N_G}} \right) \Delta z_{g_{N_G}} \right], \tag{30}$$

$$\dot{W}_{s_{j-1},s_j} = -\frac{1}{\Delta t_{\text{Thermo}}} \max \left( 0, W_{s_j} \frac{\ell_{s_j} - 0.1}{0.9} \right) \quad , \text{ for } j > 1. \tag{31}$$

Surface runoff of liquid water is simulated using a simple extinction function, applied only at the top most temporary surface water layer:

$$\dot{W}_{s_{N_S},o} = \ell_{s_{N_S}} W_{s_{N_S}} \exp \left( -\frac{\Delta t_{\text{Thermo}}}{t_{\text{Runoff}}} \right), \tag{32}$$

where $t_{\text{Runoff}}$ is a user-defined e-folding decay time, usually on the order of a few minutes to a few hours (Table S4).

In addition to the water fluxes due to sub-surface drainage, surface runoff and the transport of water between layers, we must account for the associated enthalpy fluxes. Enthalpy fluxes due to sub-surface drainage and surface runoff are defined based on the water flux and the temperature of the layers where water is lost, by applying the definition of enthalpy (Supplement S4):

$$\dot{H}_{g_1,o} = \dot{W}_{g_1,o} \, q_\ell \left( T_{g_1} - T_{\ell 0} \right), \tag{33}$$

$$\dot{H}_{s_{N_S},o} = \dot{W}_{s_{N_S},o} \, q_\ell \left( T_{s_{N_S}} - T_{\ell 0} \right), \tag{34}$$

where $q_\ell$ is the specific heat of liquid water (Table S3), and $T_{\ell 0}$ is defined in Eq. (S47). The enthalpy flux between two adjacent layers is solved similarly, but it must account for the sign of the flux in order to determine the water temperature of the donor

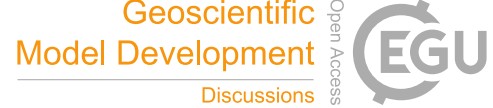



layer:

$$
\dot{H}_{x_{j-1},x_j} =
\begin{cases}
\dot{W}_{x_{j-1},x_j}\, q_\ell \left(T_{x_j} - T_{\ell 0}\right) & , \text{if } \dot{W}_{x_{j-1},x_j} < 0 \\
\dot{W}_{x_{j-1},x_j}\, q_\ell \left(T_{x_{j-1}} - T_{\ell 0}\right) & , \text{if } \dot{W}_{x_{j-1},x_j} \geq 0
\end{cases},
\tag{35}
$$

where the subscript $x_j$ represents either soil ($g_j$) or temporary surface water ($s_j$).

### 4.2 Precipitation and vegetation dripping

In ED-2.2, precipitating water from rain and snow increases the water storage of the thermodynamic systems, as rainfall can be intercept by the canopy, or reach the ground. This influx of water also affects the enthalpy storage because the enthalpy associated with precipitation, although no heat exchange is directly associated with precipitation.

To determine the partitioning of total incoming precipitation ($\dot{W}_{\infty,a}$) into interception by each cohort ($\dot{W}_{a,t_k}$) and direct interception by the ground (throughfall, $\dot{W}_{a,s_{N_S}}$), we use the fraction of open canopy ($\mathcal{O}$) and the total plant area index of each cohort ($\Pi_{t_k}$):

$$
\dot{W}_{a,t_k} = (1 - \mathcal{O})\, \dot{W}_{\infty,a}\, \frac{\Pi_{t_k}}{\sum_{k'=1}^{N_T} \Pi_{t_{k'}}},
\tag{36}
$$

$$
\dot{W}_{a,s_{N_S}} = \mathcal{O}\, \dot{W}_{\infty,a},
\tag{37}
$$

$$
\mathcal{O} = \prod_{k=1}^{N_T} (1 - X_{t_k}),
\tag{38}
$$

where $\Pi_{t_k} = \Lambda_{t_k} + \Omega_{t_k}$ is the total plant area index, $\Lambda_{t_k}$ and $\Omega_{t_k}$ being the leaf and wood area indices, both defined from PFT-dependent allometric relations (Supplement S16); $X_{t_k}$ is the crown area index of each cohort, also defined in Supplement S16. Throughfall precipitation is always placed on the topmost temporary surface water layer. In case no temporary surface water layer exists, a new layer is created, although it may be extinct in case all water is able to percolate down to the top soil layer.

Precipitation is a mass flux, but it also has an associated enthalpy flux ($\dot{H}_{\infty,a}$) that must be partitioned and incorporated to the cohorts and temporary surface water. Similarly to the water exchange between soil layers, the enthalpy flux associated with rainfall uses the definition of enthalpy (Supplement S4). Because precipitation can fall as rain, snow, or a mix of both, we parameterize the precipitation phase according to the air temperature. Rain is only allowed when the free-air temperature $T_a$ is above the water triple point ($T_3 = 273.16\,\mathrm{K}$); in this case, the rain temperature is always assumed to be at $T_a$. Pure snow occurs when the free-air temperature is below $T_3$, and likewise snow temperature is assumed to be $T_a$. When free air temperature is only slightly above $T_3$, a mix of rain and snow occurs, with the rain temperature assumed to be $T_a$ and snow temperature assumed to be $T_3$:

$$
\dot{H}_{\infty,a} = \dot{W}_{\infty,a} \left[ (1 - \ell_a)\, q_i \min(T_3, T_a) + \ell_a\, q_\ell \left(T_a - T_{\ell 0}\right) \right],
\tag{39}
$$

where ($q_i; q_\ell$) are the specific heats of ice and liquid, respectively, and $T_{\ell 0}$ is temperature at which supercooled water would have enthalpy equal to zero (Eq. S47). The fraction of precipitation that falls as rain $\ell_a$ is based on the Jin et al. (1999)



parameterization, slightly modified to make the function continuous:

$$
\ell_a = \begin{cases}
1.0 & \text{, if } T_a > 275.66\text{K} \\
0.4 + 1.2\left(T_a - T_3 - 2.0\right) & \text{, if } 275.16\,K < T_a \leq 275.66\text{K} \\
0.2\left(T_a - T_3\right) & \text{, if } T_3 < T_a \leq 275.16\text{K} \\
0.0 & \text{, if } T_a \leq T_3
\end{cases}
. \tag{40}
$$

The enthalpy flux associated with precipitation is then partitioned into canopy interception ($\dot{H}_{a,t_k}$) and throughfall ($\dot{H}_{a,s_{N_S}}$) using the same scaling factor as in Eq. (37) and Eq. (36):

$$
\dot{H}_{a,t_k} = (1-\mathcal{O})\,\dot{H}_{\infty,a}\,\frac{\Pi_{t_k}}{\sum_{k'=1}^{N_T}\Pi_{t_{k'}}}, \tag{41}
$$

$$
\dot{H}_{a,s_{N_S}} = \mathcal{O}\,\dot{H}_{\infty,a}. \tag{42}
$$

Leaves and branches can accumulate only a finite amount of water on their surfaces, proportional to their total area. When incoming precipitation rates are too high (or more rarely when dew or frost formation is excessive), any water amount that exceeds the holding capacity is lost to the ground as canopy dripping. Similarly to incoming precipitation, the excess water lost through dripping also has an associated enthalpy that must be taken into account, although dripping has no associated heat flux. The canopy dripping fluxes of water ($\dot{W}_{t_k,s_{N_S}}$) and the associated enthalpy ($\dot{H}_{t_k,s_{N_S}}$) are defined such that the leaves and
branches lose the excess water within one time step:

$$
\dot{W}_{t_k,s_{N_S}} = -\frac{1}{\Delta t_{\text{Thermo}}}\max\left(0, W_{t_k} - \hat{w}_{\max}\,\Pi_{t_k}\right), \tag{43}
$$

$$
\dot{H}_{t_k,s_{N_S}} = \dot{W}_{t_k,s_{N_S}}\left[(1-\ell_{t_k})\,q_i T_{t_k} + \ell_{t_k}\left(T_{t_k} - T_{\ell 0}\right)\right], \tag{44}
$$

where $\ell_{t_k}$ is the liquid fraction of surface water on top of cohort $k$ and $\hat{w}_{\max}$ is the cohort holding capacity, which is an adjustable parameter (Table S4) but typically is of the order of $0.05 - 0.40\,\text{kg}_{\text{W}}\,\text{m}_{\text{Leaf+Wood}}^{-2}$ (Wohlfahrt et al., 2006).

### 4.3  Radiation model

The radiation budget is solved using a multi-layer version of the two-stream model (Sellers, 1985; Liou, 2002; Medvigy, 2006) applied to three broad spectral bands: photosynthetically active radiation (PAR, wave lengths between 0.4 and $0.7\,\mu\text{m}$), near infrared radiation (NIR, wave lengths between 0.7 and $5.0\,\mu\text{m}$) and thermal infrared radiation (TIR, wave lengths between 5.0 and $40\,\mu\text{m}$).

#### 4.3.1  Canopy radiation profile

For each spectral band $m$, the canopy radiation scheme assumes that each cohort corresponds to one layer of vegetation within the canopy, and within each layer the optical and thermal properties are assumed constant. For all bands, the top boundary condition for each band is provided by the meteorological forcing (Table 2). In the cases of PAR ($m = 1$) and NIR ($m = 2$),

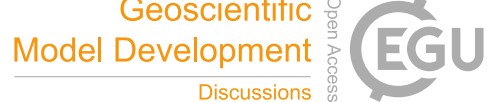

the downward irradiance is comprised of a beam (direct) and isotropic (diffuse) components, whereas TIR ($m = 3$) is assumed
to be all diffuse. Direct irradiance that is intercepted by the cohorts can be either back-scattered or forward-scattered as diffuse
radiation, and direct radiation reflected by the ground is assumed to be entirely diffuse.

Following Sellers (1985), the extinction of downward direct irradiance and the two-stream model for hemispheric diffuse
irradiance for each of the spectral bands ($m = 1, 2, 3$) is given by:

$$\underbrace{\mu_k^{\odot} \frac{\mathrm{d}\dot{Q}_{mk}^{\odot}}{\mathrm{d}\tilde{\Pi}}}_{\substack{\text{Downward direct} \\ \text{profile}}} = \underbrace{-\dot{Q}_{mk}^{\odot}}_{\text{Interception}}, \tag{45}$$

$$\underbrace{\overline{\mu}_k \frac{\mathrm{d}\dot{Q}_{mk}^{\Downarrow}}{\mathrm{d}\tilde{\Pi}}}_{\substack{\text{Downward diffuse} \\ \text{profile}}} = \underbrace{-\dot{Q}_{mk}^{\Downarrow}}_{\text{Interception}} + \underbrace{(1-\beta_{mk})\,\varsigma_{mk}\,\dot{Q}_{mk}^{\Downarrow}}_{\substack{\text{Forward scattering} \\ \text{(downward diffuse)}}} + \underbrace{\beta_{ik}\,\varsigma_{mk}\,\dot{Q}_{mk}^{\Uparrow}}_{\substack{\text{Backscattering} \\ \text{(upward diffuse)}}} + \underbrace{\frac{\overline{\mu}_k}{\mu_k^{\odot}}\varsigma_{mk}\left(1-\beta_{mk}^{\odot}\right)\dot{Q}_{mk}^{\odot}}_{\substack{\text{Forward scattering} \\ \text{(downward direct)}}} + \underbrace{(1-\varsigma_{mk})\,\dot{Q}_{mk}^{\blacklozenge}}_{\text{Emission}}, \tag{46}$$

$$\underbrace{-\overline{\mu}_k \frac{\mathrm{d}\dot{Q}_{mk}^{\Uparrow}}{\mathrm{d}\tilde{\Pi}}}_{\substack{\text{Upward diffuse} \\ \text{profile}}} = -\underbrace{\dot{Q}_{mk}^{\Uparrow}}_{\text{Interception}} + \underbrace{(1-\beta_{mk})\,\varsigma_{mk}\,\dot{Q}_{mk}^{\Uparrow}}_{\substack{\text{Forward scattering} \\ \text{(upward diffuse)}}} + \underbrace{\beta_{mk}\,\varsigma_{mk}\,\dot{Q}_{mk}^{\Downarrow}}_{\substack{\text{Backscattering} \\ \text{(downward diffuse)}}} + \underbrace{\frac{\overline{\mu}_k}{\mu_k^{\odot}}\varsigma_{mk}\,\beta_{mk}^{\odot}\,\dot{Q}_{mk}^{\odot}}_{\substack{\text{Backscattering} \\ \text{(downward direct)}}} + \underbrace{(1-\varsigma_{mk})\,\dot{Q}_{mk}^{\blacklozenge}}_{\text{Emission}}, \tag{47}$$

where index $k \in \{1, 2, \ldots, N_T\}$ corresponds to each cohort $k$ or its lower interface (Fig. 4); interface $N_T + 1$ is immediately
above the tallest cohort; $\dot{Q}_{mk}^{\odot}$ is the downward direct irradiance incident at interface $k$; ($\dot{Q}_{mk}^{\Downarrow}$ and $\dot{Q}_{mk}^{\Uparrow}$) are the downward
and upward (hemispheric) diffuse irradiances incident at interface $k$; $\varsigma_{mk}$ is the scattering coefficient, and thus $(1-\varsigma_{mk})$ is
the absorptivity; $\beta_{mk}^{\odot}$ and $\beta_{mk}$ are the backscattered fraction of scattered direct and diffuse irradiances, respectively; $\tilde{\Pi}$ is the
effective cumulative plant area index, assumed zero at the top of each layer, and increasing downwards ($\tilde{\Pi}_k$ is the total for
layer $k$); $\mu_k^{\odot}$ and $\overline{\mu}_k$ are the inverse of the optical depth per unit of effective plant area index for direct and diffuse radiation,
respectively; and $\dot{Q}_{mk}^{\blacklozenge}$ is the irradiance emitted by a black body at the same temperature as the cohort ($T_{t_k}$).

Equations (45)-(47) simplify for each spectral band. First, $\dot{Q}_{m=3\,k}^{\odot} \equiv 0$, because we assume that all incoming TIR irradiance
is diffuse. Likewise, the black-body emission $\dot{Q}_{mk}^{\blacklozenge} = \dot{Q}_{1k}^{\blacklozenge} = 0$ for the PAR ($m = 1$) and NIR ($m = 2$) bands, because thermal
emission is negligible at these wave lengths. The black-body emission for the TIR band is defined as

$$\dot{Q}_{m=3\,k}^{\blacklozenge} = \sigma_{\mathrm{SB}}\,T_{t_k}^4, \tag{48}$$

where $\sigma_{\mathrm{SB}}$ is the Stefan-Boltzmann constant (Table S3). Note that for emission of TIR radiation (45-47), we assume that
emissivity is the same as absorptivity (Kirchhoff's law; Liou, 2002), hence the $1-\varsigma$ term.

The effective plant area index $\tilde{\Pi}_k$ is the total area (leaves and branches) that is corrected to account for that leaves are not
uniformly distributed in the layer. It is defined as $\tilde{\Pi}_k = \Omega_k + f_{\mathrm{Clump}_k}\Lambda_k$, where $f_{\mathrm{Clump}_k}$ is the PFT-dependent clumping index
(Chen and Black, 1992, default values in Table S5), $\Lambda_{t_k}$ is the leaf area index and $\Omega_{t_k}$ is the wood area index. $\tilde{\Pi}$ is assumed
zero at the top of each layer, increasing downwards.

The optical properties of the leaf layers — optical depth and scattering parameters for direct and diffuse radiation for each
of the three spectral bands — are assumed constant within each layer. These properties are determined from PFT-dependent



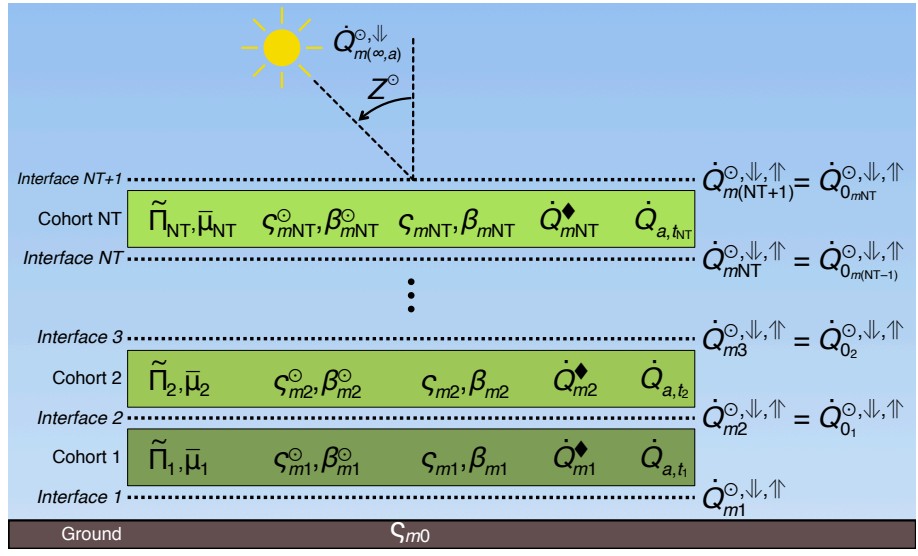

**Figure 4.** Schematic of the radiation module for a patch with $N_T$ cohorts, showing the grid arrangement of the irradiance profiles relative to the cohort positions. Index $k$ corresponds to each cohort of the interface beneath each cohort, $m$ corresponds to each spectral band; $\left(\dot{Q}^{\odot}_{m(\infty,a)}, \dot{Q}^{\odot}_{m(\infty,a)}\right)$ are the incoming direct and diffuse irradiance (Table 2); $Z^{\odot}$ is the Sun's zenith angle; $\left(\dot{Q}^{\odot}_{mk}, \dot{Q}^{\Downarrow}_{mk}, \dot{Q}^{\Uparrow}_{mk}\right)$ are the downward direct, downward diffuse, and upward diffuse irradiances (Eq. 45-47); $\tilde{\Pi}_k$ is the effective plant area index (Eq. S85); $(\varsigma_{mk}; \beta_{mk})$ are the scattering coefficients and the backscattering fraction for diffuse irradiance (Eq. S86,S87); $\left(\varsigma^{\odot}_{mk}; \beta^{\odot}_{mk}\right)$ are the scattering coefficients and the backscattering fraction for direct irradiance (Eq. S89,S90); $\dot{Q}^{\blacklozenge}_{mk}$ is the black-body irradiance (Eq. 48); and $\dot{Q}^{\odot}_{a,t_k}$ is the net absorbed irradiance (Eq. 49).

characteristics such as mean orientation factor, spectral band-dependent reflectivity, transmissivity, and emissivity (Supplement S9). Because the properties are constant within each layer, it is possible to analytically solve the full profile of both direct and diffuse radiation, using the solver described in Supplement S10.

Once the profiles of $\dot{Q}^{\odot}_{mk}$, $\dot{Q}^{\Downarrow}_{mk}$ and $\dot{Q}^{\Uparrow}_{mk}$ are determined, we obtain the irradiance that is absorbed by each cohort $\dot{Q}_{a,t_k}$:

$$\dot{Q}_{a,t_k} = \sum_{m=1}^{3} \left[ \left(\dot{Q}^{\odot}_{m(k+1)} - \dot{Q}^{\odot}_{mk}\right) + \left(\dot{Q}^{\Downarrow}_{m(k+1)} - \dot{Q}^{\Downarrow}_{mk}\right) + \left(\dot{Q}^{\Uparrow}_{mk} - \dot{Q}^{\Uparrow}_{m(k+1)}\right) \right]. \tag{49}$$

This term is then used in the enthalpy budget of each cohort (Eq. 4 and Eq. 12).

### 4.3.2 Ground radiation

The ground radiation sub-model determines the irradiance emitted by the ground surface, and the profile of irradiance through the temporary surface water layers and top soil layer. Note that the ground radiation and the canopy radiation model are interdependent: the incoming radiation at the top ground layer is determined from the canopy radiation model, and the ground scattering coefficient ($\varsigma_{m0}$, see Supplement S9) is needed for the canopy-radiation bottom boundary condition (Supplement S10). However, since the scattering coefficient does not depend on the total incoming radiation, the irradiance profile can be solved



10    for a standardized amount of incoming radiation, and once the downward radiation at the bottom of the canopy has been
calculated, the absorbed irradiance for each layer can be scaled appropriately.

Black-body emission from the ground ($\dot{Q}_{m0}^{\blacklozenge}$) is calculated as an area-weighted average of the emissivities of exposed soil
and temporary surface water:

$$\dot{Q}_{m0}^{\blacklozenge} = \begin{cases} 0 & \text{, if } m \in (1,2) \\ \frac{(1-f_{\text{TSW}})(1-\varsigma_{3g})\left(\sigma_{\text{SB}}T_{g_{N_G}}^4\right)+f_{\text{TSW}}(1-\varsigma_{3s})\left(\sigma_{\text{SB}}T_{s_{N_S}}^4\right)}{(1-f_{\text{TSW}})(1-\varsigma_{3g})+f_{\text{TSW}}(1-\varsigma_{3s})} & \text{, if } m = 3 \end{cases}. \tag{50}$$

where $(1-\varsigma_{3g})$ and $(1-\varsigma_{3s})$, are, respectively, the thermal-infrared emissivities of the top soil layer and the temporary surface

5    water (Table S4), and $f_{\text{TSW}}$ is the fraction of ground covered by temporary surface water. In ED-2.2, the soil and snow scattering
coefficients for the TIR band are assumed constant, following Walko et al. (2000).

Once the irradiance profile for the canopy is determined from Eq. (45)-(47), the irradiance absorbed by each temporary
surface water layer ($j \in \{1, 2, \ldots, N_S\}$) is calculated by integrating the transmissivity profile for each layer, starting from the
top layer:

$$\dot{Q}_{a,s_j} = \begin{cases} \sum_{m=1}^2\left\{f_{\text{TSW}}\left(\dot{Q}_{m1}^{\Downarrow}+\dot{Q}_{m1}^{\odot}\right)\left[1-\exp\left(-\frac{\Delta\overline{z}_{s_{N_S}}}{\overline{\mu}_s}\right)\right]\right\}+f_{\text{TSW}}(1-\varsigma_{3s})\left(\dot{Q}_{m=3\,k=1}^{\Downarrow}-\sigma_{\text{SB}}T_{s_{N_S}}^4\right) & \text{, if } j = N_S \\ \sum_{m=1}^2\left\{f_{\text{TSW}}\left(\dot{Q}_{m1}^{\Downarrow}+\dot{Q}_{m1}^{\odot}\right)\left[\exp\left(-\frac{\sum_{j'=j+1}^{N_S}\Delta\overline{z}_{s_j}}{\overline{\mu}_s}\right)-\exp\left(-\frac{\sum_{j'=j}^{N_S}\Delta\overline{z}_{s_{j'}}}{\overline{\mu}_s}\right)\right]\right\} & \text{, otherwise} \end{cases}, \tag{51}$$

where $\overline{\mu}_s$ is the inverse of the optical depth of temporary surface water.

The irradiance absorbed by the ground is a combination of irradiance of exposed soil and irradiance that is transmitted
through all temporary surface water layers, and the net absorption of longwave radiation:

$$\dot{Q}_{a,g_{N_G}} = \sum_{m=1}^2\left\{\left[1-f_{\text{TSW}}+f_{\text{TSW}}\exp\left(-\frac{\sum_{j'=1}^{N_S}\Delta\overline{z}_{s_{j'}}}{\overline{\mu}_s}\right)\right]\left(\dot{Q}_{m1}^{\Downarrow}+\dot{Q}_{m1}^{\odot}\right)\right\}$$
$$+(1-f_{\text{TSW}})(1-\varsigma_{3g})\left(\dot{Q}_{m=3\,k=1}^{\Downarrow}-\sigma_{\text{SB}}T_{g_{N_G}}^4\right). \tag{52}$$

15    ## 4.4    Surface Layer Model

The surface layer model determines the fluxes of enthalpy, water, and carbon dioxide between the canopy air space and the
free air above. It is based on the Monin-Obukhov similarity theory (Monin and Obukhov, 1954; Foken, 2006), which has
been widely used by biosphere-atmosphere models representing a variety of biomes (e.g. Walko et al., 2000; Best et al., 2011;
Oleson et al., 2013), although this is often an extrapolation of the theory that was not originally developed for heterogeneous

20    vegetation, or tall vegetation (Foken, 2006).

In order to obtain the fluxes, we assume that the eddy diffusivity of buoyancy is the same as the diffusivity of enthalpy, water
vapor, and $CO_2$. This assumption allows us to define a single canopy conductance $G_c$ for the three variables, following the
algorithm described in Supplement S12.1. We then obtain the following equations for fluxes between canopy air space and the





free atmosphere:

$$\dot{W}_{a,c} = \rho_c\, G_c\, (w_a - w_c), \tag{53}$$

$$\dot{H}_{a,c} = \rho_c\, G_c\, \left(\tilde{h}_a - h_c\right), \tag{54}$$

$$\dot{C}_{a,c} = \frac{\mathcal{M}_C}{\mathcal{M}_d}\, \rho_c\, G_c\, (c_a - c_c), \tag{55}$$

where $\tilde{h}_a$ is the equivalent enthalpy of air at reference height $z_a$ when the air is adiabatically moved to the top of the canopy air space, using the definition of potential temperature:

$$\tilde{h}_a = h\left(\tilde{T}_a = \theta_a \left(\frac{p_c}{p_0}\right)^{\frac{\mathcal{R}}{\mathcal{M}_d\, q_{pd}}}, w_a\right) \qquad \text{, from Eq.(S44),} \tag{56}$$

where $p_0$ is the reference pressure, $\mathcal{R}$ is the universal gas constant, $q_{pd}$ is the specific heat of dry air at constant pressure, and $\mathcal{M}_d$ is the molar mass of dry air (Table S3).

Sensible heat flux between the free atmosphere and canopy air space ($\dot{Q}_{a,c}$) can be derived from the definition of enthalpy and enthalpy flux (Eq. S44 and Eq. 54), although it is not directly applied to the energy balance in the canopy air space ($\dot{H}_{a,c}$ is used instead).

$$\dot{H}_{a,c} = \rho_c\, G_c\, \left[(1-w_a)\, q_{pd}\, \tilde{T}_a + w_a\, q_{pv}\left(\tilde{T}_a - T_{v0}\right) - (1-w_c)\, q_{pd}\, T_c + w_c\, q_{pv}\,(T_c - T_{v0})\right]$$

$$= \underbrace{\rho_c\, G_c\, \left(q_{p_a}\tilde{T}_a - q_{p_c}T_c\right)}_{\dot{Q}_{a,c}} - \rho_c\, G_c\, (w_a - w_c)\, q_{pv}\, T_{v0},$$

$\dot{Q}_{a,c} = \dot{H}_{a,c} + \dot{W}_{a,c}\, q_{pv}\, T_{v0}.$ \hfill (57)

### 4.5 Heat and water exchange between surfaces and canopy air space

#### 4.5.1 Leaves and branches

Fluxes of sensible heat ($\dot{Q}_{t_k,c}$) and water vapor ($\dot{W}_{t_k,c}$) between the leaf surface and wood surface and the canopy air space follow the same principle of conductance and gradient that define the eddy fluxes between the free atmosphere and canopy air

space (Eq. 53;54). Throughout this section, we use subscripts $\lambda_k$ and $\beta_k$ to denote leaf and wood boundary layers of cohort $k$, respectively; the different subscripts are needed to differentiate fluxes coming from the leaves' intercellular space (e.g. transpiration, see also Section 4.6). Let $G_{Q\lambda_k}\ (\mathrm{m\,s^{-1}})$ and $G_{W\lambda_k}\ (\mathrm{m\,s^{-1}})$ be the conductances of heat and water between the leaf boundary layer of cohort $k$ and the canopy air space, and $G_{Q\beta_k}$ and $G_{W\beta_k}$ be the wood boundary layer counterparts. The surface sensible heat and surface water vapor fluxes are:

$$\dot{Q}_{t_k,c} = \dot{Q}_{\lambda_k,c} + \dot{Q}_{\beta_k,c} = 2\,\Lambda_k\,\dot{q}_{\lambda_k,c} + \pi\,\Omega_k\,\dot{q}_{\beta_k,c}, \tag{58}$$

$$\dot{W}_{t_k,c} = \dot{W}_{\lambda_k,c} + \dot{W}_{\beta_k,c} = \Lambda_k\,\dot{w}_{\lambda_k,c} + \Omega_k\,\dot{w}_{\beta_k,c}, \tag{59}$$

$$\dot{q}_{\lambda_k,c} = G_{Q\lambda_k}\,\rho_c\,q_{p_c}\left(T_{l_k}^{\mathrm{Sfc}} - T_c\right), \tag{60}$$





$$\dot{q}_{\beta_k,c} = G_{Q\beta_k}\,\rho_c\,q_{p_c}\left(T_{b_k}^{\text{Sfc}} - T_c\right), \tag{61}$$

$$\dot{w}_{\lambda_k,c} = G_{W\lambda_k}\,\rho_c\left(w_{l_k}^{\text{Sfc}} - w_c\right), \tag{62}$$

$$\dot{w}_{\beta_k,c} = G_{W\beta_k}\,\rho_c\left(w_{b_k}^{\text{Sfc}} - w_c\right), \tag{63}$$

where $(\dot{q}_{\lambda_k,c}; \dot{q}_{\beta_k,c}; \dot{w}_{\lambda_k,c}; \dot{w}_{\beta_k,c})$ are the leaf-surface and branch-surface heat and water fluxes by unit of leaf and branch area, respectively; the factors 2 and $\pi$ in Eq. (58) means that sensible heat is exchanged on both sides of the leaves, and on the longitudinal area of the branches, which are assumed cylindrical. Intercepted water and dew and frost formation is allowed only on one side of the leaves, and an area equivalent to a one-sided flat plate for branches, and therefore only the leaf and wood area indices are used in Eq. (59). Canopy air space temperature, specific humidity, density, and specific heat, leaf temperature, and wood temperature are determined diagnostically. We also assume that surface temperature of leaves and branches to be the same as their internal temperatures (i.e. $T_{l_k}^{\text{Sfc}} \equiv T_{l_k}$ and $T_{b_k}^{\text{Sfc}} \equiv T_{b_k}$). Specific humidity at the leaf surface $w_{l_k}^{\text{Sfc}} = w^{\equiv}\left(T_{l_k}^{\text{Sfc}}, p_c\right)$ and branch surface $w_{\beta_k}^{\text{Sfc}} = w^{\equiv}\left(T_{b_k}^{\text{Sfc}}, p_c\right)$ are assumed to be the saturation specific humidity $w^{\equiv}$ (Supplement S13).

Heat conductance for leaves and branches are based on the convective heat transfer, as described in Supplement S12.2. Further description of the theory can be found in Monteith and Unsworth (2008, Section 10.1).

### 4.5.2 Temporary surface water and soil

Sensible heat and water fluxes between the temporary surface water and soil and the canopy air space are calculated similarly to leaves and branches. Surface conductance $G_{\text{Sfc}}$ is assumed to be the same for both heat and water, and also the same for soil and temporary surface water:

$$\dot{Q}_{s_{N_S},c} = f_{\text{TSW}}\,G_{\text{Sfc}}\,\rho_c\,q_{p_c}\left(T_{s_{N_S}} - T_c\right), \tag{64}$$

$$\dot{W}_{s_{N_S},c} = f_{\text{TSW}}\,G_{\text{Sfc}}\,\rho_c\left(w_{s_{N_S}} - w_c\right), \tag{65}$$

$$\dot{Q}_{g_{N_G},c} = (1 - f_{\text{TSW}})\,G_{\text{Sfc}}\,\rho_c\,q_{p_c}\left(T_{g_{N_G}} - T_c\right), \tag{66}$$

$$\dot{W}_{g_{N_G},c} = (1 - f_{\text{TSW}})\,G_{\text{Sfc}}\,\rho_c\,q_{p_c}\left(w_{g_{N_G}} - w_c\right), \tag{67}$$

Specific humidity for temporary surface water is computed exactly as leaves and branches, $w_{s_{N_S}} = w^{\equiv}\left(T_{s_{N_S}}, p_c\right)$ (Supplement S13). For soils the specific humidity also accounts for the soil moisture and the sign of the flux, using a method similar to Avissar and Mahrer (1988):

$$w_{g_{N_G}} = \begin{cases} s_g \exp\left(\dfrac{\mathcal{M}_w\,g\,\Psi_{g_{N_G}}}{\mathcal{R}\,T_{g_{N_G}}}\right) w^{\equiv}\left(T_{g_{N_G}}, p_c\right) + (1 - s_g)\,w_c & \text{, if } w^{\equiv}\left(T_{g_{N_G}}, p_c\right) > w_c \\ w^{\equiv}\left(T_{g_{N_G}}, p_c\right) & \text{, if } w^{\equiv}\left(T_{g_{N_G}}, p_c\right) \le w_c \end{cases}, \tag{68}$$

$$s_g = \frac{1}{2}\left\{1.0 - \cos\left[\pi\,\frac{\min\left(\vartheta_{g_{N_G}}, \vartheta_{\text{Fc}}\right) - \vartheta_{\text{Re}}}{\vartheta_{\text{Fc}} - \vartheta_{\text{Re}}}\right]\right\}, \tag{69}$$

where $g$ is the gravity acceleration, $\mathcal{M}_w$ is the water molar mass, and $\mathcal{R}$ is the universal gas constant (Table S3); $T_{g_{N_G}}$, $\vartheta_{g_{N_G}}$ and $\Psi_{g_{N_G}}$ are the temperature, soil moisture and soil matric potential of the topmost soil layer, respectively; and $\vartheta_{\text{Fc}}$ and $\vartheta_{\text{Re}}$





are the soil moisture at field capacity and the residual soil moisture, respectively. The exponential term in Eq. 68 corresponds to the soil pore relative humidity derived from the Kelvin equation (Philip, 1957), and $s_g$ is the soil wetness function, which takes a similar functional form as the relative humidity term from Noilhan and Planton (1989) and the $\beta$ term from Lee and Pielke (1992). The total resistance between the surface and the canopy air space is a combination of the resistance if the surface

was bare, plus the resistance due to the vegetation, as described in Supplement S12.3.

### 4.5.3 Enthalpy flux due to evaporation and condensation

Dew and frost are formed when water in the canopy air space condenses or freezes on any surface (leaves, branches, or ground); likewise, water that evaporates and ice that sublimates from these surfaces immediately become part of the canopy air space. In terms of energy transfer, two processes occur, the phase change and the mass exchange, and both must be accounted for

the enthalpy flux. Phase change depends on the specific latent heat of vaporization ($l_{\ell v}$) and sublimation ($l_{iv}$), which are linear functions of temperature, based on Eq. (S42) and Eq. (S43):

$$l_{\ell v}\left(T\right) = l_{\ell v3} + \left(q_{pv} - q_{\ell}\right)\left(T - T_3\right), \tag{70}$$

$$l_{iv}\left(T\right) = l_{iv3} + \left(q_{pv} - q_i\right)\left(T - T_3\right), \tag{71}$$

where $l_{\ell v3}$ and $l_{iv3}$ are the specific latent heats of vaporization and sublimation at the water triple point ($T_3$), $q_{pv}$ is the specific

heat of water vapor at constant pressure, and $q_i$ and $q_{\ell}$ are the specific heats of ice and liquid water, respectively (Table S3). The temperature for phase change must be the surface temperature because this is where the phase change occurs. In the most generic case, if a surface $x$ at temperature $T_x$ and a liquid water fraction $\ell_x$, the total enthalpy flux between the surface and canopy air space $\dot{H}_{x,c}$ associated with the water flux $W_{x,c}$ is:

$$\dot{H}_{x,c} = \dot{W}_{x,c} \left\{ \underbrace{\left[\left(1 - \ell_x\right)q_i T_x + \ell_x q_{\ell}\left(T_x - T_{\ell 0}\right)\right]}_{\text{Enthalpy flux due to mass exchange}} + \underbrace{\left[\left(1 - \ell_x\right)l_{iv}\left(T_x\right) + \ell_x l_{\ell V}\left(T_x\right)\right]}_{\text{Enthalpy flux due to phase change}} \right\}. \tag{72}$$

By using the definitions from Eq. (S48), Eq. (72) can be further simplified to:

$$\dot{H}_{x,c} = \dot{W}_{x,c}\left[q_{pv}\left(T_x - T_{v0}\right)\right] = \dot{W}_{x,c}\underbrace{h\left(T_x, w_x = 1\right)}_{\text{Eq. (S44)}}, \tag{73}$$

which is consistent with the exchange of pure water vapor and enthalpy between the thermodynamic systems. Eq. (73) is used to determine $\dot{H}_{g_{N_G},c}$, $\dot{H}_{s_{N_S},c}$, and $\dot{H}_{t_k,c}, k \in \{1, 2, \ldots, N_T\}$.

### 4.6 Leaf physiology

In ED-2.2, leaf physiology is modeled following Farquhar et al. (1980) and Collatz et al. (1991) for $C_3$ plants; and Collatz

et al. (1992) for $C_4$ plants, and Leuning (1995) model for stomatal conductance. This sub-model ultimately determines the net leaf-level $CO_2$ uptake rate of each cohort $k$ ($\dot{A}_k$, $\mathrm{mol_C\,m_{Leaf}^{-2}\,s^{-1}}$), controlled exclusively by the leaf environment, and the corresponding water loss through transpiration ($\dot{E}_k$, $\mathrm{mol_W\,m_{Leaf}^{-2}\,s^{-1}}$).





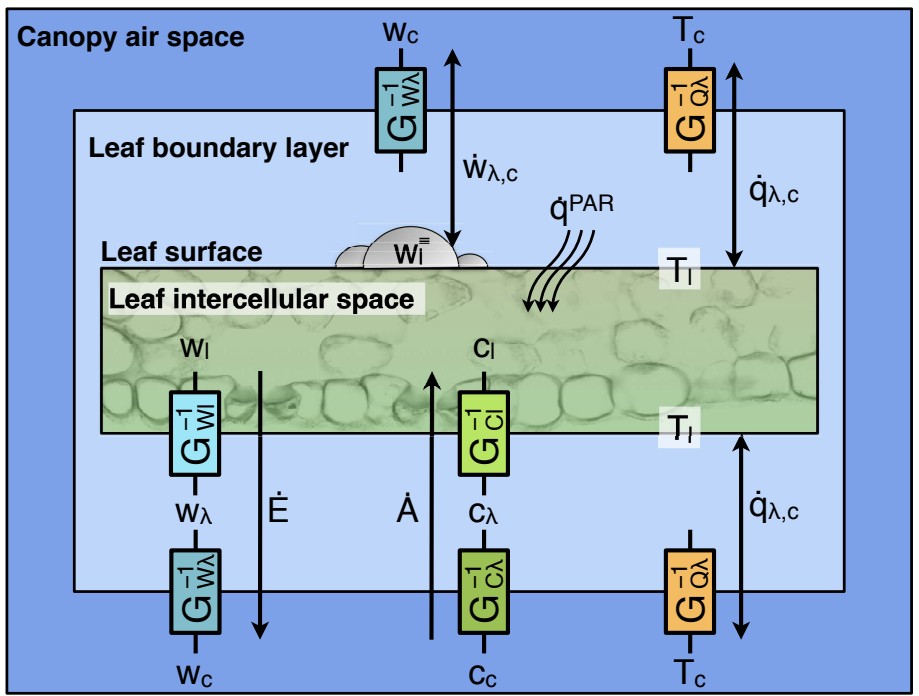

**Figure 5.** Schematic of fluxes between a leaf and the surrounding canopy air space for a hypostomatous plant during the photo period, as represented in ED-2.2. Conductances are represented by the resistances between the different environments ($G^{-1}$). Leaf-level sensible heat flux ($\dot{q}_{\lambda_k,c}$; Eq. 58) and leaf-level vapor flux between intercepted water and canopy air space ($\dot{w}_{\lambda_k,c}$; Eq. 59) are also shown for comparison. Cohort index $k$ is omitted from the figure for clarity.

The exchange of water and $CO_2$ between the leaf intercellular space and the canopy air space is mediated by the stomata and the leaf boundary layer, which imposes an additional resistance to fluxes of these substances. For simplicity, we assume that the leaf boundary layer air has low storage capacity, and thus the fluxes of any substance (water or $CO_2$) entering and exiting the boundary layer must be the same. Fluxes of water and carbon between the leaf intercellular space and the canopy air space must overcome both the stomatal resistance and the boundary layer resistance, whereas sensible heat flux and water flux from leaf surface water must overcome the boundary layer resistance only (Fig. 5). The potential fluxes of $CO_2$ and water can be written as:

$$\dot{A}_k = \hat{G}_{C\lambda_k} \left( c_c - c_{\lambda_k} \right) = \hat{G}_{Cl_k} \left( c_{\lambda_k} - c_{l_k} \right) = \frac{\hat{G}_{C\lambda_k} \hat{G}_{Cl_k}}{\hat{G}_{C\lambda_k} + \hat{G}_{Cl_k}} \left( c_c - c_{l_k} \right), \tag{74}$$

$$\dot{E}_k = \hat{G}_{W\lambda_k} \left( w_c - w_{\lambda_k} \right) = \hat{G}_{Wl_k} \left( w_{\lambda_k} - w_{l_k} \right) = \frac{\hat{G}_{W\lambda_k} \hat{G}_{Wl_k}}{\hat{G}_{W\lambda_k} + \hat{G}_{Wl_k}} \left( w_c - w_{l_k} \right), \tag{75}$$

$$w_{l_k} = w^{\equiv} \left( T_{t_k}, p_c \right) \qquad \text{(Supplement S13)}, \tag{76}$$

$$\hat{G}_{X\lambda_k} = \frac{\rho_c G_{X\lambda_k}}{\mathcal{M}_d}, \tag{77}$$





$$\hat{G}_{Xl_k} = \frac{\rho_c \, G_{Xl_k}}{\mathcal{M}_d}, \tag{78}$$

where $G_{X\lambda_k}$ and $G_{Xl_k}$ (units $\mathrm{m\,s^{-1}}$) are the leaf boundary layer and stomatal conductances for element $X$ (either water $W$ or carbon $C$), respectively; $c_{\lambda_k}$ and $w_{\lambda_k}$ are the $CO_2$ mixing ratio and the specific humidity of the leaf boundary layer, respectively; and $c_{l_k}$ and $w_{l_k}$ are the $CO_2$ and specific humidity of the leaf intercellular space, respectively. As stated in

Eq. (76), we assume the leaf intercellular space to be at water vapor saturation. The leaf boundary-layer conductances are obtained following the algorithm shown in Supplement S12.2. The net $CO_2$ assimilation flux and stomatal conductances are described below.

From Farquhar et al. (1980), the net $CO_2$ assimilation flux is defined as:

$$\dot{A}_k = \underbrace{\dot{V}_{C_k}}_{\text{Carboxylation}} - \underbrace{\frac{1}{2}\dot{V}_{O_k}}_{\substack{\text{Oxygenation} \\ \text{(Photorespiration)}}} - \underbrace{\dot{R}_k}_{\text{Day respiration}}. \tag{79}$$

Oxygenation releases $0.5\,\mathrm{mol_{CO_2}}$ for every $\mathrm{mol_{O_2}}$, hence the half multiplier, and it is related to carboxylation by means of the $CO_2$ compensation point $\Gamma_k$ (Lambers et al., 2008):

$$\dot{V}_{O_k} = \frac{2\Gamma_k}{c_{l_k}}\dot{V}_{C_k}, \tag{80}$$

where $c_{l_k}$ is the $CO_2$ mixing ratio in the leaf intercellular space. The $CO_2$ compensation point is determined after Collatz et al. (1991, 1992):

$$\Gamma_k = \begin{cases} \frac{o_\oplus}{2\,\phi} & \text{, in case cohort } k \text{ is a } C_3 \text{ plant} \\ 0 & \text{, in case cohort } k \text{ is a } C_4 \text{ plant} \end{cases}, \tag{81}$$

where $o_\oplus$ is the reference $O_2$ mixing ratio (Table S3), and $\phi$ represents the ratio between the rates of carboxylase to oxygenase and is a function of temperature. The general form of the function describing the metabolic dependence upon temperature for any variable $x$ (including $\phi$) is:

$$\mathcal{T}(T,x) = x_{15} \times \mathcal{Q}_{10_x}^{\frac{T-T_{15}}{10}}, \tag{82}$$

where $x_{15}$ is the value of variable $x$ at temperature $T_{15} = 288.15\mathrm{K}$, and $\mathcal{Q}_{10_x}$ is the parameter which describes temperature dependence (Table S4).

Because $C_4$ plants have a mechanism to concentrate $CO_2$ near the $CO_2$-fixing enzyme Rubisco (Ribulose-1,5-Biphosphate Carboxylase Oxygenase), photorespiration is nearly nonexistent in $C_4$ plants (Lambers et al., 2008), hence the assumption that $\Gamma_k$ is zero. For $C_4$ plants, the carboxylation rate under Ribulose-1,5-Biphosphate (RuBP) saturated conditions becomes the maximum capacity of Rubisco to perform the carboxylase function ($\dot{V}_{C_k} = \dot{V}_{C_k}^{\max}$). For $C_3$, this rate is unattainable even

under RuBP-saturated conditions because carboxylation and oxygenation are mutually inhibitive reactions (Lambers et al., 2008). Therefore, the maximum attainable carboxylation ($\dot{V}_{C_k} = \dot{V}_{C_k}^{\mathrm{RuBP}}$) is expressed by a modified Michaelis-Menten kinetics




equation:

$$\dot{V}_{C_k}^{\mathrm{RuBP}} = \begin{cases} \dot{V}_{C_k}^{\max} \dfrac{c_{l_k}}{c_{l_k} + \mathcal{K}_{\mathrm{ME}_k}} & \text{, if cohort } k \text{ is a C}_3 \text{ plant} \\ \dot{V}_{C_k}^{\max} & \text{, if cohort } k \text{ is a C}_4 \text{ plant} \end{cases}, \tag{83}$$

where $\mathcal{K}_{\mathrm{ME}_k} = \mathcal{K}_{C_k}(1 + o_\oplus/\mathcal{K}_{O_k})$ is the effective Michaelis constant, and $\mathcal{K}_{C_k}$ and $\mathcal{K}_{O_k}$ are the Michaelis constants for carboxylation and oxygenation, respectively. Both $\mathcal{K}_{C_k}$ and $\mathcal{K}_{O_k}$ are dependent on temperature, following Eq. (82) (default parameters in Table S4), whereas $\dot{V}_{C_k}^{\max}$ follows a modified temperature-dependent function to account for the fast decline of both productivity and respiration at low and high temperatures (Sellers et al., 1996; Moorcroft et al., 2001):

$$\mathcal{T}'(T,x) = \frac{\mathcal{T}(T,x)}{\{1 + \exp[-f_{\mathrm{Cold}}(T - T_{\mathrm{Cold}})]\}\{1 + \exp[+f_{\mathrm{Hot}}(T - T_{\mathrm{Hot}})]\}}, \tag{84}$$

where $f_{\mathrm{Cold}}$, $f_{\mathrm{Hot}}$, $T_{\mathrm{Cold}}$ and $T_{\mathrm{Hot}}$ are PFT-dependent, phenomenological parameters to reduce the function value at low and high temperatures (Table S5).

The original expression for the initial slope of the carboxylation rate under near-zero $CO_2$ ($\dot{V}_{C_k}^{\mathrm{InSl}}$) for $C_4$ plants by Collatz et al. (1992) has been modified later (e.g. Foley et al., 1996) to explicitly include $\dot{V}_{C_k}^{\max}$; this is the same expression used in ED-2.2:

$$\dot{V}_{C_k}^{\mathrm{InSl}} = k_{\mathrm{PEP}}\, \dot{V}_{C_k}^{\max}\, c_{l_k}, \tag{85}$$

where $k_{\mathrm{PEP}}$ represents the initial slope of the response curve to increasing $CO_2$; the default value in ED-2.2 (Table S4) is the same value used by Collatz et al. (1992).

From the total photosynthetically active irradiance absorbed by the cohort $\dot{Q}_{\mathrm{PAR}:a,t_k}$ (Eq. 49), we define the photon flux that is absorbed by the leaf ($\dot{q}_k^{\mathrm{PAR}}$, $\mathrm{mol\,m_{Leaf}^{-2}\,s^{-1}}$):

$$\dot{q}_k^{\mathrm{PAR}} = \frac{1}{\mathrm{Ein}}\frac{f_{\mathrm{Clump}_k}}{\tilde{\Pi}_k}\dot{Q}_{\mathrm{PAR}:a,t_k}, \tag{86}$$

where Ein is the average photon-specific energy in the PAR band ($0.4-0.7\,\mu$m; Table S3). Even though a high fraction $\epsilon_k^\star$ of the absorbed irradiance is used to transport electrons needed by the light reactions of photosynthesis (Lambers et al., 2008), only a fraction of the irradiance absorbed by the leaf is absorbed by the chlorophyll; in addition, the number electrons needed by each carboxylation and oxygenation reaction poses an additional restriction to the total carboxylation rate. The product of these three factors is combined into a single scaling factor for total absorbed PAR, the quantum yield ($\epsilon_k$), which is a PFT-dependent property in ED-2.2 (Table S5). The maximum carboxylation rate under light limitation $\dot{V}_{C_k}^{\mathrm{PAR}}$ is:

$$\dot{V}_{C_k}^{\mathrm{PAR}} = \epsilon_k \dot{q}_k^{\mathrm{PAR}}\frac{1}{1+\dfrac{\dot{V}_{O_k}}{\dot{V}_{C_k}}} = \begin{cases} \epsilon_k \dot{q}_k^{\mathrm{PAR}}\dfrac{c_{l_k}}{c_{l_k}+2\Gamma_k} & \text{, if cohort } k \text{ is a C}_3 \text{ plant} \\ \epsilon_k \dot{q}_k^{\mathrm{PAR}} & \text{, if cohort } k \text{ is a C}_4 \text{ plant} \end{cases}. \tag{87}$$





Carboxylation may also be limited by the export rate of starch and sucrose that is synthesized by triose phosphate, especially when $CO_2$ concentration is high combined with high irradiance, at low temperatures, or $O_2$ concentration is low (von Caemmerer, 2000; Lombardozzi et al., 2018). This limitation was not included in ED-2.2.

Day respiration comprises all leaf respiration terms that are not dependent on photosynthesis, and it is mostly due to mitochondrial respiration; it is currently represented as a function of the maximum carboxylation rate, following Foley et al. (1996):

$$\dot{R}_k = f_R \dot{V}_{C_k}^{\max}, \tag{88}$$

where $f_R$ is a PFT-dependent parameter (Table S5).

Stomatal conductance is controlled by plants and is a result of a trade-off between the amount of carbon that leaves can uptake and the amount of water that plants may lose. Leuning (1995) proposed a semi-empirical stomatal conductance expression for water based on these trade-offs:

$$\hat{G}_{Wl_k} = \begin{cases} \hat{G}_{Wl_k}^{\varnothing} + \dfrac{M_k \dot{A}_k}{(c_{\lambda_k} - \Gamma_k)\left(1 + \dfrac{w_{l_k} - w_{\lambda_k}}{\Delta w_k}\right)} & \text{, if } \dot{A}_k > 0 \\[1em] \hat{G}_{Wl_k}^{\varnothing} & \text{, if } \dot{A}_k \leq 0 \end{cases}, \tag{89}$$

where $\hat{G}_{Wl_k}^{\varnothing}$ is the residual conductance when stomata are closed, $M_k$ is the slope of the stomatal conductance function, and $\Delta w_k$ is an empirical coefficient controlling conductance under severe leaf-level water deficit; all of them are PFT-dependent parameters (Table S5). From Cowan and Troughton (1971), stomatal conductance of $CO_2$ is estimated by the ratio $f_{Gl}$ between the diffusivities of water and $CO_2$ in the air (Table S4):

$$\hat{G}_{Wl_k} = f_{Gl} \hat{G}_{Cl_k}. \tag{90}$$

Variables $w_{l_k}$, $\dot{V}_{C_k}^{\max}$, $\dot{R}_k$, $\phi_k$, $\mathcal{K}_{O_k}$, $\mathcal{K}_{C_k}$, $\Gamma_k$, and $\mathcal{K}_{ME_k}$ are functions of leaf temperature and canopy air space pressure, and thus can be determined directly. In constrast, nine variables are unknown for each limitation case as well as for the case when the stomata are closed: $\dot{E}_k$, $\dot{A}_k$, $\dot{V}_{C_k}$, $\dot{V}_{O_k}$, $c_{l_k}$, $c_{\lambda_k}$, $w_{\lambda_k}$, $\hat{G}_{Wl_k}$, and $\hat{G}_{Cl_k}$. The remaining unknown variables are determined numerically, following the algorithm described in Supplement S14.

The stomatal conductance model by Leuning (1995) (Eq. 89) is regulated by leaf vapor pressure deficit, however, Eq. (74) and Eq. (75) do not account for soil moisture limitation on photosynthesis. To represent this additional effect, we define a soil-moisture dependent scaling factor ($f_{Wl_k}$, Supplement S15) to reduce productivity and transpiration as soil available water decreases. Because stomatal conductance cannot be zero, the scaling factor $f_{Wl_k}$ interpolates between the fully closed case and the solution without soil moisture limitation, yielding to the actual fluxes of $CO_2$ ($\dot{C}_{l_k,c}$, $\mathrm{kg_C\,m^{-2}\,s^{-1}}$) and water ($\dot{W}_{l_k,c}$, $\mathrm{kg_W\,m^{-2}\,s^{-1}}$):

$$\dot{C}_{l_k,c} = -\not{b}_k \mathcal{M}_C \Lambda_k \left[(1 - f_{Wl_k})\dot{A}_k^{\varnothing} + f_{Wl_k} \dot{A}_k\right], \tag{91}$$





$$\dot{W}_{l_k,c} = \mathrm{p}_k\,\mathcal{M}_w\,\Lambda_k\left[(1 - f_{Wl_k})\,\dot{E}_k^{\varnothing} + f_{Wl_k}\,\dot{E}_k\right], \tag{92}$$

where $\mathrm{p}_k$ is 1 if the PFT is hypostomatous or 2 if the PFT is amphistomatous or needleleaf. Alternatively, Xu et al. (2016) implemented a process-based plant hydraulics scheme that solves the soil-stem-leaf water flow in ED-2.2; details of this implementation are available in the referred paper.

For simplicity, we assume that the water content in the leaf intercellular space and the plant vascular system are constant, therefore the amount of water lost by the intercellular space through transpiration always matches the amount of water absorbed by roots. Plants may extract water from all layers to which they have access, and the amount of water extracted from each layer is proportional to the available water in the layer relative to the total available water:

$$\sum_{j=j_{0k}}^{N_G} \dot{W}_{g_j,l_k} = \dot{W}_{l_k,c}, \tag{93}$$

$$\dot{W}_{g_j,l_k} = \dot{W}_{l_k,c}\,\frac{W_{g_j}^{\star} - W_{g_{j+1}}^{\star}}{W_{g_{j0}}^{\star}}, \tag{94}$$

and $W_{g(N_G+1)}^{\star} \equiv 0$. The net water flux in the leaf intercellular space due to transpiration is assumed to be zero, however the associated net energy flux cannot be zero. Water enters the leaf intercellular space as liquid water at the soil temperature, reaches thermal equilibrium with leaves, and is lost to the canopy air space as water vapor at the leaf temperature. Therefore, the enthalpy flux between the soil layers and the cohort is calculated similarly to Eq. (35), whereas the enthalpy flux between

the leaf intercellular space and the canopy air space is solved similarly to Eq. (73):

$$\dot{H}_{g_j,l_k} = \dot{W}_{g_j,l_k}\,q_\ell\left(T_{g_j} - T_{\ell 0}\right), \tag{95}$$

$$\dot{H}_{l_k,c} = \dot{W}_{l_k,c}\,q_{pv}\left(T_{t_k} - T_{v0}\right). \tag{96}$$

### 4.7   Non-leaf autotrophic respiration

Respiration from fine roots is defined using a phenomenological function of temperature that has the same functional form

as leaf respiration (Moorcroft et al., 2001). Because roots are allowed in multiple layers, and in ED-2.2 roots have a uniform distribution of mass throughout the profile, the total respiration ($\dot{C}_{r_k,c}$: $\mathrm{kg_C\,m^{-2}\,s^{-1}}$) is the integral of the contribution from each soil layer, weighted by the layer thickness:

$$\dot{C}_{r_k,c} = C_{r_k}\,\frac{\sum_{j=j_{0k}}^{N_G}\left[\mathcal{T}'\left(T_{g_j},r_{r_k}\right)\Delta z_{g_j}\right]}{\sum_{j=j_{0k}}^{N_G}\Delta z_{g_j}}, \tag{97}$$

where $r_{r_k}$ ($\mathrm{s^{-1}}$) is the PFT-dependent decay rate due to root respiration (Table S5), and $\mathcal{T}'$ is the same temperature-dependent function from Eq. (84); default parameters are listed in Table S5.

     Total storage respiration is a combination of two terms: a phenomenological term that represents the long-term turnover rate of the accumulated storage pool (individual-based $\dot{R}_{n_k}$ or flux-based $\dot{C}_{n_k,c}$), assumed constant (Medvigy et al., 2009), and a

term related to the losses associated with the assimilated carbon for growth and maintenance of the living tissues (individual-based $\dot{R}_{\Delta_k}$ or flux-based $\dot{C}_{\Delta_k,c}$, Amthor, 1984). The latter is a strong function of the plant metabolic rate, which has strong



daily variability hence is a function of the daily carbon balance:

$$\dot{C}_{n_k,c} = \tau_{n_k} C_{n_k}, \tag{98}$$

$$\dot{C}_{\Delta_k,c} = \tau_{\Delta_k} C_{\Delta_k}, \tag{99}$$

where $(\tau_{n_k}, \tau_{\Delta_k})$ are the PFT-dependent decay rates associated with storage turnover and consumption for growth, respectively (Table S5); and $C_{\Delta_k}$ ($\mathrm{kg_C\,m^{-2}}$) is the total accumulated carbon from the previous day as defined in Eq. (25). The transport from non-structural storage and the accumulated carbon for maintenance, growth and, storage is summarized in Supplement S2.

## 4.8 Heterotrophic respiration

Heterotrophic respiration comes from the decomposition of carbon in the three soil/litter carbon pools. For each carbon pool
$e_j; j \in (1, 2, 3)$, we determine the maximum carbon loss based on the characteristic decay rate, which corresponds to the typical half-life for metabolic litter ($e_1$); structural litter ($e_2$); and slow soil organic matter ($e_3$) determined from Bolker et al. (1998):

$$\dot{C}_{e_j,c} = C_{e_j} f_{he_j} B_{e_j} \mathcal{E}_T\left(\overline{T}_{g_{20}}\right) \mathcal{E}_{\vartheta'}\left(\overline{\vartheta}'_{20}\right), \tag{100}$$

where $f_{h\mathbf{e}}$ is the fraction of decay that is lost through respiration (Table S4), and by definition $f_{he_3}$ must be always one (slow soil carbon can only be lost through heterotrophic respiration); $B_{\mathbf{e}}$ are the decay rates at optimal conditions, based on Bolker
et al. (1998) (Table S4); $\overline{T}_{g_{20}}$ and $\overline{\vartheta}'_{20}$ are the average temperature and relative soil moisture of the top $0.2\,\mathrm{m}$ of soil; the relative soil moisture for each layer is defined as:

$$\vartheta'_{g_j} = \frac{\vartheta_{g_j} - \vartheta_{\mathrm{Re}}}{\vartheta_{\mathrm{Po}} - \vartheta_{\mathrm{Re}}}; \tag{101}$$

and $\mathcal{E}_T(\overline{T}_{g_{20}})$ and $\mathcal{E}_{\vartheta'}(\overline{\vartheta}'_{20})$ are functions that reduces the decomposition rate due to temperature or soil moisture under extreme conditions:

$$\mathcal{E}_T(\overline{T}_{g_{20}}) = \frac{1}{\left\{1 + \exp\left[-e_{\mathrm{Cold}}\left(\overline{T}_{g_{20}} - T_{g_{\mathrm{Cold}}}\right)\right]\right\}\left\{1 + \exp\left[-e_{\mathrm{Hot}}\left(\overline{T}_{g_{20}} - T_{g_{\mathrm{Hot}}}\right)\right]\right\}}, \tag{102}$$

$$\mathcal{E}_{\vartheta'}(\overline{\vartheta}'_{20}) = \frac{1}{\left\{1 + \exp\left[-e_{\mathrm{Dry}}\left(\overline{\vartheta}'_{20} - \vartheta'_{\mathrm{Dry}}\right)\right]\right\}\left\{1 + \exp\left[+e_{\mathrm{Wet}}\left(\overline{\vartheta}'_{20} - \vartheta'_{\mathrm{Wet}}\right)\right]\right\}}, \tag{103}$$

where $(e_{\mathrm{Cold}}; T_{g_{\mathrm{Cold}}})$, $(e_{\mathrm{Hot}}; T_{g_{\mathrm{Hot}}})$, $(e_{\mathrm{Dry}}; \vartheta'_{\mathrm{Dry}})$ and $(e_{\mathrm{Wet}}; \vartheta'_{\mathrm{Wet}})$ are phenomenological parameters to decrease decomposition rates at low and high temperatures, and dry and saturated soils, respectively (Table S4). The decay fraction from fast and structural soil carbon that is not lost through heterotrophic respiration is transported to the slow soil carbon (Supplement S3).

## 5 Results

### 5.1 Conservation of energy, water, and carbon dioxide

The ED-2.2 simulations show an excellent conservation of the total energy, water, and carbon (Fig. 6). In the example simulation for one patch at GYF, the accumulated deviation from perfect closure (residual) of the energy budget over 50 years was $0.1\%$





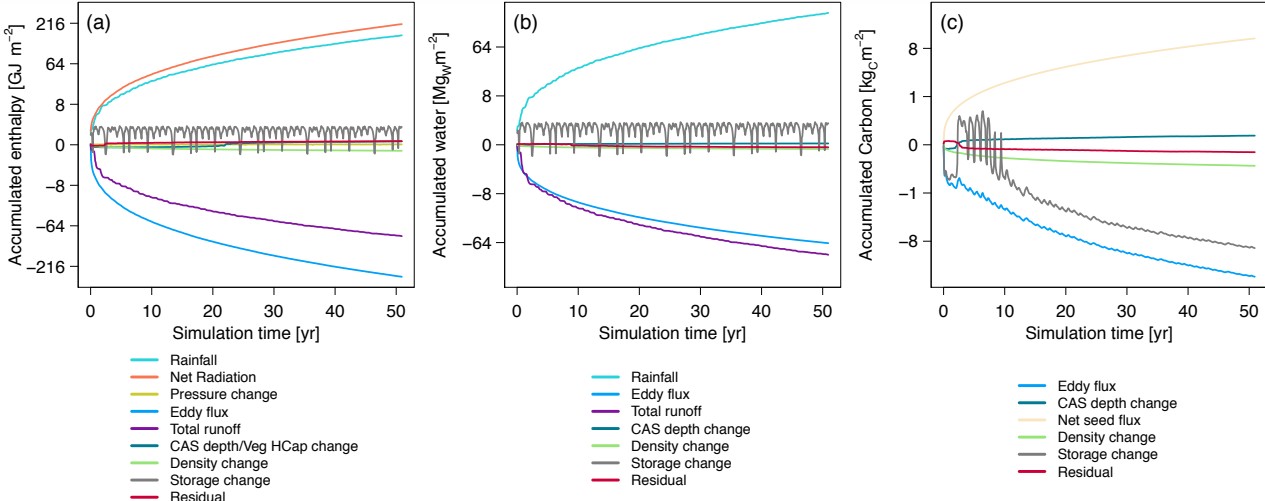

**Figure 6.** Example of (a) enthalpy, (b) water, and (c) carbon conservation assessment in ED-2.2, for a single-patch simulation at GYF for 50 years. Terms are presented as the cumulative contribution to the change storage. Total storage is the combination of canopy air space, cohorts, temporary surface water and soil layers in the case enthalpy and water, and canopy air space, cohorts, seed bank, and soil carbon pools in the case of carbon. Positive (negative) values mean accumulation (loss) by the combined storage pool over the time. Pressure change accounts for changes in enthalpy when pressure from the meteorological forcing is updated, and density change accounts for changes in mass to ensure the ideal gas law. Canopy air space (CAS) change and vegetation heat capacity (Veg Hcap) change reflect the addition/subtraction of carbon, water, and enthalpy due to the vegetation dynamics modifying the canopy air space depth and the total heat capacity of the vegetation due to biomass accumulation or loss. Storage change is the net gain or loss of total storage, and residual corresponds to the deviation from the perfect closure. Note that we present the $y$ axis in cube root scale to improve visualization of the smallest terms.

of the total enthalpy storage — sum of enthalpy stored at the canopy air space, cohorts, temporary surface water and soil layers (Fig. 6a), which is ten times less than the tolerance error accepted in the solver of the ordinary differential equations (ODEs) that describe the energy budget in ED2, and $0.002\%$ of the accumulated losses through eddy flux, the largest cumulative flux of enthalpy. Results for the water budget were even better, with maximum accumulated residuals of $0.04\%$ of the total water stored in the ED-2.2 thermodynamic systems, or $0.0006\%$ of the total water input by precipitation (Fig. 6b), and the accumulated residual of carbon was $0.008\%$ of the total carbon storage or $0.017\%$ the total accumulated loss through eddy flux.

The conservation of energy and water of ED-2.2 also represents a substantial improvement from previous versions of the model. We carried out additional decadal-long simulations with ED-2.2 and two former versions of the model (ED-2.0.12 and ED-2.1) and the most similar configuration possible among versions, and found that cumulative residual of enthalpy relative to eddy flux loss decreased from $15.2\%$ (ED-2.0.12) or $5.7\%$ (ED-2.1) to $6.1 \cdot 10^{-5}\%$ (ED-2.2) (Fig. S3a-c). Similarly, the cumulative violation of perfect water budget closure, relative to total precipitation input, decreased from $3.4\%$ (ED-2.0.12) or $1.1\%$ (ED-2.1) to $1.2 \cdot 10^{-4}\%$ (ED-2.2) (Fig. S3d-f).





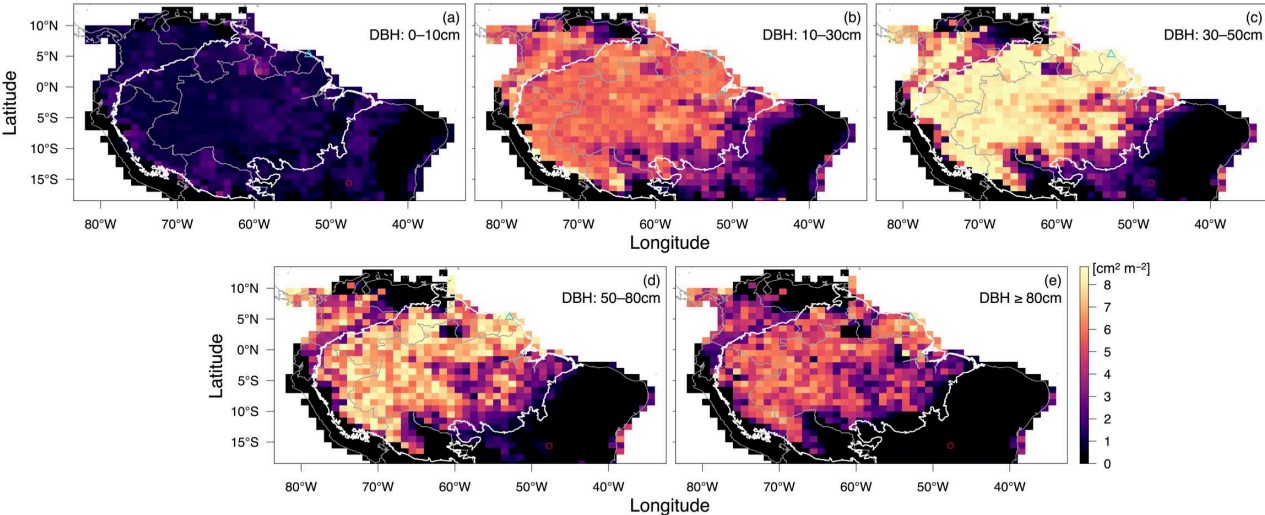

**Figure 7.** Simulated distribution of size-dependent basal area across tropical South America, aggregated for the following diameter at breast height (DBH) bins: (a) $0 - 10\,$cm; (b) $10 - 30\,$cm; (c) $30 - 50\,$cm; (d) $50 - 80\,$cm; (e) $\geq 80\,$cm. Maps were obtained from the final state of a 500-year simulation (1500–2000), initialized with near-bare ground conditions, active fires, and with prescribed land use changes between 1900 and 2000. Points indicate the location of the example sites (Fig. 8): (blue triangle) Paracou (GYF), a tropical forest site; (red circle) Brasília (BSB), a woody savanna site. White contour is the domain of the Amazon biome, and grey contours are the political borders.

## 5.2 Simulated ecosystem heterogeneity

Because ED-2.2 accounts for the vertical distribution of the plant community and the local heterogeneity of ecosystems, it is possible to describe the structural variability of ecosystems using continuous metrics. To illustrate this, we show the results of a

5-century simulation $(1500-2002)$ carried out for tropical South America, starting from near-bare ground conditions and driven by the Princeton Global Meteorological Forcing (Sheffield et al., 2006, ; $1969-2008$), and with active fires (Supplement S2.4). For the last 100 years, we also prescribed land use changes derived from Hurtt et al. (2006) and Soares-Filho et al. (2006). The distribution of basal area binned by diameter at breast height (DBH) classes show high variability across the domain, and even within biome boundaries (Fig. 7). For example, larger trees (DBH $\geq 50\,$cm) are nearly absent outside the Amazon

biome, with the exception of more humid regions such as the Atlantic Forest along the Brazilian coast, western Colombia, and Panama (Fig. 7d,e). In contrast, in seasonally dry areas as the Brazilian cerrado, intermediate-sized trees ($10 \leq \mathrm{DBH} < 50\,$cm) contribute the most to the basal area (e.g. areas near site BSB, Fig. 7b,c). Even within the Amazon ecoregion, basal area shows variability in the contribution of trees with different sizes, including the areas outside the arc of deforestation along the southern and eastern edges of the biome (Fig. 7). Similarly, the abundance of different plant functional groups shows great variability

across the region, with dominance of grasses and early-successional tropical trees in deforested regions and in drier areas in the Brazilian Cerrado, whereas late-succesional tropical trees dominating the tropical forests, albeit with lower dominance in parts of Central Amazonia (Fig. S4).



The variability of forest structural and functional composition observed in regional simulations emerge from both the competition among cohorts in the local microenvironment and the environmental controls on the disturbance regime. In Fig. 8
we present the impact of different disturbance regimes modulating the predicted ecosystem structure and composition for two sites: Paracou (GYF), a tropical forest region in French Guiana, and Brasília (BSB), a woody savanna site in Central Brazil. Both sites were simulated for 500 years using a 40-year meteorological forcing developed from local meteorological observations, following the methodology described in Longo et al. (2018); we allowed fires to occur but for simplicity we did not prescribe land use change. After 500 years of simulation, the structure at the two sites are completely different, with large,
late-successional trees dominating the canopy at GYF (Fig. 8a) and open areas with shorter, mostly early-successional trees dominating the landscape at BSB (Fig. 8b). For GYF, the structural and functional composition is achieved only after 200 years of simulation, whereas in BSB a dynamic steady state caused by the strong fire regime is achieved in about 100 years (Fig. S5). At both sites, early sucessional trees dominate the canopy at recently disturbed areas (Fig. 8c,d) with late-successional (GYF) or mid-successional trees (BSB) increasing in size only at the older patches ($> 30$ years, Fig. 8c,d), and the variation of basal area as a function of age since last disturbance show great similarity at both sites (Fig. 8e). However, the disturbance regimes are markedly different: at GYF, fires never occurred and disturbance was driven exclusively by tree fall (prescribed at $1.11\%, \mathrm{yr}^{-1}$), whereas fires substantially increase the disturbance rates at BSB (average fire return interval of 19.3 years). Consequently, old-growth patches (older than 100 years) are inexistent at BSB and abundant at GYF (Fig. S5f). In addition,
5 the high disturbance regime at BSB meant that large trees and late-sucessional trees (slow growers) failed to establish, but succeeded and maintained a stable population at GYF (Fig. S5).

The impacts of simulating structurally and functionally diverse ecosystems are also observed in the fluxes of energy, water, carbon, and momentum. For example, in Fig. 9 we present the monthly average fluxes from the last 40 years of simulation at GYF, along with the interannual variability of the fluxes aggregated to the stand-level (hereafter *stand variability*, error
10 bars) and the interannual variability of the fluxes accounting for the patch probability (herafter *patch variability*, colors in the background). In all cases, the patch variability far exceeded the stand variability. In the case of sensible heat, stand variable was between 39 and 64% of the patch variability (Fig. 9a). The stand-to-patch variability ratio was similar for both friction velocity ($19 - 39\%$) and water fluxes ($17 - 44\%$) (Fig. 9b,c). In the case of gross primary productivity the relevance of patch variability was even higher, with stand-to-patch variability ratio ranging from $3.7\%$ during the dry season to $17\%$ during the
15 wet season (Fig. 9d). Importantly, the broader range of fluxes across patches in the site can be entirely attributed to structural and functional diversity, because all patches were driven by the same meteorological forcing.

## 6 Discussion

### 6.1 Conservation of biophysical and biogeochemical properties

As demonstrated in Section 5.1, it is possible to represent the long-term, large-scale dynamics of heterogeneous and functionally diverse plant canopy while still accurately conserving the fluxes of carbon, water and energy fluxes that occur the
5 ecosystem. ED-2.2 exhibits excellent conservation of energy, water, and carbon dioxide even in multi-decadal scales. After 50





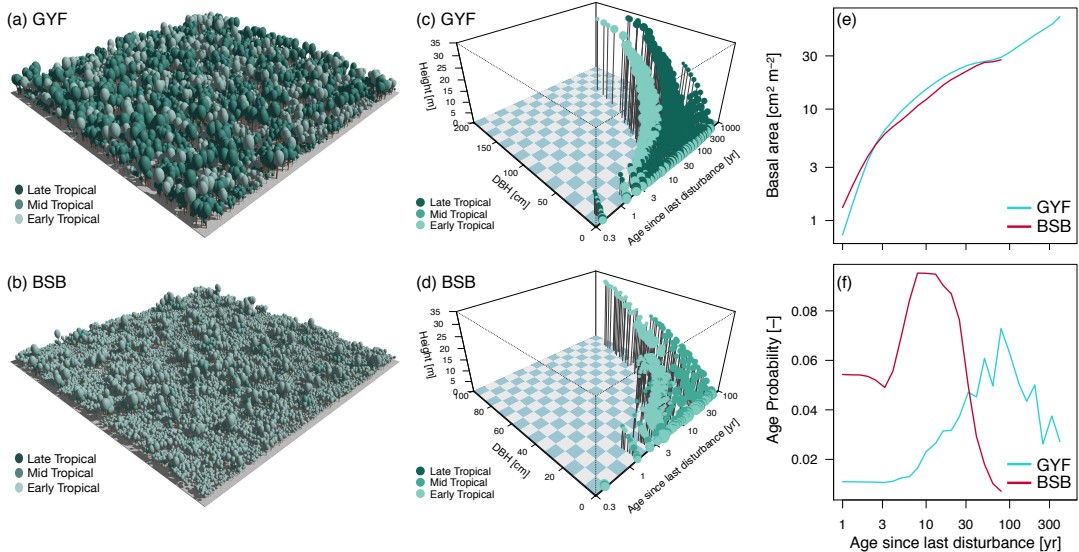

**Figure 8.** Example of size, age, and functional structure simulated by ED-2.2, after 500 years of simulation using local meteorological forcing and active fires. (a-b) Individual realization of simulated stands for sites (a) Paracou (GYF, tropical forest); (b) Brasília (BSB, woody savanna), using POV-Ray. The number of individuals shown is proportional to the simulated stem density, the distribution in local communities is proportional to the patch area, the crown size and stem height are proportional to the cohort size, and the crown color indicates the functional group. (c-d) Distribution of cohorts as a function of size (diameter at breast height (DBH) and height), as function of age since last disturbance (patch age) for sites (c) GYF and (d) BSB. Crown sizes are proportional to the logarithm of the stem density within each patch. (e,f) Patch-specific properties as a function of age since last disturbance (patch age) for sites GYF and BSB after 500 years of simulation: (e) basal area and (f) probability density function of age (patch area). See Fig. 7 for the location of both example sites.

years of simulation, the accumulated residuals from perfect closure never exceeded 0.1% of the total energy, water, and carbon stored in the pools resolved by the model (Fig. 6), which is significantly less than the error accepted in each time step (1%).

The model's excellent conservation of these three key properties is possible because the ordinary differential equations are written directly in terms of the variables that we sought to conserve, thus reducing the effects of non-linearities. A key feature that facilitates the model's high level of energy conservation is the use of enthalpy as the primary state variables within the model. This contrasts with most terrestrial biosphere models, which use temperature as their energy state variable (e.g. Best et al., 2011; Oleson et al., 2013). By using enthalpy, the model can seamlessly incorporate energy storage changes caused by rapid changes in water content and consequently heat capacity. It also reduces errors near phase changes (freezing or melting), when changes in energy may not correspond to changes in temperature. The main contribution to the remaining residual errors in carbon, water, and energy fluxes comes from the linearization of the prognostic equations due to changes in density at the canopy air space (Eq. 18-19;23). The magnitude of these residuals would likely be further reduced by using the bulk enthalpy, water content, and carbon dioxide content in the canopy air space as the state variables instead of the specific enthalpy, specific humidity and $CO_2$ mixing ratio.





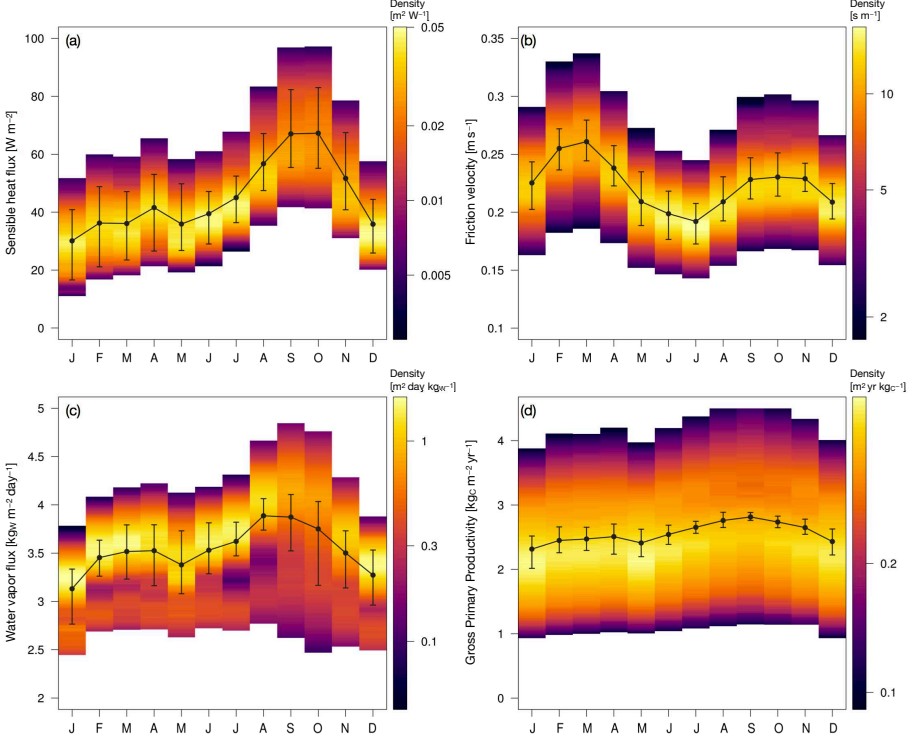

**Figure 9.** Monthly averages and variability of fluxes attributable to meteorological conditions and plant community heterogeneity combined with interannual variability. Results are shown for GYF, a tropical forest site for (a) sensible heat flux; (b) friction velocity (momentum flux); (c) water vapor flux; and (d) gross primary productivity. The variability was calculated for the last 40 years of a 500-year simulation starting from near-bare ground. Points correspond to the 40-year monthly averages for the entire stand, line bars correspond to the $2.5 - 97.5\%$ quantile of monthly averages aggregated at the stand level (stand interannual variability), and background colors represent the 40-year probability density function of monthly means for each simulated patch, and scaled by the area of each patch (patch interannual variability). Density function colors outside the $2.5 - 97.5\%$ quantile interval are not shown. Note that the density function scale is logarithmic. See Fig. 7 for the location of the example site.

Unlike most existing terrestrial biosphere models, in ED-2.2 we explicitly include the dynamic storage of energy, water, and carbon dioxide in the canopy air space. Canopy air space storage is particularly important in tall, dense tropical forests; accounting for this storage term, as well as the energy storage of vegetation allows a more realistic representation of the fluxes between the ecosystem and the air above (see also Haverd et al., 2007). In addition, the separation of the ecosystem fluxes in the model into eddy fluxes and change in canopy air space storage allows a thorough evaluation of the model's ability to represent both the total exchange and the ventilation of water, energy and carbon in and out of the ecosystem with eddy covariance towers, as shown in the companion paper (Longo et al., 2019).

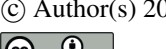



## 6.2 Heterogeneity of ecosystems

It has been long advocated that terrestrial biosphere models must incorporate demographic processes and ecosystem heterogeneity to improve their predictive ability in a changing world (Moorcroft, 2006; Purves and Pacala, 2008; Evans, 2012; Fisher et al., 2018). In ED-2.2, we aggregate individuals and forest communities according to similar characteristics (Fig. 1). For example, individuals are only aggregated into cohorts if they are of similar size, same functional group, and live in comparable micro-environments. Likewise, local plant communities are aggregated only if their disturbance history and their vertical structure are similar. The level of aggregation of ED-2.2 still allows mechanistic representation of ecological processes such as how individuals' access and competition for resources vary depending on their size, adaptation, and presence of other individuals. This approach allows representing a broad range of structure and composition of ecosystems (Fig. 7,S4), as opposed to simplified biome classification.

Previous analysis by Levine et al. (2016) has shown that the dynamic, fine-scale heterogeneity and functional diversity of the plant canopy in ED-2.2 is essential for capturing macro-scale patterns in tropical forest properties. Specifically, Levine et al. (2016) found that ED-2.1 was able to characterize the smoother observed transition in tropical forest biomass across a dry-season length gradient in the Amazon, whereas a highly aggregated (big-leaf like) version of ED-2.1 predicted abrupt shifts in biomass, which is commonly observed in many dynamic global vegetation models (e.g. Good et al., 2011). Results from two related studies have shown that the incorporation of sub-grid scale heterogeneity and diversity within ED-2 also improves its ability to correctly capture the responses of terrestrial ecosystems to environmental perturbation. First, in an assessment of the ability of four terrestrial biosphere models to capture the impact of rainfall changes on biomass in Amazon forests (Powell et al., 2013), ED-2.1 was the only model that captured the timing and average magnitude of above-ground biomass loss that was observed in two experimental drought treatments while all three big-leaf model formulations predicted minimal impacts of the drought experiment. Second, a recent analysis by Longo et al. (2018) on the impact of recurrent droughts in the Amazon found that drought-induced carbon losses in ED-2.2 arose mostly from the death of canopy trees, a characteristic that is consistent with field and remote sensing observations of drought impacts in the region (Phillips et al., 2010; Yang et al., 2018).

Importantly, since its inception, the ED model accounts for the disturbance-driven horizontal heterogeneity of ecosystems (Moorcroft et al., 2001). As demonstrated in Moorcroft et al. (2001), the continuous development of treefall gaps is fundamental explaining the long-term trajectory of biomass accumulation in tropical forests; for example, by representing both recently disturbed and old-growth fragments of forests, it is possible to simulate micro-environments where either shade-intolerant plants thrive or slow-growing, shade-tolerant individuals dominate the canopy (Fig. 8a,c). Moreover, ED-2.2 can also represent dynamic and diverse disturbance regimes, which ultimately mediate the regional variation of ecosystem properties. For example, tropical forests and woody savannas may share similarities in local communities with similar age since disturbance (Fig. 8e); however, because fire disturbances frequently affect large areas in the savannas, fragments of old-growth vegetation are nearly absent in these regions (Fig. 8f), which creates an environment dominated mostly by smaller trees (Fig. S5c).

Furthermore, the heterogeneity of ecosystems in ED-2.2 is integrated across all time scales, because we solve the biophysical and biogeochemical cycles for each cohort and each patch separately (Fig. 2-3). While solving the cycles at sub-grid scale adds



complexity, it also improves the characterization of heterogeneity of available water and energy for plants of different sizes,
even within the same stand: for example, the light profile and soil water availability are not only determined by meteorological
conditions, but also by the number of individuals, their height and their rooting depth, and their traits and trade-offs that deter-
mine their ability to extract soil moisture or assimilate carbon. As a result, the variability in ecosystem functioning represented
by ED-2.2 is significantly increased relative to the variability that a highly aggregated model based on the average ecosystem

structure would be able to capture (Fig. 9).

## 6.3   Current and Future Developments

In this manuscript, we focused on describing the biophysical and biogeochemical core of the ED-2.2 model, and appraising its
ability to represent both short-term (intra-annual and interannual) and long-term (decades to century) processes. However, the
ED-2.2 community is continuously developing and improving the model. In this section we summarize some of the recent and

ongoing model developments being built on top of the ED-2.2 dynamic core.

Terrestrial biosphere models still show significant uncertainties in representing photosynthesis due to missing processes and
inconsistencies in parameter estimations (Rogers et al., 2017). We are currently implementing the carboxylation limitation
by the maximum electron transport rate and by the triose phosphate utilization (von Caemmerer, 2000; Lombardozzi et al.,
2018), and constrained by observations (Norby et al., 2017), and incorporating nitrogen and phosphorus limitation. In addition,

the model has also been recently updated to mechanistically represent plant hydraulics, and first results indicate a significant
improvement of the the model's prediction of water use efficiency and water stress in tropical forests in Central America (Xu
et al., 2016). Also, to better represent the dynamics of soil carbon in ED-2.2, we are implementing and optimizing a more
detailed version of the CENTURY model (Bolker et al., 1998).

To improve the representation of surface and soil water dynamics, the model has been coupled with a hydrological routine

model that accounts for lateral flux of water as a function of terrain characteristics and simulates river discharge (Pereira et al.,
2017; Arias et al., 2018). Moreover, an integrated approach of hydraulic routing based on TOPMODEL (Walko et al., 2000;
Beven and Freer, 2001), which allows exchange of water and internal energy exchange between different sites as a function of
topographic characteristics, is being implemented in ED-2.2.

The ED-2.2 model framework is designed to simulate functionally diverse ecosystems but trait values within each functional

group are fixed. To account for the observed plasticity in many leaf traits, a new parameterization of leaf trait variation as
function of the light level, based on the parameterization by Lloyd et al. (2010) and (Xu et al., 2017) is being implemented.
In addition, the ED-2.2 model has also been recently updated to represent the light competition and parasite-host relationships
between lianas and trees (di Porcia e Brugnera et al., 2019), and it is currently being extended to incorporate plant functional
types from different biogeographic regions, such as temperate semi-arid shrublands (Pandit et al., 2018), as well as boreal
ecosystems, building on previous works using ED-1 (Ise et al., 2008).

Anthropogenic forest degradation is a pervasive throughout the tropics (Lewis et al., 2015). To improve the model's ability
to represent damage and recovery from degradation, we are implementing a selective logging module that represents the direct

impact of felling of marketable individuals, and accounts the damage associated with skid trails, roads and decks, which are



modulated by logging intensity and logging techniques (Pereira Jr. et al., 2002; Feldpausch et al., 2005). In addition, the original fire model has been recently improved to account for size- and bark-thickness-dependent survivorship (Trugman et al., 2018), and is being developed to account for natural and anthropogenic drivers of ignition, fire intensity, fire spread and fire duration (Thonicke et al., 2010; Le Page et al., 2015).

The complexity and sophistication of ED-2.2 also creates important scientific challenges. For example, the multiple processes for functionally diverse ecosystems represented by the model also requires a large number of parameters, with some of them being highly uncertain given the scarcity of data. To explore the effect of parameter uncertainty on model results and leverage the growing number of observations, the ED-2.2 model has been fully integrated with the Predictive Ecosystem Analyzer (LeBauer et al., 2013; Dietze et al., 2014), a hierarchical-Bayesian-based framework that constrains model parameters

based on available data and quantifies the uncertainties on model predictions due to parameter uncertainty.

    Importantly, the need to incorporate terrestrial ecosystem heterogeneity in Earth System Models has been long advocated (e.g. Moorcroft, 2006; Purves et al., 2008; Evans, 2012), but only recently global models have been incorporating ecological mechanisms that allow representing functionally diverse and heterogeneous biomes at global scale without relying on artificial climate envelopes. One example is the Functionally Assembled Terrestrial Ecosystem Simulation (FATES; Fisher et al., 2015),

which incorporated the patch and cohort structure of ED-2.2 into the Community Land Model (CLM; Oleson et al., 2013) framework.

## 7   Conclusions

ED-2.2 represents a significant advance in how to integrate a variety of processes ranging across multiple time scales in heterogeneous landscapes: it retains all the detailed representation of the long-term dynamics of functionally diverse, spatially

heterogeneous landscapes and long-term dynamics from the original ED ecosystem model (Moorcroft et al., 2001; Hurtt et al., 2002; Albani et al., 2006), but also solves for the associated energy, water, and $CO_2$ fluxes of plants living in horizontally and vertically stratified micro-environments within the plant canopy, which was initially implemented by Medvigy et al. (2009) (ED-2) by adapting the big-leaf land surface model LEAF-3 (Walko et al., 2000) to the cohort-based structure of ED-2.

    The results presented in the model description demonstrated that ED-2.2 has an excellent conservation of carbon, energy,

and water, even over multi-decadal scales (Fig. 6). Importantly, the current formulation of the model allows us to represent functional and structural diversity both at local and regional scales (Fig. 7-8;S4-S5), and the effect of the heterogeneity on energy, water, carbon, and momentum fluxes (Fig. 9). In the companion paper, we use data from eddy covariance towers, forest inventory, bottom-up estimates of carbon cycles and remote sensing products to assess the strengths and limitations of the current model implementation (Longo et al., 2019).

    This manuscript focused on the milestone updates in the energy, water, and carbon cycle within the ED-2.2 framework, but the model continues to be actively developed. Some of the further developments include implementing more mechanisms that

influence photosynthesis and water cycle such as plant hydraulics, nutrient cycling, expanding the plant functional diversity including trait plasticity and lianas, as well as expanding the types of natural and anthropogenic disturbances. ED-2.2 is a




collaborative, open-source model that is readily available from its repository, and the scientific community is encouraged to use the model and contribute with new model developments.

*Code availability.* The ED-2.2 software and further developments are publicly available. The most up-to-date source code, post-processing
`R` scripts, and an open discussion forum are available on https://github.com/EDmodel/ED2. The code described in this manuscript, along with a wiki-based technical manual, is stored as a permanent release at https://github.com/mpaiao/ED2/releases/tag/rev-85 and permanently stored at https://dx.doi.org/10.5281/zenodo.2579481.

*Author contributions.* M.L., R.G.K., D.M.M., M.C.D., Y.K., R.L.B., S.C.W. and P.R.M. designed the ED-2.2 model. M.L., R.G.K, D.M.M., N.M.L, M.C.D., Y.K., A.L.S.S., K.Z., C.R. and P.R.M. developed the model. M.L., R.G.K., N.M.L. and A.L.S.S. carried out the ED-2.2
simulations. M.L., R.G.K., D.M.M., N.M.L., M.C.D., Y.K., A.L.S.S. and P.R.M wrote the paper.

*Competing interests.* The authors declare no competing interests.

*Acknowledgements.* The research was partially carried out at the Jet Propulsion Laboratory, California Institute of Technology, under a contract with the National Aeronautics and Space Administration. We thank Miriam Johnston and Luciana Alves for suggestions that improved the manuscript; Alexander Antonarakis, Fabio Berzaghi, Istem Fer, Miriam Johnston, Geraldine Klarenberg, Robert Kooper, Manfredo
Brugnera, Afshin Pourmokhtarian, Thomas Powell, Daniel Scott, Shawn Serbin, Alexey Shiklomanov, Anna Trugman, Toni Viskari, and Xiangtao Xu for contributing to the code development. The model simulations were carried out at the Odyssey cluster, supported by the FAS Division of Science, Research Computing Group at Harvard University. M.L. was supported by Conselho Nacional de Desenvolvimento Científico e Tecnológico (CNPq, grant 200686/2005-4), NASA Earth and Space Science Fellowship (NNX08AU95H) and National Science Foundation (NSF, grant OISE-0730305, Amazon-PIRE). R.G.K was supported by a National Science Foundation Grant ATM-0449793 and
National Aeronautics and Space Administration Grant NNG06GD63G. A.L.S.S. was supported as a Giorgio Ruffolo Fellow in the Sustainability Science Program at Harvard University, for which support from Italy's Ministry for Environment, Land and Sea is gratefully acknowledged.





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
