# Peer review of "The biophysics, ecology, and biogeochemistry of functionally diverse, vertically- and horizontally-heterogeneous ecosystems: the Ecosystem Demography Model, version 2.2 — Part 1: Model description"

_Geoscientific Model Development, 2019_

## Short Comment (SC1) · 16 Apr 2019

Thank you for an interesting and helpful paper.

In the introduction the authors briefly outline how ED-2.2 evolved from ED-2 (lines 25-35), and comparisons against results from ED-2.0.12 and ED-2.1 are made in several places in the Results section. But it's unclear exactly how ED-2.2 is different than those previous versions. What exactly changed between those versions? What new

processes are simulated?

I think clearly identifying the differences would help the reader understand what motivates this particular model description paper and how it's new and different than the existing model description paper for ED-2 (Medvigy et al. 2009); as well as help the reader understand how those different results might arise between ED-2.0.12, ED-2.1 and ED-2.2.

---

## Author Comment (AC1) · 2 May 2019

Dear Dr. Kim,

Thank you for your positive feedback and suggestions on how to improve the motivation of this manuscript.

First, we would like to clarify that the developments presented in our manuscript reflect

changes since ED-2.0, as the technical description of ED-2.0.12 and ED-2.1 were not published and they are part of a continuous model development effort. In addition, while ED-2.0 solved the energy and water cycles at sub-daily time scales, the paper by Medvigy et al. (2009) does not describe the implementation of these cycles in the ED framework. Our manuscript describes, for the first time, the fundamental equations that govern the energy, water, and $CO_2$ dynamics in ED-2 (Section 3), and how we obtain each flux that is accounted for in the fundamental equations (Section 4).

Below we summarize the main developments between ED-2.0, ED-2.0.12, ED-2.1, and ED-2.2, which should clarify the differences between the versions.

ED–2.0

This is the first version of the ED model that seeks to solve the energy and water cycles independently for each patch and cohort. Most of the model biophysics was adapted from the LEAF-2 land surface model that is part of the Regional Atmospheric Model System (RAMS; Walko et al., 2000). The photosynthesis solver mostly follows ED-1 description (Moorcroft et al., 2001).

ED–2.0.12

- The ED-2.0 code was partially written in C (legacy from ED-1), and partially written in Fortran (legacy from RAMS). This created many challenges to correctly transfer information between codes written in two languages. By the ED-2.0.12 version, we had rewritten most of the code in Fortran (except for a few file handling functions that remain in C).

- We also switched the standard output to HDF5 (as opposed to ASCII), which allowed us to efficiently store many output variables segregated by cohorts and

patches.

We used this version as reference in the manuscript because it was the last version of ED-2 that used temperature as prognostic variable for leaf and canopy air space. In addition, this version had a code structure that was similar to ED-2.1 and ED-2.2, which helped to perform the inter-version comparison of energy and water conservation.

ED–2.1

- Leaf internal energy and canopy air space enthalpy replaced temperature as the prognostic variables. This change simplifies the solution of the energy balance, as we solve changes in energy directly from the energy balance. It also reduces the errors in energy conservation under high fluxes of water (which rapidly modifies the heat capacity of the pools) and eliminates the singularity of surface water energy at $0^\circ$C, when internal energy changes due to freezing or melting water, but temperature may not change.

- We replaced the ED-2.0 heat capacity for vegetation (based on total vegetation LAI for each patch) with cohort-specific heat capacity, which is scaled with the cohort leaf and wood biomass. To be consistent with the thermodynamic definition of extensive property, heat capacity must be linearly related with the mass of each component of the system (e.g. Dufour and van Mieghem, 1975), and the ED-2.1 and later versions are consistent with this definition.

- We replaced the LEAF-2-based surface layer model (Louis 1979) with the parameterization by Beljaars and Holtslag (1991), as the latter parameterization improved numerical stability of eddy covariance fluxes under stable conditions.

- We included an option to prescribe the silt, clay, and sand fractions of soils and use the general equations by Cosby et al. (1984) and Monteith and Unsworth

(2008) to determine the hydraulic and thermal properties of soils. In ED-2.0, soils were assigned one of the 12 fixed soil texture classes originally defined in LEAF-2 (Walko et al., 2000). This option allows using site-specific soil texture characteristics in ED-2 simulations.

- We implemented a capability to save the entire ecosystem and thermodynamic state of the model in HDF5 files, which can be used to resume interrupted simulations and yield exactly the same results of uninterrupted simulations. This option is useful when carrying out simulations with long runtime that are interrupted before reaching the end (e.g. power outage or computer clusters with queueing systems that restrict the maximum runtime for individual jobs).

ED-2.2

- We identified an important inconsistency in the definition of enthalpy. To be a true thermodynamic state variable, the values of enthalpy given a thermodynamic state must be path-independent (e.g. Dufour and van Mieghem, 1975). We identified that assuming latent heat of vaporization to be constant prevented the definition of enthalpy to be path-independent. We re-defined latent heat to be a linear function of temperature and derived a new equation for enthalpy. The new definition of enthalpy is described in detail in supplement S4, where we also demonstrate that the new definition of enthalpy meets the thermodynamic definition of state variable.

- We identified and included missing components of the energy cycle.

  - The transfer of internal energy of transpired water from soils to leaves before applying the enthalpy loss through transpiration (Eq. 95 and 96). Ignoring this transfer of energy caused errors in soil temperature dynamics, often leading to unrealistic values in shallow layers.

– The energy exchange associated with phase change (evaporation, transpiration, dew/frost formation) was corrected. Because evaporated/transpired (condensed) water effectively leaves (enters) soils and leaves, we must also account for the transfer of internal energy at the evaporated (condensed) surface. Ignoring this term leads to unrealistic variations of temperature, especially in surfaces with low heat capacity such as cohorts with low LAI.

– We implemented detailed conservation verification during the model execution, which now reports any violation of energy, water, and carbon conservation, generates detailed output of the violation, and interrupts the simulation.

• The photosynthesis solver was rewritten to allow temperature-dependent functions to be expressed as functions of $Q_{10}$. We retained the original Arrhenius-based functions as legacy options, but the new option increases the options for assimilating data into the model. In addition, the current $Q_{10}$-based parameters fix the low temperature optimum in tropical plants previously noted by Rogers et al. (2017). Importantly, we rewrote the photosynthesis solver to ensure that it would always converge to a unique solution for net assimilation rate, stomatal conductance, and intercellular carbon dioxide concentration given the environmental conditions. We describe the algorithm in detail in the supplement (S14).

• We implemented a complete mechanistic representation of leaf boundary layer conductance (Supplement S12.2) based on Monteith and Unsworth (2008), which accounts for leaf and branch characteristics for each cohort, and accounts for both the type of convection (free or forced) and the type of flow (laminar or turbulent). The original formulation was partially based on Monteith and Unsworth (2008), but it always assumed laminar flow and imposed a scaling factor based on total patch LAI, which led to numeric instabilities in sparsely vegetated areas.

• We implemented ground-to-canopy conductance formulations (Sellers et al., 1986; Massman, 1997; Massman and Weil, 1999) that accounts for the cumu-

lative drag profile of vegetated areas obtained from the cohort structure, as well as the stability of the surface layer (Supplement 12.3).

- We replaced the version control to GitHub, which makes the new code developments readily available to the scientific community and encourages users to post issues, bug fixes and pull requests to the main code repository in open and collaborative forums.

- We implemented a shared-memory parallelization of many subroutines that solve the energy, water and carbon cycles at sub-daily scale. These subroutines comprise most of the processing time in ED-2. The parallelization approach was written to take any number of cores, and it distributes patches per core based on core availability, and accounts for patch age to improve processing balance amongst cores.

We must point out that the list above describes the key regarding the energy, water, and carbon balance, and developments to improve efficiency and reproducibility of results, but it is not complete. The full list would be lengthy for a manuscript, but is available through the GitHub logs.

We will incorporate a summarized list of main developments between ED-2.0 and ED-2.2 in the revised version of this manuscript.

Best regards,

Marcos Longo, on behalf of the co-authors.

**References**

Beljaars, A. C. M. and Holtslag, A. A. M.: Flux Parameterization over Land Surfaces for Atmospheric Models, J. Appl. Meteor., 30, 327–341, 1991.

Cosby, B. J., Hornberger, G. M., Clapp, R. B., and Ginn, T. R.: A Statistical Exploration of the Relationships of Soil Moisture Characteristics to the Physical Properties of Soils, Water Resour. Res., 20, 682–690, 1984.

Dufour, L. and van Mieghem, J.: Thermodynamique de l'Atmosphère, Institut Royal Météorologique de Belgique, Gembloux, Belgium, 2 edn., 1975.

Massman, W. J.: An analytical one-dimensional model of momentum transfer by vegetation of arbitrary structure, Boundary-Layer Meteorol., 83, 407–421, 1997.

Massman, W. J. and Weil, J. C.: An Analytical one-Dimensional Second-Order Closure Model of Turbulence Statistics and the Lagrangian Time Scale Within and Above Plant Canopies of Arbitrary Structure, Boundary-Layer Meteorol., 91, 81–107, 1999.

Medvigy, D. M., Wofsy, S. C., Munger, J. W., Hollinger, D. Y., and Moorcroft, P. R.: Mechanistic scaling of ecosystem function and dynamics in space and time: Ecosystem Demography model version 2, J. Geophys. Res.-Biogeosci., 114, G01 002, 2009.

Monteith, J. L. and Unsworth, M. H.: Principles of Environmental Physics, Academic Press, London, 3rd edition edn., 2008.

Moorcroft, P. R., Hurtt, G. C., and Pacala, S. W.: A method for scaling vegetation dynamics: The Ecosystem Demography model (ED), Ecol. Monogr., 71, 557–586, 2001.

Rogers, A., Medlyn, B. E., Dukes, J. S., Bonan, G., von Caemmerer, S., Dietze, M. C., Kattge, J., Leakey, A. D. B., Mercado, L. M., Niinemets, U., Prentice, I. C., Serbin, S. P., Sitch, S., Way, D. A., and Zaehle, S.: A roadmap for improving the representation of photosynthesis in Earth system models, New Phytol., 213, 22–42, 2017.

Sellers, P. J., Mintz, Y., Sud, Y. C., and Dalcher, A.: A Simple Biosphere Model (SIB) for Use within General Circulation Models, J. Atmos. Sci., 43, 505–531, 1986.

Walko, R. L., Band, L. E., Baron, J., Kittel, T. G. F., Lammers, R., Lee, T. J., Ojima, D., Pielke, R. A., Taylor, C., Tague, C., Tremback, C. J., and Vidale, P. L.: Coupled Atmosphere–Biophysics–Hydrology Models for Environmental Modeling, J. Appl. Meteor., 39, 931–944, 2000.

---

## Referee Comment (RC1) · Ian Baker (Referee) · 12 Jun 2019

**Review:** The biophysics, ecology and biogeochemistry of functionally diverse, vertically- and horizontally-heterogeneous ecosystems. The Ecosystem Demography Model, version 2.2- Part 1: Model description

By Longo et al.

**Larger Impressions**

This paper describes the code used in ED2.2. That's pretty much it. There isn't really any 'new science' here, and in fact any text not devoted to explaining equations is just showing that the model gives reasonable results and conserves energy (there are a few paragraphs showing differences in size/age distributions for two tropical sites with different disturbance regimes). It can be hard to get a paper like this through review, but the authors are lucky to have me for a reviewer. I know the value of papers like this (e.g. the BATS NCAR Technical Manual by Dickinson et al., the LSM Technical Note by Bonan, and the Sellers SiB papers from 1986 and 1996). However, the people reviewing the methods paper want to see results, and vice versa, and they want the paper to be short. But these 'code papers' have value for the people who use models, and I appreciate that because I am one of them.

I understand there is no way to combine the Parts 1 and 2 of the ED2.2 paper. This weighty tome already comes in at over 100 pages (paper plus supplements for Part 1), yet it is critical to make the code information available for people who will use the model. Some might suggest a technical manual (like is done for CLM), and the authors have in fact done this; I took a quick look at the wiki, and I think it has very useful information for users, but it doesn't really lay out the rational for the code. Also, the authors want to get journal citations and credit for the work they've done, and I don't blame them one bit.

I'm not going to download the code and study it line by line to see if the explanations make sense. There is no way I could do that and get a review back in under a year. Therefore, it is incumbent upon the authors to *very carefully* go over the manuscript and check for typos in the equations as they appear in the paper.

Initially, I thought that perhaps I was the wrong person to review this paper. I have years of experience with SiB and CLM, but none with ED. But then I realized that makes me the perfect person to review; someone familiar with ED will already know much of the material. But if *I* can understand how ED2.2 works after reading the paper, then the authors have done their job. And I think they've succeeded. I feel fairly comfortable, for the most part, about the ED2.2 framework after reading the paper (multiple times). This paper will be useful for researchers learning or developing ED, and other models, in the future.

My formal recommendation is to accept the paper for publication, with minor revisions. I don't need to see it again.

I really like the use of enthalpy; that is an innovative way to demonstrate conservation of energy, and I'm not sure it has been used before.

I'd like to see more emphasis on what is new in ED2.2 (final paragraph in the Introduction). A bullet list might draw the reader's eye to the new features in this version of the model.

**Specific Comments**

(many of these are suggestions for grammar, and in some cases need not be implemented exactly as I suggest. They are just places where I noted typos and grammar issues. I also apologize for location indicators; in my copy there were new line numbers on each page, and after about page 26 I found a line numbered 5 at the bottom of the page sometimes.)

- Abstract, line 11: "out and is presented"
- Page 2, lines 5-15: This description of generational advances in model development does not align exactly with Sellers et al. (1997). I think it would be helpful to acknowledge the Sellers paper and put the descriptions here in that context.
- Page 3, line 7: SiB does not have an explicitly layered canopy or sunlit/shaded leaves separately treated.
- Page 3, lines 12-13: I'm confused here. I thought models were transitioning from broadly-defined 'biomes' to a PFT-based mosaic structure. This sentence says the opposite.
- Page 6: The full set of PFTs is not listed. In table S5 we're shown parameter values for the tropical grasses and trees used here, but if this paper is going to be the 'go to' manual for ED2.2, all PFTs should be listed in a table somewhere. Don't worry about the extra length-this paper is already incredibly long.
- Page 7, line 15: I'd like to see the index k introduced here. I had to wade through a bit of text in the supplements before I realized that k addressed cohorts (this might also have to do with the fact that I had a hard time seeing k in the lettering in Figure 1. It might be helpful to have a small table showing the indexes used to address sites, patches, and cohorts. By the time I had read the paper several times I think I had it figured out, but a more explicit explanation might be helpful.
- Page 9, lines 31-32: How do you specify $CO_2$ mole fraction on the timescale of the model? I'm not aware of CarbonTracker or GlobalView products that give that kind of resolution, and products with temporal averaging will cause issues with your carbon exchange during diurnal cycles (I think Jih-Wang Wang et al.,2007, talks about this). I don't see any mention of $CO_2$ drivers in the wiki either. We've always calculated atmosphere-CAS $CO_2$ exchange using a constant atmospheric value, and the flux can be easily scaled during a mesoscale- GCM- or transport-model application when a time-varying atmospheric $CO_2$ value is available in the lowest atmospheric level. This may be a recommendation more appropriate for the github wiki, but I think the authors need to explain to the user how to deal with it.

- Page 10, lines 12-13: "aboveground part each cohort" Huh? I think there is some re-wording needed here.
- Page 11, line 11: "components on the right-hand side"
- Page 15, line 1: I'm not sure I understand exactly what j-prime means. I think I know, as in there is no sub-surface runoff from any soil level above the bottom one, and no ground evaporation from layers below the surface, but this is not made explicit to the reader.
- Page 10, line 3: I'm not sure that holding energy, enthalpy, and water fluxes to zero is consistent with the explanation given on page 9, lines 24-26. If free drainage is allowed out of the bottom of the soil column, won't W-dot g0,g1 be nonzero? This needs to be made more clear.
- Page 16, lines 1-2: If layer Ns+1 does not exist, why mention it at all? Does it exist in the code as a placeholder? If so, that should be stated.
- Page 17, line 3 or so "that changes we obtain" could be "then we obtain"
- Page 21, line 3: "because the enthalpy" could be "due to the enthalpy"
- Page 28, line 22: "surface x is at temperature T with a liquid"
- Page 28, line 5 (bottom of page): "and Leuning" could be "and the Leuning"
- Page 31, lines 9-15: This temperature restriction is similar to what we've used in SiB for years. We also have a frost 'delay' term, where plants do not rebound immediately to photosynthesize during periods where temperatures may go below freezing (think spring in higher latitudes). I'd be happy to share it with you. Also, we have a humidity restriction term.
- Page 33, water extraction by roots: OK, so plants can extract water from all layers "to which they have access" (which, in a 3-layer soil I imagine is all of them), but roots have a uniform mass distribution. I think I might know why this is done. In the real world, I would expect a shorter/younger cohort to be less deeply rooted than an older/taller cohort, and grasses to be shallower rooted still. I also imagine that when this was done in ED, the short/young/grass cohorts might have died due to lack of water because the old/tall trees took it all. This is fine, but you can't have it both ways. In Section 6.2, "Heterogeneity of ecosystems" the authors claim ED 2.2 "…improves the characterization of heterogeneity…by the number of individuals, their height and rooting depth, and their traits and trade-offs that determine their ability to extract soil moisture…" which contradicts what is described in Section 4.6. These stories need to be made consistent.
- Page 37: "nonexistent"
- Page 37: "stand-level" is not defined in the paper. Does this mean polygon, site, or something else? Also, I'm not sure the significance of the paragraph comparing *stand variability* to *patch variability*. What does it mean?
- Page 38: "density in the canopy air space"
- Page 39: SiB has had a prognostic CAS since 2003 (Baker et al), based on Vidale and Stockli (2005). Just sayin'.
- Page 40, line 12: "access to and competition for"
- Page 40, lines 29-30: "is fundamental to explaining"

- Page 41, lines ?: "degradation is pervasive"
- Page 42, line 29: "has excellent conservation"

On to the Supplements!  (I do not have specific line numbers in my supplements file; I'll just have to do my best with explaining where the comments address)

- Table S2: might help to add bulk specific enthalpy, and Temporary Surface Water.
- S2: What is a "leaf elongation factor", and how is it determined? There is a long equation to describe $s_{lk}$, but we aren't told what it *means*.
- S7: is the 'b' term the Clapp and Hornberger b? If not C+H, where does the value come from?
- S7, field capacity: I've seen several definitions for determining field capacity from things like moisture potential. Is there a reference for what is being used here?
- S9: "contains contributions from reflectance and transmittance"
- S12.1: Are you really able to avoid the 'material surface at the top of the CAS' problem under stable conditions? This has been a problem for years, and may be worth a publication of its own. If you've already written it, advertise it here.
- S15: soil moisture limitation on photosynthesis. There's been a lot of work done on this with regard to the fact that individual plants maintain photosynthesis as soil dries down from wilt point, until suddenly closing stomates (Colello et al., 1998; Kim et al., 2010). This behavior, while well-known on the plant scale, is problematic when imposed on the ecosystem scale, as it frequently results in binary, or 'on-off' behavior. Many methods have been utilized to deal with it (e.g. Laio et al., 2001; Porporato et al., 2001, 2002: Rodriguez-Iturbe 2000; Baker et al., 2008, 2013; Wood et al., 1992, to name just a few ). I'd like to see more explanation of what you're doing. A graph showing how stress is imposed, from field capacity to wilt point, would be helpful. Is stress imposed in a linear fashion, or does it behave like the *btran* function in CLM? Is this function based on previous research (which should be cited), or something incorporated specially for ED2.2? If so, why?

**Figures**
- Figure 1: White text was difficult for me to read. It might be worth sacrificing the pretty clouds/sky background for something more simple. Or maybe just use red lettering.
- Figure 2: caption should say "dashed yellow arrows"
- Figure 3: caption should say there are 3 cohorts shown.

Nice paper, people. Good work.

Ian Baker
Colorado State University

**References**

Baker, I.T., A.S. Denning, N. Hanan, L. Prihodko, P.-L. Vidale, K. Davis and P. Bakwin (2003), Simulated and observed fluxes of sensible and latent heat and CO2 at the WLEF-TV Tower using SiB2.5. Glob. Change Biol., 9, 1262-1277.

Baker, I.T., L. Prihodko, A.S. Denning, M. Goulden, S. Milller, H. da Rocha (2008), Sea- sonal Drought Stress in the Amazon: Reconciling Models and Observations. J. Geophys. Res., 113, G00B01, doi:10.1029/2007JG000644.

Baker, I.T., H.R. da Rocha, N. Restrepo-Coupe, R. Sťockli, L.S. Borma, O.M. Cabral, A.O. Manzi, A.D. Nobre, S.C. Wofsy, S.R. Saleska, M.L. Goulden, S.D. Miller, F.L. Cardoso, A.S. Denning (2013), Surface ecophysiological behavior across vegetation and moisture gradients in Amazonia. Agr. Forest Meteorol., 182-183, 177-188, doi: http://dx.doi.org/10.1016/j.agformet.2012.11.015.

Bonan, G.B, 1996: A land surface model (LSM version 1.0) for ecological, hydrological, and atmospheric studies: Technical description and user's guide. NCAR/TN-417+STR, 150pp.

Colello, G.D. and Grivet, C., P.J. Sellers, J.A. Berry (1998), Modeling of Energy, Water and CO2 Flux in a Temperate Grassland Ecosystem with SiB2: May-October 1987. J. Atmos. Sci., 55, 1141-1169, 01 April 1998.

Clapp, R.B. and Hornberger, G.M. (1978), Empirical equations for some soil hydraulic properties, Water Resour. Res., 14(4), 601-604.

Dickinson, R.E., A. Henderson-Sellers, P.J. Kennedy, 1993: Biosphere-Atmosphere Transfer Scheme (BATS) Version 1e as coupled to the NCAR Community Climate Model. NCAR/TN-38+STR, 72pp.

Haynes, K.D., I.T. Baker, A.S. Denning, R. Sťockli, K. Schaefer, E.Y. Lokupitiya, J.M. Haynes (2019). Representing ecosystems using dynamic prognostic phenology based on biological growth stages: Part 1. Implementation in the Simple Biosphere Model (SiB4). Accepted for Pulication in J. Adv. Mod. Earth Sy.

Laio, F., A. Porporato, L. Ridolfi, I. Rodriguez-Iturbe (2001), Plants in water-controlled ecosystems: active role in hydrologic processes and resppnse to water stress II. Proba- bilistic soil moisture dynamics. Adv. Water Resour., 24, 707-723.

Porporato, A., F. Laio, L. Ridolfi, I. Rodriguez-Iturbe (2001), Plants in water-controlled ecosystems: active role in hydrologic processes and response to water stress III. Vegetation water stress. Adv. Water Resour., 24, 725-744.

Porporato, A., P. D'Odorico, F. Laio, L. Ridolfi, I. Rodriguez-Iturbe (2002), Ecohydrology of water-controlled ecosystems. Adv. Water Resour., 25, 1335-1348.

Rodriguez-Iturbe, I. (2000), Ecohydrology: A hydrologic perspective of climate-soil- vegetation dynamics. Water Resour. Res., 36 (1), 3-9.

Sellers, P.J., R.E. Dickinson, D.A. Randall, A.K. Betts, F.J. Hall, J.A. Berry, G.J. Collatz, A.S. Denning, H.A. Mooney, C.A. Nobre, N. Sato, C.B. Field, A. Henderson-Sellers (1997), Modeling the Exchanges of Energy, Water, and Carbon Between Continents and the Atmosphere. Science, 275, 502-509.

Sellers, P.J. and Y Mintz, Y.C. Sud and A. Dalcher, 1986: A Simple Biosphere Model (SiB) for Use within General Circulation Models. Journal of the Atmospheric Sciences, 43(6), 505-531.

Sellers, P.J., D.A. Randall, G.J. Collatz, J.A. Berry, C.B. Field, D.A. Da- zlich, C. Zhang, G.D. Collelo, and L. Bounoua,1996: A Revised Land Surface Parameteriztion (SiB2) for Atmospheric GCMs. Part I: Model Formulation. Journal of Climate, 9(4), 676-705

Vidale, P.L. and Sťockli, R., 2005: Prognostic Canopy Air Space solutions for Land Surface Exchanges. Theoretical and Applied Climatology, 80, 245- 257, doi:10.1007/s00704-004-0103-2

Wang, J.-W., A. S. Denning, L. Lu, I. T. Baker, K. D. Corbin, and K. J. Davis (2007), Observations and simulations of synoptic, regional, and local variations in atmospheric CO2. J. Geophys. Res., 112, D04108,doi:10.1029/2006JD007410, 2007

Wood, E.F., D.P. Lettenmeier, V.G. Zartarian (1992), A land-surface hydrology param- eterization with subgrid variability for general circulation models, J. Geophys. Res., 97(D3), 2717-2728.

---

## Referee Comment (RC2) · Stefan Olin (Referee) · 8 Jul 2019

The model description did not leave out any details, which is a very good thing and it is not very common for many of the existing model description papers. The downside of that is of course that the manuscript is rather long, too long in my opinon. One thing I miss from the very thorough walkthrough of vegetation models in the introduction are references to the DGVMs that are closer to ED such as LPJ-GUESS (for disclosure, I

am an LPJ-GUESS developer).

The text is easy to read, and the refernces to equations, sections and tables are good. One comment regarding the referencing to equations is that it should be concistent, for example on line 14, page 7. In my opinon it should read: Eq. 2-3 cannot ... . Another comment I have regarding the equations (or symbols) is the sometimes odd choice of symbols. Like Eq. 36-38, why choose the same symbol for a variable that you are using as an operator?, that is very confusing. The same goes for the use of exp instead of e, and on the note of the letter e, you are using it as pool ($e_j$) and as a scaling factor ($e_Hot$), I'd say that it is better to use the letter e as the mathematical constant it is, and then use some other symbol to denote your pools. And for your factors, use q or f. On the same topic of mathematical operators as variables, in Eq. 76, maybe something went wrong, there is a definition character instead of an equal sign. And again, why use operators as super scripts, just adds confusion. And likewise, in Eq. 56, is that an equal sign as a superscript or do you have an assignment within the equation? Or is it a pre-request? Either way, that equation is confusing.

With such an explicit formulation of the exchange of heat and water I find it rather strange that the incoming water does not have an explicit energy level specified. If 15 deg. C water lands on a surface that is 25 deg. C, there would be a cooling taking place. Maybe this is of minor importance in the Amazon, but in colder places this would matter. Or did I totally misread what is written in the beginning of Sect. 4.2, if so, I suggest you clarify this.

In the first paragraph of the discussion you are writing that you have demonstrated a functional diverse canopy, from the supplements I get that you have three PFTs along one functional trait axis.

Results are not really discussed nor shown, but one result that there is much focus on is the closed energy budget. Is it really closed if there is a 0.01 deviation? Is there not a great risk of error propagation if the bar is set that low? In LPJ-GUESS we are

concerned if the mass balance is off by $10^{-12}$.

Some specific comments in addition to those spotted by Ian Baker: Line 3, page 41: remove the 'a'. Line 3, page 21: intercepted instead of intercept. Line 1, page 33: What is a decay rate due to respiration? Do you mean turnover? Page 34: GYF is not defined, comes later.

―――――――――――――――

---

## Author Comment (AC2) · 1 Aug 2019

**Author's response to reviews of "*The biophysics, ecology, and biogeochemistry of functionally diverse, vertically- and horizontally-heterogeneous ecosystems: the Ecosystem Demography Model, version 2.2 – Part 1: Model description*"**

We would like to thank Dr. Baker and Dr. Olin for reviewing this manuscript, and providing constructive feedback on our model description.  We also thank Dr. Kim for the suggestion to include the differences between the ED-2 versions in this manuscript (see separate response, submitted earlier).

Below we a detailed point-by-point response to comments, suggestions, and questions raised by each referee, and how we plan to address them in the revised manuscript.  We also include proposed text modifications following the referees' suggestions ("quoted blue text").

**Responses to Dr. Ian Baker**

*IB Comment 01: Larger Impressions.  This paper describes the code used in ED2.2. That's pretty much it. There isn't really any 'new science' here, and in fact any text not devoted to explaining equations is just showing that the model gives reasonable results and conserves energy (there are a few paragraphs showing differences in size/age distributions for two tropical sites with different disturbance regimes). It can be hard to get a paper like this through review, but the authors are lucky to have me for a reviewer. I know the value of papers like this (e.g. the BATS NCAR Technical Manual by Dickinson et al., the LSM Technical Note by Bonan, and the Sellers SiB papers from 1986 and 1996). However, the people reviewing the methods paper want to see results, and vice versa, and they want the paper to be short. But these 'code papers' have value for the people who use models, and I appreciate that because I am one of them.*

> Response: We are glad the Reviewer (Dr. Baker) appreciates the value of this model description manuscript. We believe that the ED-2 model needs a detailed technical description The GMD *model description paper* type expects: "detailed, complete, rigorous, and accessible" descriptions of the model, and to present examples of model output and model verification.  We are glad that the Reviewer identified all these elements in this manuscript.

*IB Comment 02: I understand there is no way to combine the Parts 1 and 2 of the ED2.2 paper. This weighty tome already comes in at over 100 pages (paper plus supplements for Part 1), yet it is critical to make the code information available for people who will use the model. Some might suggest a technical manual (like is done for CLM), and the authors have in fact done this; I took a quick look at the wiki, and I think it has very useful information for users, but it doesn't really lay out the rational for the code. Also, the authors want to get journal citations and credit for the work they've done, and I don't blame them one bit.*

> Response: We are glad that the Reviewer agrees in the value of a manuscript describing the model in detail.  The online wiki explains how to run the ED-2.2 model whereas, as the reviewer correctly pointed out, this manuscript describes the rationale

and the parameterizations needed to solve the energy, water, and carbon cycles within the ED-2.2 framework.

*IB Comment 03: I'm not going to download the code and study it line by line to see if the explanations make sense. There is no way I could do that and get a review back in under a year. Therefore, it is incumbent upon the authors to very carefully go over the manuscript and check for typos in the equations as they appear in the paper.*

> Response: As per the Reviewer's suggestion, we will carefully revise the paper and look for typos.

*IB Comment 04: Initially, I thought that perhaps I was the wrong person to review this paper. I have years of experience with SiB and CLM, but none with ED. But then I realized that makes me the perfect person to review; someone familiar with ED will already know much of the material. But if I can understand how ED2.2 works after reading the paper, then the authors have done their job. And I think they've succeeded. I feel fairly comfortable, for the most part, about the ED2.2 framework after reading the paper (multiple times). This paper will be useful for researchers learning or developing ED, and other models, in the future.*

> Response: We thank the Reviewer for thinking that the paper was useful to understand the concepts of ED-2.2.  We sought to make the manuscript accessible to a broad community that may or may not be familiar with the ED-2.2 model.

*IB Comment 05: My formal recommendation is to accept the paper for publication, with minor revisions. I don't need to see it again.*

> Response: We thank the Reviewer for the encouraging recommendation.

*IB Comment 06: I really like the use of enthalpy; that is an innovative way to demonstrate conservation of energy, and I'm not sure it has been used before.*

Response: We also prefer the equations to be based on energy (or enthalpy) as the response variable, as this simplifies the tracking of energy and helps to enforce energy/enthalpy conservation. We too are not aware of energy/enthalpy being used as the energy-related state variable, and thus represents a novel feature ED-2.2 model.

*IB Comment 07: I'd like to see more emphasis on what is new in ED2.2 (final paragraph in the Introduction). A bullet list might draw the reader's eye to the new features in this version of the model.*

> Response: This suggestion is also in line with Dr. John Kim's iterative comment.  We propose to modify the next-to-last paragraph of the introduction to briefly describe and itemize the main changes between ED-2.0 and ED-2.2:

"The original ED model formulation was an off-line ecosystem model describing the coupled carbon and water fluxes of a heterogeneous tropical forest ecosystem (Moorcroft et al., 2001). Subsequently Medvigy et al. (2009) applied a similar approach to develop the Ecosystem Demography model version 2 (ED-2) that describes coupled carbon, water and energy fluxes of the land surface.  Since then, the ED-2 model has been continuously developed to improve several aspects of the model (see Supplement S1 for further information): (1) the conservation and thermodynamic representation of

energy, water, and carbon cycles of the ecosystems; (2) the representation of several components of the energy, water, and carbon cycles, including the canopy radiative transfer, aerodynamic conductances and eddy fluxes, and leaf physiology (photosynthesis); (3) the structure of the code, including efficient data storage, code parallelization, and version control and code availability. ED-2 has been used in many studies including offline simulations (e.g. Medvigy et al., 2009; Antonarakis et al., 2011; Kim et al. 2012, Zhang et al., 2015; Castanho et al., 2016; Levine et al., 2016), or interactively with a regional atmospheric model (e.g. Knox et al., 2015; Swann et al., 2015)."

In addition, we will include a new Supplement (S1) that will describe the timeline of changes between ED-2.0 and ED-2.2, based on our previous reply to John Kim (https://doi.org/10.5194/gmd-2019-45-AC1).

**"S1 ED-2 developments since ED-2.0 and ED-2.2**

In this Supplement, we list the main developments in the Ecosystem Demography Model version 2 (ED-2), with focus on mentioned in this manuscript (Fig. S3). The complete list of implementations, improvements, and code fixes are available on the GitHub website (https://github.com/ EDmodel/ED2).

**S1.1 Version 2.0 (ED-2.0)**

This is the version described in Medvigy (2006); Medvigy et al. (2009), and it is the first version of the ED model that implements energy and water cycles at sub-daily scale. The biophysics core was adapted from the LEAF-2 land surface model (Walko et al., 2000), which is part of the Regional Atmospheric Model System (RAMS). The main differences in the ED-2.0 biophysics core include (1) solution of the energy and water cycle for each cohort and patch; (2) use of 4th order Runge-Kutta solver to improve numerical stability. In addition, this version allowed leaf phenology to be prescribed from external data (Supplement S3.1.1). The photosynthesis solver was largely the same as in ED-1.0 (Moorcroft et al., 2001).

**S1.2 Version 2.0.12 (ED-2.0.12)**

Most developments between ED-2.0 and ED-2.0.12 relate to code organization and structure. ED-2.0 was partly written in C (legacy from ED-1) and partly written in Fortran (legacy from LEAF-2). To simplify the code and ensure data were correctly transferred between subroutines, we rewrote most of the code in Fortran. The only exceptions were a few file handling functions that remained in C because we could not find equivalent functions in Fortran.

In addition, this version uses Hierarchical Data Format 5 (HDF5) format and libraries (The HDF Group, 2016) to generate model outputs. HDF5 allows a more efficient framework to output variables in the dynamic patch and cohort structures. It also introduced an XML model parameter input file, rather than relying solely on hard-coded defaults, which makes it easier to perform model calibration, sensitivity analyses, and ensemble error propagation.  Importantly, this was the last version of ED-2 that used temperature as prognostic variable for leaves and canopy air space.

**S1.3 Version 2.1 (ED-2.1)**

Most ED-2.1 developments aimed at improving the energy cycle representation in ED-2.1. Importantly, leaf enthalpy and canopy air space enthalpy replaced temperature as the prognostic energy-related variables (Eq. 4; Sec. 3.2.3-3.2.4). The main advantages of energy-based prognostic equations include: (1) simplification the numeric integration, as total energy changes must be equivalent to net energy flux; (2) improved conservation of energy when water fluxes are large and cause rapid changes in heat capacity of the thermodynamic systems; (3) elimination of singularity at the water's fusion point (0°C, when enthalpy changes due to freezing or melting, but the temperature remains the same.

To ensure the model was thermodynamically consistent, we also: (1) implemented a mechanistic representation of heat capacity for vegetation (leaves and branches, Supplement S6.2) that is scaled with leaf and branch biomass (e.g. Dufour and van Mieghem, 1975); and (2) replaced the original LEAF-2-based surface layer model (that was based on Louis, 1979) with the parameterization by Beljaars and Holtslag (1991), as the latter parameterization improved numerical stability of eddy covariance fluxes under thermally stable conditions; (3) included an option to prescribe silt, clay, and sand fractions to define site-specific soil texture characteristics (Supplement S8) instead of the original ED-2.0 implementation that required soils to be assigned to one of the 12 fixed classes originally defined in LEAF-2 (Walko et al., 2000); (iv) implemented the capability of saving the entire ecosystem and thermodynamic state of the model into HDF5 files, which can be used to stop and start simulations and yield the same results of uninterrupted simulations, a desirable feature for simulations with long runtimes.

**S1.4 Version 2.2 (ED-2.2)**

The ED-2.2 version implemented several improvements and fixed inconsistencies in the representation of the energy, water, and carbon dioxide cycles. First, we redefined enthalpy (S5), to ensure that it would be a true thermodynamic state variable (i.e. path independent, see Dufour and van Mieghem, 1975), by making latent heat of vaporization a linear function of temperature (Eq. 72-73). Moreover, we identified missing components of the energy cycle that precluded the conservation: (1) the transfer of internal energy from soils to leaves before transpiration (Eq. 97); (2) the enthalpy exchange associated with vaporization and condensation also accounts the mass transfer of water between the thermodynamic systems (e.g. Eq. 75; 98. Furthermore, to ensure results from ED-2.2 consistently conserve mass and energy, we implemented detailed conservation verification during the model execution, which now reports any violation of energy, water, and carbon conservation, generates detailed output of the violation, and interrupts the simulation. Finally, to improve computational efficiency of the energy, water, and carbon cycle solvers at sub-daily time steps, we implemented a shared-memory parallelization of the most computationally-intensive subroutines. The parallelization was written to allow users to select any number of cores (depending on core availability), and to account for patch ages in order to balance the load among cores.

In addition, we rewrote the photosynthesis to allow temperature-dependent functions to be expressed as functions of $Q_{10}$. We retained the original Arrhenius-based functions as legacy options, but the new option increases the options for assimilating data into the

model. The current $Q_{10}$-based parameters fix the low-temperature optimum in tropical plants previously noted by Rogers et al. (2017). Importantly, we rewrote the photosynthesis solver to ensure that it would always converge to a unique solution for net assimilation rate, stomatal conductance, and intercellular carbon dioxide concentration given the environmental conditions (Supplement S15).

The ED-2.2 version also includes improvements in the representation of conductances between different thermodynamic systems. First, the leaf boundary-layer conductance now accounts for differences in leaf and branch characteristics of each cohort, and to account for both free and forced convection under both laminar and turbulent flow (Supplement S13.2). Second, we implemented ground-to-canopy conductance formulations (Sellers et al., 1986; Massman, 1997; Massman and Weil, 1999) that account for the cumulative drag profile of vegetated areas obtained from the cohort structure, as well as the stability of the surface layer (Supplement S13.3).

Finally, in ED-2.2 we replaced the version control to GitHub, which makes the new code developments readily available to the scientific community and encourages users to post issues, code fixes and model improvements and developments to the main code repository in open and collaborative forums."

*Specific Comments*

*(many of these are suggestions for grammar, and in some cases need not be implemented exactly as I suggest. They are just places where I noted typos and grammar issues. I also apologize for location indicators; in my copy there were new line numbers on each page, and after about page 26 I found a line numbered 5 at the bottom of the page sometimes.)*

*IB Comment 08: Abstract, line 11: "out and is presented"*

> Response: We will modify the text as suggested.

*IB Comment 09: Page 2, lines 5-15: This description of generational advances in model development does not align exactly with Sellers et al. (1997). I think it would be helpful to acknowledge the Sellers paper and put the descriptions here in that context.*

> Response: As suggested, we will cite Sellers et al. (1997) and revise our narrative to be consistent with Sellers et al. (1997):

"As described by Sellers et al. (1997), the first generation of land surface models (LSMs) were limited to provide boundary conditions to atmospheric models, and only solved a simplified energy and water budget, and accounted for the effects of surface on frictional effects on near-surface winds (e.g. Manabe et al., 1965; Somerville et al., 1974). These models, however, did not account for the active role of vegetation. The second generation of LSMs considered the active role of vegetation and represented the spectral properties of the canopy, the changes in roughness of vegetated surfaces, and the biophysical controls on evaporation and transpiration (Sellers et al., 1997); examples of these models include NCAR/BATS (Dickinson et al., 1986) and SiB (Sellers et al., 1986). The increasing recognition of the role of vegetation in mediating the exchanges of carbon, water and energy between the land and the atmosphere led to the third generation of LSMs, which incorporated explicit representations of plant photosynthesis,

and resulting dynamics of terrestrial carbon uptake, turnover and release within terrestrial ecosystems (Sellers et al., 1997); examples of such models included LSM (Bonan, 1995) and SiB2 (Sellers et al., 1996). While the fluxes of carbon, water and energy predicted by these models would change in response to changes in their climate forcing, the biophysical and biogeochemical properties of the ecosystem within each climatological grid cell was prescribed, and thus did not change over time."

*IB Comment 10: Page 3, line 7: SiB does not have an explicitly layered canopy or sunlit/shaded leaves separately treated.*

> Response: As suggested, we will remove the reference to SiB.

*IB Comment 11: Page 3, lines 12-13: I'm confused here. I thought models were transitioning from broadly-defined 'biomes' to a PFT-based mosaic structure. This sentence says the opposite.*

> Response: We intended to say that the lack of functional diversity and the mechanistic representation of ecological processes such as competition cause models to predict ecosystems comprised of single homogeneous vegetation types. We will rewrite the sentence to clarify this point:

"The lack of significant variability in resource conditions limits the range of environmental niches within the climatological grid cells of terrestrial biosphere, and makes the coexistence between PFTs difficult.  Consequently, models typically predict ecosystems comprised of single homogeneous vegetation types (Moorcroft, 2003, 2006)."

*IB Comment 12: Page 6: The full set of PFTs is not listed. In table S5 we're shown parameter values for the tropical grasses and trees used here, but if this paper is going to be the 'go to' manual for ED2.2, all PFTs should be listed in a table somewhere. Don't worry about the extra length-this paper is already incredibly long.*

> Response: As suggested by the reviewer, we will include the parameter values for all the default PFTs. Specifically, we will add a separate table for temperate PFTs, revise the Leaf phenology supplement (formerly S2, now S3)  to include descriptions of the cold-deciduous leaf phenology, and update Supplement S16 (Allometric equations) to include non-tropical PFTs.

**"S3.1 Leaf phenology**

The phenological strategy of the plant functional types, can be evergreen, drought-deciduous, or cold-deciduous. The plant's phenology strategy is defined by two functions: (i) the leaf elongation factor ($\hat{e}_{lk}$), defined as the ratio between the environmentally-constrained leaf biomass and the potential (maximum) leaf biomass, and Iii) the rate of leaf shedding ($\omega_{lk}(t)$). These functions can either be prognosed or prescribed from observations.

**S3.1.1 Evergreen Plants**

For evergreen PFTs, the elongation factor is always 1, the rate of leaf shedding ($\omega_{lk}(t)$) is zero, and their rate of leaf turnover is governed by the PFT-dependent leaf turnover

parameter ($\tau_l$, see Eq. S5, and Tables S5 and S6). The leaf phenology of tropical trees can also be represented by an empirical model that is driven by the seasonality of light availability (see Supplement S2 and Kim et al., 2012).

**S3.1.2 Drought-deciduous tropical phenology**

For drought-deciduous tropical plant functional types, the leaf elongation factor is governed by:

$$e_{l_k} = \begin{cases} 1 & , \text{if } s_{l_k} \geq 1 \\ s_{l_k} & , \text{if } 0.05 \leq s_{l_k} < 1 \\ 0 & , \text{if } s_{l_k} < 0.05 \end{cases} \tag{S4}$$

$$s_{l_k} = \frac{1}{|z_{r_k}| \, \Delta t_{\text{El}}} \int_{t'-\Delta_{\text{Phen}}}^{t'} \left( \sum_{j=j(z_{r_k})}^{N_G} \left\{ \frac{\max\left[0, \Psi_{g_j}(t') + \frac{1}{2}\left(z_{g_j} + z_{g_{j+1}}\right) - \Psi_{\text{Wp}}\right]}{\Psi_{\text{Ld}} - \Psi_{\text{Wp}}} \right\} \right) dt, \tag{S5}$$

where $s_{l_k}$ is a running average of soil moisture accessed by cohort $k$ (normalized by the difference between $\Psi_{\text{Ld}}$, and $\Psi_{\text{Wp}}$, the difference between soil matric potential below which plants start shedding leaves and the soil matric potential at wilting point), $z_{r_k}$ is the rooting depth of cohort $k$ (Supplement S19), $\Delta t_{El}$ is the time scale for changes in phenology (assumed to be 10 days), $j(z_{r_k})$ is the soil layer containing the deepest roots of cohort $k$, and $z_{g_j}$ is the depth of soil layer $j$, ($z_{g_{N_g+1}} \equiv 0$). Leaf shedding occurs whenever soil is drier than the threshold defined by $\Psi_{\text{Ld}}$ and drought conditions are increasing. Specifically:

$$\omega_{l_k} = \frac{1}{\Delta t_{\text{Phen}}} \max\left[0, \frac{C_{l_k}}{C_{l_k}^{\bullet}} - f_{\text{El}}\right]. \tag{S6}$$

**S3.1.3 Cold-deciduous phenology**

The prognostic cold-deciduous leaf phenology approach is a thermal sum and chilling sum-based model identical to that of Albani et al. (2006), which, in turn, is based on Botta et al. (2000). At each site, growing degree-days (GDD) are accumulated during the extended growing season ($t_{GS}$, January–August for the Northern Hemisphere, and July–February for the Southern Hemisphere), and the chilling days (CHD) in the extended senescing season ($t_{SS}$, November-June for the Northern Hemisphere, and May-December for the Southern Hemisphere):

$$\text{GDD}\,(t) = \sum_{t'=t_{\text{GS}}(0)}^{t} \max\left(0, \overline{T}_c\left(t'\right) - T_{\text{Phen}}\right), \tag{S5}$$

$$\text{CHD}\,(t) = \begin{cases} 0 & \text{, if } \overline{T}_c(t) \geq T_{\text{Phen}}, \text{ or } t \notin t_{\text{SS}} \\ \text{CHD}\,(t - \Delta t_{\text{Phen}}) + 1 & \text{, otherwise} \end{cases}, \tag{S6}$$

where $\overline{T}_C$ is the daily average canopy air space temperature, $\Delta t_{\text{Phen}}$ = 1 day is the phenology time step (Table 2), $t_{GS}(0)$ is the beginning of the growing season, and $T_{\text{Phen}}$ = 278.15 K (0°C) is the leaf phenology temperature threshold (Albani et al., 2006). The valued elongation factor $\hat{e}_{lk}$ is then determined by the following series of conditions:

$$\hat{e}_{l_k}(t) = \begin{cases} 0 & \text{, if } \overline{T}_s(t) < 275.15 \text{ K} \\ 0 & \text{, if } \overline{T}_s(t) < 284.30 \text{ K and } t_{\odot} < 655 \text{ min} \\ 1 & \text{, if GDD} \geq -68.0 + 638.0\exp\left[-0.01\,\text{CHD}\,(t)\right] \\ \hat{e}_{l_k}(t - \Delta t_{\text{Phen}}) & \text{, otherwise} \end{cases}, \tag{S7}$$

where t⊙ is the daytime duration."

If desired, cold-deciduous phenology can be prescribed rather than prognosed, as described in Medvigy et al. (2009) and Viskari et al. (2015). The timing of leaf onset and leaf senescence are empirically determined from either field observations or from remote sensing (e.g. Zhang et al., 2003) by fitting the following curves, which are then used to determine $\hat{e}_{lk}$ in the model:

$$\hat{e}_{l_k} = \begin{cases} \dfrac{1}{1 + (y_0 t)^{y_1}} & \text{, if } t \in t_{\text{GS}} \\ \dfrac{1}{1 + (y_2 t)^{y_3}} & \text{, if } t \in t_{\text{SS}} \end{cases}, \tag{S4}$$

where $y_0, y_1, y_2$, and $y_3$ are empirical parameters, determined from data prior to running the ED-2.2 model and provided to the model as inputs; $t$ is the time, provided as day of year (i.e. 1 for January 1st, 365 for December 31, and 366 for December 31 in leap years); $t_{\text{GS}}$ is the extended growing season (e.g. January-July for the Northern Hemisphere, and July-January for the Southern Hemisphere); and $t_{\text{SS}}$ is the senescing season (e.g. August-December for the Northern Hemisphere, February-June for the Southern Hemisphere).

*IB Comment 13: Page 7, line 15: I'd like to see the index k introduced here. I had to wade through a bit of text in the supplements before I realized that k addressed cohorts (this might also have to do with the fact that I had a hard time seeing k in the lettering in Figure 1. It might be helpful to have a small table showing the indexes used to address sites, patches, and cohorts. By the time I had read the paper several times I think I had it figured out, but a more explicit explanation might be helpful.*

> Response: As suggested, we will include a new Table 1 with indices associated with patches, cohorts, disturbance types and PFTs. The new Table 1 caption will also refer to

existing Table S1 that lists all the subscripts used in the manuscript. We will also revise the text to describe the subscripts, and refer to the new Tables 1 and S1.

"Equation (2) and Eq. (3) cannot be solved analytically (except for trivial cases) and therefore were solved numerically using the method of characteristics. The age distribution is discretized into a series of patches (subscript $u$, $u \in 1,2,..., N_P$; Table 1) of similar age and same disturbance type, and the population of plants living in each patch u is discretized into a series of size-cohorts (subscript $k$, $k \in 1, 2,..., N_T$ ; Table 1) containing plants of similar size and of the same PFT (Fig. 1)."

*IB Comment 14: Page 9, lines 31-32: How do you specify CO2 mole fraction on the timescale of the model? I'm not aware of CarbonTracker or GlobalView products that give that kind of resolution, and products with temporal averaging will cause issues with your carbon exchange during diurnal cycles (I think Jih-Wang Wang et al.,2007, talks about this). I don't see any mention of CO2 drivers in the wiki either. We've always calculated atmosphere-CAS CO2 exchange using a constant atmospheric value, and the flux can be easily scaled during a mesoscale- GCM- or transport-model application when a time- varying atmospheric CO2 value is available in the lowest atmospheric level. This may be a recommendation more appropriate for the github wiki, but I think the authors need to explain to the user how to deal with it.*

> Response: Above-canopy $CO_2$ is read-in as part of the meteorological drivers read routines. It therefore must be provided by the user, along with other meteorological variables. Further details on the initialization procedure can be found on ED2 site GitHub (https://github.com/EDmodel/ED2/wiki/Drivers). Ideally, atmospheric $CO_2$ should be provided at comparable temporal and spatial resolution to other meteorological variables, to avoid the issues raised by the reviewer. In the case of coupled ED2-BRAMS simulations (e.g. Knox et al. 2015; Swann et al. 2015), this happens automatically since $CO_2$ is explicitly tracked as an atmospheric state variable whose value at the lower boundary value is passed into ED2 thereby eliminating the need to scale the fluxes.

In offline simulations, $CO_2$ data are not generally available at comparable spatial scale or similar temporal resolution as other meteorological drivers. For these cases, ED-2.2 allows the user to provide time-varying $CO_2$ that is constant in space (this option is available to all meteorological drivers, as ED-2.2 can be driven by micrometeorological tower data). Alternatively, the user can set a constant $CO_2$ mixing ratio when no $CO_2$ data are available.

In line with the Reviewer's suggestion, we will add a sentence clarifying that ED-2.2 can read $CO_2$ data when available. We will also explain that ED-2.2 can read data from a single site (eddy flux tower) and gridded drivers (re-analysis):

"Meteorological drivers can be either at a single location (e.g. eddy covariance towers), or gridded meteorological drivers such as reanalysis (e.g. Dee et al., 2011; Gelaro et al., 2017) or bias-corrected products based on reanalysis (e.g. Sheffield et al., 2006; Weedon et al., 2014). Whenever available, $CO_2$ must be provided at comparable temporal and spatial resolution as other meteorological drivers; otherwise, it is possible to provide spatially homogeneous, time-variant $CO_2$, or constant $CO_2$, although this may increase uncertainties in the model predictions (e.g. Wang et al., 2007). Alternatively, the meteorological forcing (including $CO_2$) may be provided directly by BRAMS (Knox et al., 2015; Swann et al., 2015)."

*IB Comment 15: Page 10, lines 12-13: "aboveground part each cohort" Huh? I think there is some re- wording needed here.*

> Response: As suggested, we will replace this text with "leaves and branchwood portion of each cohort", to clarify the meaning of the sentence.

*IB Comment 16: Page 11, line 11: "components on the right-hand side"*

> Response: As suggested, we will correct this typographical error.

*IB Comment 17: Page 15, line 1: I'm not sure I understand exactly what j-prime means. I think I know, as in there is no sub-surface runoff from any soil level above the bottom one, and no ground evaporation from layers below the surface, but this is not made explicit to the reader.*

> Response: We used $j'$ to refer to two different soil layers ($g_j$ and $g_{j'}$). For example, in the third term on the right-hand side of Eq. (8) ($\delta_{gjgj'}$) is 0 for all layers except layer $g_1$. We did not want to add yet another index because it could be even more confusing and therefore plan to keep the $j'$ notation; however, in light of the Reviewer's suggestion, we will modify the text to clarify the meaning of $j'$:

"where $\delta g_j g_{j'}$ is the Kronecker delta for comparing two soil layers $g_j$ and $g_{j'}$ (1 if $g_j = g_{j'}$; 0 otherwise),…"

We will make a similar modification to the text after Eq. (11):

"where $\delta s_j s_{j'}$ is the Kronecker delta for comparing two TSW layers $s_j$ and $s_{j'}$ (1 if $s_j = s_{j'}$, 0 otherwise),…"

*IB Comment 18: Page 10, line 3: I'm not sure that holding energy, enthalpy, and water fluxes to zero is consistent with the explanation given on page 9, lines 24-26. If free drainage is allowed out of the bottom of the soil column, won't W-dot g0,g1 be nonzero? This needs to be made more clear.*

> Response: We believe the Reviewer was referring to Page 15, line 3. In the original submission, we had defined sub-surface runoff (drainage) as a separate term ($\dot{H}_{g1,o}$ ; $\dot{W}_{g1,o}$), and thus we defined $\dot{H}_{g0,g1}$ and $\dot{W}_{g0,g1}$ to be zero in section 3.2.1. However, we agree with the Reviewer that this is confusing, and also is inconsistent with the text in section 4.1. To remove the ambiguity, we will refer to sub-surface runoff fluxes as $\dot{H}_{g0,g1}$ and $\dot{W}_{g0,g1}$ and use subscript $o$ exclusively for surface runoff. Specifically, we propose the following three changes:

(i) Re-define the flux notation at the beginning of section 3.2:

"For any variable $X$ with that has flux between a system $m$ and a system $n$, we assume that $\dot{X}_{m,n} > 0$ when the net flux goes from system $m$ to system $n$, and that $\dot{X}_{m,n} = -\dot{X}_{n,m}$. Arrows in Fig. 2 indicate the directions allowed in ED-2.2."

(ii) Replace the notation $g_1,o$ with $g_1,g_0$ in Table 4 (which will become Table 5), and in section 4.1 (Hydrology sub-model and ground energy exchange).

(iii) Remove the sub-surface runoff term from equations (6) and (7), and describe that the terms $\dot{H}_{g0,g1}$ and $\dot{W}_{g0,g1}$ are the (negative) sub-surface runoff fluxes:

In the equations above, we assume $\dot{Q}_{g0,g1}$ to be zero (i.e. bottom boundary in thermal equilibrium); $\dot{H}_{g1,g0}$ = - $\dot{H}_{g0,g1}$, and $\dot{W}_{g1,g0}$ = - $\dot{W}_{g0,g1}$ to be sub-surface runoff fluxes (see section 4.1). In addition, $\dot{Q}_{gNG,gNG+1}$, $\dot{H}_{gNG,gNG+1}$, $\dot{W}_{gNG,gNG+1}$ are equivalent to ($\dot{Q}_{gNG,s1}$, $\dot{H}_{gNG,s1}$ $\dot{W}_{gNG,S1}$), which are the fluxes between the top-most soil layer and the bottom-most temporary surface water layer (please see also section 3.2.2)."

*IB Comment 19: Page 16, lines 1-2: If layer Ns+1 does not exist, why mention it at all? Does it exist in the code as a placeholder? If so, that should be stated.*

> Response: We made the comment because the second term on the right-hand sides of Equations (9–11) for layer $s_j = s_{NS}$ implies the existence of such layers. We will rewrite the sentence to clarify this:

"When solving Eq. (9)-(11) for layer $s_{NS}$, we assume the terms $\dot{Q}_{sj,sj+1}$, $\dot{H}_{sj,sj+1}$ and $\dot{W}_{gNG,gNG+1}$ to be all zero, as layer $N_{S+1}$ does not exist."

*IB Comment 20: Page 17, line 3 or so "that changes we obtain" could be "then we obtain"*

> Response: As suggested, we will replace "that changes we obtain" with "then we obtain."

*IB Comment 21: Page 21, line 3: "because the enthalpy" could be "due to the enthalpy"*

> Response: As suggested, we will replace "because the enthalpy" with "due to the enthalpy."

*IB Comment 22: Page 28, line 22: "surface x is at temperature T with a liquid"*

> Response: As suggested, we will replace "and a liquid" with "with a liquid."

*IB Comment 23: Page 28, line 5 (bottom of page): "and Leuning" could be "and the Leuning"*

> Response: As suggested, we will insert the missing word "the" before "Leuning."

*IB Comment 24: Page 31, lines 9-15: This temperature restriction is similar to what we've used in SiB for years. We also have a frost 'delay' term, where plants do not rebound immediately to photosynthesize during periods where temperatures may go below freezing (think spring in higher latitudes). I'd be happy to share it with you. Also, we have a humidity restriction term.*

> Response: We thank the Reviewer for kindly offering to share the parameterization. We will consider implementing this parameterization in future versions of ED-2.

*IB Comment 25: Page 33, water extraction by roots: OK, so plants can extract water from all layers "to which they have access" (which, in a 3-layer soil I imagine is all of them), but roots have a uniform mass distribution. I think I might know why this is done. In the real world, I would expect a shorter/younger cohort to be less deeply rooted than an older/taller cohort, and grasses to be shallower rooted still. I also imagine that when this was done in ED, the*

*short/young/grass cohorts might have died due to lack of water because the old/tall trees took it all. This is fine, but you can't have it both ways. In Section 6.2, "Heterogeneity of ecosystems" the authors claim ED 2.2 "...improves the characterization of heterogeneity...by the number of individuals, their height and rooting depth, and their traits and trade-offs that determine their ability to extract soil moisture..." which contradicts what is described in Section 4.6. These stories need to be made consistent.*

> Response: We first must clarify that ED-2.2 does not necessarily have three soil layers: the user can specify how many layers to use and the thickness of each layer. This confusion may have arisen from Figure 2, which depicted three layers. In light of the Reviewer's comment, we will modify the diagram to show a generic number of layers. Not all cohorts have access to all layers: access is determined by the root allometry. We agree that Sections 4.6 and 6.2 should be consistent and we believe that the ambiguity will be resolved by redrawing Figure 2, and rephrasing the text in the beginning of section 3.2.1 to the following:

"In ED-2.2, the soil characteristics (number of soil layers, thickness of each soil layer and total soil depth, soil texture, soil color) are defined by the user, and assumed constant throughout the simulation."

A few additional factors must be accounted for in the below-ground competition. (1) Even though the biomass of each cohort is uniformly distributed, the amount of water extracted by each cohort from each layer is proportional to the available water in each layer (i.e., if the shallower soils are drier, deep-rooted plants will remove most of the water from the wettest soil layers. This is described in Eq.s 95–96. (2) Because ED-2.2 represents cohorts with different sizes in the same patch, when large and small trees coexist, the water demand of small cohorts is typically reduced, because these cohorts are light-limited. (3) Small cohorts have generally lower carbon demand because the biomass of their living tissues, and consequently their maintenance costs, is smaller than the live biomass of large trees. Therefore, it is not necessarily true that the higher soil moisture stress of small cohorts will lead to their extinction. In ED-2.2, the success or failure of plants to survive droughts is governed by tradeoff between soil moisture stress, size-specific carbon and water demands, and maintenance costs under drought scenarios. These interactions have been previously explored in Longo et al. (2018). The results indicated increases in the abundance of small-sized plants under more frequent and extended droughts, as high-mortality of large, water-demanding trees improved light conditions in the understory.

*IB Comment 26:* *Page 37: "nonexistent"*

> Response: As suggested, we will replace "inexistent" with "non-existent."

*IB Comment 27:* *Page 37: "stand-level" is not defined in the paper. Does this mean polygon, site, or something else? Also, I'm not sure the significance of the paragraph comparing stand variability to patch variability. What does it mean?*

> Response: The notation was indeed confusing, and we will remove "stand" from the paragraph, and it replace with "polygon" to be consistent with the rest of the narrative. The goal of the analysis in this section was to compare and quantify the impact of structural variability of complex ecosystems such as forests with the climate variability.

Climate variability is represented by most dynamic global vegetation models, but structural variability cannot be properly represented by big-leaf models. The goal here was to show that this variability is relatively important. We will re-write the paragraph to clarify these points:

"The impacts of simulating structurally and functionally diverse ecosystems are also observed in the fluxes of energy, water, carbon, and momentum. For example, in Fig. 9 we show the monthly average fluxes from the last 40 years of simulation at GYF, along with the inter-annual variability of the fluxes aggregated to the polygon-level (hereafter *polygon variability*, error bars) and the inter-annual variability of the fluxes of the different patches within polygons (hereafter *patch variability*, colors in the background). The polygon-level variability can be thought as the variability attributable exclusively to climate variability, whereas the patch variability also incorporates the impact of the structural and compositional heterogeneity on the degree of variability. Most highly aggregated ("big-leaf") models characterize the polygon-level variability, but not the patch variability. However, in all cases, the patch variability far exceeded the polygon variability, indicating that structural and compositional variability is as important as the inter-annual variability in complex ecosystems. In the case of sensible heat, the polygon variability was between 39 and 64% of the patch variability (Fig. 9a). The polygon-to-patch variability ratio was similar for both friction velocity (19 − 39%) and water fluxes (17 − 44%) (Fig. 9b,c). In the case of gross primary productivity, the relevance of patch variability was even higher, with polygon-to-patch variability ratio ranging from 3.7% during the dry season to 17% during the wet season (Fig. 9d). Importantly, the broader range of fluxes across patches in the site can be entirely attributed to structural and functional diversity, because all patches were driven by the same meteorological forcing."

*IB Comment 28: Page 38: "density in the canopy air space"*

> Response: As suggested, we will replace "density at the canopy air space" with "density in the canopy air space"

*IB Comment 29: Page 39: SiB has had a prognostic CAS since 2003 (Baker et al), based on Vidale and Stockli (2005). Just sayin'.*

> Response: Thank you for the additional references, we will re-phrase the sentence to include SiB as another model that represents canopy air space:

"Unlike most existing terrestrial biosphere models (but see SiB2, e.g. Baker et al., 2003; Vidale and Stöckli, 2005), in ED-2.2 we explicitly include the dynamic storage of energy, water, and carbon dioxide in the canopy air space."

*IB Comment 30: Page 40, line 12: "access to and competition for"*

> Response: We will modify the text as suggested.

*IB Comment 31: Page 40, lines 29-30: "is fundamental to explaining"*

> Response: We will modify the text as suggested.

*IB Comment 32: Page 41, lines ?: "degradation is pervasive"*

> > Response: As suggested, we will correct the sentence to "degradation is pervasive."

*IB Comment 33: Page 42, line 29: "has excellent conservation"*

> > Response: We will replace this sentence with "has a high degree of conservation."

*On to the Supplements! (I do not have specific line numbers in my supplements file; I'll just have to do my best with explaining where the comments address)*

*IB Comment 34: Table S2: might help to add bulk specific enthalpy, and Temporary Surface Water.*

> > Response: As suggested, we will add bulk specific enthalpy and temporary surface water to Table S2.

*IB Comment 35: S2: What is a "leaf elongation factor", and how is it determined? There is a long equation to describe slk, but we aren't told what it means.*

> > Response: We will modify the first paragraph of the Leaf phenology appendix (formerly S2, now S3) to clarify the meaning of leaf elongation factor:

> "The phenological strategy of the plant functional types, can be evergreen, drought-deciduous, or cold-deciduous. The plant's phenology strategy is defined by two functions: (i) the leaf elongation factor ($\hat{e}_{lk}$), defined as the ratio between the environmentally-constrained leaf biomass and the potential (maximum) leaf biomass, and the rate of leaf shedding ($\omega_{lk}(t)$) which either be prognosed, or prescribed from observations."

> We will also include a description of $s_{lk}$ beneath Eq.s S4 and S5 (please see our response to comment 12 above).

*IB Comment 36: S7: is the 'b' term the Clapp and Hornberger b? If not C+H, where does the value come from?*

> > Response: Yes, the Reviewer is correct. We will include the citation to Clapp and Hornberger (1978) when $b$ is first defined:

> "... $b$ is the slope of the logarithmic water retention curve (Clapp and Hornberger, 1978)..."

> We will also include the information that the parameterization of soil matric potential is derived from Clapp and Hornberger (1978):

> "The equation that describes soil matric potential as a function of soil moisture is taken from Clapp and Hornberger (1978); soil hydraulic conductivity..."

*IB Comment 37: S7, field capacity: I've seen several definitions for determining field capacity from things like moisture potential. Is there a reference for what is being used here?*

> Response: We used the definition by Romano and Santini (2002) and references therein. We will include the reference to Romano and Santini (2002) and the rationale in the revised manuscript.

"Field capacity ($\vartheta_{Fc}$) is often defined from soil matric potential (e.g. Hodnett and Tomasella, 2002; Saxton and Rawls, 2006). However, this definition is based on field measurements and the definition of $\vartheta_{Fc}$ from soil matric potential can substantially across studies, with values ranging from −0.1 kPa to −0.5 kPa (Romano and Santini, 2002). In ED-2.2, we follow Romano and Santini (2002) and define field capacity in terms of hydraulic conductivity, and assume that the drainage flux of water becomes negligible at hydraulic conductivity of 0.1 kgW m$^{-2}$ day$^{-1}$."

*IB Comment 38:* S9: "contains contributions from reflectance and transmittance"

> Response: As suggested, we replaced "contribution" with "contributions." We did not replace "contain" with "contains" as we are referring to "both the bulk diffuse backscattering ($\beta_{mk}$) and forward scattering ($1-\beta_{mk}$)."

*IB Comment 39:* S12.1: Are you really able to avoid the 'material surface at the top of the CAS' problem under stable conditions? This has been a problem for years, and may be worth a publication of its own. If you've already written it, advertise it here.

> Response: We did not develop any new parameterization that avoids the flux underestimation under stable conditions. Instead, we implemented the Beljaars and Holtslag's (1991) empirical formulation of the flux profile functions. Beljaars and Holtslag (1991) found that their empirical model resulted in more mixing under stable conditions, so this is their result, not ours. We rewrote the paragraph to clarify these points:

"The ED-2.2 model uses the empirical parameterization of the originally developed by Beljaars and Holtslag (1991). For the unstable cases, Beljaars and Holtslag (1991) used the Businger-Dyer flux profile equations (Businger et al., 1971). For the stable cases, Beljaars and Holtslag (1991) implemented an empirical formulation that improved the vertical mixing between the canopy air space and the air above under stable conditions:"

*IB Comment 40:* S15: soil moisture limitation on photosynthesis. There's been a lot of work done on this with regard to the fact that individual plants maintain photosynthesis as soil dries down from wilt point, until suddenly closing stomates (Colello et al., 1998; Kim et al., 2010). This behavior, while well-known on the plant scale, is problematic when imposed on the ecosystem scale, as it frequently results in binary, or 'on-off' behavior. Many methods have been utilized to deal with it (e.g. Laio et al., 2001; Porporato et al., 2001, 2002: Rodriguez-Iturbe 2000; Baker et al., 2008, 2013; Wood et al., 1992, to name just a few ). I'd like to see more explanation of what you're doing. A graph showing how stress is imposed, from field capacity to wilt point, would be helpful. Is stress imposed in a linear fashion, or does it behave like the btran function in CLM? Is this function based on previous research (which should be cited), or something incorporated specially for ED2.2? If so, why?

> Response: The formulation of the soil moisture stress is mostly based on previous versions of ED (Moorcroft et al., 2001). The only difference in ED-2.2 is that we re-defined the soil water term to be a function of soil matric potential, similar to CLM

(Oleson et al., 2013), because the response of the wilting factor to drying the new formulation is slightly more gradual than the original formulation. We will modify the text to include these citations:

"To account for soil water stress, we define a phenomenological scaling function $f_{Wlk}$ (wilting factor). The functional form of $f_{Wlk}$ follows the previous versions of ED (Moorcroft et al., 2001; Medvigy et al., 2009). However, in ED-2.2 we define water availability ($W_g{}^*$) in terms of soil matric potential, similarly to CLM (Oleson et al., 2013), which produces a more gradual transition from no-stress conditions to completely closed stomata as soil moisture approaches the wilting point (Fig. S9)."

We will also include a new Supplemental Figure that shows the response of the soil moisture stress using both the original ED formulation and ED-2.2, which highlights the less steep transition from no stress to extreme stress when soils are dry. The proposed figure and caption are shown below:

[Figure]

"Figure S9: Example of the wilting factor ($f_{Wlk}$, Eq. S192) response to soil moisture change for the original implementation in ED (ED-1.0 and ED-2.0, Moorcroft et al., 2001; Medvigy et al., 2009) and the ED-2.2 model approach. Results here are shown for the idealized case with constant soil moisture profile in a 3-m deep, sandy clay loam soil, for a mid-successional tropical cohort with default parameters (Table S5 with diameter at breast height of 30 cm and leaf area index of 1 $m^2_{Leaf}$ $m^{-2}$, non-limited leaf-level transpiration rate $\dot{E}_k = 9.0$ $kg_W$ $m_{Leaf}^{-2}$ $day^{-1}$. Values are shown for soil moisture columns ranging from wilting point ($\vartheta_{Wp};\Psi_{Wp}$) to field capacity ($\vartheta_{Fc};\Psi_{Fc}$)."

We should also clarify that the soil moisture stress is calculated for each cohort and each patch, and as explained before, plants with different sizes and PFT have different rooting depths, and may have different water use strategies, expressed through different stomatal

conductance parameters. The predicted response to soil water depletion in ED-2.2 does not show the "on-off" response. For example, in Figure S14 of Longo et al. (2018), we found that the soil moisture stress factor $(1-f_{Wlk})$ would increase more quickly for small cohorts (DBH < 10cm) than larger cohorts (DBH > 35cm) under recurrent fire regimes. However, that did not cause stomatal conductance to decrease for the small cohorts because once large trees started to die, the small trees experienced more light that compensated the additional soil moisture stress.

The diversity of cohort responses to droughts are further enhanced when using TOPMODEL (one of our current developments, as explained in Section 6.3). This approach accounts for both edaphic heterogeneity and lateral moisture transfer, and can represent that mesic lowlands do not dry as quickly as ridges, for example. Finally, please note that a version of ED-2 has been developed that mechanistically solves the plant hydraulics (Xu et al., 2016). This is currently not integrated with the ED-2.2 described here; however, work is underway to include this new formulation into the main distribution of the model on GitHub.

*Figures*

*IB Comment 41: Figure 1: White text was difficult for me to read. It might be worth sacrificing the pretty clouds/sky background for something more simple. Or maybe just use red lettering.*

> Response: In light of the Reviewer's feedback, we will increase the contrast between the background and the text, and replace the cloud picture to avoid white letters on white background. We will also increase the contrast in Figures 2 and 3. In addition, we will change the symbols in the regional maps (Figures 7 and S4), following a suggestion we received for Part 2.

*IB Comment 42: Figure 2: caption should say "dashed yellow arrows"*

> Response: We will correct the caption as suggested. We also noticed that the colors were incorrect in Figure 3 caption, and will correct these as well.

*IB Comment 43: Figure 3: caption should say there are 3 cohorts shown.*

> Response: In line with the changes described in our response to *IB Comment 25*, we will update Figure 3 to show a generic number ($N_T$) of cohorts.

*IB Comment 44: Nice paper, people. Good work.*

> Response: Thank you Ian for reviewing the manuscript and for your constructive thoughts and feedback.

**Responses to Dr. Stefan Olin**

*SO Comment 1: The model description did not leave out any details, which is a very good thing and it is not very common for many of the existing model description papers. The downside of that is of course that the manuscript is rather long, too long in my opinon.*

> Response: We are glad that the Reviewer thinks the model description is detailed.  Our goal was indeed to describe how the processes are actually solved within the ED-2.2 framework; we believe that providing the complete description allows ED-2.2 users to understand the rationale behind each module, and researchers using other models to understand and reproduce our methods should they decide to implement our developments in other models.  We tried to keep the main text as concise as possible, by keeping only the main description of the fluxes in the main text, and algorithms and details on other variables (specific heat, conductances, definition of state variables, among others) in the Supplements. However, because the model solves many processes at multiple time scales, the final manuscript is still very long.

*SO Comment 2: One thing I miss from the very thorough walkthrough of vegetation models in the introduction are references to the DGVMs that are closer to ED such as LPJ-GUESS (for disclosure, I am an LPJ-GUESS developer).*

> Response: Following the Reviewer's suggestion, we included a new paragraph in the introduction describing the emergence of cohort-based models and included references to some of them, including LPJ-GUESS.

"The need to represent vegetation structure in terrestrial biosphere models, without the computational burden required to simulate every tree at regional and global scales, led to the development of cohort-based models (Fisher et al., 2018). In the cohort-based approach, individual trees are grouped according to their size (e.g. height or diameter at breast height); functional groups, which can be defined along trait axes (e.g. Reich et al., 1997; Wright et al., 2004; Fortunel et al., 2012), and micro-environment conditions (e.g. whether plants are living in a gap, recently burned fragment, or in a patch of old-growth forest). Over the past two decades, many cohort-based models have emerged, including the Ecosystem Demography Model (ED, Moorcroft et al., 2001; Hurtt et al., 2002; Albani et al., 2006; Medvigy et al., 2009); the Lund-Potsdam-Jena General Ecosystem Simulator (LPJ-GUESS, Smith et al., 2001; Ahlström et al., 2012; Lindeskog et al., 2013); and the Land Model version 3 with Perfect Plasticity Approximation (LM3-PPA, Weng et al., 2015), and the Functionally-Assembled Terrestrial Ecosystem Simulator (FATES, Fisher et al., 2015; Huang et al., 2019). Because these models also represent functional diversity and heterogeneity of micro-environments, the ecosystem's structure, diversity and functioning also emerge from the interactions between plants with different life strategies under different resource availability, albeit at a lesser extent than individual-based models (Fisher et al., 2018)."

*SO Comment 3: The text is easy to read, and the refernces to equations, sections and tables are good. One comment regarding the referencing to equations is that it should be concistent, for example on line 14, page 7. In my opinon it should read: Eq. 2-3 cannot ... .*

> Response: We believe our equation references are consistent with the GMD style guidelines.  We do not have a strong preference and we will change the notation if the editor considers the reference style suggested by the Reviewer more appropriate.

*SO Comment 4: Another comment I have regarding the equations (or symbols) is the sometimes odd choice of symbols. Like Eq. 36-38, why choose the same symbol for a variable that you are using as an operator?, that is very confusing.*

> Response: We sometimes resorted on less conventional symbols because we used all letters of the Latin and Greek alphabet (lower case, upper case, and calligraphy, see Table S2). However, we agree with the reviewer that the use of $\Pi$ for plant area index in Eq. 36–37 is potentially confusing. To address this issue we will replace the plant area index symbol from $\Pi$ to $\Phi$ (upper case), and plant area density to $\phi$ (lower case), and use the symbol $\varpi$ (originally used for plant area density) for oxygenase:carboxylase rate. We tried to restrict odd symbols to variables that were only used in specific equations, to avoid distractions.

*SO Comment 5: The same goes for the use of exp instead of e, and on the note of the letter e, you are using it as pool (ej) and as a scaling factor (eH ot), I'd say that it is better to use the letter e as the mathematical constant it is, and then use some other symbol to denote your pools. And for your factors, use q or f.*

> Response: Both "*e*" and "*exp*" standard mathematical notation to refer to an exponential function.  Regarding the scaling factors, we believe the Reviewer was referring to Eq. 103 and 104.  However, we agree that using "*e*" is this context is potentially confusing, and therefore we will replace it with "*f-hat*" ($f_{Cold}$ *and* $f_{Hot}$ are already used in Eq. 85, and *q* is used for specific heat). We will also replace *e* for the elongation factor (Supplements) with *ê*. We did not use *f* in this case because it would lead to too many levels of subscripts, and no other letter was available.  Regarding the soil carbon pools, we think that the subscripts make them clearly distinct from actual variables, and we plan to keep them the same, as all other Latin letters have already been used for something else.

*SO Comment 6: On the same topic of mathematical operators as variables, in Eq. 76, maybe something went wrong, there is a definition character instead of an equal sign. And again, why use operators as super scripts, just adds confusion.*

> Response: We believe that the Reviewer was referring to Eq. 75.  We agree that the choice of the equivalent sign to indicate phase equilibrium (also known as saturation) was confusing, and that the superscript was also confusing. In light of the Reviewer's comment, we will refer to phase-equilibrium partial pressure as $p_{Sat}$ and to phase-equilibrium specific humidity as $w_{Sat}$.

*SO Comment 7: And likewise, in Eq. 56, is that an equal sign as a superscript or do you have an assignment within the equation? Or is it a pre-request? Either way, that equation is confusing.*

> Response: In light of the Reviewer's suggestion, we will split Eq. 56 into two equations to eliminate the ambiguity.

*SO Comment 8: With such an explicit formulation of the exchange of heat and water I find it rather strange that the incoming water does not have an explicit energy level specified. If 15 deg. C water lands on a surface that is 25 deg. C, there would be a cooling taking place. Maybe this is of minor importance in the Amazon, but in colder places this would matter. Or did I totally misread what is written in the beginning of Sect. 4.2, if so, I suggest you clarify this.*

> Response: Precipitation has an explicit energy level associated, and this energy input is defined in the paragraph that starts on Page 21, line 15, including equation 39. Most meteorological drivers (including eddy covariance towers) do not provide precipitation

temperature.  Therefore, we assumed it to be the same as the temperature of the air above canopy ($T_a$, Eq. 39), which is not the same as the leaf or ground temperature; therefore the model does account for the cooling effect of precipitation.  We will rewrite the paragraph to make these important points more explicit:

"Precipitation is a mass flux, but it also has an associated enthalpy flux ($\dot{H}$) that must be partitioned and incorporated to the cohorts and temporary surface water. Similar to the water exchange between soil layers, the enthalpy flux associated with rainfall uses the definition of enthalpy (Supplement S5). Because precipitation temperatures are seldom available in meteorological drivers (either from towers or gridded meteorological forcing data sets), we assume that precipitation temperature is closely associated with the free-air temperature $(T_a)$, and we use $T_a$ to determine whether the precipitation falls as rain, snow, or a mix of both. Importantly, the use of free-air temperature partly accounts for the thermal difference between precipitation temperature and the temperature of intercepted surfaces…"

In addition to the temperature difference, precipitation in the form of snow has varying density, which in turn affects the density of developing snowpack.  The dynamics of the snowpack was missing in the original manuscript, and we think it is an important process for non-tropical simulations.  To avoid extending the main manuscript, we will include the description of the snowpack depth dynamics as a new supplement, and indicate this in the text:

"In the case of liquid TSW, the layer thickness of the single layer is defined as $\Delta z = \rho_\ell^{-1} W_{s1}$, where $\rho_\ell$ is the density of liquid water (Table S3). In the case of snowpack development, the snow density and the layer thickness of the TSW are solved as described in Supplement S7. The thickness of each layer of snow ($\Delta z_{sj}$) is defined using the same algorithm as LEAF-2 (Walko et al., 2000) and described in Supplement S7."

The new Supplement that briefly describes the snowpack depth dynamics:

**"S7 Snowpack depth dynamics**

In addition to enthalpy and total water, we must also track the changes in snowpack depth of each layer ($\Delta z_{sj}$) and density ($\rho_{sj}$) over time. The ordinary differential equation that governs changes in depth over time is defined as:

$$\frac{\mathrm{d}\Delta z_{s_j}}{\mathrm{d}t} = \begin{cases} \underbrace{\rho_{wa}\dot{W}_{a,s_j}}_{\substack{\text{Throughfall} \\ \text{precipitation} \\ \text{(4.2)}}} + \underbrace{\left(\sum_{k=1}^{N_T}\rho_{wt_k}\dot{W}_{t_k,s_j}\right)}_{\substack{\text{Canopy dripping} \\ \text{from cohorts} \\ \text{(4.2)}}} - \underbrace{\rho_{ws_j}\dot{W}_{s_j,o}}_{\substack{\text{Surface runoff} \\ \text{(4.1)}}} - \underbrace{\rho_{wx}\dot{W}_{s_j,c}}_{\substack{\text{Surface water} \\ \text{evaporation} \\ \text{(4.5.2 and 4.5.3)}}} - \underbrace{\delta_{s_1 s_j}\rho_{ws_j}\dot{W}_{s_1,g_{N_G}}}_{\substack{\text{Surface water} \\ \text{percolation} \\ \text{(4.1)}}} & \text{, if } s_j = s_{N_S} \\[2em] 0 & \text{, otherwise} \end{cases}$$

$$\text{(S68)}$$

$$\rho_{ws_j} = \frac{W_{s_j}}{\Delta z_{s_j}} \tag{S69}$$

$$\rho_{wx} = \begin{cases} \rho_{ws_{N_S}} & \text{, if } \dot{W}_{s_{N_S},c} \geq 0 \\ \rho_{wc} & \text{, if } \dot{W}_{s_{N_S},c} < 0 \end{cases} \tag{S70}$$

where $\delta s_j s_{j'}$ is the Kronecker delta for comparing two TSW layers $s_j$ and $s_{j'}$ (1 if $s_j = s_{j'}$, 0 otherwise), $\rho_{wa}$ is the precipitation density, $\rho_{wtk}$ is the canopy interception density, $\rho_{wc}$ is the density of condensing water vapor. Precipitation density is defined based on Jin et al. (1999), but slightly modified to make it continuous:

$$\rho_{wa} = \frac{\rho_{ia}\rho_\ell}{\ell_a\rho_{ia} + (1-\ell_a)\rho_\ell}, \tag{S71}$$

$$\rho_{ia} = \begin{cases} 169.16 & \text{, if } T_a > 275.16 \text{ K} \\ 50. + 1.7\,(T_a - 258.16)^{1.5} & \text{, if } 258.16 \text{ K} < T_a \leq 275.66 \text{ K} \\ 50. & \text{, if } T_a \leq 258.16 \text{ K} \end{cases} \tag{S72}$$

where $\rho_\ell$ is the density of liquid water (Table S3). For the canopy dripping flux, water density is similar to Eq. (S71), except that we assume the density of frozen water to be the same as frost density ($\rho_*$, Table S3). A similar assumption is done for water condensing from canopy air space, with the additional assumption that the liquid fraction of condensation is the same as the liquid fraction of the top TSW layer:

The maximum allowed number of snow layers is determined by the user, but the actual number of snow layers is dynamically determined, following the same algorithm as Walko et al. (2000). Multiple layers only exist when ice is present, otherwise a single layer ($N_S = 1$) is enforced. When ice is present, the model selects NS to be the maximum number of layers that satisfies

$$\rho_{wt_k} = \frac{\rho_* \rho_\ell}{\ell_{t_k} \rho_* + \left(1 - \ell_{t_k}\right) \rho_\ell}, \tag{S73}$$

$$\rho_{wc} = \frac{\rho_* \rho_\ell}{\ell_{s_{N_S}} \rho_* + \left(1 - \ell_{s_{N_S}}\right) \rho_\ell}. \tag{S74}$$

$W_{sj} \geq 5$ kg$_W$ m$^{-2}$ for all layers $s_j$, $j \in 1,2,...,N_S$, to ensure numerical stability. The layer thickness distribution ($\Delta z_{sj}$) for any given $N_S$ is defined as:

$$\Delta z_{s_j} = z_s \frac{2^{\min(j-1,N_S-j)}}{2^{\lfloor \frac{N_S+1}{2} \rfloor} + 2^{\lfloor \frac{N_S}{2} \rfloor} - 2}, \tag{S75}$$

$$z_s = \sum_{j=1}^{N_S} \Delta z_{s_j}, \tag{S76}$$

where $z_s$ is the total depth of the snow, and $\lfloor x \rfloor$ is the floor function (i.e. the nearest integer value to $x$ that is not greater than $x$). The layer distribution described by Eq. (S75) ensures that the layers near the ground and near the canopy air space are thinner than the intermediate layers, to improve the representation of exchanges between the snowpack and the canopy air space, soils, and incoming irradiance (Walko et al., 2000)."

*SO Comment 9:* In the first paragraph of the discussion you are writing that you have demonstrated a functional diverse canopy, from the supplements I get that you have three PFTs along one functional trait axis.

> Response: The reviewer is correct that there are three default tree PFTs for tropical South America, and for the purposes of this manuscript we only used these three tree PFTs. However, in ED-2.2 model users can specify additional PFTs, or modify the existing PFTs, using XML files that are read during the initialization, and several published ED2 studies have modified or added tropical PFTs according to the scientific questions (e.g. Xu et al. 2016, Trugman et al. 2016, Feng et al. 2018).

This information was missing in our manuscript, and we will include the following paragraph in Section 2.3 (Model inputs):

"*Plant functional types.* The user must specify which plant functional types (PFTs) are allowed to occur in any given simulation. ED-2.2 has a list of default PFTs, with parameters described in Tables S5-S6. Alternatively, the user can modify the parameters of existing PFTs or define new PFTs through an extensible markup language (XML) file, which is read during the model initialization."

We will also update the discussion (Section 6.2) and point readers to these previous studies that have used non-default PFTs in the tropics:

"In this manuscript, we presented the functional diversity using only the default tropical plant functional types (PFTs), which describe the functional diversity along a single

functional trait axis of broadleaf tropical trees. However, the ED-2.2 framework allows users to easily modify the traits and trade-offs of existing PFTs, or include new functional groups; previous studies using ED-2.2 have leveraged this feature of the code to define PFTs according to the research question both in the tropics (e.g. Xu et al., 2016; Trugman et al., 2018; Feng et al., 2018) and in the extra-tropics (e.g. Raczka et al., 2018; Bogan et al., 2019)."

*SO Comment 10: Results are not really discussed nor shown, but one result that there is much focus on is the closed energy budget. Is it really closed if there is a 0.01 deviation? Is there not a great risk of error propagation if the bar is set that low? In LPJ-GUESS we are concerned if the mass balance is off by $10^{-12}$.*

> Response: We agree with the reviewer that we should impose stricter tolerances by default, and we will modify the default to be $10^{-5}$ and update the code available in the permanent repository. We cannot impose a tolerance as strict as LPJ-GUESS: to reduce the size of the output files, variables in ED-2.2 are stored as single precision (truncation error of the order of $10^{-7}$), even though the biophysics solver uses double precision. We tested the model with the new tolerance, and the model ran without problems. We will revise the paragraph to include the updated information:

"The ED-2.2 simulations show a high degree of conservation of the total energy, water, and carbon (Fig. 6). In the example simulation for one patch at Paracou, French Guiana (GYF), a tropical forest site, the accumulated deviation from perfect closure (residual) of the energy budget over 50 years (2,629,800 time steps) was 0.1% of the total enthalpy storage — sum of enthalpy stored at the canopy air space, cohorts, temporary surface water and soil layers (Fig. 6a), and 0.002% of the accumulated losses through eddy flux, the largest cumulative flux of enthalpy. Results for the water budget were even better, with maximum accumulated residuals of 0.04% of the total water stored in the ED-2.2 thermodynamic systems, or 0.0006% of the total water input by precipitation (Fig. 6b), and the accumulated residual of carbon was 0.008% of the total carbon storage or 0.017% the total accumulated loss through eddy flux. The average absolute residual errors by time step, relative to the total storage, ranged from $3.6 \cdot 10^{-11}$ (carbon) to $3.8 \cdot 10^{-10}$ (energy), and thus orders of magnitude less than the truncation error of single-precision numbers ($1.2 \cdot 10^{-7}$) and the model tolerance for each time step ($1.2 \cdot 10^{-5}$)."

We had already included text in the discussion proposing areas of improvement of closure, but we will insert the following sentence acknowledging that the closure should be improved before the suggestions for improvement:

"Nonetheless, the residual errors in ED-2.2 are larger than the error of each time steps after integrating the model over multiple decades (Fig. 6), which suggests that the errors may have a systematic component that deserves further investigation."

Regarding the absence of results, we only showed model verification and some examples of simulations in this paper, in line with the GMD guidelines for model description manuscript. The companion manuscript (Longo et al., 2019), also in review for GMD, has extensive model evaluation for tropical South America.

*SO Comment 11: Some specific comments in addition to those spotted by Ian Baker: Line 3, page 41: remove the 'a'.*

> Response: As suggested, we will remove the "a".

*SO Comment 12: Line 3, page 21: intercepted instead of intercept.*

> Response: As suggested, we will replace "intercept" with "intercepted."

*SO Comment 13: Line 1, page 33: What is a decay rate due to respiration? Do you mean turnover?*

> Response: The original sentence was incorrect. This section is describing metabolic fine-root respiration, not turnover. We will replace the sentence with the following:

[revised manuscript text omitted]

---

## Author Response (AR1)

Dear Dr. Müller,

Thank you, the reviewers Dr. Baker and Dr. Olin for the in-depth and constructive feedback on our first version of the manuscript "The biophysics, ecology, and biogeochemistry of functionally diverse, vertically- and horizontally-heterogeneous ecosystems: the Ecosystem Demography Model, version 2.2 — Part 1: Model description" (gmd-2019-45). We also thank Dr. Kim for the iterative comment and suggestion to include the differences between the ED-2 versions in this manuscript.

Below we include the detailed, point-by-point response to each comment and question raised by each referee. In our responses, line and page numbers correspond to the enclosed annotated manuscript and annotated supporting information. In the annotated files, blue text was added, and  was removed.

Sincerely,
Marcos Longo.

**Responses to Dr. Ian Baker**

*IB Comment 01: Larger Impressions. This paper describes the code used in ED2.2. That's pretty much it. There isn't really any 'new science' here, and in fact any text not devoted to explaining equations is just showing that the model gives reasonable results and conserves energy (there are a few paragraphs showing differences in size/age distributions for two tropical sites with different disturbance regimes). It can be hard to get a paper like this through review, but the authors are lucky to have me for a reviewer. I know the value of papers like this (e.g. the BATS NCAR Technical Manual by Dickinson et al., the LSM Technical Note by Bonan, and the Sellers SiB papers from 1986 and 1996). However, the people reviewing the methods paper want to see results, and vice versa, and they want the paper to be short. But these 'code papers' have value for the people who use models, and I appreciate that because I am one of them.*

> Response: We are glad that the Reviewer appreciates the value of this model description manuscript. We believe that the ED-2 model needs a detailed technical description The GMD *model description paper* type expects: "detailed, complete, rigorous, and accessible" descriptions of the model, and to present examples of model output and model verification. We are glad that the Reviewer identified all these elements in this manuscript.

*IB Comment 02: I understand there is no way to combine the Parts 1 and 2 of the ED2.2 paper. This weighty tome already comes in at over 100 pages (paper plus supplements for Part 1), yet it is critical to make the code information available for people who will use the model. Some might suggest a technical manual (like is done for CLM), and the authors have in fact done this; I took a quick look at the wiki, and I think it has very useful information for users, but it doesn't really lay out the rational for the code. Also, the authors want to get journal citations and credit for the work they've done, and I don't blame them one bit.*

> Response: We are glad that the Reviewer agrees in the value of a manuscript describing the model in detail. The online wiki explains how to run the ED-2.2 model

whereas, as the reviewer correctly pointed out, this manuscript describes the rationale and the parameterizations needed to solve the energy, water, and carbon cycles within the ED-2.2 framework.

*IB Comment 03: I'm not going to download the code and study it line by line to see if the explanations make sense. There is no way I could do that and get a review back in under a year. Therefore, it is incumbent upon the authors to very carefully go over the manuscript and check for typos in the equations as they appear in the paper.*

> Response: As per the Reviewer's suggestion, we are revising the paper and looked for typos. We identified some typos (Eq. 4, Eq. S5, Eq. 59, Eq. S87), and will continue to revise the equations to ensure they are correct.

*IB Comment 04: Initially, I thought that perhaps I was the wrong person to review this paper. I have years of experience with SiB and CLM, but none with ED. But then I realized that makes me the perfect person to review; someone familiar with ED will already know much of the material. But if I can understand how ED2.2 works after reading the paper, then the authors have done their job. And I think they've succeeded. I feel fairly comfortable, for the most part, about the ED2.2 framework after reading the paper (multiple times). This paper will be useful for researchers learning or developing ED, and other models, in the future.*

> Response: We thank the Reviewer for thinking that the paper was useful to understand the concepts of ED-2.2. We sought to make the manuscript accessible to a broad community that may or may not be familiar with the ED-2.2 model.

*IB Comment 05: My formal recommendation is to accept the paper for publication, with minor revisions. I don't need to see it again.*

> Response: We thank the Reviewer for the encouraging recommendation.

*IB Comment 06: I really like the use of enthalpy; that is an innovative way to demonstrate conservation of energy, and I'm not sure it has been used before.*

Response: We also prefer the equations to be based on energy (or enthalpy) as the response variable, as this simplifies the tracking of energy and helps to enforce energy/enthalpy conservation. We too are not aware of energy/enthalpy being used as the energy-related state variable, and thus represents a novel feature ED-2.2 model.

*IB Comment 07: I'd like to see more emphasis on what is new in ED2.2 (final paragraph in the Introduction). A bullet list might draw the reader's eye to the new features in this version of the model.*

> Response: This suggestion is also in line with Dr. John Kim's iterative comment. We modified the next-to-last paragraph of the introduction to briefly describe and itemize the main changes between ED-2.0 and ED-2.2 (p. 5, l. 23–29 of the annotated manuscript).

In addition, we included a new Supplement (S1) that will describe the timeline of changes between ED-2.0 and ED-2.2, based on our previous reply to John Kim (https://doi.org/10.5194/gmd-2019-45-AC1) (p. S26–S28 of the annotated supplement).

*Specific Comments*

*(many of these are suggestions for grammar, and in some cases need not be implemented exactly as I suggest. They are just places where I noted typos and grammar issues. I also apologize for location indicators; in my copy there were new line numbers on each page, and after about page 26 I found a line numbered 5 at the bottom of the page sometimes.)*

*IB Comment 08: Abstract, line 11: "out and is presented"*

> > Response: We modified the text as suggested  (p. 1, l. 11 of the annotated manuscript).

*IB Comment 09: Page 2, lines 5-15: This description of generational advances in model development does not align exactly with Sellers et al. (1997). I think it would be helpful to acknowledge the Sellers paper and put the descriptions here in that context.*

> > Response: As suggested, we cited Sellers et al. (1997) and revised our narrative to be consistent with Sellers et al. (1997) (p. 2, l. 13–23 of the annotated manuscript).

*IB Comment 10: Page 3, line 7: SiB does not have an explicitly layered canopy or sunlit/shaded leaves separately treated.*

> > Response: As suggested, we removed the reference to SiB  (p. 2, l. 17–18 of the annotated manuscript).

*IB Comment 11: Page 3, lines 12-13: I'm confused here. I thought models were transitioning from broadly-defined 'biomes' to a PFT-based mosaic structure. This sentence says the opposite.*

> > Response: We intended to say that the lack of functional diversity and the mechanistic representation of ecological processes such as competition cause models to predict ecosystems comprised of single homogeneous vegetation types. We rewrote the sentence to clarify this point (p. 3, l. 23–24 of the annotated manuscript).

*IB Comment 12: Page 6: The full set of PFTs is not listed. In table S5 we're shown parameter values for the tropical grasses and trees used here, but if this paper is going to be the 'go to' manual for ED2.2, all PFTs should be listed in a table somewhere. Don't worry about the extra length-this paper is already incredibly long.*

> > Response: As suggested by the reviewer, we will include the parameter values for all the default PFTs. Specifically, we added a separate table for temperate PFTs  (New Table S6), revised the Leaf phenology supplement  (New Supplement subsection S3.1.3) to include descriptions of the cold-deciduous leaf phenology, and updated Supplement (Allometric equations) (formerly S16, now S18) to include non-tropical PFTs (p. S69–S72 of the annotated supplement).

*IB Comment 13: Page 7, line 15: I'd like to see the index k introduced here. I had to wade through a bit of text in the supplements before I realized that k addressed cohorts (this might also have to do with the fact that I had a hard time seeing k in the lettering in Figure 1. It might be helpful to have a small table showing the indexes used to address sites, patches, and cohorts. By the time I had read the paper several times I think I had it figured out, but a more explicit explanation might be helpful.*

> Response: As suggested, we included a new Table 1 with indices associated with patches, cohorts, disturbance types and PFTs (p. 8 of the annotated manuscript). The new Table 1 caption also refers to existing Table S1 that lists all the subscripts used in the manuscript. We also revised the text to describe the subscripts, and referred to the new Tables 1 and S1 (p. 6, l. 29; p. 7, l. 5; p. 8, l. 15; p.9, l. 1 of the annotated manuscript).

*IB Comment 14: Page 9, lines 31-32: How do you specify CO2 mole fraction on the timescale of the model? I'm not aware of CarbonTracker or GlobalView products that give that kind of resolution, and products with temporal averaging will cause issues with your carbon exchange during diurnal cycles (I think Jih-Wang Wang et al.,2007, talks about this). I don't see any mention of CO2 drivers in the wiki either. We've always calculated atmosphere-CAS CO2 exchange using a constant atmospheric value, and the flux can be easily scaled during a mesoscale- GCM- or transport-model application when a time- varying atmospheric CO2 value is available in the lowest atmospheric level. This may be a recommendation more appropriate for the github wiki, but I think the authors need to explain to the user how to deal with it.*

> Response: Above-canopy $CO_2$ is read-in as part of the meteorological drivers read routines. It therefore must be provided by the user, along with other meteorological variables. Further details on the initialization procedure can be found on ED2 site GitHub (https://github.com/EDmodel/ED2/wiki/Drivers). Ideally, atmospheric $CO_2$ should be provided at comparable temporal and spatial resolution to other meteorological variables, to avoid the issues raised by the reviewer. In the case of coupled ED2-BRAMS simulations (e.g. Knox et al. 2015; Swann et al. 2015), this happens automatically since $CO_2$ is explicitly tracked as an atmospheric state variable whose value at the lower boundary value is passed into ED2 thereby eliminating the need to scale the fluxes.

In offline simulations, $CO_2$ data are not generally available at comparable spatial scale or similar temporal resolution as other meteorological drivers. For these cases, ED-2.2 allows the user to provide time-varying $CO_2$ that is constant in space (this option is available to all meteorological drivers, as ED-2.2 can be driven by micrometeorological tower data). Alternatively, the user can set a constant $CO_2$ mixing ratio when no $CO_2$ data are available.

In line with the Reviewer's suggestion, we added text clarifying that ED-2.2 can read $CO_2$ data when available. We also explained that ED-2.2 can read data from a single site (eddy flux tower) and gridded drivers (reanalysis) (p. 11, l. 14–18 of the annotated manuscript).

*IB Comment 15: Page 10, lines 12-13: "aboveground part each cohort" Huh? I think there is some re- wording needed here.*

> Response: As suggested, we replaced this text with "leaves and branchwood portion of each cohort", to clarify the meaning of the sentence (p. 12, l. 3–4 of the annotated manuscript).

*IB Comment 16: Page 11, line 11: "components on the right-hand side"*

> Response: As suggested, we corrected this typographical error (p. 13, l. 1 of the annotated manuscript).

*IB Comment 17: Page 15, line 1: I'm not sure I understand exactly what j-prime means. I think I know, as in there is no sub-surface runoff from any soil level above the bottom one, and no ground evaporation from layers below the surface, but this is not made explicit to the reader.*

> Response: We used $j'$ to refer to two different soil layers ($g_j$ and $g_{j'}$). For example, in the third term on the right-hand side of Eq. (8) ($\delta_{gjgj'}$) is 0 for all layers except layer $g_1$. We did not want to add yet another index because it could be even more confusing and therefore plan to keep the $j'$ notation; however, in light of the Reviewer's suggestion, we modified the text to clarify the meaning of $j'$ (p. 15, l. 21 of the annotated manuscript). We made a similar modification to the text after Eq. (11) (p. 17, l. 16 of the annotated manuscript).

*IB Comment 18: Page 10, line 3: I'm not sure that holding energy, enthalpy, and water fluxes to zero is consistent with the explanation given on page 9, lines 24-26. If free drainage is allowed out of the bottom of the soil column, won't W-dot g0,g1 be nonzero? This needs to be made more clear.*

> Response: We believe the Reviewer was referring to Page 15, line 3. In the original submission, we had defined sub-surface runoff (drainage) as a separate term (H-dot$_{g1,o}$ ; W-dot$_{g1,o}$), and thus we defined H-dot$_{g0,g1}$ and W-dot$_{g0,g1}$ to be zero in section 3.2.1. However, we agree with the Reviewer that this is confusing and also is inconsistent with the text in section 4.1. To remove the ambiguity, we will refer to sub-surface runoff fluxes as H-dot$_{g0,g1}$ and W-dot$_{g1,g0}$, and use subscript $o$ exclusively for surface runoff. Specifically, we made the following three changes:

(i) Re-define the flux notation at the beginning of section 3.2 (p. 14, l. 11 – p. 15, l. 2 of the annotated manuscript).

(ii) Replace the notation $g_1,o$ with $g_1,g_0$ in Table 5 (formerly Table 4), and in section 4.1 (Hydrology sub-model and ground energy exchange) (p. 22, l. 22 of the annotated manuscript, Eq. 29 and Eq. 33).

(iii) Remove the sub-surface runoff term from equations (6) and (7), and describe that the terms H-dot$_{g0,g1}$ and W-dot$_{g0,g1}$ are the (negative) sub-surface runoff fluxes (p. 16, l. 1 – p. 17, l. 3 of the annotated manuscript).

*IB Comment 19: Page 16, lines 1-2: If layer Ns+1 does not exist, why mention it at all? Does it exist in the code as a placeholder? If so, that should be stated.*

> Response: We made the comment because the second term on the right-hand sides of Equations (9–11) for layer $s_j = s_{NS}$ implies the existence of such layers. We rewrote the sentence to clarify this (p. 17, l. 20–22 of the annotated manuscript).

*IB Comment 20: Page 17, line 3 or so "that changes we obtain" could be "then we obtain"*

> Response: As suggested, we replaced "that changes we obtain" with "then we obtain" (p. 19, l. 7–8 of the annotated manuscript).

*IB Comment 21: Page 21, line 3: "because the enthalpy" could be "due to the enthalpy"*

> Response: As suggested, we replaced "because the enthalpy" with "due to the enthalpy" (p. 23, l. 22 of the annotated manuscript).

*IB Comment 22: Page 28, line 22: "surface x is at temperature T with a liquid"*

> Response: As suggested, we replaced "and a liquid" with "with a liquid" (p. 32, l. 4 of the annotated manuscript).

*IB Comment 23: Page 28, line 5 (bottom of page): "and Leuning" could be "and the Leuning"*

> Response: As suggested, we inserted the missing word "the" before "Leuning" (p. 32, l. 13 of the annotated manuscript).

*IB Comment 24: Page 31, lines 9-15: This temperature restriction is similar to what we've used in SiB for years. We also have a frost 'delay' term, where plants do not rebound immediately to photosynthesize during periods where temperatures may go below freezing (think spring in higher latitudes). I'd be happy to share it with you. Also, we have a humidity restriction term.*

> Response: We thank the Reviewer for kindly offering to share the parameterization. We will consider implementing this parameterization in future versions of ED-2.

*IB Comment 25: Page 33, water extraction by roots: OK, so plants can extract water from all layers "to which they have access" (which, in a 3-layer soil I imagine is all of them), but roots have a uniform mass distribution. I think I might know why this is done. In the real world, I would expect a shorter/younger cohort to be less deeply rooted than an older/taller cohort, and grasses to be shallower rooted still. I also imagine that when this was done in ED, the short/young/grass cohorts might have died due to lack of water because the old/tall trees took it all. This is fine, but you can't have it both ways. In Section 6.2, "Heterogeneity of ecosystems" the authors claim ED 2.2 "...improves the characterization of heterogeneity...by the number of individuals, their height and rooting depth, and their traits and trade-offs that determine their ability to extract soil moisture..." which contradicts what is described in Section 4.6. These stories need to be made consistent.*

> Response: We first must clarify that ED-2.2 does not necessarily have three soil layers: the user can specify how many layers to use and the thickness of each layer. This confusion may have arisen from Figure 2, which depicted three layers. In light of the Reviewer's comment, we will modify the diagram to show a generic number of layers. Not all cohorts have access to all layers: access is determined by the root allometry. We agree that Sections 4.6 and 6.2 should be consistent and we believe that the ambiguity will be resolved by redrawing Figure 2, and rephrasing the text in the beginning of section 3.2.1 (p. 15, l. 8–9 of the annotated manuscript).

A few additional factors must be accounted for in the below-ground competition. (1) Even though the biomass of each cohort is uniformly distributed, the amount of water extracted by each cohort from each layer is proportional to the available water in each layer (i.e., if the shallower soils are drier, deep-rooted plants will remove most of the water from the wettest soil layers. This is described in Eq. 95–96. (2) Because ED-2.2 represents cohorts with different sizes in the same patch, when large and small trees coexist, the water demand of small cohorts is typically reduced, because these cohorts are light-limited. (3) Small cohorts have generally lower carbon demand because the

biomass of their living tissues, and consequently their maintenance costs, is smaller than the live biomass of large trees.  Therefore, it is not necessarily true that the higher soil moisture stress of small cohorts will lead to their extinction.  In ED-2.2, the success or failure of plants to survive droughts is governed by tradeoff between soil moisture stress, size-specific carbon and water demands, and maintenance costs under drought scenarios.  These interactions have been previously explored in Longo et al. (2018). The results indicated increases in the abundance of small-sized plants under more frequent and extended droughts, as high-mortality of large, water-demanding trees improved light conditions in the understory.

*IB Comment 26: Page 37: "nonexistent"*

> Response: As suggested, we replaced "inexistent" with "nonexistent" (p. 41, l. 9 of the annotated manuscript).

*IB Comment 27: Page 37: "stand-level" is not defined in the paper. Does this mean polygon, site, or something else? Also, I'm not sure the significance of the paragraph comparing stand variability to patch variability. What does it mean?*

> Response: The notation was indeed confusing, and we will remove "stand" from the paragraph, and it replace with "polygon" to be consistent with the rest of the narrative. The goal of the analysis in this section was to compare and quantify the impact of structural variability of complex ecosystems such as forests with the climate variability. Climate variability is represented by most dynamic global vegetation models, but structural variability cannot be properly represented by big-leaf models.  The goal here was to show that this variability is relatively important. We re-wrote the paragraph to clarify these points (p. 41, l. 12 – p. 42, l. 8 of the annotated manuscript).

*IB Comment 28: Page 38: "density in the canopy air space"*

> Response: As suggested, we replaced "density at the canopy air space" with "density in the canopy air space" (p. 44, l. 3 of the annotated manuscript).

*IB Comment 29: Page 39: SiB has had a prognostic CAS since 2003 (Baker et al), based on Vidale and Stockli (2005). Just sayin'.*

> Response: Thank you for the additional references, we rephrased the sentence to include SiB as another model that represents canopy air space (p. 44, l. 6 of the annotated manuscript).

*IB Comment 30: Page 40, line 12: "access to and competition for"*

> Response: We modified the text as suggested (p. 44, l. 21 of the annotated manuscript).

*IB Comment 31: Page 40, lines 29-30: "is fundamental to explaining"*

> Response: We modified the text as suggested (p. 45, l. 11 of the annotated manuscript).

*IB Comment 32: Page 41, lines ?: "degradation is pervasive"*

> Response: As suggested, we corrected the sentence to "degradation is pervasive" (p. 46, l. 18 of the annotated manuscript).

*IB Comment 33: Page 42, line 29: "has excellent conservation"*

> Response: We replaced this sentence with "has a high degree of conservation" (p. 47, l. 10–11 of the annotated manuscript).

*On to the Supplements! (I do not have specific line numbers in my supplements file; I'll just have to do my best with explaining where the comments address)*

*IB Comment 34: Table S2: might help to add bulk specific enthalpy, and Temporary Surface Water.*

> Response: As suggested, we added bulk specific enthalpy and temporary surface water to Table S2.

*IB Comment 35: S2: What is a "leaf elongation factor", and how is it determined? There is a long equation to describe slk, but we aren't told what it means.*

> Response: We modified the first paragraph of the Leaf phenology supplement (formerly S2, now S3) to clarify the meaning of leaf elongation factor (p. S30, l. 13–17 of the annotated supplement).  We also included the meaning of $s_{lk}$ after the equation (p. S31, l. 1–2 of the annotated supplement).

*IB Comment 36: S7: is the 'b' term the Clapp and Hornberger b? If not C+H, where does the value come from?*

> Response: Yes, the Reviewer is correct.  We included the citation to Clapp and Hornberger (1978) when *b* is first defined (p. S46, l. 17–18 of the annotated supplement). We also included the information that the parameterization of soil matric potential is derived from Clapp and Hornberger (1978) (p. S46, l. 20–21 of the annotated supplement).

*IB Comment 37: S7, field capacity: I've seen several definitions for determining field capacity from things like moisture potential. Is there a reference for what is being used here?*

> Response: We used the definition by Romano and Santini (2002) and references therein.  We included the reference to Romano and Santini (2002) and the rationale in the revised manuscript (p. S47, l. 9–14 of the annotated supplement).

*IB Comment 38: S9: "contains contributions from reflectance and transmittance"*

> Response: As suggested, we replaced "contribution" with "contributions" (p. S50, l. 5 of the annotated supplement). We did not replace "contain" with "contains" as we are referring to "both the bulk diffuse backscattering ($\beta_{mk}$) and forward scattering ($1-\beta_{mk}$)" (p. S50, l. 5 of the annotated supplement).

*IB Comment 39: S12.1: Are you really able to avoid the 'material surface at the top of the CAS' problem under stable conditions? This has been a problem for years, and may be worth a publication of its own. If you've already written it, advertise it here.*

> Response: We did not develop any new parameterization that avoids the flux underestimation under stable conditions.  Instead, we implemented the Beljaars and Holtslag's (1991) empirical formulation of the flux profile functions.  Beljaars and Holtslag (1991) found that their empirical model resulted in more mixing under stable conditions, so this is their result, not ours. We rewrote the paragraph to clarify these points (p. S59, l. 9–14 of the annotated supplement).

*IB Comment 40: S15: soil moisture limitation on photosynthesis. There's been a lot of work done on this with regard to the fact that individual plants maintain photosynthesis as soil dries down from wilt point, until suddenly closing stomates (Colello et al., 1998; Kim et al., 2010). This behavior, while well-known on the plant scale, is problematic when imposed on the ecosystem scale, as it frequently results in binary, or 'on-off' behavior. Many methods have been utilized to deal with it (e.g. Laio et al., 2001; Porporato et al., 2001, 2002: Rodriguez-Iturbe 2000; Baker et al., 2008, 2013; Wood et al., 1992, to name just a few ). I'd like to see more explanation of what you're doing. A graph showing how stress is imposed, from field capacity to wilt point, would be helpful. Is stress imposed in a linear fashion, or does it behave like the btran function in CLM? Is this function based on previous research (which should be cited), or something incorporated specially for ED2.2? If so, why?*

> Response: The formulation of the soil moisture stress is mostly based on previous versions of ED (Moorcroft et al., 2001).  The only difference in ED-2.2 is that we re-defined the soil water term to be a function of soil matric potential, similar to CLM (Oleson et al., 2013), because the response of the wilting factor to drying the new formulation is slightly more gradual than the original formulation.  We modified the text to include these citations (p. S68, l. 15–20 of the annotated supplement). We also include a new Supplemental Figure S9 that shows the response of the soil moisture stress using both the original ED formulation and ED-2.2, which highlights the less steep transition from no stress to extreme stress when soils are dry.

We should also clarify that the soil moisture stress is calculated for each cohort and each patch, and as explained before, plants with different sizes and PFT have different rooting depths, and may have different water use strategies, expressed through different stomatal conductance parameters. The predicted response to soil water depletion in ED-2.2 does not show the "on-off" response.  For example, in Figure S14 of Longo et al. (2018), we found that the soil moisture stress factor $(1-f_{Wlk})$ would increase more quickly for small cohorts (DBH < 10cm) than larger cohorts (DBH > 35cm) under recurrent fire regimes.  However, that did not cause stomatal conductance to decrease for the small cohorts because once large trees started to die, the small trees experienced more light that compensated the additional soil moisture stress.

The diversity of cohort responses to droughts are further enhanced when using TOPMODEL (one of our current developments, as explained in Section 6.3).  This approach accounts for both edaphic heterogeneity and lateral moisture transfer, and can represent that mesic lowlands do not dry as quickly as ridges, for example.  Finally, please note that a version of ED-2 has been developed that mechanistically solves the plant hydraulics (Xu et al., 2016). This is currently not integrated with the ED-2.2 described here; however, work is underway to include this new formulation into the main distribution of the model on GitHub.

*Figures*

*IB Comment 41: Figure 1: White text was difficult for me to read. It might be worth sacrificing the pretty clouds/sky background for something more simple. Or maybe just use red lettering.*

> Response: In light of the Reviewer's feedback, we increased the contrast between the background and the text, and replaced the cloud picture to avoid white letters on white background.  We also increased the contrast in Figures 2 and 3.  In addition, we changed the symbols in the regional maps (Figures 7 and S4), following a suggestion we received for Part 2.

*IB Comment 42: Figure 2: caption should say "dashed yellow arrows"*

> Response: We corrected the caption as suggested. We also noticed that the colors were incorrect in Figure 3 caption, and corrected these as well.

*IB Comment 43: Figure 3: caption should say there are 3 cohorts shown.*

> Response: In line with the changes described in our response to *IB Comment 25*, we updated Figure 3 to show a generic number ($N_T$) of cohorts.

*IB Comment 44: Nice paper, people. Good work.*

> Response: Thank you Ian for reviewing the manuscript and for your constructive thoughts and feedback.

**Responses to Dr. Stefan Olin**

*SO Comment 1: The model description did not leave out any details, which is a very good thing and it is not very common for many of the existing model description papers. The downside of that is of course that the manuscript is rather long, too long in my opinon.*

> Response: We are glad that the Reviewer thinks the model description is detailed.  Our goal was indeed to describe how the processes are actually solved within the ED-2.2 framework; we believe that providing the complete description allows ED-2.2 users to understand the rationale behind each module, and researchers using other models to understand and reproduce our methods should they decide to implement our developments in other models.  We tried to keep the main text as concise as possible, by keeping only the main description of the fluxes in the main text, and algorithms and details on other variables (specific heat, conductances, definition of state variables, among others) in the Supplements. However, because the model solves many processes at multiple time scales, the final manuscript is still very long.

*SO Comment 2: One thing I miss from the very thorough walkthrough of vegetation models in the introduction are references to the DGVMs that are closer to ED such as LPJ-GUESS (for disclosure, I am an LPJ-GUESS developer).*

> Response: Following the Reviewer's suggestion, we included a new paragraph in the introduction describing the emergence of cohort-based models and included references to some of them, including LPJ-GUESS (p. 4, l. 24 – p.5, l. 2 of the annotated manuscript).

*SO Comment 3: The text is easy to read, and the refernces to equations, sections and tables are good. One comment regarding the referencing to equations is that it should be concistent, for example on line 14, page 7. In my opinon it should read: Eq. 2-3 cannot ... .*

> Response: We believe our equation references are consistent with the GMD style guidelines. We do not have a strong preference and we will change the notation if the editor considers the reference style suggested by the Reviewer more appropriate.

*SO Comment 4: Another comment I have regarding the equations (or symbols) is the sometimes odd choice of symbols. Like Eq. 36-38, why choose the same symbol for a variable that you are using as an operator?, that is very confusing.*

> Response: We sometimes resorted on less conventional symbols because we used all letters of the Latin and Greek alphabet (lower case, upper case, and calligraphy, see Table S2). However, we agree with the reviewer that the use of $\Pi$ for plant area index in Eq. 36–37 is potentially confusing. To address this issue we replaced the plant area index symbol from $\Pi$ to $\Phi$ (upper case), and plant area density to $\phi$ (lower case), and use the symbol $\varpi$, (originally used for plant area density) for oxygenase:carboxylase rate. We tried to restrict odd symbols to variables that were only used in specific equations, to avoid distractions.

*SO Comment 5: The same goes for the use of exp instead of e, and on the note of the letter e, you are using it as pool ($e_j$) and as a scaling factor ($e_H$ ot), I'd say that it is better to use the letter e as the mathematical constant it is, and then use some other symbol to denote your pools. And for your factors, use q or f.*

> Response: Both "e" and "exp" standard mathematical notation to refer to an exponential function. Regarding the scaling factors, we believe the Reviewer was referring to Eq. 103 and 104. However, we agree that using "e" is this context is potentially confusing, and therefore we replaced it with "*f-hat*" ($f_{Cold}$ *and* $f_{Hot}$ are already used in Eq. 85, and *q* is used for specific heat). We also replaced *e* for the elongation factor (Supplements) with $\hat{e}$. We did not use *f* in this case because it would lead to too many levels of subscripts, and no other letter was available. Regarding the soil carbon pools, we think that the subscripts make them clearly distinct from actual variables, and we kept them the same, as all other Latin letters have already been used for something else.

*SO Comment 6: On the same topic of mathematical operators as variables, in Eq. 76, maybe something went wrong, there is a definition character instead of an equal sign. And again, why use operators as super scripts, just adds confusion.*

> Response: We agree that the choice of the equivalent sign to indicate phase equilibrium (also known as saturation) was confusing, and that the superscript was also confusing. In light of the Reviewer's comment, we replaced the previous notation and now refer to phase-equilibrium partial pressure as $p_{Sat}$ and to phase-equilibrium specific humidity as $w_{Sat}$ (Eq. 70 and Eq. 78).

*SO Comment 7: And likewise, in Eq. 56, is that an equal sign as a superscript or do you have an assignment within the equation? Or is it a pre-request? Either way, that equation is confusing.*

> Response: In light of the Reviewer's suggestion, we split Eq. 56 into two equations (Eq. 56 and 57) to eliminate the ambiguity.

*SO Comment 8: With such an explicit formulation of the exchange of heat and water I find it rather strange that the incoming water does not have an explicit energy level specified. If 15 deg. C water lands on a surface that is 25 deg. C, there would be a cooling taking place. Maybe this is of minor importance in the Amazon, but in colder places this would matter. Or did I totally misread what is written in the beginning of Sect. 4.2, if so, I suggest you clarify this.*

> Response: Precipitation has an explicit energy level associated, and this energy input is defined in the paragraph that starts on Page 24, line 10 of the annotated manuscript, including equation 39. Most meteorological drivers (including eddy covariance towers) do not provide precipitation temperature. Therefore, we assumed it to be the same as the temperature of the air above canopy ($T_a$, Eq. 39), which is not the same as the leaf or ground temperature; therefore, the model does account for the cooling effect of precipitation. We rewrote the paragraph to make these important points more explicit (p. 24, l. 10–21 of the annotated manuscript).

In addition to the temperature difference, precipitation in the form of snow has varying density, which in turn affects the density of developing snowpack. The dynamics of the snowpack was missing in the original manuscript, and we think it is an important process for non-tropical simulations. To avoid extending the main manuscript, we included the description of the snowpack depth dynamics as a new supplement S7, and indicated this in the text (p. 18, l. 1–4 of the annotated manuscript).

*SO Comment 9: In the first paragraph of the discussion you are writing that you have demonstrated a functional diverse canopy, from the supplements I get that you have three PFTs along one functional trait axis.*

> Response: The reviewer is correct that there are three default tree PFTs for tropical South America, and for the purposes of this manuscript we only used these three tree PFTs. However, in ED-2.2 model users can specify additional PFTs, or modify the existing PFTs, using XML files that are read during the initialization, and several published ED2 studies have modified or added tropical PFTs according to the scientific questions (e.g. Xu et al. 2016, Trugman et al. 2016, Feng et al. 2018).

This information was missing in our manuscript, and we included a paragraph in Section 2.3 (Model inputs) (p. 11, l. 21–24 of the annotated manuscript).

We also updated the discussion (Section 6.2) and point readers to these previous studies that have used non-default PFTs in the tropics (p. 44, l. 23–28 of the annotated manuscript).

*SO Comment 10: Results are not really discussed nor shown, but one result that there is much focus on is the closed energy budget. Is it really closed if there is a 0.01 deviation? Is there not a great risk of error propagation if the bar is set that low? In LPJ-GUESS we are concerned if the mass balance is off by $10^{-12}$.*

> Response: We agree with the reviewer that we should impose stricter tolerances by default, and we modified the default to be $10^{-5}$ and updated the code available in the

permanent repository information (p. 47, l. 24–25 of the annotated manuscript). We cannot impose a tolerance as strict as LPJ-GUESS: to reduce the size of the output files, variables in ED-2.2 are stored as single precision (truncation error of the order of $10^{-7}$), even though the biophysics solver uses double precision. We tested the model with the new tolerance, and the model ran without problems. We revised the paragraph to include the updated information (p. 39, l. 6–17 of the annotated manuscript).

We had already included text in the discussion proposing areas of improvement of closure, but we inserted a sentence acknowledging that the closure should be improved before the suggestions for improvement (p. 43, l. 10 – p. 44, l. 1 of the annotated manuscript).

Regarding the absence of results, we only showed model verification and some examples of simulations in this paper, in line with the GMD guidelines for model description manuscript. The companion manuscript (Longo et al., 2019, https://doi.org/10.5194/gmd-2019-71), also in review for GMD, has extensive model evaluation for tropical South America.

*SO Comment 11: Some specific comments in addition to those spotted by Ian Baker: Line 3, page 41: remove the 'a'.*

> Response: As suggested, we removed the "a" (p. 46, l. 18 of the annotated manuscript).

*SO Comment 12: Line 3, page 21: intercepted instead of intercept.*

> Response: As suggested, we replaced "intercept" with "intercepted" (p. 23, l. 21 of the annotated manuscript).

*SO Comment 13: Line 1, page 33: What is a decay rate due to respiration? Do you mean turnover?*

> Response: The original sentence was incorrect. This section is describing metabolic fine-root respiration, not turnover. We replaced the sentence for clarification (p. 37, l. 24–25 of the annotated manuscript).

*SO Comment 14: Page 34: GYF is not defined, comes later.*

> Response: We rewrote the text to explain that GYF stands for Paracou, French Guiana (p. 39, l. 7 of the annotated manuscript).

**Response to iterative comment by Dr. John Kim**

*JK Comment 1: Thank you for an interesting and helpful paper. In the introduction the authors briefly outline how ED-2.2 evolved from ED-2 (lines 25- 35), and comparisons against results from ED-2.0.12 and ED-2.1 are made in several places in the Results section. But it's unclear exactly how ED-2.2 is different than those previous versions. What exactly changed between those versions? What new processes are simulated?*

*I think clearly identifying the differences would help the reader understand what mo- tivates this particular model description paper and how it's new and different than the existing model description paper for ED-2 (Medvigy et al. 2009); as well as help the reader understand how those different results might arise between ED-2.0.12, ED-2.1 and ED-2.2.*

> Response: We thank Dr. Kim for the iterative comment.

We would like to clarify that the developments presented in our manuscript reflect changes since ED-2.0, as the technical description of ED-2.0.12 and ED-2.1 were not published and they are part of a continuous model development effort. In addition, while ED-2.0 solved the energy and water cycles at sub-daily time scales, the paper by Medvigy et al. (2009) does not describe the implementation of these cycles in the ED framework. Our manuscript describes, for the first time, the fundamental equations that govern the energy, water, and $CO_2$ dynamics in ED-2 (Section 3), and how we obtain each flux that is accounted for in the fundamental equations (Section 4).

We agree that a more explicit explanation of these improvements was needed. We included additional text in the introduction (p. 5, l. 23–29 of the annotated manuscript), and a detailed description of the model improvements since Medvigy et al. (2009) as a New Supplement S1 (p. S26–S28 of the annotated supplement).

[revised manuscript text omitted]
_{\text{Cold}}(T - T_{\text{Cold}})]\}\{1 + \exp[+f_{\text{Hot}}(T - T_{\text{Hot}})]\}}, \tag{86}$$

where $f_{\text{Cold}}$, $f_{\text{Hot}}$, $T_{\text{Cold}}$ and $T_{\text{Hot}}$ are PFT-dependent, phenomenological parameters to reduce the function value at low and high temperatures (Tables S5-S6).

5    The original expression for the initial slope of the carboxylation rate under near-zero $CO_2$ ($\dot{V}_{C_k}^{\text{InSl}}$) for $C_4$ plants by Collatz et al. (1992) has been modified later (e.g. Foley et al., 1996) to explicitly include $\dot{V}_{C_k}^{\max}$; this is the same expression used in ED-2.2:

$$\dot{V}_{C_k}^{\text{InSl}} = k_{\text{PEP}} \, \dot{V}_{C_k}^{\max} \, c_{l_k}, \tag{87}$$

where $k_{\text{PEP}}$ represents the initial slope of the response curve to increasing $CO_2$; the default value in ED-2.2 (Table S4) is the
10   same value used by Collatz et al. (1992).

From the total photosynthetically active irradiance absorbed by the cohort $\dot{Q}_{\text{PAR:}a,t_k}$ (Eq. 49), we define the photon flux that is absorbed by the leaf ($\dot{q}_k^{\text{PAR}}$, $\text{mol m}_{\text{Leaf}}^{-2}\,\text{s}^{-1}$):

$$\dot{q}_k^{\text{PAR}} = \frac{1}{\text{Ein}} \frac{f_{\text{Clump}_k}}{\tilde{\Pi\Phi}_k} \dot{Q}_{\text{
[revised manuscript text omitted]

[Figure]

Figure S9: Example of the wilting factor ($f_{Wl_k}$, Eq. S201) response to soil moisture change for the original implementation in ED (ED-1.0 and ED-2.0, Moorcroft et al., 2001; Medvigy et al., 2009) and the ED-2.2 model approach. Results here are shown for the idealized case with constant soil moisture profile in a 3-m deep, sandy clay loam soil, for a mid-successional tropical cohort with default parameters (Table S5 with diameter at breast height of 30 cm and leaf area index of 1 $m_{Leaf}^2 m^{-2}$, non-limited leaf-level transpiration rate $\dot{E}_k = 9.0$ $kg_W m_{Leaf}^{-2} day^{-1}$. Values are shown for soil moisture columns ranging from wilting point ($\vartheta_{Wp}; \Psi_{Wp}$) to field capacity ($\vartheta_{Fc}; \Psi_{Fc}$).

[revised manuscript text omitted]

**S1  ED-2 developments since ED-2.0 and ED-2.2**

In this Supplement, we list the main developments in the Ecosystem Demography Model version 2 (ED-2), with focus on mentioned in this manuscript (Fig. S3). The complete list of implementations, improvements, and code fixes are available on the GitHub website (https://github.com/EDmodel/ED2).

**S1.1  Version 2.0 (ED-2.0)**

This is the version described in Medvigy (2006); Medvigy et al. (2009), and it is the first version of the ED model that implements energy and water cycles at sub-daily scale. The biophysics core was adapted from the LEAF-2 land surface model (Walko et al., 2000), which is part of the Regional Atmospheric Model System (RAMS). The main differences in the ED-2.0 biophysics core include (1) solution of the energy and water cycle for each cohort and patch; (2) use of 4th order Runge-Kutta solver to improve numerical stability. In addition, this version allowed leaf phenology to be prescribed from external data (Supplement **??**). The photosynthesis solver was largely the same as in ED-1.0 (Moorcroft et al., 2001).

**S1.2  Version 2.0.12 (ED-2.0.12)**

Most developments between ED-2.0 and ED-2.0.12 relate to code organization and structure. ED-2.0 was partly written in C (legacy from ED-1) and partly written in Fortran (legacy from LEAF-2). To simplify the code and ensure data were correctly transferred between subroutines, we rewrote most of the code in Fortran. The only exceptions were a few file handling functions that remained in C because we could not find equivalent functions in Fortran.

In addition, this version uses Hierarchical Data Format 5 (HDF5) format and libraries (The HDF Group, 2016) to generate model outputs. HDF5 allows a more efficient framework to output variables in the dynamic patch and cohort structures. It also introduced an XML model parameter input file, rather than relying solely on hard-coded defaults, which makes it easier to perform model calibration, sensitivity analyses, and ensemble error propagation. Importantly, this was the last version of ED-2 that used temperature as prognostic variable for leaves and canopy air space.

**S1.3   Version 2.1 (ED-2.1)**

Most ED-2.1 developments aimed at improving the energy cycle representation in ED-2.1. Importantly, leaf enthalpy and canopy air space enthalpy replaced temperature as the prognostic variables (Eq. 4; Sec. 3.2.3–3.2.4). The main advantages of energy-related prognostic equations include: (1) simplification the numeric integration, as total energy changes must be equivalent to net energy flux; (2) improved conservation of energy when water fluxes are large and cause rapid changes in heat capacity of the thermodynamic systems; (3) elimination of singularity at the water's fusion point (0 °C, when enthalpy changes due to freezing or melting, but the temperature remains the same.

To ensure the model was thermodynamically consistent, we also: (1) implemented a mechanistic representation of heat capacity for vegetation (leaves and branches, Supplement S6.2) that is scaled with leaf and branch biomass (e.g. Dufour and van Mieghem, 1975); (2) replaced the original LEAF-2-based surface layer model (that was based on Louis, 1979) with the parameterization by Beljaars and Holtslag (1991), as the latter parameterization improved numerical stability of eddy covariance fluxes under thermally stable conditions; (3) included an option to prescribe silt, clay, and sand fractions to define site-specific soil texture characteristics (Supplement S9) instead of the original ED-2.0 implementation that required soils to be assigned to one of the 12 fixed classes originally defined in LEAF-2 (Walko et al., 2000); (4) implemented the capability of saving the entire ecosystem and thermodynamic state of the model into HDF5 files, which can be used to stop and start simulations and yield the same results of uninterrupted simulations, a desirable feature for simulations with long runtimes.

**S1.4   Version 2.2 (ED-2.2)**

The ED-2.2 version implemented several improvements and fixed inconsistencies in the representation of the energy, water, and carbon dioxide cycles. First, we redefined enthalpy (S5), to ensure that it would be a true thermodynamic state variable (i.e. path independent, see Dufour and van Mieghem, 1975), by making latent heat of vaporization a linear function of temperature (Eq. 72–73). Moreover, we identified missing components of the energy cycle that precluded the conservation: (1) the transfer of internal energy from soils to leaves before transpiration (Eq. 97); (2) the enthalpy exchange associated with vaporization and condensation also accounts the mass transfer of water between the thermodynamic systems (e.g. Eq. 75; 98). Furthermore, to ensure results from ED-2.2 consistently conserve mass and energy, we implemented detailed conservation verification during the model execution, which now reports any violation of energy, water, and carbon conservation, generates detailed output of the violation, and interrupts the simulation. Finally, to improve computational efficiency of the energy, water, and carbon cycle

solvers at sub-daily time steps, we implemented a shared-memory parallelization of the most computationally-intensive subroutines. The parallelization was written to allow users to select any number of cores (depending on core availability), and to account for patch ages in order to balance the load among cores.

In addition, we rewrote the photosynthesis to allow temperature-dependent functions to be expressed as functions of $Q_{10}$. We retained the original Arrhenius-based functions as legacy options, but the new option increases the options for assimilating data into the model. The current $Q_{10}$-based parameters fix the low-temperature optimum in tropical plants previously noted by Rogers et al. (2017). Importantly, we rewrote the photosynthesis solver to ensure that it would always converge to a unique solution for net assimilation rate, stomatal conductance, and intercellular carbon dioxide concentration given the environmental conditions (Supplement S16).

The ED-2.2 version also includes improvements in the representation of conductances between different thermodynamic systems. First, the leaf boundary-layer conductance now accounts for differences in leaf and branch characteristics of each cohort, and to account for both free and forced convection under both laminar and turbulent flow (Supplement S14.2). Second, we implemented ground-to-canopy conductance formulations (Sellers et al., 1986; Massman, 1997; Massman and Weil, 1999) that account for the cumulative drag profile of vegetated areas obtained from the cohort structure, as well as the stability of the surface layer (Supplement S14.3).

Finally, in ED-2.2 we replaced the version control to GitHub, which makes the new code developments readily available to the scientific community and encourages users to post issues, code fixes and model improvements and developments to the main code repository in open and collaborative forums.

**S2 Boundary conditions for the ecosystem dynamics equations**

The boundary conditions for Eq. (2) and (3) are:

$$\underbrace{n_{fq}\left(\mathbf{C}_{f_0},a,t\right)}_{\text{Recruit}} = \frac{1}{\mathbf{g}_{f_0}\cdot\mathbf{1}}\left\{\underbrace{\int_{\mathbf{C_{f_0}}}^{\infty}\left(1-f_{\delta_f}\right)\varrho_f\,n_{fq}\mathrm{d}\mathbf{C}}_{\text{Local recruitment}}+\underbrace{\sum_{q'=1}^{N_Q}\left[\int_{\mathbf{C_{f_0}}}^{\infty}\int_0^{\infty}f_{\delta_f}\,\varrho_f\,n_{FQ'X'Y}\alpha_{fq'}\,\mathrm{d}a\,\mathrm{d}\mathbf{C}\right]}_{\text{Non-local, random dispersal}}\right\},$$

(S1)

$$\underbrace{n_{fq}\left(\mathbf{C}_f,0,t\right)}_{\text{Population at new gap}} = \underbrace{\sum_{q'=1}^{N_Q}\left[\int_0^{\infty}\hat{\sigma}_{fq'}\,n_{fq'}\,\alpha_{q'}\,\mathrm{d}a\right]}_{\text{Disturbance Survivors}},$$

(S2)

$$\underbrace{\alpha_q(0,t)}_{\text{Probability of new gap}} = \underbrace{\sum_{q'=1}^{N_Q} \lambda_{q'q} \alpha_{q'} \mathrm{d}a,}_{\text{Disturbance rates}} \qquad (S3)$$

where $\mathbf{C}_{f_0}$ is the size of the smallest individual of PFT $f$; $\mathbf{g}_{f_0}$ is the growth rate for individuals of PFT $f$ with size $\mathbf{C}_{f_0}$; $\mathbf{1}$ is the unity vector for size; $\varrho_f$ is the recruitment rate, which may depend on the PFT, size, and carbon balance; $f_{\delta_f}$ is the fraction of recruits of PFT $f$ that are randomly dispersed instead of locally recruited; and $\hat{\sigma}_{fq}$ is size-dependent survivorship probability for a PFT $f$ following a disturbance of type $q$ (for a complete list of subscripts and variable meanings, refer to Tables S1 and S2). Both $\mathbf{g}_f$ and $m_f$ are functions of the plant size and the individual's carbon balance. The individual's carbon balance depends on the environment perceived by each individual; in turn, the environment perceived by each individual is modulated by both the plant community living in the same gap and the general landscape environment. Likewise, the disturbance rates may be affected by the local plant community in the gap and the regional landscape environment.

**S3    Long-term carbon dynamics and relation with carbon balance**

**S3.1    Leaf phenology**

Leaf shedding rates ($\omega_{l_k}$) depend on the cohort's life strategy (evergreen or deciduous). In case of deciduous trees, the rates are modulated by the difference between the fully-flushed leaf biomass given size ($C^\bullet$, Supplement S16) and the maximum leaf biomass given environmental constrains, expressed through the leaf elongation factor ($e_{l_k}$). For cold-deciduous cohorts, $e_{l_k}$ is determined either from a prognostic model (Botta et al., 2000; Albani et al., 2006) or prescribed from MODIS-based estimates or from ground observations (Medvigy et al., 2009). For drought-deciduous cohorts, it is determined by the following parameterization:

$$e_{l_k} = \begin{cases} 1 & , \text{if } s_{l_k} \geq 1 \\ s_{l_k} & , \text{if } 0.05 \leq s_{l_k} < 1 \, , \\ 0 & , \text{if } s_{l_k} < 0.05 \end{cases}$$

$$s_{l_k} = \frac{1}{|z_{r_k}| \, \Delta t_{\mathrm{El}}} \int_{t'-\Delta_{\mathrm{Phen}}}^{t'} \left( \sum_{j=j(z_{r_k})}^{N_G} \left\{ \frac{\max\left[0, \Psi_{g_j}(t') + \frac{1}{2}\left(z_{g_j} + z_{g_{j+1}}\right) - \Psi_{\mathrm{Wp}}\right]}{\Psi_{\mathrm{Ld}} - \Psi_{\mathrm{Wp}}} \right\} \right) \mathrm{d}t,$$

$$\omega_{l_k} = \frac{1}{\Delta t_{Phen}} \max\left[0, \frac{C_{l_k}}{C_{l_k}^{\bullet}} - f_{El}\right].$$

In addition to the cold-deciduous and drought-deciduous strategies, leaf phenology of tropical trees can also be represented by an empirical model that is driven by the seasonality of light availability (Kim et al., 2012); this approach, however, was not used in the model evaluation because the empirical model requires site-specific parameters to describe the seasonality of leaf flushing and leaf shedding, and this approach has been tested in only one site so far.

The phenological strategy of the plant functional types, can be evergreen, drought-deciduous, or cold-deciduous. The plant's phenology strategy is defined by two functions: (i) the leaf elongation factor ($\hat{e}_{l_k}$), defined as the ratio between the environmentally-constrained leaf biomass and the potential (maximum) leaf biomass, and the rate of leaf shedding ($\omega_{l_k}(t)$) which either be prognosed, or prescribed from observations.

**S3.1.1  Evergreen plants**

For evergreen PFTs, the elongation factor is always 1, the rate of leaf shedding ($\omega_{l_k}(t)$) is zero, and their rate of leaf turnover is governed by the PFT-dependent leaf turnover parameter ($\tau_{l_k}$, see Eq. S12, and Tables S5-S6). The leaf phenology of tropical trees can also be represented by an empirical model that is driven by the seasonality of light availability (see Kim et al., 2012).

**S3.1.2  Drought-deciduous tropical phenology**

The drought-deciduous phenology assumes that leaf flushing and leaf senescence are controlled by the water availability in the rooting zone. The elongation factor $\hat{e}_{l_k}$ is determined by the following parameterization:

$$\hat{e}_{l_k} = \begin{cases} 1 & \text{, if } s_{l_k} \geq 1 \\ s_{l_k} & \text{, if } 0.05 \leq s_{l_k} < 1 \,, \\ 0 & \text{, if } s_{l_k} < 0.05 \end{cases} \tag{S4}$$

$$s_{l_k} = \frac{1}{|z_{r_k}| \, \Delta t_{\mathrm{El}}} \int_{t'-\Delta t_{\mathrm{El}}}^{t'} \left( \sum_{j=j(z_{r_k})}^{N_G} \left\{ \frac{\max\left[0, \Psi_{g_j}(t') + \frac{1}{2}\left(z_{g_j} + z_{g_{j+1}}\right) - \Psi_{\mathrm{Wp}}\right]}{\Psi_{\mathrm{Ld}} - \Psi_{\mathrm{Wp}}} \right\} \right) dt, \qquad \text{(S5)}$$

where $s_{l_k}$ is a 10-day running average of soil moisture accessed by cohort $k$ (normalized by the difference between the water potential threshold and the wilting point), $z_{r_k}$ is the rooting depth of cohort $k$ (Supplement S18), $\Delta t_{\mathrm{El}}$ is the time scale for changes in phenology (assumed to be 10 days), $j(z_{r_k})$ is the soil layer containing the deepest roots of cohort $k$, $\Psi_{g_j}$ is the soil matric potential at soil layer $j$, $\Psi_{\mathrm{Ld}}$ is the soil matric potential below which plants start shedding leaves (assumed $-1.2$ MPa), $\Psi_{\mathrm{Wp}}$ is the soil matric potential at the wilting point, and $z_{g_j}$ is the depth of soil layer $j$, $(z_{g_{N_G+1}} \equiv 0)$. Leaf shedding occurs whenever soil is drier than the threshold defined by $\Psi_{\mathrm{Ld}}$ and drought conditions are increasing. Specifically:

$$\omega_{l_k} = \frac{1}{\Delta t_{\mathrm{Phen}}} \max\left[0, \frac{C_{l_k}}{C_{l_k}^\bullet} - f_{\mathrm{El}}\right]. \qquad \text{(S6)}$$

**S3.1.3 Cold-deciduous phenology**

The prognostic cold-deciduous leaf phenology approach is a thermal sum and chilling sum-based model identical to that of Albani et al. (2006), which, in turn, is based on Botta et al. (2000). At each patch, growing degree-days (GDD) are accumulated during the extended growing season ($\mathfrak{T}_{\mathrm{GS}}$, January–August for the Northern Hemisphere, and July–February for the Southern Hemisphere), and the chilling days (CHD) in the extended senescing season ($\mathfrak{T}_{\mathrm{SS}}$, November–June for the Northern Hemisphere, and May–December for the Southern Hemisphere):

$$\mathrm{GDD}\,(t) = \sum_{t'=t_{\mathrm{GS}}(0)}^{t} \max\left(0, \overline{T}_c\,(t') - T_{\mathrm{Phen}}\right), \qquad \text{(S7)}$$

$$\mathrm{CHD}\,(t) = \begin{cases} 0 & \text{, if } \overline{T}_c\,(t) \geq T_{\mathrm{Phen}}, \text{ or } t \notin \mathfrak{T}_{\mathrm{SS}} \\ \mathrm{CHD}\,(t - \Delta t_{\mathrm{Phen}}) + 1 & \text{, otherwise} \end{cases}, \qquad \text{(S8)}$$

where $\overline{T}_c$ is the daily average canopy air space temperature, $\Delta t_{\mathrm{Phen}} = 1$ day is the phenology time step (Table 2), $t_{\mathrm{GS}}(0)$ is the beginning of the growing season, and $T_{\mathrm{Phen}} = 278.15$ K ($5\,^\circ$C) is the leaf phenology threshold (Albani et al., 2006). The valued elongation factor $\hat{e}_{l_k}$ is then determined by the following series of conditions:

$$
\hat{e}_{l_k}(t) =
\begin{cases}
0 & \text{, if } \overline{T}_s(t) < 275.15 \text{ K} \\
0 & \text{, if } \overline{T}_s(t) < 284.30 \text{ K and } t_\odot < 655 \text{ min} \\
1 & \text{, if GDD} \geq -68.0 + 638.0 \exp\left[-0.01\,\text{CHD}(t)\right] \\
\hat{e}_{l_k}(t - \Delta t_{\text{Phen}}) & \text{, otherwise}
\end{cases}, \tag{S9}
$$

where $t_\odot$ is the daytime duration.

If desired, cold-deciduous phenology can be prescribed rather than prognosed, as described in Medvigy et al. (2009) and Viskari et al. (2015). The timing of leaf onset and leaf senescence are empirically determined from either field observations or from remote sensing (e.g. Zhang et al., 2003) by fitting the following curves, which are then used to determine $\hat{e}_{l_k}$ in the model:

$$
\hat{e}_{l_k} =
\begin{cases}
\dfrac{1}{1 + (y_0 t)^{y_1}} & \text{, if } t \in \mathfrak{T}_{\text{GS}} \\[2ex]
\dfrac{1}{1 + (y_2 t)^{y_3}} & \text{, if } t \in \mathfrak{T}_{\text{SS}}
\end{cases}, \tag{S10}
$$

where $y_0$, $y_1$, $y_2$, and $y_3$ are empirical parameters, determined from data prior to running the ED-2.2 model and provided to the model as inputs; $t$ is the time, provided as day of year (i.e. 1 for January 1st, 365 for December 31 in non-leap years, and 366 for December 31 in leap years); $\mathfrak{T}_{\text{GS}}$ is the extended growing season (e.g. January–July for the Northern Hemisphere, July–January for the Southern Hemisphere); and $\mathfrak{T}_{\text{SS}}$ is the senescing season (e.g. August–December for the Northern Hemisphere, February–June for the Southern Hemisphere).

[revised manuscript text omitted]
}^{\mathrm{DI}} = \lambda_{\mathrm{TF}} \cdot \left[ N \cdot H \cdot \mathit{VOLTI} \cdot \mathit{YA} \cdot \left( 1 \cdot H \cdot \frac{\rho_{t_k}}{\rho_{\mathrm{LTR}}} \right) \right] \cdot \lambda_{\mathrm{TF}} + \frac{1}{æ}, \tag{S28}$$

$$\tag{S29}$$

 where $æ$ is a PFT-specific term to account for the excess mortality in addition to the background mortality due to plant life span (Tables S5–S6). For tropical broadleaf trees, $æ$ is parameterized following Moorcroft et al. (2001):

$$æ = \frac{0.0933 \, \rho_{\mathrm{LTR}}}{\lambda_{\mathrm{TF}} \, (\rho_{\mathrm{LTR}} - \rho_{t_k})}, \tag{S30}$$

where $\rho_{t_k}$ ( g cm$^{-3}$) is the wood density of tropical broadleaf cohort $k$ (Table S5), and $\rho_{\mathrm{LTR}}$ is the wood density for late-successional, tropical broadleaf trees (Table S5).

Mortality due to cold or frost is also determined through a phenomenological parameterization that linearly increases mortality when the monthly mean canopy air space temperature $\overline{T}_c$ falls below a temperature threshold (Albani et al., 2006):

$$m_{t_k}^{\mathrm{CF}} = 3.0 \max \left[ 0, \min \left( 1, 1 - \frac{\overline{T}_c - T_{F_k}}{5} \right) \right], \tag{S31}$$

where $T_{F_k}$ is a cold temperature threshold that represents the plant hardiness to cold, currently assumed to be 275.65 K for all tropical plants.

Mortality due to fire in ED-2.2 follows the original implementation by Moorcroft et al. (2001), and assumes that while fire depends on local scale dryness, once it ignites, it can spread throughout the entire site. Unlike other mortality rates, here we take multiple patches into account (patches are denoted by subscript $u$). First, let $\lambda_{u,u_0}^{\mathrm{FR}}$ be the disturbance rate associated with fires affecting patch $u$ (and creating patch $u_0$), defined as in Moorcroft et al. (2001):

$$\lambda_{u,u_0}^{\mathrm{FR}} = \mathcal{I} \sum_{u=1}^{N_P} \sum_{k=1}^{N_{T_u}} \left\{ \left[ C_{ul_k} + f_{\mathrm{AG}_{uk}} \left( C_{u\sigma_k} + C_{uh_k} \right) \right] \mathcal{Y}_u \, \alpha_u \right\}, \tag{S32}$$

where $N_P$ is the number of patches, $N_{T_u}$ is the number of cohorts in patch $u$, $\mathcal{Y}_u$ is the binary

ignition function, $\alpha_u$ is the relative area of patch $u$, and  $\mathcal{I} = 0.5\,\mathrm{m^2\,kgC^{-1}\,yr^{-1}}$ is a phenomenological parameter that controls fire intensity, and $f_{\mathrm{AG}_{uk}}$ is the fraction of the tissue that is above ground ( Tables S5–S6).

The ignition switch is defined in terms of the dryness of the environment, following the original formulation by Moorcroft et al. (2001), which uses soil moisture to estimate dryness:

$$
\mathcal{Y}_u =
\begin{cases}
1 & ,\text{ if } \left( \frac{1}{|z_{\mathrm{Fr}}|} \int_{z_{\mathrm{Fr}}}^{0} \vartheta_g \, \mathrm{d}z \right) < \vartheta_{\mathrm{Fr}} \\
0 & ,\text{ otherwise}
\end{cases}
, \tag{S33}
$$

where $z_{\mathrm{Fr}}$ is the maximum soil depth to consider when assessing dryness and $\vartheta_{\mathrm{Fr}}$ is the average soil moisture below which ignition occurs. Both $z_{\mathrm{Fr}}$ and $\vartheta_{\mathrm{Fr}}$ are adjustable parameters; default values are $z_{\mathrm{Fr}} = -0.50\,\mathrm{m}$ and $\vartheta_{\mathrm{Fr}} = \vartheta\,(\Psi_{\mathrm{Fr}})$ $(\Psi_{\mathrm{Fr}} = -1.4\,\mathrm{MPa})$. Once the fire disturbance rate is determined, mortality rate can be determined from the definition of disturbance rate (c.f. Moorcroft et al., 2001):

Comment: Original equation was technically correct, but not intuitive.

$$
m_{ut_k}^{\mathrm{FR}} = \ln \left[ \frac{1}{\hat{\sigma}_{ut_k}^{\mathrm{FR}} + \left(1 - \hat{\sigma}_{ut_k}^{\mathrm{FR}}\right) \exp\left(-\lambda_{u,u_0}^{\mathrm{FR}} \Delta t_{\mathrm{PD}}\right)} \right], \tag{S34}
$$

[revised manuscript text omitted]

**S7  Snowpack depth dynamics**

In addition to enthalpy and total water, we must also track the changes in snowpack depth of each layer ($\Delta z_{s_j}$) and density ($\rho_{s_j}$) over time. The ordinary differential equation that governs changes in depth over time is defined as:

$$\frac{\mathrm{d}\Delta z_{s_j}}{\mathrm{d}t} = \begin{cases} \underbrace{\rho_{wa}\dot{W}_{a,s_j}}_{\substack{\text{Throughfall} \\ \text{precipitation} \\ (4.2)}} + \underbrace{\left(\sum_{k=1}^{N_T}\rho_{wt_k}\dot{W}_{t_k,s_j}\right)}_{\substack{\text{Canopy dripping} \\ \text{from cohorts} \\ (4.2)}} - \underbrace{\rho_{ws_j}\dot{W}_{s_j,o}}_{\substack{\text{Surface runoff} \\ (4.1)}} - \underbrace{\rho_{wx}\dot{W}_{s_j,c}}_{\substack{\text{Surface water} \\ \text{evaporation} \\ (4.5.2) \text{ and } (4.5.3)}} - \underbrace{\delta_{s_1 s_j}\rho_{ws_j}\dot{W}_{s_1,g_{N_G}}}_{\substack{\text{Surface water} \\ \text{percolation} \\ (4.1)}} & \text{, if } s_j = s_{N_S} \\[1em] 0 & \text{, otherwise} \end{cases}$$

, (S68)

$$\rho_{ws_j} = \frac{W_{s_j}}{\Delta z_{s_j}} \tag{S69}$$

$$\rho_{wx} = \begin{cases} \rho_{ws_{N_S}} & \text{, if } \dot{W}_{s_{N_S},c} \geq 0 \\ \rho_{wc} & \text{, if } \dot{W}_{s_{N_S},c} < 0 \end{cases} \tag{S70}$$

where $\delta_{s_j s_{j'}}$ is the Kronecker delta for comparing two TSW layers $s_j$ and $s_{j'}$ (1 if $s_j = s_{j'}$, 0 otherwise), $\rho_{wa}$ is the precipitation density, $\rho_{wt_k}$ is the canopy interception density, $\rho_{wc}$ is the density of condensing water vapor. Precipitation density is defined based on Jin et al. (1999), but slightly modified to make it continuous:

$$\rho_{wa} = \frac{\rho_{ia}\rho_\ell}{\ell_a\rho_{ia} + (1 - \ell_a)\rho_\ell}, \tag{S71}$$

$$\rho_{ia} = \begin{cases} 169.16 & \text{, if } T_a > 275.16 \text{ K} \\ 50. + 1.7(T_a - 258.16)^{1.5} & \text{, if } 258.16 \text{ K} < T_a \leq 275.66 \text{ K} , \\ 50. & \text{, if } T_a \leq 258.16 \text{ K} \end{cases} \tag{S72}$$

where $\rho_\ell$ is the density of liquid water (Table S3). For the canopy dripping flux, water density is similar to Eq. (S71), except that we assume the density of frozen water to be the same as frost density ($\rho_*$, Table S3). A similar assumption is done for water condensing from canopy air space, with the additional assumption that the liquid fraction of condensation is the same as the liquid fraction of the top TSW layer:

$$\rho_{wt_k} = \frac{\rho_*\rho_\ell}{\ell_{t_k}\rho_* + (1 - \ell_{t_k})\rho_\ell}, \tag{S73}$$

$$\rho_{wc} = \frac{\rho_*\rho_\ell}{\ell_{s_{N_S}}\rho_* + \left(1 - \ell_{s_{N_S}}\right)\rho_\ell}. \tag{S74}$$

The maximum allowed number of snow layers is determined by the user, but the actual number of snow layers is dynamically determined, following the same algorithm as Walko et al. (2000). Multiple layers only exist when ice is present, otherwise a single layer ($N_S = 1$) is

enforced. When ice is present, the model selects $N_S$ to be the maximum number of layers that satisfies $W_{s_j} \geq 5\,\mathrm{kg_W\,m^{-2}}$ for all layers $s_j, j \in 1, 2, \ldots, N_S$, to ensure numerical stability. The layer thickness distribution ($\Delta z_{s_j}$) for any given $N_S$ is defined as:

$$\Delta z_{s_j} = z_s \frac{2^{\min(j-1, N_S-j)}}{2^{\lfloor \frac{N_S+1}{2} \rfloor} + 2^{\lfloor \frac{N_S}{2} \rfloor} - 2}, \tag{S75}$$

$$z_s = \sum_{j=1}^{N_S} \Delta z_{s_j}, \tag{S76}$$

where $z_s$ is the total depth of the snow, and $\lfloor x \rfloor$ is the floor function (i.e. the nearest integer value to $x$ that is not greater than $x$). The layer distribution described by Eq. (S75) ensures that the layers near the ground and near the canopy air space are thinner than the intermediate layers, to improve the representation of exchanges between the snowpack and the canopy air space, soils, and incoming irradiance (Walko et al., 2000).

**S8 Canopy-Air-Space Pressure**

[revised manuscript text omitted]

The equation that describes soil matric potential as a function of soil moisture is taken from Clapp and Hornberger (1978); soil hydraulic conductivity is defined after Brooks and Corey (1964), with an additional correction term applied to hydraulic conductivity to reduce conductivity in case the soil is partially or completely frozen:

$$\Psi = \Psi_{\text{Po}} \left( \frac{\vartheta_{\text{Po}}}{\vartheta} \right)^b, \tag{S86}$$

$$\Upsilon_{\Psi} = \left[ 10^{-7(1-\ell)} \right] \Upsilon_{\Psi_{\text{Po}}} \left( \frac{\cancel{\vartheta_{\text{Po}}}\,\vartheta}{\vartheta_{\text{Po}}} \right)^{2b+3}, \tag{S87}$$

where $\Psi_{\text{Po}}$ and $\Upsilon_{\Psi_{\text{Po}}}$ are the soil-texture dependent, matric potential and hydraulic conductivity at bubbling pressure, assumed to be the same as porosity ($\vartheta_{\text{Po}}$); and $\ell$ is the fraction of liquid water of soil moisture.

Comment: Repetitive text, defined in the previous paragraph.

Additional reference points are determined using the above equations combined with Eq. (S86) and (S87). The permanent wilting point $\vartheta_{\text{Wp}}$ and residual soil moisture $\vartheta_{\text{Re}}$ are defined as the soil moisture when soil matric potential is equivalent to $-1.5$ and $-3.1$ MPa, respectively:

$$\vartheta_{\text{Wp}} = \vartheta_{\text{Po}} \cdot \left( -\frac{g\,\rho_\ell\,\Psi_{\text{Po}}}{1.5 \cdot 10^6} \right)^{\frac{1}{b}}, \tag{S88}$$

$$\vartheta_{\text{Re}} = \vartheta_{\text{Po}} \cdot \left( -\frac{g\,\rho_\ell\,\Psi_{\text{Po}}}{3.1 \cdot 10^6} \right)^{\frac{1}{b}}, \tag{S89}$$

where $g$ is the gravity acceleration and $\rho_\ell$ is the density of liquid water (Table S3).

Field capacity ($\vartheta_{\text{Fc}}$) is often defined from soil matric potential (e.g. Hodnett and Tomasella, 2002; Saxton and Rawls, 2006). However, this definition is based on field measurements and the definition of $\vartheta_{\text{Fc}}$ from soil matric potential can substantially across studies, with values ranging from $-0.1$ kPa to $-0.5$ kPa (Romano and Santini, 2002). In ED-2.2, we follow Romano and Santini (2002) and define field capacity in terms of hydraulic conductivity, and assume that the drainage flux of water becomes negligible at hydraulic conductivity of 0.1 kg$_{\text{W}}$ m$^{-2}$ day$^{-1}$:

$$\vartheta_{\text{Fc}} = \vartheta_{\text{Po}} \cdot \left( \frac{1.16 \cdot 10^{-9}}{\Upsilon_{\Psi_{\text{Po}}}} \right)^{\frac{1}{2b+3}}. \tag{S90}$$

Soil thermal conductivity at soil layer $j$ ($\Upsilon_{\text{Q}_{g_j}}$) is a function of the soil texture and soil moisture, and is determined using the *de Vries* weighted average of conductivities of each constituent of the soil (e.g. Parlange et al., 1998):

$$\Upsilon_{\text{Q}_{g_j}} = \frac{\sum\limits_{\kappa=0}^{3} \left[ \left( \frac{3\Upsilon_{\text{Q}_\ell}}{2\Upsilon_{\text{Q}_\ell} + \Upsilon_{\text{Q}_\kappa}} \right) \mathcal{V}_\kappa \left( z_{g_j} \right) \Upsilon_{\text{Q}_\kappa} \right] + \vartheta_{g_j} \Upsilon_{\text{Q}_\ell}}{\sum\limits_{\kappa=0}^{3} \left[ \left( \frac{3\Upsilon_{\text{Q}_\ell}}{2\Upsilon_{\text{Q}_\ell} + \Upsilon_{\text{Q}_\kappa}} \right) \mathcal{V}_\kappa \left( z_{g_j} \right) \right] + \vartheta_{g_j}}, \tag{S91}$$

$$\mathcal{V}_\kappa \left( z_{g_j} \right) = \begin{cases} \vartheta_{\text{Po}} - \vartheta_{g_j} & \kappa = 0 \\ \mathcal{V}_\kappa^{\text{Dry}} \left( 1 - \vartheta_{\text{Po}} \right) & \kappa \neq 0 \end{cases}, \tag{S92}$$

where $\mathcal{V}_\kappa \left( z_{g_j} \right)$ is the volumetric fraction for soil components air, sand, silt, and clay ($\kappa = 0, 1, 2, 3$, respectively) at soil layer $j$; $\Upsilon_{\text{Q}_\kappa}$ is the thermal conductivity for air, sand, silt, and clay (Table S7), respectively; $\Upsilon_{\text{Q}_\ell}$ is the thermal conductivity of water (Table S3); $\mathcal{V}_\kappa^{\text{Dry}}$ is the dry matter volumetric fraction; and $\vartheta_{\text{Po}}$ is the soil porosity. In Eq. (S91), the weights are the product between the volumetric fraction and a function that represents both the ratio of the thermal gradient of the soil constituents and the thermal gradient of water and the shape of each soil constituent (Camillo and Schmugge, 1981); in ED-2.2 we assume all particles to be spherical.

**S10  Thermal and hydraulic properties of temporary surface water**

The fraction of ground covered by the temporary surface water ($f_{TSW}$) is determined following Niu and Yang (2007), with the same coefficients used in the Community Land Model (NCAR-CLM Oleson et al., 2013):

$$f_{TSW} = \begin{cases} 0 & \text{if } N_S = 0 \\ \tanh\left[\frac{\sum_{j=1}^{N_S} z_{s_j}}{2.5 z_{0\varnothing}} \left(\frac{\overline{\rho}_s}{\rho_\circledR}\right)^{-1.0}\right] & \text{if } N_S > 0 \end{cases}, \tag{S93}$$

$$\overline{\rho}_s = \frac{\sum_{j=1}^{N_S} W_{s_j}}{\sum_{j=1}^{N_S} z_{s_j}}, \tag{S94}$$

where $N_S$ is the number of temporary surface water layers, $z_{s_j}$ ( m) is the vertical position of the temporary surface water layer $j$; $W_{s_j}$ ( $\mathrm{kg\,m^{-2}}$) is the water mass of temporary surface water layer $j$, $z_{0\varnothing}$ is the bare soil roughness (Table S4); $\rho_\circledR$ is the reference density of fresh snow (Table S3).

The thermal conductivity of each temporary surface water layer ($\Upsilon_{Q_{s_j}}$) is a function of the layer temperature $T_{s_j}$ and the bulk layer density $\rho_{ws_j}$ (Eq. S69), and is found using the same parameterization as LEAF-2 (Walko et al., 2000):

$$\Upsilon_{Q_{s_j}} = y_0 \cdot \left[y_1 + y_2 \frac{W_{s_j}}{\Delta z_{s_j}} \rho_{ws_j} + y_3 \left(\frac{W_{s_j}}{\Delta z_{s_j}} \rho_{ws_j}\right)^2 + y_4 \left(\frac{W_{s_j}}{\Delta z_{s_j}} \rho_{ws_j}\right)^3\right] \cdot \exp\left(y_5 T_{s_j}\right), \tag{S95}$$

where $(y_0; y_1; y_2; y_3; y_4; y_5) = (1.093 \times 10^{-3}; 0.03; 3.03 \times 10^{-4}; -1.77 \times 10^{-7}; 2.25 \times 10^{-9}; 0.028)$ are empirical constants.

**S11  Optical properties of vegetation, soil, and temporary surface water.**

[revised manuscript text omitted]

$$\varsigma_{m0} = (1 - f_{\mathrm{TSW}}) \, \varsigma_{R_{mg}} + f_{\mathrm{TSW}} \, \varsigma_{R_{ms}} \left( 1 + \varsigma_{T_{ms}} \, \varsigma_{R_{mg}} \right), \tag{S106}$$

where $f_{\mathrm{TSW}}$ is the fraction of ground covered by temporary surface water, $\varsigma_{R_{mg}}$ is the reflectance of the top soil layer; and $\varsigma_{R_{ms}}$ and $\varsigma_{T_{ms}}$ are the reflectance and transmittance of the temporary surface water, respectively. Soil reflectance is a function of the soil color and volumetric soil moisture at the topmost layer, determined from the same parameterization and soil color classes as in Oleson et al. (2013):

$$\varsigma_{R_{mg}} = \min \left[ \varsigma_{R_m}^{\mathrm{Po}} + 0.11 - 0.40\,\vartheta_{g_{N_G}}, \varsigma_{R_m}^{\mathrm{Re}} \right], \tag{S107}$$

where $\varsigma_{R_m}^{\mathrm{Re}}$ and $\varsigma_{R_m}^{\mathrm{Po}}$ are the soil color-dependent reflectance for dry and saturated soils, respectively.

The temporary surface water reflectance $\varsigma_{R_{ms}}$ depends on the liquid fraction, snow grain size and age, impurities, and the direction of incoming radiation, but here we simply assume a

linear interpolation of soil reflectance at saturation and pure snow reflectance ($\varsigma_{R_{ms}}^{\circledast}$; Table S4), assumed constant for each band:

$$\varsigma_{R_{ms}} = \varsigma_{R_{ms}}^{\circledast} + \ell_{s_{N_S}} \left( \varsigma_{R_m}^{\text{Po}} - \varsigma_{R_{ms}}^{\circledast} \right). \tag{S108}$$

Following Verseghy (1991) and Walko et al. (2000), the transmissivity of intercepted irradiance for PAR and NIR is solved following Beer's law, with a direction-independent extinction coefficient:

$$\varsigma_{T_{ms}} = \begin{cases} \exp\left( -\dfrac{\sum_{j=1}^{N_S} \Delta \bar{z}_{s_j}}{f_{\text{TSW}} \bar{\mu}_s} \right) & , \text{if } m \in (1,2) \\ 0 & , \text{if } m = 3 \end{cases}, \tag{S109}$$

where $\bar{\mu}_s = 0.05\,\text{m}$ is the inverse of the optical depth per unit of temporary surface water depth, defined here to be the same coefficient used by Verseghy (1991) and Walko et al. (2000), and the additional $f_{\text{TSW}}^{-1}$ term accounts for the clumping of the temporary surface water, when the water does not cover all ground. Temporary surface water is assumed to be opaque for the TIR band ($m = 3$), following Walko et al. (2000).

**S12 Solving the two-stream linear system of canopy radiation in ED-2.2.**

Because we assume that the optical properties are constant within each layer, it is possible to find an analytical solution for the full profile of direct and diffuse radiation. First, let $\dot{Q}_{mk}^{\odot}$, $\dot{Q}_{mk}^{\Downarrow}$, and $\dot{Q}_{mk}^{\Uparrow}$ be the solution for band $m$ and interface $k$ immediately beneath the cohort (i.e. at $\tilde{\Phi} = \tilde{\Phi}_k$), and $\dot{Q}_{0_{mk}}^{\odot}$, $\dot{Q}_{0_{mk}}^{\Downarrow}$, and $\dot{Q}_{0_{mk}}^{\Uparrow}$ be the solution for band $m$ and interface $k$ immediately above the cohort (i.e. at $\tilde{\Phi} = 0$), as shown in Fig. 4. The direct radiation profile within each layer is simply given by:

$$\dot{Q}_{mk}^{\odot} = \dot{Q}_{0_{mk}}^{\odot} \exp\left( -\frac{\tilde{\Phi}_k}{\mu_k^{\odot}} \right), \tag{S110}$$

$$\dot{Q}_{0_{mk}}^{\odot} = \dot{Q}_{m(k+1)}^{\odot}, \tag{S111}$$

$$\dot{Q}_{m(N_T+1)}^{\odot} = \dot{Q}_{m(\infty,a)}^{\odot}, \tag{S112}$$

where $\dot{Q}_{m(\infty,a)}^{\odot}$ is the above-canopy, incoming direct radiation for band $m$ and serves as the top

boundary condition. Because the value at interface $N_T + 1$ is known, it is possible to determine all levels by integrating the layers from top to bottom.

For the diffuse components, an analytic solution can be found by defining two auxiliary variables $\dot{Q}^+_{mk} \equiv \dot{Q}^\Downarrow_{mk} + \dot{Q}^\Uparrow_{mk}$ and $\dot{Q}^-_{mk} = \dot{Q}^\Downarrow_{mk} - \dot{Q}^\Uparrow_{mk}$. By subtracting (adding) Eq. (46) from (to) Eq. (47), and using Eq. (S110)-(S112) we obtain

$$\frac{\mathrm{d}\dot{Q}^+_{mk}}{\mathrm{d}\tilde{\Phi}} = -\frac{1 - (1 - 2\beta_{mk})\,\varsigma_{mk}}{\overline{\mu}_k}\,\dot{Q}^-_{mk} + \frac{\left(1 - 2\beta^\odot_{mk}\right)\varsigma_{mk}}{\mu^\odot_{mk}}\,\dot{Q}^\odot_{m(k+1)}, \tag{S113}$$

$$\frac{\mathrm{d}\dot{Q}^-_{mk}}{\mathrm{d}\tilde{\Phi}} = -\frac{1 - \varsigma_{mk}}{\overline{\mu}_k}\,\dot{Q}^+_{mk} + \frac{\varsigma_{mk}}{\mu^\odot_k}\,\dot{Q}^\odot_{m(k+1)} + \frac{2\,(1 - \varsigma_{mk})}{\overline{\mu}_k}\,\dot{Q}^\blacklozenge_{mk}. \tag{S114}$$

By differentiating Eq. (S113) and Eq. (S114) and substituting the first derivatives by Eq. (S114) and Eq. (S113), we obtain two independent, second-order ordinary differential equations:

$$\frac{\mathrm{d}^2\dot{Q}^+_{mk}}{\mathrm{d}\tilde{\Phi}^2} = \varkappa^2_{mk}\,\dot{Q}^+_{mk} + \kappa^+_{mk}\exp\left(-\frac{\tilde{\Phi}}{\mu^\odot_k}\right) - 2\varkappa^2_{mk}\,\dot{Q}^\blacklozenge_{ik}, \tag{S115}$$

$$\frac{\mathrm{d}^2\dot{Q}^-_{mk}}{\mathrm{d}\tilde{\Phi}^2} = -\varkappa^2_{mk}\,\dot{Q}^-_{mk} + \kappa^-_{mk}\exp\left(-\frac{\tilde{\Phi}}{\mu^\odot_k}\right), \tag{S116}$$

where

$$\varkappa^2_{mk} = \frac{\left[1 - (1 - 2\beta_{mk})\,\varsigma_{mk}\right](1 - \varsigma_{mk})}{\overline{\mu}^2_k}, \tag{S117}$$

$$\kappa^+_{mk} = -\left[\frac{1 - (1 - 2\beta_{mk})\,\varsigma_{mk}}{\overline{\mu}_k} + \frac{1 - 2\beta^\odot_{mk}}{\mu^\odot_k}\right]\frac{\varsigma_{mk}\,\dot{Q}^\odot_{m(k+1)}}{\mu^\odot_k}, \tag{S118}$$

$$\kappa^-_{mk} = -\left[\frac{(1 - \varsigma_{mk})\left(1 - 2\beta^\odot_{mk}\right)}{\overline{\mu}_k} + \frac{1}{\mu^\odot_k}\right]\frac{\varsigma_{mk}\,\dot{Q}^\odot_{m(k+1)}}{\mu^\odot_k}. \tag{S119}$$

The solution of Eq. (S115)-(S116) is the combination of the homogeneous and the particular solution, and can be determined analytically:

$$\dot{Q}^+_{mk}\left(\tilde{\Phi}\right) = x^{+-}_{mk}\exp\left(-\varkappa_{mk}\tilde{\Phi}\right) + x^{++}_{mk}\exp\left(+\varkappa_{mk}\tilde{\Phi}\right) + \frac{\kappa^+\mu^{\odot 2}_k}{1 - \varkappa^2_{mk}\mu^{\odot 2}_k}\exp\left(-\frac{\tilde{\Phi}}{\mu^\odot_k}\right) + 2\dot{Q}^\blacklozenge_{mk} \tag{S120}$$

$$\dot{Q}^-_{mk}\left(\tilde{\Phi}\right) = x^{--}_{mk}\exp\left(-\varkappa_{mk}\tilde{\Phi}\right) + x^{-+}_{mk}\exp\left(+\varkappa_{mk}\tilde{\Phi}\right) + \frac{\kappa^-\mu^{\odot 2}_k}{1 - \varkappa^2_{mk}\mu^{\odot 2}_k}\exp\left(-\frac{\tilde{\Phi}}{\mu^\odot_k}\right) \tag{S121}$$

where $x_{mk}^{+-}$, $x_{mk}^{++}$, $x_{mk}^{--}$, and $x_{mk}^{-+}$ are coefficients to be determined. We can reduce the number of coefficients to two by differentiating Eq. (S120)-(S121) and comparing them to Eq. (S113)-(S114), and using the fact that they must be equal for any $\tilde{\Phi}$, $\mu_k^{\odot}$, $\varkappa_{mk}$, and $\dot{Q}_{mk}^{\blacklozenge}$. We call these parameters $x_{m(2k-1)}$ and $x_{m(2k)}$, $k \in \{1,2,\ldots,N_T\}$. By further recalling the definition of $\dot{Q}_{mk}^+$ and $\dot{Q}_{mk}^-$, we obtain the profile of downward and upward diffuse irradiances:

$$\dot{Q}_{mk}^{\Downarrow}\left(\tilde{\Phi}\right) = x_{m(2k-1)} Đ_{mk}^+ \exp\left(-\varkappa_{mk}\tilde{\Phi}\right) + x_{m(2k)} Đ_{mk}^- \exp\left(+\varkappa_{mk}\tilde{\Phi}\right) + Þ_{mk}^+ \exp\left(-\frac{\tilde{\Phi}}{\mu_k^{\odot}}\right) + \dot{Q}_{mk}^{\blacklozenge},$$
(S122)

$$\dot{Q}_{mk}^{\Uparrow}\left(\tilde{\Phi}\right) = x_{m(2k-1)} Đ_{mk}^- \exp\left(-\varkappa_{mk}\tilde{\Phi}\right) + x_{m(2k)} Đ_{mk}^+ \exp\left(+\varkappa_{mk}\tilde{\Phi}\right) + Þ_{mk}^- \exp\left(-\frac{\tilde{\Phi}}{\
[revised manuscript text omitted]

 The ED-2.2 model uses the empirical parameterization of the originally developed by Beljaars and Holtslag (1991). For the unstable cases, Beljaars and Holtslag (1991) used the Businger-Dyer flux profile equations (Businger et al., 1971). For the stable cases, Beljaars and Holtslag (1991) implemented an empirical formulation that improved the vertical mixing between the canopy air space and the air above under stable conditions:

$$\psi_U(\zeta) = \begin{cases} 2\ln\left[ \frac{1+Y(\zeta)}{2} \right] + \ln\left[ \frac{1+Y^2(\zeta)}{2} \right] - 2\arctan[Y(\zeta)] + \frac{\pi}{2} & \text{, if Ri}_B < 0 \\ y_1\zeta + y_2\left( \zeta - \frac{y_3}{y_4} \right) \exp(-y_4\zeta) + \frac{y_2 y_3}{y_4} & \text{, if Ri}_B \geq 0 \end{cases}, \tag{S153}$$

$$\psi_\Theta(\zeta) = \begin{cases} 2\ln\left[ \frac{1+Y^2(\zeta)}{2} \right] & \text{, if Ri}_B < 0 \\ 1 - \left( 1 - \frac{y_1}{y_5}\zeta \right)^{y_5} + x_2\left( \zeta - \frac{y_3}{y_4} \right) \exp(-y_4\zeta) + \frac{y_2 y_3}{y_4} & \text{, if Ri}_B \geq 0 \end{cases}, \tag{S154}$$

$$Y(\zeta) = \sqrt[4]{1 - y_6\zeta}, \tag{S155}$$

where $\mathbf{y} = \left( -1; -\frac{2}{3}; 5; 0.35; \frac{3}{2}; 13 \right)$ are empirical and adjustable parameters. Equation (S152) cannot be solved analytically, therefore $\zeta^\star$ is calculated using a root-finding technique. Once $\zeta^\star$ is determined, we can find $u^\star$ using Eq. (S150), and define the canopy conductance $G_c$ ($\text{m s}^{-1}$) using Eq. (S151) as the starting point, similarly to Oleson et al. (2013):

$$G_c = \frac{u^\star \theta_V^\star}{\theta_{V_a} - \theta_{V_c}} = \frac{\kappa u^\star}{\text{Pr}\left[ \ln\left( \frac{\zeta^\star}{\zeta_0} \right) - \psi_\Theta(\zeta^\star) + \psi_\Theta(\zeta_0) \right]}. \tag{S156}$$

**S14.2 Derivation leaf and wood boundary layer conductances**

Following Monteith and Unsworth (2008), convection can be of two types: forced convection, which depends on mechanic mixing associated with the fluid velocity; and free convection, which

is due to buoyancy of the boundary layer fluid. Although convection is often dominated by either forced or free convection, in ED-2.2 we always assume that the total conductance is a simple combination of forced and free convection conductances as if they were parallel:

$$G_{Qx_k} = G_{Qx_k}^{\text{Free}} + G_{Qx_k}^{\text{Forced}}, \tag{S157}$$

where $x_k$ can be either the leaf ($\lambda_k$) or the branch wood ($\beta_k$) boundary layer. For each convective regime, we define the conductance in terms of the Nusselt number Nu, a dimensionless number that corresponds to the ratio between heat exchange through convection and conduction:

$$G_{Qx_k} = \frac{\eta_c \, \text{Nu}}{x^\star}. \tag{S158}$$

where $\eta_c$ is the thermal diffusivity of canopy air space and $x^\star$ is the characteristic size of the obstacle. For leaves, the characteristic size $x_{\lambda_k}^\star$ is a PFT-dependent constant corresponding to the typical leaf width , whereas for branch wood the typical size $x_{\beta_k}^\star$ is assumed to be the typical diameter of twigs ( Tables S5–S6).

Free convection is a result of the thermal gradient between the obstacle surface and the fluid, and this is normally expressed in terms of the Grashof number Gr, a dimensionless index that relates buoyancy and viscous forces. In ED-2.2 we use the same empirical functions as Monteith and Unsworth (2008), using flat plate geometry for leaves and horizontal cylinder geometry for branch wood:

$$\text{Nu}_{\lambda_k}^{(\text{Free})} = \max \left[ \underbrace{0.50 \, \text{Gr}_{\lambda_k}^{\frac{1}{2}}}_{\text{Laminar}}, \underbrace{0.13 \, \text{Gr}_{\lambda_k}^{\frac{1}{3}}}_{\text{Turbulent}} \right], \tag{S159}$$

$$\text{Nu}_{\beta_k}^{(\text{Free})} = \max \left[ \underbrace{0.48 \, \text{Gr}_{\beta_k}^{\frac{1}{2}}}_{\text{Laminar}}, \underbrace{0.09 \, \text{Gr}_{\beta_k}^{\frac{1}{3}}}_{\text{Turbulent}} \right], \tag{S160}$$

$$\text{Gr}_{x_k} = \frac{\varepsilon_c \, g \, \left( x_{x_k}^\star \right)^3}{v_c^2} \, |T_{x_k} - T_c|, \tag{S161}$$

[revised manuscript text omitted]
^{\#}{}_{\mathrm{Sat}}(T) = \min\left[p_{vi}^{\#}(T), p_{v\ell}^{\#}(T)\right]. \tag{S185}$$

Both $p_{vi}^{\#}$ and $p_{v\ell}^{\#}$ are defined after the parameterization by ([Murphy and Koop, 2005]), which have high degree of accuracy ($< 0.05\%$) between 123 K and 332 K, and thus includes all the range of near-surface temperatures solved by ED-2.2:

$$p_{vi}^{\#}(T) = \exp\left[9.550426 - \frac{5723.265}{T} + 3.53068 \ln(T) - 0.00728332\,T\right], \tag{S186}$$

$$p_{v\ell}^{\#}(T) = \exp\{Y_1(T) + Y_2(T)\tanh[0.0415(T-218.8)]\}, \tag{S187}$$

$$Y_1(T) = 54.842763 - \frac{6763.22}{T} - 4.210\ln(T) + 0.000367\,T, \tag{S188}$$

$$Y_2(T) = 53.878 - \frac{1331.22}{T} - 9.44523\ln(T) + 0.014025\,T. \tag{S189}$$

Importantly, Eq. (S186) and Eq. (S187) yield the same value (within $4.1 \cdot 10^{-6}\%$ accuracy) at the water's triple point, which guarantees continuity of Eq. (S185).

The saturation specific humidity $w^{=}$ is obtained using Eq. (S185) and the definition of specific humidity:

$$w^{\#}{}_{\text{Sat}}(T,p) = \frac{\mathcal{M}_w\, p^{\#}{}_{\text{Sat}}(T)}{\mathcal{M}_d\,[p - p^{\#}{}_{\text{Sat}}(T)] + \mathcal{M}_w\, p^{\#}{}_{\text{Sat}}(T)}, \tag{S190}$$

5 where $\mathcal{M}_d$ and $\mathcal{M}_w$ are the molar masses of dry air and water, respectively (Tab S3).

**S16  Solver for the $CO_2$ assimilation rates and transpiration**

Variables $w_{l_k}$, $\dot{V}_{C_k}^{\max}$, $\dot{R}_k$, $\not{\varpi}_k$, $\mathcal{K}_{O_k}$, $\mathcal{K}_{C_k}$, $\Gamma_k$, and $\mathcal{K}_{\text{ME}_k}$ are functions of leaf temperature and canopy air space pressure, and thus can be determined directly. In constrast, nine variables are unknown for each limitation case as well as for the case when the stomata are closed: $\dot{E}_k$, $\dot{A}_k$, $\dot{V}_{C_k}$,
10 $\dot{V}_{O_k}$, $c_{l_k}$, $c_{\lambda_k}$, $w_{\lambda_k}$, $\hat{G}_{Wl_k}$, and $\hat{G}_{Cl_k}$. To solve the remaining unknowns, we first substitute Eq. (82) and either Eq. (85), Eq. (87) or Eq. (89) into Eq. (81) and write a general functional form for $\dot{A}_k$, similarly to Medvigy (2006), that is a function of only one unknown, $c_{l_k}$:

$$\dot{A}_k(c_{l_k}) = \frac{F_k^{\text{A}}\, c_{l_k} + F_k^{\text{B}}}{F_k^{\text{C}}\, c_{l_k} + F_k^{\text{D}}} - \dot{R}_k, \tag{S191}$$

where parameters $F$ depend on the limitation and the photosynthetic pathway, as shown in Table S9.
15 We then combine Eq. (76) and Eq. (S170) to eliminate $\hat{G}_{Cl_k}$ and $c_{\lambda_k}$, and write an alternative equation for $\hat{G}_{Wl_k}$:

$$\hat{G}_{Wl_k} = \frac{f_{Gl}\,\hat{G}_{W\lambda_k}\,\dot{A}_k}{\hat{G}_{W\lambda_k}\,(c_c - c_{l_k}) - f_{G\lambda}\,\dot{A}_k}. \tag{S192}$$

To eliminate $c_{\lambda_k}$ and $w_{\lambda_k}$ from Eq. (91), we use Eq. (76) and Eq. (77). Then, we eliminate $\hat{G}_{Wl_k}$ by replacing the left hand side of Eq. (91) by the alternative Eq. (S192), yielding to the following

function $\mathcal{F}(c_{l_k})$ for which we seek the solution $\mathcal{F}(c_{l_k}) = 0$:

$$\mathcal{F}(c_{l_k}) = \mathcal{F}_1(c_{l_k})\,\mathcal{F}_2(c_{l_k})\,\mathcal{F}_3(c_{l_k}) - 1, \tag{S193}$$

$$\mathcal{F}_1(c_{l_k}) = \frac{\left(f_{Gl} - f_{G\lambda}\,\dfrac{\hat{G}^{\varnothing}_{Wl_k}}{\hat{G}_{W\lambda_k}}\right)\dot{A}_k - \hat{G}^{\varnothing}_{Wl_k}\left(c_c - c_{l_k}\right)}{m_k\dot{A}_k}, \tag{S194}$$

$$\mathcal{F}_2(c_{l_k}) = \frac{\hat{G}_{W\lambda_k}\left(c_c - \Gamma_k\right) - f_{G\lambda}\dot{A}_k}{\hat{G}_{W\lambda_k}\left(c_c - c_{l_k}\right) + \left(f_{Gl} - f_{G\lambda}\right)\dot{A}_k}, \tag{S195}$$

$$\mathcal{F}_3(c_{l_k}) = 1 + \frac{w_c - w_{l_k}}{\Delta w_k}\,\frac{\hat{G}_{W\lambda_k}\left(c_c - c_{l_k}\right) - f_{G\lambda}\dot{A}_k}{\hat{G}_{W\lambda_k}\left(c_c - c_{l_k}\right) + \left(f_{Gl} - f_{G\lambda}\right)\dot{A}_k}. \tag{S196}$$

For the limitation cases in which Eq. (S191) does not depend on $c_{l_k}$, Eq. (S193) is reduced to a quadratic equation. For the other cases, Eq. (S193) becomes a fifth-order polynomial, which cannot be solved algebraically. Nevertheless, Eq. (S193) is still convenient because it highlights the range of plausible solutions, corresponding to the singularities associated with $\mathcal{F}_1$ and $\mathcal{F}_2$ — the singularities associated with $\mathcal{F}_3$ requires $c_{l_k}$ to exceed $c_c$, which could be only achieved with negative $\hat{G}_{l_k w}$ or $\dot{A}_k < -\dot{M}_k$, and none of them are meaningful. Function $\mathcal{F}_1$ is singular when $\dot{A}_k = 0$; from Eq. (S192), this would $\hat{G}_{Wl_k}$ to be 0, unless $c_{l_k} = c_c$. Function $\mathcal{F}_2$ is singular when $\dot{A}_k = \hat{G}_{C\lambda_k}\left(c_c - c_{l_k}\right)$; from Eq. (S192), this happens only when $c_{l_k} = c_c$ or at $\lim_{\hat{G}_{Wl_k}\to\infty}$. The singularities for when $c_c \neq c_{l_k}$ are obtained by substituting Eq. (S191) into Eq. (76), and by taking the $\lim_{\hat{G}_{Wl_k}\to 0}\left(\dot{A}_k\right)$ and $\lim_{\hat{G}_{Wl_k}\to\infty}\left(\dot{A}_k\right)$:

$$c_{l_k}^{\min} + \frac{F_k^{D}\dot{M}_k - F_k^{B}}{F_k^{C}\dot{M}_k - F_k^{A}} = 0, \tag{S197}$$

$$\left(c_{l_k}^{\max}\right)^2 + \frac{\hat{G}_{C\lambda_k}F_k^{D} + F_k^{B} - F_k^{C}\left(\hat{G}_{C\lambda_k}c_c + \dot{M}_k\right)}{\hat{G}_{C\lambda_k}F_k^{C}}c_{l_k}^{\max} + \frac{F_k^{B} - F_k^{D}\left(\hat{G}_{C\lambda_k}c_c + \dot{M}_k\right)}{\hat{G}_{C\lambda_k}F_k^{C}} = 0. \tag{S198}$$

[revised manuscript text omitted]

$$z_{t_k} = \min\left\{35.0, 61.7\left[1 - \exp\left(-0.0352 \cdot \text{DBH}_k^{0.694}\right)\right]\right\}$$

$$z_{t_k}^{-} = \begin{cases} \max\left(0.05, 0.01\, z_{t_k}\right) & \text{if cohort } k \text{ is grass} \\ \max\left(0.05, z_{t_k} - 0.31\, z_{t_k}^{1.098}\right) & \text{, if cohort } k \text{ is tree} \end{cases}.$$

The tree height of any cohort $k$ ($z_{t_k}$) is determined through a modified Weibull function:

$$z_{t_k} = \min\left\{z_{t_{max}}, \mathcal{Z}_0 + \mathcal{Z}_\infty\left[1 - \exp\left(-\mathcal{Z}_1 \cdot \text{DBH}_k^{\mathcal{Z}_2}\right)\right]\right\}, \tag{S204}$$

where $\mathcal{Z}_0$, $\mathcal{Z}_1$, $\mathcal{Z}_2$, and $\mathcal{Z}_\infty$ are PFT-dependent coefficients; and $z_{t_{max}}$ is the maximum tree height, imposed to avoid excessive extrapolation of the allometric equations for carbon stocks. The coefficients are shown in Tables S5-S6; coefficients for tropical trees are provided by Poorter et al. (2006) allometric equation for moist forests in Bolivia; coefficients for temperate trees are from Albani et al. (2006).

The tree height at the bottom of the crown ($z_{t_k}^-$) is based on Poorter et al. (2006), and it is currently applied to tropical, subtropical, and temperate trees. For grasses, we fix the height to 1% 
[revised manuscript text omitted]

Romano, N. and Santini, A.: Field, in: Methods of Soil Analysis: Part 4 Physical Methods, edited by Dane, J. H. and Topp, G. C., SSSA Book Series 5.4, chap. 3.3.3, pp. 721–738, Soil Science Society of America, Madison, WI, 2002.

Sauer, T. and Norman, J.: Simulated canopy microclimate using estimated below-canopy soil surface transfer coefficients, Agric. For. Meteorol., 75, 135–160, doi:10.1016/0168-1923(94)02208-2, 1995.

Saxton, K. E. and Rawls, W. J.: Soil Water Characteristic Estimates by Texture and Organic Matter for Hydrologic Solutions, Soil Sci. Soc. Am. J., 70, 1569–1578, doi:10.2136/sssaj2005.0117, 2006.

Sellers, P. J.: Canopy reflectance, photosynthesis and transpiration, Int. J. Remote Sens., 6, 1335–1372, doi:10.1080/01431168508948283, 1985.

Sellers, P. J., Mintz, Y., Sud, Y. C., and Dalcher, A.: A Simple Biosphere Model (SIB) for Use within General Circulation Models, J. Atmos. Sci., 43, 505–531, doi:10.1175/1520-0469(1986)043<0505:ASBMFU>2.0.CO;2, 1986.

Sellers, P. J., Randall, D. A., Collatz, G. J., Berry, J. A., Field, C. B., Dazlich, D. A., Zhang, C., Collelo, G. D., and Bounoua, L.: A Revised Land Surface Parameterization (SiB2) for Atmospheric GCMS. Part I: Model Formulation, J. Climate, 9, 676–705, doi:10.1175/1520-0442(1996)009<0676:ARLSPF>2.0.CO;2, 1996.

Shaw, R. H. and Pereira, A.: Aerodynamic roughness of a plant canopy: A numerical experiment, Agric. For. Meteorol., 26, 51–65, doi:10.1016/0002-1571(82)90057-7, 1982.

Shinozaki, K., Yoda, K., Hozumi, K., and Kira, T.: A quantitative analysis of plant form – the pipe model theory. I. Basic analyses, Jpn. J. Ecol., 14, 97–105, doi:10.18960/seitai.14.3_97, 1964a.

Shinozaki, K., Yoda, K., Hozumi, K., and Kira, T.: A quantitative analysis of plant form – the pipe model theory. II. Further evidence of the theory and its application in forest ecology, Jpn. J. Ecol., 14, 133–139, doi:10.18960/seitai.14.4_133, 1964b.

Stull, R. B.: An introduction to boundary layer meteorology, vol. 13 of *Atmospheric and Oceanographic Sciences Library*, Springer Netherlands, Dordrecht, Netherlands, doi:10.1007/978-94-009-3027-8, 1988.

The HDF Group: Hierarchical data format, version 5, URL http://www.hdfgroup.org/HDF5/, 2016.

Verseghy, D. L.: Class—A Canadian land surface scheme for GCMS. I. Soil model, Intl. J. Climatol., 11, 111–133, doi:10.1002/joc.3370110202, 1991.

Viskari, T., Hardiman, B., Desai, A. R., and Dietze, M. C.: Model-data assimilation of multiple phenological observations to constrain and predict leaf area index, Ecol. Appl., 25, 546–558, doi:10.1890/14-0497.1, 2015.

Walko, R. L., Band, L. E., Baron, J., Kittel, T. G. F., Lammers, R., Lee, T. J., Ojima, D., Pielke, R. A., Taylor, C., Tague, C., Tremback, C. J., and Vidale, P. L.: Coupled Atmosphere–Biophysics–Hydrology Models for Environmental Modeling, J. Appl. Meteor., 39, 931–944, doi:10.1175/1520-0450(2000)039<0931:CABHMF>2.0.CO;2, 2000.

Wohlfahrt, G. and Cernusca, A.: Momentum Transfer By A Mountain Meadow Canopy: A Simulation Analysis Based On Massman's (1997) Model, Boundary-Layer Meteorol., 103, 391–407, doi:10.1023/A:1014960912763, 2002.

Wright, I. J., Reich, P. B., Westoby, M., Ackerly, D. D., Baruch, Z., Bongers, F., Cavender-Bares, J., Chapin, T., Cornelissen, J. H. C., Diemer, M., Flexas, J., Garnier, E., Groom, P. K., Gulias, J., Hikosaka, K., Lamont, B. B., Lee, T., Lee, W., Lusk, C., Midgley, J. J., Navas, M.-L., Niinemets, U., Oleksyn, J., Osada, N., Poorter, H., Poot, P., Prior, L., Pyankov, V. I., Roumet, C., Thomas, S. C., Tjoelker, M. G., Veneklaas, E. J., and Villar, R.: The worldwide leaf economics spectrum, Nature, 428, 821–827, doi:10.1038/nature02403, 2004.

Wright, S. J., Jaramillo, M. A., Pavon, J., Condit, R., Hubbell, S. P., and Foster, R. B.: Reproductive size thresholds in tropical trees: variation among individuals, species and forests, J. Trop. Ecol., 21, 307–315, doi:10.1017/S0266467405002294, 2005.

Zhang, X., Friedl, M. A., Schaaf, C. B., Strahler, A. H., Hodges, J. C. F., Gao, F., Reed, B. C., and Huete, A.: Monitoring vegetation phenology using MODIS, Remote Sens. Environ., 84, 471–475, doi:10.1016/S0034-4257(02)00135-9, 2003.

---

## Author Response (AR2)

Dear Dr. Müller,

Thank you for the additional assessment on our manuscript "*The biophysics, ecology, and biogeochemistry of functionally diverse, vertically- and horizontally-heterogeneous ecosystems: the Ecosystem Demography Model, version 2.2 — Part 1: Model description*" (gmd-2019-45), and for accepting our manuscript for publication.

We carefully checked the equations one more time (as suggested by Dr. Ian Baker), and identified a few typographical errors. We also found and fixed minor typographical and grammar errors in the text and in two figures (Fig. 4 and Fig. S1), rewrote a few sentences to improve clarity, and changed the font size of some figures to improve legibility. None of these changes affected any of the results, interpretation of the results, or conclusions.

Below we enclose the annotated manuscript and annotated supporting information for reference. In the annotated text, blue text was added, and  was removed.

Sincerely,
Marcos Longo

[revised manuscript text omitted]
}'\left(T, x\right) = \frac{\mathcal{T}\left(T, x\right)}{\left\{1 + \exp\left[-f_{\text{Cold}}\left(T - T_{\text{Cold}}\right)\right]\right\}\left\{1 + \exp\left[+f_{\text{Hot}}\left(T - T_{\text{Hot}}\right)\right]\right\}}, \tag{86}$$

where $f_{\text{Cold}}$, $f_{\text{Hot}}$, $T_{\text{Cold}}$ and $T_{\text{Hot}}$ are PFT-dependent, phenomenological parameters to reduce the function value at low and high temperatures (Tables S5-S6).

The original expression for the initial slope of the carboxylation rate under near-zero $CO_2$ ($\dot{V}_{C_k}^{\text{InSl}}$) for $C_4$ plants by Collatz et al. (1992) has been modified later (e.g. Foley et al., 1996) to explicitly include $\dot{V}_{C_k}^{\max}$; this is the same expression used in ED-2.2:

$$\dot{V}_{C_k}^{\text{InSl}} = k_{\text{PEP}}\,\dot{V}_{C_k}^{\max}\,c_{l_k}, \tag{87}$$

where $k_{\text{PEP}}$ represents the initial slope of the response curve to increasing $CO_2$; the default value in ED-2.2 (Table S4) is the same value used by Collatz et al. (1992).

From the total photosynthetically active irradiance absorbed by the cohort $\dot{Q}_{\text{PAR}:a,t_k}$ (Eq. 49), we define the photon flux that is absorbed by the leaf ($\dot{q}_k^{\text{PAR}}, \text{mol}\,\text{m}_{\text{Leaf}}^{-2}\,\text{s}^{-1}$):

$$\dot{q}_k^{\text{PAR}} = \frac{1}{\text{Ein}}\frac{f_{\text{Clump}_k}}{\tilde{\Phi}_k}\,\dot{Q}_{\text{
[revised manuscript text omitted]

[Figure]

Figure S9: Example of the wilting factor ($f_{Wl_k}$, Eq. S201) response to soil moisture change for the original implementation in ED (ED-1.0 and ED-2.0, Moorcroft et al., 2001; Medvigy et al., 2009) and the ED-2.2 model approach. Results here are shown for the idealized case with constant soil moisture profile in a 3-m deep, sandy clay loam soil, for a mid-successional tropical cohort with default parameters (Table S5), with diameter at breast height of 30 cm and leaf area index of 1 $m^2_{Leaf} m^{-2}$, non-limited leaf-level transpiration rate $\dot{E}_k = 9.0$ $kg_W m^{-2}_{Leaf} day^{-1}$. Values are shown for soil moisture columns ranging from wilting point ($\vartheta_{Wp}; \Psi_{Wp}$) to field capacity ($\vartheta_{Fc}; \Psi_{Fc}$).

[revised manuscript text omitted]

Table S5: List of default parameters that depend on plant functional type (PFT) used in ED-2.2  for tropical and subtropical regions, for default ED-2.2 PFTs. The  PFTs are C$_4$ tropical grass (C4G), C$_3$ tropical grass (C3G); early successional tropical tree (ETR); mid-successional tropical tree (MTR); late-successional tropical tree (LTR); subtropical conifers (ARC); additional PFTs can be specified by the user and provided directly to ED-2.2 through extensible markup language (XML). Spectral-dependent parameters $x$ are provided as vectors ($x_{\text{PAR}}$; $x_{\text{NIR}}$; $x_{\text{TIR}}$), corresponding to the visible (photosynthetically active), nearinfrared, and thermal infrared, respectively. The default parameters for temperate PFTs are shown in Table S6. The values of constants and default global parameters are shown in Table S3.

| Symbol | PFT-specific value | | | | | | Units | Description |
|---|---|---|---|---|---|---|---|---|
| | C4G | C3G | ETR | MTR | LTR | ARC | | |
| $\alpha$ | 15.15 | 15.15 | 16.22 | 31.58 | $\infty$ | 900.09 | yr | Aging factor for density-independent mortality |
| $B_{Wl}$ | 1.85 | 1.85 | 1.85 | 1.85 | 1.85 | 1.85 | — | Water:oven-dry-biomass ratio for leaves |
| $B_{Wb}$ | 0.70 | 0.70 | 0.70 | 0.70 | 0.70 | 0.70 | — | Water:oven-dry-biomass ratio for wood |
| $C_{0l}$ | 0.158 | 0.158 | 0.418 | 0.560 | 0.701 | 0.410 | — | Scaling coefficient for leaf biomass allometry |
| $C_{1l}$ | 0.975 | 0.975 | 0.975 | 0.975 | 0.975 | 0.975 | — | Exponent coefficient for leaf biomass allometry |
| $C_{0h}$ | 0.0627 | 0.0627 | 0.166 | 0.222 | 0.282 | 0.163 | — | Scaling coefficient for heartwood biomass allometry (sub-canopy) |
| $C_{1h}$ | 2.432 | 2.432 | 2.432 | 2.432 | 2.432 | 2.432 | — | Exponent coefficient for heartwood biomass allometry (sub-canopy) |
| $C_{2h}$ | 0.0647 | 0.0647 | 0.172 | 0.230 | 0.291 | 0.168 | — | Scaling coefficient for heartwood biomass allometry (canopy) |
| $C_{3h}$ | 2.426 | 2.426 | 2.426 | 2.426 | 2.426 | 2.426 | — | Exponent coefficient for heartwood biomass allometry (canopy) |
| $f_{\text{AG}}$ | 0.70 | 0.70 | 0.70 | 0.70 | 0.70 | 0.70 | — | Fraction of above-ground biomass |
| $f_{\text{Cold}}$ | 0.40 | 0.40 | 0.40 | 0.40 | 0.40 | 0.40 | — | Decay parameter to down-regulate metabolism at cold temperatures |
| $f_{\text{Clump}}$ | 1.00 | 1.00 | 0.80 | 0.80 | 0.80 | 0.735 | — | Clumping index |
| $f_{\text{Hot}}$ | 0.40 | 0.40 | 0.40 | 0.40 | 0.40 | 0.40 | — | Decay parameter to down-regulate metabolism at hot temperatures |
| $f_n$ | 0.00 | 0.00 | 0.10 | 0.10 | 0.10 | 0.10 | — | Fraction of carbon storage retained in storage pool |
| $f_r$ | 1.00 | 1.00 | 1.00 | 1.00 | 1.00 | 1.00 | — | Fine-root:leaf biomass ratio |
| $f_R$ | 0.035 | 0.015 | 0.015 | 0.015 | 0.015 | 0.015 | — | Respiration:carboxylation ratio |
| $f_\varrho$ | 1.0 | 1.0 | 0.30 | 0.30 | 0.30 | 0.30 | — | Fraction of carbon allocation to reproduction at maturity |
| $f_\sigma$ | 3900 | 3900 | 3900 | 3900 | 3900 | 3900 | $m^3\,kg_C^{-1}$ | Sapwood:leaf biomass scaling factor |
| $\hat{G}_{lw}^{\varnothing}$ | 0.01 | 0.01 | 0.01 | 0.01 | 0.01 | 0.001 | $\text{mol}\,\text{m}^{-2}\,\text{s}^{-1}$ | Residual conductance (closed stomata) |
| $\hat{G}_r$ | 900 | 900 | 600 | 600 | 600 | 600 | $\text{m}^2\,\text{kg}_C^{-1}\,\text{yr}^{-1}$ | Scaling factor for fine root conductance |
| $\ell_h$ | 1.0 | 1.0 | 1.0 | 1.0 | 1.0 | 0.79 | — | Fraction of lignified tissues (sapwood and hardwood) |
| $\ell_l$ | 0.0 | 0.0 | 0.0 | 0.0 | 0.0 | 0.00 | — | Fraction of lignified tissues (leaves and fine roots) |
| $M$ | 7.2 | 9.0 | 9.0 | 9.0 | 9.0 | 7.2 | — | Slope factor for stomatal conductance |
| $m_\varrho$ | 0.95 | 0.95 | 0.95 | 0.95 | 0.95 | 0.95 | $\text{mo}^{-1}$ | Loss rate of reproductive tissues |
| $Q_{10}(\dot{V}_C)$ | 2.40 | 2.40 | 2.40 | 2.40 | 2.40 | 2.40 | — | Temperature dependence factor for carboxylation rate |

**Table S5: (Continued)**

| Symbol | PFT-specific value | | | | | | Units | Description |
|---|---|---|---|---|---|---|---|---|
| | C4G | C3G | ETR | MTR | LTR | ARC | | |
| $Q_{10}(r_r)$ | 2.40 | 2.40 | 2.40 | 2.40 | 2.40 | 2.40 | — | Temperature dependence factor for fine root respiration |
| $q_l^{(OD)}$ | 3218 | 3218 | 3218 | 3218 | 3218 | 3218 | $J\,kg^{-1}\,K^{-1}$ | Specific heat of oven-dry leaf biomass |
| $q_b^{(OD)}$ | 1217 | 1217 | 1217 | 1217 | 1217 | 1217 | $J\,kg^{-1}\,K^{-1}$ | Specific heat of oven-dry wood biomass |
| $r_{r15}$ | 0.246 | 0.246 | 0.246 | 0.246 | 0.246 | 0.246 | $s^{-1}$ | Fine-root respiration rate at $15\,^\circ$C |
| SLA | 22.70 | 22.70 | 16.02 | 11.65 | 9.66 | 6.32 | $m^2\,kg_C^{-1}$ | Specific leaf area |
| $T_{Cold}$ | 288.15 | 283.15 | 283.15 | 283.15 | 283.15 | 277.86 | K | Cold temperature threshold for metabolic activity |
| $T_F$ | 275.65 | 275.65 | 275.65 | 275.65 | 275.65 | 258.15 | K | Temperature threshold for plant hardiness to frost |
| $T_{Hot}$ | 318.15 | 318.15 | 318.15 | 318.15 | 318.15 | 318.15 | K | Hot temperature threshold for metabolic activity |
| $b$ | 1 | 1 | 1 | 1 | 1 | 2 | — | Number of sides of leaf with stomata |
| $\dot{V}_{C15}^{max}$ | 12.5 | 18.75 | 18.75 | 12.5 | 6.25 | 15.62 | $\mu mol_C\,m^{-2}\,s^{-1}$ | Maximum carboxylation rate at $15\,^\circ$C |
| $x_\beta^\star$ | 0.05 | 0.05 | 0.05 | 0.05 | 0.05 | 0.05 | m | Typical obstacle size for branches and twigs |
| $x_\lambda^\star$ | 0.05 | 0.05 | 0.10 | 0.10 | 0.10 | 0.05 | m | Typical leaf width |
| $\mathcal{Z}_0$ | 0 | 0 | 0 | 0 | 0 | 0 | m | Offset parameter for tree height allometry |
| $\mathcal{Z}_1$ | 0.0352 | 0.0352 | 0.0352 | 0.0352 | 0.0352 | 0.0352 | — | Slope coefficient for leaf biomass allometry |
| $\mathcal{Z}_2$ | 0.694 | 0.694 | 0.694 | 0.694 | 0.694 | 0.694 | — | Exponent coefficient for leaf biomass allometry |
| $\mathcal{Z}_\infty$ | 61.7 | 61.7 | 61.7 | 61.7 | 61.7 | 61.7 | m | Asymptote height (relative to $\mathcal{Z}_0$ at $\lim_{DBH\to\infty}$) |
| $z_{t\,max}$ | 1.5 | 1.5 | 35.0 | 35.0 | 35.0 | 35.0 | m | Maximum attainable height |
| $z_t^{Repro}$ | 1.5 | 1.5 | 18.0 | 18.0 | 18.0 | 18.0 | m | Plant height at reproductive maturity |
| $\Delta q_b^{Bond}$ | 63.10 | 63.10 | 63.10 | 63.10 | 63.10 | 63.10 | $J\,kg^{-1}\,K^{-1}$ | Specific heat associated with bonding between wood and water |
| $\Delta w$ | 0.016 | 0.016 | 0.016 | 0.016 | 0.016 | 0.016 | $mol_w\,mol^{-1}$ | Leaf water deficit down-regulation parameter (stomatal conductance) |
| $\epsilon$ | 0.055 | 0.080 | 0.080 | 0.080 | 0.080 | 0.08 | — | Quantum yield |
| $\rho_t$ | — | — | 0.53 | 0.71 | 0.90 | 0.52 | $g\,cm^{-3}$ | Wood density |
| $\hat\sigma^{FR}$ | 0.0 | 0.0 | 0.0 | 0.0 | 0.0 | 0.00 | — | Survivorship to fire disturbance |
| $\hat\sigma^{TF}(z_t<10\,m)$ | 0.25 | 0.25 | 0.10 | 0.10 | 0.10 | 0.10 | — | Survivorship of small trees to tree fall disturbance |
| $\hat\sigma^{TF}(z_t\geq10\,m)$ | 0.0 | 0.0 | 0.0 | 0.0 | 0.0 | 0.0 | — | Survivorship of large trees to tree fall disturbance |
| $\varsigma_R^{Leaf}$ | (0.100;0.400;0.040) | (0.100;0.400;0.040) | (0.100;0.400;0.050) | (0.100;0.400;0.050) | (0.100;0.400;0.050) | (0.090;0.577;0.030) | — | Leaf reflectance |
| $\varsigma_R^{Wood}$ | (0.160;0.250;0.040) | (0.110;0.250;0.040) | (0.110;0.250;0.100) | (0.110;0.250;0.100) | (0.110;0.250;0.100) | (0.110;0.250;0.100) | — | Wood reflectance |
| $\varsigma_T^{Leaf}$ | (0.050;0.200;0.000) | (0.050;0.200;0.000) | (0.050;0.200;0.000) | (0.050;0.200;0.000) | (0.050;0.200;0.000) | (0.050;0.248;0.000) | — | Leaf transmittance |
| $\varsigma_T^{Wood}$ | (0.028;0.248;0.000) | (0.001;0.001;0.000) | (0.001;0.001;0.000) | (0.001;0.001;0.000) | (0.001;0.001;0.000) | (0.001;0.001;0.000) | — | Wood transmittance |
| $\tau_l$ | 2.0 | 2.0 | 1.0 | 0.50 | 0.33 | 0.042 | $yr^{-1}$ | Leaf turnover rate |
| $\tau_n$ | 0.333 | 0.333 | 0.167 | 0.167 | 0.167 | 0.167 | $yr^{-1}$ | Storage turnover rate |
| $\tau_r$ | 2.0 | 2.0 | 1.0 | 0.50 | 0.33 | 0.042 | $yr^{-1}$ | Fine-root turnover rate |
| $\tau_\Delta$ | 0.333 | 0.333 | 0.333 | 0.333 | 0.333 | 0.450 | $dy^{-1}$ | Growth respiration factor |
| $\chi$ | 0.00 | 0.00 | 0.10 | 0.10 | 0.10 | 0.01 | — | Mean orientation factor |

Table S6: List of default parameters that depend on plant functional type (PFT) used in ED-2.2 for temperate regions, for default ED-2.2 PFTs. The PFTs are C$_3$ temperate grass (C3T); mid-latitude ("Northern") pines (NPN); subtropical ("Southern") pines (SPN); late-successional conifers (LCN), early-successional hardwood tree (EHW); mid-succesional hardwood tree (MHW), late-successional tropical tree (LHW); additional PFTs can be specified by the user and provided directly to ED-2.2 through extensible markup language (XML). Spectral-dependent parameters $x$ are provided as vectors ($x_{PAR}$; $x_{NIR}$; $x_{TIR}$), corresponding to the visible (photosynthetically active), nearinfrared, and thermal infrared, respectively. The default parameters for tropical and subtropical PFTs are shown in Table S3. The values of constants and default global parameters are shown in Table S5.

| Symbol | PFT-specific value | | | | | | | Units | Description |
|---|---|---|---|---|---|---|---|---|---|
| | C3T | NPN | SPN | LCN | EHW | MHW | LHW | | |
| $\alpha$ | 15.15 | 294.74 | 232.56 | 424.30 | 162.76 | 262.61 | 233.64 | yr | Aging factor for density-independent mortality |
| $B_{Wl}$ | 2.50 | 2.50 | 2.50 | 2.50 | 2.50 | 2.50 | 2.50 | — | Water:oven-dry-biomass ratio for leaves |
| $B_{Wb}$ | 0.7 | 0.7 | 0.7 | 0.7 | 0.7 | 0.7 | 0.7 | — | Water:oven-dry-biomass ratio for wood |
| $C_{0l}$ | 0.0800 | 0.0240 | 0.0240 | 0.0454 | 0.0129 | 0.0480 | 0.0170 | — | Scaling coefficient for leaf biomass allometry |
| $C_{1l}$ | 1.0000 | 1.8990 | 1.8990 | 1.6829 | 1.7477 | 1.4550 | 1.7310 | — | Exponent coefficient for leaf biomass allometry |
| $C_{0h}$ | $1\times10^{-5}$ | 0.1470 | 0.1470 | 0.1617 | 0.0265 | 0.1617 | 0.2350 | — | Scaling coefficient for heartwood biomass allometry (sub-canopy) |
| $C_{1h}$ | 1.0000 | 2.2380 | 2.2380 | 2.1536 | 2.9595 | 2.4572 | 2.2518 | — | Exponent coefficient for heartwood biomass allometry (sub-canopy) |
| $C_{2h}$ | $1\times10^{-5}$ | 0.1470 | 0.1470 | 0.1617 | 0.0265 | 0.1617 | 0.2350 | — | Scaling coefficient for heartwood biomass allometry (canopy) |
| $C_{3h}$ | 1.0000 | 2.2380 | 2.2380 | 2.1536 | 2.9595 | 2.4572 | 2.2518 | — | Exponent coefficient for heartwood biomass allometry (canopy) |
| $f_{AG}$ | 0.70 | 0.70 | 0.70 | 0.70 | 0.70 | 0.70 | 0.70 | — | Fraction of above-ground biomass |
| $f_{Cold}$ | 0.40 | 0.40 | 0.40 | 0.40 | 0.40 | 0.40 | 0.40 | — | Decay parameter to down-regulate metabolism at cold temperatures |
| $f_{Clump}$ | 0.840 | 0.735 | 0.735 | 0.735 | 0.840 | 0.840 | 0.840 | — | Clumping index |
| $f_{Hot}$ | 0.40 | 0.40 | 0.40 | 0.40 | 0.40 | 0.40 | 0.40 | — | Decay parameter to down-regulate metabolism at hot temperatures |
| $f_n$ | 0.00 | 0.00 | 0.00 | 0.00 | 0.00 | 0.00 | 0.00 | — | Fraction of carbon storage retained in storage pool |
| $f_r$ | 1.0000 | 0.3463 | 0.3463 | 0.3463 | 1.1274 | 1.1274 | 1.1274 | — | Fine-root:leaf biomass ratio |
| $f_R$ | 0.02 | 0.02 | 0.02 | 0.02 | 0.02 | 0.02 | 0.02 | — | Respiration:carboxylation ratio |
| $f_\varrho$ | 1.0 | 1.0 | 0.30 | 0.30 | 0.30 | 0.30 | 0.30 | — | Fraction of carbon allocation to reproduction at maturity |
| $f_\sigma$ | 3900 | 3900 | 3900 | 3900 | 3900 | 3900 | 3900 | $m^3\,kg_C^{-1}$ | Sapwood:leaf biomass scaling factor |
| $\hat{G}_{lw}^{\varnothing}$ | 0.020 | 0.001 | 0.001 | 0.001 | 0.020 | 0.020 | 0.020 | $molm^{-2}\,s^{-1}$ | Residual conductance (closed stomata) |
| $\hat{G}_r$ | 160 | 150 | 150 | 150 | 150 | 150 | 150 | $m^2\,kg_C^{-1}\,yr^{-1}$ | Scaling factor for fine root conductance |
| $\pounds_h$ | 1.00 | 0.79 | 0.79 | 0.79 | 0.79 | 0.79 | 0.79 | — | Fraction of lignified tissues (sapwood and hardwood) |
| $\pounds_l$ | 0.0 | 0.0 | 0.0 | 0.0 | 0.0 | 0.0 | 0.0 | — | Fraction of lignified tissues (leaves and fine roots) |
| $M$ | 8.0 | 6.4 | 6.4 | 6.4 | 6.4 | 6.4 | 6.4 | — | Slope factor for stomatal conductance |
| $m_\varrho$ | 0.95 | 0.95 | 0.95 | 0.95 | 0.95 | 0.95 | 0.95 | $mo^{-1}$ | Loss rate of reproductive tissues |
| $Q_{10}(\dot{V}_C)$ | 2.40 | 2.40 | 2.40 | 2.40 | 2.40 | 2.40 | 2.40 | — | Temperature dependence factor for carboxylation rate |
| $Q_{10}(r_r)$ | 2.40 | 2.40 | 2.40 | 2.40 | 2.40 | 2.40 | 2.40 | — | Temperature dependence factor for fine root respiration |
| $q_l^{(OD)}$ | 3218 | 3218 | 3218 | 3218 | 3218 | 3218 | 3218 | $Jkg^{-1}\,K^{-1}$ | Specific heat of oven-dry leaf biomass |
| $q_b^{(OD)}$ | 1217 | 1217 | 1217 | 1217 | 1217 | 1217 | 1217 | $Jkg^{-1}\,K^{-1}$ | Specific heat of oven-dry wood biomass |
| $r_{15}$ | 0.246 | 0.246 | 0.246 | 0.246 | 0.246 | 0.246 | 0.246 | $s^{-1}$ | Fine-root respiration rate at 15°C |

**Table S6: (Continued)**

| Symbol | PFT-specific value | | | | | | | Units | Description |
|---|---|---|---|---|---|---|---|---|---|
| | C3T | NPN | SPN | LCN | EHW | MHW | LHW | | |
| SLA | 22.70 | 6.0 | 9.0 | 10.0 | 30.0 | 24.2 | 60.0 | $m^2\,kg_C^{-1}$ | Specific leaf area |
| $T_{Cold}$ | 277.86 | 277.86 | 277.86 | 277.86 | 277.86 | 277.86 | 277.86 | K | Cold temperature threshold for metabolic activity |
| $T_F$ | 193.15 | 193.15 | 263.15 | 213.15 | 193.15 | 253.15 | 253.15 | K | Temperature threshold for plant hardiness to frost |
| $T_{Hot}$ | 318.15 | 318.15 | 318.15 | 318.15 | 318.15 | 318.15 | 318.15 | K | Hot temperature threshold for metabolic activity |
| $b$ | 1 | 2 | 2 | 2 | 1 | 1 | 1 | — | Number of sides of leaf with stomata |
| $V_{C15}^{max}$ | 18.30 | 11.35 | 11.35 | 4.54 | 20.39 | 17.45 | 6.98 | $\mu mol_C\,m^{-2}\,s^{-1}$ | Maximum carboxylation rate at 15 °C |
| $x_\beta^\star$ | 0.05 | 0.05 | 0.05 | 0.05 | 0.05 | 0.05 | 0.05 | m | Typical obstacle size for branches and twigs |
| $x_\lambda^\star$ | 0.05 | 0.05 | 0.05 | 0.05 | 0.05 | 0.05 | 0.05 | m | Typical leaf width |
| $z_0$ | 0.0 | 1.3 | 1.3 | 1.3 | 1.3 | 1.3 | 1.3 | m | Offset parameter for tree height allometry |
| $z_1$ | 0.7500 | 0.0388 | 0.0388 | 0.0444 | 0.0653 | 0.0496 | 0.0540 | — | Slope coefficient for leaf biomass allometry |
| $z_2$ | 1 | 1 | 1 | 1 | 1 | 1 | 1 | — | Exponent coefficient for leaf biomass allometry |
| $z_\infty$ | 0.478 | 27.140 | 27.140 | 22.790 | 22.680 | 25.180 | 23.387 | m | Asymptote height (relative to $Z_0$ at $\lim_{DBH\to\infty}$) |
| $z_t^{max}$ | 0.454 | 27.113 | 27.113 | 22.767 | 22.657 | 25.155 | 23.364 | m | Maximum attainable height |
| $z_t^{Repro}$ | 0.45 | 18.0 | 18.0 | 18.0 | 18.0 | 18.0 | 18.0 | m | Plant height at reproductive maturity |
| $\Delta q_b^{Bond}$ | 63.10 | 63.10 | 63.10 | 63.10 | 63.10 | 63.10 | 63.10 | $J\,kg^{-1}\,K^{-1}$ | Specific heat associated with bonding between wood and water |
| $\Delta w$ | 0.016 | 0.016 | 0.016 | 0.016 | 0.016 | 0.016 | 0.016 | $mol_w\,mol^{-1}$ | Leaf water deficit down-regulation parameter (stomatal conductance) |
| $\epsilon$ | 0.080 | 0.080 | 0.080 | 0.080 | 0.080 | 0.080 | 0.080 | — | Quantum yield |
| $\rho_l$ | — | — | — | — | — | — | — | $g\,cm^{-3}$ | Wood density (currently not used) |
| $\hat\sigma^{FR}$ | 0.0 | 0.0 | 0.0 | 0.0 | 0.0 | 0.0 | 0.0 | — | Survivorship to fire disturbance |
| $\hat\sigma^{TF}(z_t < 10\,m)$ | 0.25 | 0.10 | 0.10 | 0.10 | 0.10 | 0.10 | 0.10 | — | Survivorship of small trees to tree fall disturbance |
| $\hat\sigma^{TF}(z_t \geq 10\,m)$ | 0.0 | 0.0 | 0.0 | 0.0 | 0.0 | 0.0 | 0.0 | — | Survivorship of large trees to tree fall disturbance |
| $\varsigma_R^{Leaf}$ | (0.110;0.577,0.040) | (0.110;0.577,0.030) | | | (0.110;0.577,0.050) | | | — | Leaf reflectance |
| $\varsigma_R^{Wood}$ | (0.160;0.110,0.040) | (0.110;0.250;0.100) | | | | | | — | Wood reflectance |
| $\varsigma_T^{Leaf}$ | | (0.160;0.248;0.000) | | | | | | — | Leaf transmittance |
| $\varsigma_T^{Wood}$ | (0.028;0.248;0.000) | (0.001;0.001;0.000) | | | | | | — | Wood transmittance |
| $\tau_l$ | 2.0 | 0.333 | 0.333 | 0.333 | — | — | — | $yr^{-1}$ | Leaf turnover rate (not applicable for hardwoods, which are deciduous) |
| $\tau_n$ | 0.000 | 0.000 | 0.000 | 0.000 | 0.624 | 0.624 | 0.624 | $yr^{-1}$ | Storage turnover rate |
| $\tau_r$ | 2.0 | 3.927 | 4.118 | 3.800 | 5.773 | 5.083 | 5.071 | $yr^{-1}$ | Fine-root turnover rate |
| $\tau_\Delta$ | 0.333 | 0.450 | 0.450 | 0.450 | — | — | — | $dy^{-1}$ | Growth respiration factor (not applicable for hardwoods, which are deciduous) |
| $\chi$ | 0.00 | 0.00 | 0.00 | 0.00 | 0.00 | 0.00 | 0.00 | — | Mean orientation factor |

Table S7: List of soil component properties (air, sand, silt, and clay), used to derive most soil-texture dependent properties. Most parameters are based on Monteith and Unsworth (2008); values for silt were unavailable and assumed to be intermediate between sand and clay. The volumetric fractions of the default soil texture types in ED-2.2 are listed in Table S8.

| Symbol | Soil components | | | | Units | Description |
| | Air | Sand | Silt | Clay | | |
| --- | --- | --- | --- | --- | --- | --- |
| $q$ | 1010 | 800 | 850 | 900 | $J\,kg^{-1}\,K^{-1}$ | Specific heat |
| $\rho$ | 1.200 | 2660 | 2655 | 2650 | $kg\,m^{-3}$ | Bulk density |
| $\Upsilon_Q$ | 0.025 | 8.80 | 5.87 | 2.92 | $W\,m^{-1}\,K^{-1}$ | Thermal conductivity |

Table S8: List of volumetric fractions of sand, silt, and clay ($f_V$) for the default soil texture types in ED-2.2 (Fig. S6). Component-specific properties of soils are listed in Table S7.

| Class | Description | Volumetric fractions | | |
| --- | --- | --- | --- | --- |
| | | Sand | Silt | Clay |
| Sa | Sand | 0.920 | 0.050 | 0.030 |
| LSa | Loamy sand | 0.825 | 0.115 | 0.060 |
| SaL | Sandy loam | 0.660 | 0.230 | 0.110 |
| SiL | Silt loam | 0.200 | 0.640 | 0.160 |
| L | Loam | 0.410 | 0.420 | 0.170 |
| SaCL | Sandy clay loam | 0.590 | 0.140 | 0.270 |
| SiCL | Silty clay loam | 0.100 | 0.560 | 0.340 |
| CL | Clayey loam | 0.320 | 0.340 | 0.340 |
| SaC | Sandy clay | 0.520 | 0.060 | 0.420 |
| SiC | Silty clay | 0.060 | 0.470 | 0.470 |
| C | Clay | 0.200 | 0.200 | 0.600 |
| Si | Silt | 0.075 | 0.875 | 0.050 |
| CC | Heavy clay | 0.100 | 0.100 | 0.800 |
| CSa | Clayey sand | 0.375 | 0.100 | 0.525 |
| CSi | Clayey silt | 0.125 | 0.350 | 0.525 |

Table S9: Coefficients used in Eq. (S191) for each limitation and photosynthetic path. The special case in which the stomata are closed is also shown for reference.

| Case | C$_3$ photosynthesis | | | | C$_4$ photosynthesis | | | |
|---|---|---|---|---|---|---|---|---|
| | $F^A$ | $F^B$ | $F^C$ | $F^D$ | $F^A$ | $F^B$ | $F^C$ | $F^D$ |
| Closed stomata ($\dot{A}_k^{\varnothing}$) | 0 | 0 | 0 | 1 | 0 | 0 | 0 | 1 |
| RuBP-saturated ($\dot{A}_k^{\text{RuBP}}$) | $\dot{V}_{C_k}^{\max}$ | $-\dot{V}_{C_k}^{\max}\Gamma_k$ | 1 | $\mathcal{K}_{\text{ME}_k}$ | 0 | $\dot{V}_{C_k}^{\max}$ | 0 | 1 |
| CO$_2$-limited ($\dot{A}_k^{\text{InSl}}$) | $\dot{V}_{C_k}^{\max}$ | $-\dot{V}_{C_k}^{\max}\Gamma_k$ | 1 | $\mathcal{K}_{\text{ME}_k}$ | $k_{\text{PEP}}\dot{V}_{C_k}^{\max}$ | 0 | 0 | 1 |
| Light-limited ($\dot{A}_k^{\text{PAR}}$) | $\epsilon_k\,\dot{q}_k$ | $-\epsilon_k\,\dot{q}_k\Gamma_k$ | 1 | $2\Gamma_k$ | 0 | $\epsilon_k\,\dot{q}_k$ | 0 | 1 |

**S1 ED-2 developments since ED-2.0 and ED-2.2**

In this Supplement, we list the main developments in the Ecosystem Demography Model version 2 (ED-2), with focus on mentioned in this manuscript (Fig. S3). The complete list of implementations, improvements, and code fixes are available on the GitHub website (`https://github.com/EDmodel/ED2`).

**S1.1 Version 2.0 (ED-2.0)**

This is the version described in Medvigy (2006); Medvigy et al. (2009), and it is the first version of the ED model that implements energy and water cycles at sub-daily scale. The biophysics core was adapted from the LEAF-2 land surface model (Walko et al., 2000), which is part of the Regional Atmospheric Model System (RAMS). The main differences in the ED-2.0 biophysics core include (1) solution of the energy and water cycle for each cohort and patch; (2) use of $4^{th}$ order Runge-Kutta solver to improve numerical stability. In addition, this version allowed leaf phenology to be prescribed from external data (Supplement S3.1.3). The photosynthesis solver was largely the same as in ED-1.0 (Moorcroft et al., 2001).

**S1.2 Version 2.0.12 (ED-2.0.12)**

Most developments between ED-2.0 and ED-2.0.12 relate to code organization and structure. ED-2.0 was partly written in C (legacy from ED-1) and partly written in Fortran (legacy from LEAF-2). To simplify the code and ensure data were correctly transferred between subroutines, we rewrote most of the code in Fortran. The only exceptions were a few file handling functions that remained in C because we could not find equivalent functions in Fortran.

In addition, this version uses Hierarchical Data Format 5 (HDF5) format and libraries (The HDF Group, 2016) to generate model outputs. HDF5 allows a more efficient framework to output variables in the dynamic patch and cohort structures. It also introduced an XML model parameter input file, rather than relying solely on hard-coded defaults, which makes it easier to perform model calibration, sensitivity analyses, and ensemble error propagation. Importantly, this was the last version of ED-2 that used temperature as prognostic variable for leaves and canopy air space.

**S1.3 Version 2.1 (ED-2.1)**

Most ED-2.1 developments aimed at improving the energy cycle representation in ED-2.1. Leaf enthalpy and canopy air space enthalpy replaced temperature as the prognostic variables (Eq. 4; Sec. 3.2.3-3.2.4). The main advantages of energy-related prognostic equations include: (1) simplification the numeric integration, as total energy changes must be equivalent to net energy flux; (2) improved conservation of energy when water fluxes are large and cause rapid changes in heat capacity of the thermodynamic systems; (3) elimination of singularity at the water's fusion point ($0\,°C$), when enthalpy changes due to freezing or melting, but the temperature remains the same.

To ensure the model was thermodynamically consistent, we also: (1) implemented a mechanistic representation of heat capacity for vegetation (leaves and branches, Supplement S6.2) that is scaled with leaf and branch biomass (e.g. Dufour and van Mieghem, 1975); (2) replaced the original LEAF-2-based surface layer model (that was based on Louis, 1979) with the parameterization by Beljaars and Holtslag (1991), as the latter parameterization improved numerical stability of eddy covariance fluxes under thermally stable conditions; (3) included an option to prescribe silt, clay, and sand fractions to define site-specific soil texture characteristics (Supplement S9) instead of the original ED-2.0 implementation that required soils to be assigned to one of the 12 fixed classes originally defined in LEAF-2 (Walko et al., 2000); (4) implemented the capability of saving the entire ecosystem and thermodynamic state of the model into HDF5 files, which can be used to stop and start simulations and yield the same results of uninterrupted simulations, a desirable feature for simulations with long runtimes.

**S1.4 Version 2.2 (ED-2.2)**

The ED-2.2 version implemented several improvements and fixed inconsistencies in the representation of the energy, water, and carbon dioxide cycles. First, we redefined enthalpy (S5), to ensure that it would be a true thermodynamic state variable (i.e. path independent, see Dufour and van Mieghem, 1975), by making latent heat of vaporization a linear function of temperature (Eq. 72-73). Moreover, we identified missing components of the energy cycle that precluded the conservation: (1) the transfer of internal energy from soils to leaves before transpiration (Eq. 97); (2) the enthalpy exchange associated with vaporization and condensation also accounts the mass transfer of water between the thermodynamic systems (e.g. Eq. 75; 98). Furthermore, to ensure results from ED-2.2 consistently conserve mass and energy, we implemented detailed conservation verification during the model execution, which now reports any violation of energy, water, and carbon conservation, generates detailed output of the violation, and interrupts the simulation. Finally, to improve computational efficiency of the energy, water, and carbon cycle solvers at sub-daily time steps, we implemented a shared-memory parallelization of the most computationally-intensive subroutines. The parallelization was written to allow users to select any number of cores (depending on core availability), and it accounts for patch ages in order to balance the load among cores.

In addition, we rewrote the photosynthesis to allow temperature-dependent functions to be expressed as functions of $Q_{10}$. We retained the original Arrhenius-based functions as legacy options, but the new option increases the options for assimilating data into the model. The current $Q_{10}$-based parameters fix the low-temperature optimum in tropical plants previously noted by Rogers et al. (2017). Importantly, we rewrote the photosynthesis solver to ensure that it would always converge to a unique solution for net assimilation rate, stomatal conductance, and intercellular carbon dioxide concentration given the environmental conditions (Supplement S16).

The ED-2.2 version also includes improvements in the representation of conductances between different thermodynamic systems. First, the leaf boundary-layer conductance now accounts for differences in leaf and branch characteristics of each cohort, and to account for both free and forced convection under both laminar and turbulent flow (Supplement S14.2). Second, we implemented ground-to-canopy conductance formulations (Sellers et al., 1986; Massman, 1997; Massman and Weil, 1999) that account for the cumulative drag profile of vegetated areas obtained from the cohort structure, as well as the stability of the surface layer (Supplement S14.3).

Finally, in ED-2.2 we replaced the version control to GitHub, which makes the new code developments readily available to the scientific community and encourages users to post issues, code fixes and model improvements and developments to the main code repository in open and collaborative forums.

**S2 Boundary conditions for the ecosystem dynamics equations**

The boundary conditions for Eq. (2) and (3) are:

$$\underbrace{n_{fq}\left(\mathbf{C}_{f_0},a,t\right)}_{\text{Recruit}} = \frac{1}{\mathbf{g}_{f_0}\cdot\mathbf{1}}\left\{\underbrace{\int_{\mathbf{C}_{\mathbf{f_0}}}^{\infty}\left(1-f_{\delta_f}\right)\,\varrho_f\,n_{fq}\mathrm{d}\mathbf{C}}_{\text{Local recruitment}}+\underbrace{\sum_{q'=1}^{N_Q}\left[\int_{\mathbf{C}_{\mathbf{f_0}}}^{\infty}\int_0^{\infty}f_{\delta_f}\,\varrho_f\,n_{FQ'X'Y}\alpha_{fq'}\,\mathrm{d}a\,\mathrm{d}\mathbf{C}\right]}_{\text{Non-local, random dispersal}}\right\},$$

(S1)

$$\underbrace{n_{fq}\left(\mathbf{C}_f,0,t\right)}_{\text{Population at new gap}} = \underbrace{\sum_{q'=1}^{N_Q}\left[\int_0^{\infty}\hat{\sigma}_{fq'}\,n_{fq'}\,\alpha_{q'}\,\mathrm{d}a\right]}_{\text{Disturbance Survivors}},$$

(S2)

$$\underbrace{\alpha_q(0,t)}_{\text{Probability of new gap}} = \underbrace{\sum_{q'=1}^{N_Q} \lambda_{q'q} \alpha_{q'} \mathrm{d}a,}_{\text{Disturbance rates}} \tag{S3}$$

where $\mathbf{C}_{f_0}$ is the size of the smallest individual of PFT $f$; $\mathbf{g}_{f_0}$ is the growth rate for individuals of PFT $f$ with size $\mathbf{C}_{f_0}$ ; $\mathbf{1}$ is the unity vector for size; $\varrho_f$ is the recruitment rate, which depends on the PFT, size, and carbon balance; $f_{\delta_f}$ is the fraction of recruits of PFT $f$ that are randomly dispersed instead of locally recruited; and $\hat{\sigma}_{fq}$ is size-dependent survivorship probability for a PFT $f$ following a disturbance of type $q$ (for a complete list of subscripts and variable meanings, refer to Tables S1 and S2). Both $\mathbf{g}_f$ and $m_f$ are functions of the plant size and the individual's carbon balance. The individual's carbon balance depends on the environment perceived by each individual; in turn, the environment perceived by each individual is modulated by both the plant community living in the same gap and the general landscape environment. Likewise, the disturbance rates may be affected by the local plant community in the gap and the regional landscape environment.

**S3 Long-term carbon dynamics and relation with carbon balance**

**S3.1 Leaf phenology**

The phenological strategy of the plant functional types~ can be evergreen, drought-deciduous, or cold-deciduous. The plant's phenology strategy is defined by two functions: (i) the leaf elongation factor ($\hat{e}_{l_k}$), defined as the ratio between the environmentally-constrained leaf biomass and the potential (maximum) leaf biomass, and the rate of leaf shedding ($\omega_{l_k}(t)$) which can either be prognosed, or prescribed from observations.

**S3.1.1 Evergreen plants**

For evergreen PFTs, the elongation factor is always 1, the rate of leaf shedding ($\omega_{l_k}(t)$) is zero, and their rate of leaf turnover is governed by the PFT-dependent leaf turnover parameter ($\tau_{l_k}$, see Eq. S12, and Tables S5–S6). The leaf phenology of tropical trees can also be represented by an empirical model that is driven by the seasonality of light availability (see Kim et al., 2012).

**S3.1.2 Drought-deciduous tropical phenology**

The drought-deciduous phenology assumes that leaf flushing and leaf senescence are controlled by the water availability in the rooting zone. The elongation factor $\hat{e}_{l_k}$ is determined by the following parameterization:

$$
\hat{e}_{l_k} =
\begin{cases}
1 & , \text{if } s_{l_k} \geq 1 \\
s_{l_k} & , \text{if } 0.05 \leq s_{l_k} < 1 , \\
0 & , \text{if } s_{l_k} < 0.05
\end{cases}
\tag{S4}
$$

$$
s_{l_k} = \frac{1}{|z_{r_k}| \, \Delta t_{\text{El}}} \int_{t'-\Delta t_{\text{El}}}^{t'} \left( \sum_{j=j(z_{r_k})}^{N_G} \left\{ \frac{\max\left[0, \Psi_{g_j}(t') + \frac{1}{2}\left(z_{g_j} + z_{g_{j+1}}\right) - \Psi_{\text{Wp}}\right]}{\Psi_{\text{Ld}} - \Psi_{\text{Wp}}} \right\} \right) dt,
\tag{S5}
$$

5 where $s_{l_k}$ is a 10-day running average of soil moisture accessed by cohort $k$ (normalized by the difference between the water potential threshold and the wilting point), $z_{r_k}$ is the rooting depth of cohort $k$ (Supplement S18), $\Delta t_{\text{El}}$ is the time scale for changes in phenology (assumed to be 10 days), $j(z_{r_k})$ is the soil layer containing the deepest roots of cohort $k$, $\Psi_{g_j}$ is the soil matric potential at soil layer $j$, $\Psi_{\text{Ld}}$ is the soil matric potential below which plants start shedding leaves 10 (assumed $-1.2$ MPa), $\Psi_{\text{Wp}}$ is the soil matric potential at the wilting point, and $z_{g_j}$ is the depth of soil layer $j$ ($z_{g_{N_G+1}} \equiv 0$, otherwise $z_g$ is negative). Leaf shedding occurs whenever soil is drier than the threshold defined by $\Psi_{\text{Ld}}$ and drought conditions are increasing. Specifically:

$$
\omega_{l_k} = \frac{1}{\Delta t_{\text{Phen}}} \max\left[0, \frac{C_{l_k}}{C_{l_k}^{\bullet}} - f_{\text{El}}\right].
\tag{S6}
$$

**S3.1.3 Cold-deciduous phenology**

The prognostic cold-deciduous leaf phenology approach is a thermal sum and chilling sum-based 15 model identical to that of Albani et al. (2006), which, in turn, is based on Botta et al. (2000). At each patch, growing degree-days (GDD) are accumulated during the extended growing season ($\mathfrak{T}_{\text{GS}}$, January–August for the Northern Hemisphere, and July–February for the Southern Hemisphere), and the chilling days (CHD) in the extended senescing season ($\mathfrak{T}_{\text{SS}}$, November–June for the Northern Hemisphere, and May–December for the Southern Hemisphere):

$$
\text{GDD}(t) =
\begin{cases}
0 & , \text{if } t \notin \mathfrak{T}_{\text{GS}} \\
\sum_{t'=t_{\text{GS}}(0)}^{t} \max\left(0, \overline{T}_c(t') - T_{\text{Phen}}\right) & , \text{otherwise}
\end{cases},
\tag{S7}
$$

$$\text{CHD}(t) = \begin{cases} 0 & \text{, if } \overline{T}_c(t) \geq T_{\text{Phen}}, \text{ or } t \notin \mathfrak{T}_{\text{SS}} \\ \text{CHD}(t - \Delta t_{\text{Phen}}) + 1 & \text{, otherwise} \end{cases}, \tag{S8}$$

where $\overline{T}_c$ is the daily average canopy air space temperature, $\Delta t_{\text{Phen}} = 1$ day is the phenology time step (Table 2), $t_{\text{GS}}(0)$ is the beginning of the growing season, and $T_{\text{Phen}} = 278.15$ K ($5\,^\circ$C) is the leaf phenology threshold (Albani et al., 2006). The valued elongation factor $\hat{e}_{l_k}$ is then determined by the following series of conditions:

$$\hat{e}_{l_k}(t) = \begin{cases} 0 & \text{, if } \overline{T}_s(t) < 275.15 \text{ K} \\ 0 & \text{, if } \overline{T}_s(t) < 284.30 \text{ K and } t_\odot < 655 \text{ min} \\ 1 & \text{, if GDD} \geq -68.0 + 638.0 \exp\left[-0.01\,\text{CHD}(t)\right] \\ \hat{e}_{l_k}(t - \Delta t_{\text{Phen}}) & \text{, otherwise} \end{cases}, \tag{S9}$$

where $t_\odot$ is the daytime duration.

If desired, cold-deciduous phenology can be prescribed rather than prognosed, as described in Medvigy et al. (2009) and Viskari et al. (2015). The timing of leaf onset and leaf senescence are empirically determined from either field observations or from remote sensing (e.g. Zhang et al., 2003) by fitting the following curves, which are then used to determine $\hat{e}_{l_k}$ in the model:

$$\hat{e}_{l_k} = \begin{cases} \dfrac{1}{1 + (y_0 t)^{y_1}} & \text{, if } t \in \mathfrak{T}_{\text{GS}} \\ \dfrac{1}{1 + (y_2 t)^{y_3}} & \text{, if } t \in \mathfrak{T}_{\text{SS}} \end{cases}, \tag{S10}$$

where $y_0$, $y_1$, $y_2$, and $y_3$ are empirical parameters, determined from data prior to running the ED-2.2 model and provided to the model as inputs; $t$ is the time, provided as day of year (i.e. 1 for January $1^{\text{st}}$, 365 for December 31 in non-leap years, and 366 for December 31 in leap years); $\mathfrak{T}_{\text{GS}}$ is the extended growing season (e.g. January–July for the Northern Hemisphere, July–January for the Southern Hemisphere); and $\mathfrak{T}_{\text{SS}}$ is the senescing season (e.g. August–December for the Northern Hemisphere, February–June for the Southern Hemisphere).

**S3.2 Carbon allocation to living tissues and non-structural carbon**

The accumulated carbon balance ($C_{\Delta_k}$, Eq. 25) over the phenology time step $\Delta t_{\text{Phen}}$ is used to update the non-structural carbon storage ($C_{n_k}$) as well as the changes in carbon stocks of living tissues (leaves: $C_{l_k}$; fine roots $C_{r_k}$ and sapwood $C_{\sigma_k}$) due to carbon allocation, turnover losses, and phenology. Changes in living tissues and non-structural carbon are interdependent and described

by the following system of equations (see also Medvigy et al., 2009; Kim et al., 2012):

$$\frac{dC_{n_k}}{dt} = \frac{1}{\Delta t_{\text{Phen}}} \left[ \int_{t-\Delta t_{\text{Phen}}}^{t} \frac{dC_{\Delta_k}}{dt} \, dt' \right] + \left( f_{\text{LD}} \, \omega_{l_k} - \gamma_{l_k} \right) C_{l_k} - \gamma_{r_k} C_{r_k} - \gamma_{\sigma_k} C_{\sigma_k} - \tau_{n_k} C_{n_k}, \quad \text{(S11)}$$

$$\frac{dC_{l_k}}{dt} = \left( \gamma_{l_k} - \tau_{l_k} - \omega_{l_k} \right) C_{l_k}, \quad \text{(S12)}$$

$$\frac{dC_{r_k}}{dt} = \left( \gamma_{r_k} - \tau_{r_k} \right) C_{r_k}, \quad \text{(S13)}$$

$$\frac{dC_{\sigma_k}}{dt} = \gamma_{\sigma_k} C_{\sigma_k}, \quad \text{(S14)}$$

where $\hat{e}_{l_k}$ is the elongation factor (Supplement S3.1); $f_{\text{LD}}$ is the fraction of carbon retained from active leaf drop as storage, currently assumed to be 0.5; $(\gamma_{l_k}; \gamma_{r_k}; \gamma_{\sigma_k})$ are the growth rates of leaves, fine roots, and sapwood, respectively; $(\tau_{l_k}; \tau_{r_k}; \tau_{n_k})$ are the background turnover rates of leaves, fine roots, and non-structural carbon, and are typically assumed constant (Tables S5-S6; but see Kim et al., 2012); and $\omega_{l_k}$ is the phenology-driven leaf shedding rate (Supplement S3.1).

The allocation to living tissues depends on whether the plant carbon balance and environmental conditions are favorable for growing, and it is proportional to the amount of carbon needed by each pool to reach the expected carbon stock given size and environmental constrains (Supplement S18). First, let $\left( C_{l_k}^{\odot}; C_{r_k}^{\odot}; C_{\sigma_k}^{\odot} \right)$ be the biomass increment needed to bring leaves, fine roots, and sapwood, respectively to the expected carbon stock given the plant size and PFT $\left( C_{l_k}^{\bullet}; C_{r_k}^{\bullet}; C_{\sigma_k}^{\bullet} \right)$:

$$C_{l_k}^{\odot} = \max \left[ 0, \hat{e}_{l_k} C_{l_k}^{\bullet} - C_{l_k} \left( 1 - \tau_{l_k} \Delta t_{\text{Phen}} \right) \right], \quad \text{(S15)}$$

$$C_{r_k}^{\odot} = \max \left[ 0, C_{r_k}^{\bullet} - C_{r_k} \left( 1 - \tau_{r_k} \Delta t_{\text{Phen}} \right) \right], \quad \text{(S16)}$$

$$C_{\sigma_k}^{\odot} = \max \left[ 0, C_{\sigma_k}^{\bullet} - C_{\sigma_k} \right], \quad \text{(S17)}$$

$$C_{\alpha_k}^{\odot} = C_{l_k}^{\odot} + C_{r_k}^{\odot} + C_{\sigma_k}^{\odot}, \quad \text{(S18)}$$

where $C_{\alpha_k}^{\odot}$ is the biomass increment needed to bring all living tissues to expected biomass given size and PFT, and $\Delta t_{\text{Phen}}$ is the phenology time step (Table 2). Growth rates of leaves ($\gamma_{l_k}$), fine roots ($\gamma_{r_k}$) and sapwood ($\gamma_{\sigma_k}$) are proportional to the amount needed by each tissue to be brought back to the expected biomass given size and PFT, but also constrained by the amount of non-structural carbon ($C_{n_k}$) available:

$$\gamma_{l_k} = \max \left\{ 0, \frac{1}{\Delta t_{\text{Phen}}} \frac{\hat{e}_{l_k} C_{l_k}^{\odot}}{C_{\alpha_k}^{\odot}} \min \left[ \cancel{\hat{e}_{l_k}} C_{\alpha_k}^{\odot}, C_{n_k} \left( 1 - \tau_{n_k} \right) + C_{\Delta_k} \right] \right\}, \quad \text{(S19)}$$

$$\gamma_{r_k} = \max\left\{0, \frac{1}{\Delta t_{\text{Phen}}} \frac{C_{r_k}^{\odot}}{C_{\alpha_k}^{\odot}} \min\left[\cancel{\omega_{l_k}} C_{\alpha_k}^{\odot}, C_{n_k}(1-\tau_{n_k}) + C_{\Delta_k}\right]\right\}, \tag{S20}$$

$$\gamma_{\sigma_k} = \max\left\{0, \frac{1}{\Delta t_{\text{Phen}}} \frac{C_{\sigma_k}^{\odot}}{C_{\alpha_k}^{\odot}} \min\left[\cancel{\omega_{l_k}} C_{\alpha_k}^{\odot}, C_{n_k}(1-\tau_{n_k}) + C_{\Delta_k}\right]\right\}. \tag{S21}$$

When the cohorts are actively shedding leaves due to phenology, $(\gamma_{l_k}; \gamma_{r_k}; \gamma_{\sigma_k})$ are assumed to be zero. In case carbon balance is sufficiently negative to consume the entire non-structural carbon pool, carbon stocks of living tissues will be depleted and mortality rates will increase (Supplement S3.4).

**S3.3 Carbon allocation to structural tissues and reproduction**

Growth of structural $(C_{h_k})$ and reproductive $(C_{\varrho_k})$ tissues are calculated at the cohort dynamics time step ($\Delta t_{\text{CD}}$, Table 2), after the biomass of living tissues and phenology have been updated:

$$C_{h_k}(t) = C_{h_k}(t - \Delta t_{\text{CD}}) + \gamma_{h_k} C_{n_k}(t) \Delta t_{\text{CD}}, \tag{S22}$$

$$C_{\varrho_k}(t) = \varrho_{t_k} C_{n_k}(t), \tag{S23}$$

$$\gamma_{h_k} = \frac{1}{\Delta t_{\text{CD}}} - \varrho_{t_k} - \gamma_{n_k}, \tag{S24}$$

$$\varrho_{t_k} = \frac{1}{\Delta t_{\text{CD}}} \begin{cases} 0.0 & , \text{if } z_{t_k} < z_{t_k}^{\text{Repro}} \text{ or } \omega_{l_k} > 0 \\ f_\varrho & , \text{otherwise} \end{cases}, \tag{S25}$$

$$\gamma_{n_k} = \frac{1}{\Delta t_{\text{CD}}} \begin{cases} 1.0 & , \text{if } \omega_{l_k} > 0 \\ f_n & , \text{otherwise} \end{cases}, \tag{S26}$$

where $z_{t_k}$ is the cohort height (Supplement S18); $z_{t_k}^{\text{Repro}}$ is the minimum height for reproduction, currently defined as the maximum height for grasses and 18 m for  trees (based on Wright et al., 2005); $f_\varrho$ is the fraction of carbon storage allocated for reproduction when trees are above minimum reproductive height, currently defined as 1.0 for grasses and 0.3 for tropical and temperate trees (Moorcroft et al., 2001); $f_n$ is the fraction of carbon storage that is kept as storage, currently assumed to be 0 for grasses and temperate trees, and 0.1 for tropical trees; and $\omega_{l_k}$ is the phenology-driven leaf shedding rate (Supplement S3.1). The total reproduction biomass $C_{\varrho_k}$ is transferred either to the patches' seed bank or to the soil carbon pools. The fraction that is transferred to the soil carbon pools is defined in terms of a mortality factor ($m_{\varrho_k}$), by default equivalent to 95% in a month, which accounts for both the allocation to reproductive accessories (fruits, flowers, or cones), which are eventually lost, and the seedling mortality rate; the remainder $(1 - m_{\varrho_k})$ is transferred to the seed bank. Carbon storage $C_{n_k}$ is updated after carbon allocation

to structural carbon and reproduction.

**S3.4 Mortality rates**

Following Moorcroft et al. (2001) and Albani et al. (2006), the individual-based mortality rate $(m_{t_k})$ of any cohort $k$ is the sum of four terms:

$$m_{t_k} = \underbrace{m_{t_k}^{DI}}_{\substack{\text{Aging}\\ \text{(Density-Independent)}}} + \underbrace{m_{t_k}^{DD}}_{\substack{\text{Carbon starvation}\\ \text{(Density dependent)}}} + \underbrace{m_{t_k}^{CF}}_{\substack{\text{Cold/Frost}}} + \underbrace{m_{t_k}^{FR}}_{\text{Fire}}. \tag{S27}$$

As in Moorcroft et al. (2001), density-independent mortality is the component attributable to aging of the cohort, and it depends both on the typical tree fall disturbance rate $\lambda_{TF}$ (Table S4) and the cohort wood density:

$$m_{t_k}^{DI} = \lambda_{TF} + \frac{1}{\text{æ}}, \tag{S28}$$

$$\tag{S29}$$

where æ is a PFT-specific term to account for the excess mortality in addition to the background mortality due to plant life span (Tables S5-S6). For tropical broadleaf trees, æ is parameterized following Moorcroft et al. (2001):

$$\text{æ} = \frac{0.0933\,\rho_{LTR}}{\lambda_{TF}\,(\rho_{LTR} - \rho_{t_k})}, \tag{S30}$$

[revised manuscript text omitted]

**S7 Snowpack depth dynamics**

In addition to enthalpy and total water, we must also track the changes in snowpack depth of each layer ($\Delta z_{s_j}$) and density ($\rho_{s_j}$) over time. The ordinary differential equation that governs changes in depth over time is defined as:

$$\frac{d\,\mathcal{N}\!t\,/\!/\,(\Delta z_{s_j})}{dt} = \begin{cases} \underbrace{\rho_{wa}\dot{W}_{a,s_j}}_{\substack{\text{Throughfall}\\ \text{precipitation}\\ (4.2)}} + \underbrace{\left(\sum_{k=1}^{N_T} \rho_{wt_k}\dot{W}_{t_k,s_j}\right)}_{\substack{\text{Canopy dripping}\\ \text{from cohorts}\\ (4.2)}} - \underbrace{\rho_{ws_j}\dot{W}_{s_j,o}}_{\substack{\text{Surface runoff}\\ (4.1)}} - \underbrace{\rho_{wx}\dot{W}_{s_j,c}}_{\substack{\text{Surface water}\\ \text{evaporation}\\ (4.5.2 \text{ and } 4.5.3)}} - \underbrace{\delta_{s_1 s_j}\rho_{ws_j}\dot{W}_{s_1,g_{N_G}}}_{\substack{\text{Surface water}\\ \text{percolation}\\ (4.1)}} &, \text{if } s_j = s_{N_S} \\[4ex] 0 &, \text{otherwise} \end{cases}, \tag{S68}$$

$$\rho_{ws_j} = \frac{W_{s_j}}{\Delta z_{s_j}} \tag{S69}$$

$$\rho_{wx} = \begin{cases} \rho_{ws_{N_S}} &, \text{if } \dot{W}_{s_{N_S},c} \geq 0 \\ \rho_{wc} &, \text{if } \dot{W}_{s_{N_S},c} < 0 \end{cases} \tag{S70}$$

where $\delta_{s_j s_{j'}}$ is the Kronecker delta for comparing two TSW layers $s_j$ and $s_{j'}$ (1 if $s_j = s_{j'}$, 0 otherwise), $\rho_{wa}$ is the precipitation density, $\rho_{wt_k}$ is the canopy interception density, $\rho_{wc}$ is the density of condensing water vapor. Precipitation density is defined based on Jin et al. (1999), but slightly modified to make it continuous:

$$\rho_{wa} = \frac{\rho_{ia}\rho_\ell}{\ell_a\rho_{ia} + (1 - \ell_a)\,\rho_\ell}, \tag{S71}$$

$$\rho_{ia} = \begin{cases} 169.16 &, \text{if } T_a > 275.16 \text{ K} \\ 50. + 1.7\,(T_a - 258.16)^{1.5} &, \text{if } 258.16 \text{ K} < T_a \leq 275.66 \text{ K}, \\ 50. &, \text{if } T_a \leq 258.16 \text{ K} \end{cases} \tag{S72}$$

where $\rho_\ell$ is the density of liquid water (Table S3). For the canopy dripping flux, water density is similar to Eq. (S71), except that we assume the density of frozen water to be the same as frost density ($\rho_*$, Table S3). A similar assumption is done for water condensing from canopy air space, with the additional assumption that the liquid fraction of condensation is the same as the liquid fraction of the top TSW layer:

$$\rho_{wt_k} = \frac{\rho_* \rho_\ell}{\ell_{t_k} \rho_* + \left(1 - \ell_{t_k}\right) \rho_\ell}, \tag{S73}$$

$$\rho_{wc} = \frac{\rho_* \rho_\ell}{\ell_{s_{N_S}} \rho_* + \left(1 - \ell_{s_{N_S}}\right) \rho_\ell}. \tag{S74}$$

The maximum allowed number of snow layers is determined by the user, but the actual number of snow layers is dynamically determined, following the same algorithm as Walko et al. (2000). Multiple layers only exist when ice is present, otherwise a single layer ($N_S = 1$) is enforced. When ice is present, the model selects $N_S$ to be the maximum number of layers that satisfies $W_{s_j} \geq 5 \; \mathrm{kg_W \, m^{-2}}$ for all layers $s_j, j \in 1, 2, \ldots, N_S$, to ensure numerical stability. The layer thickness distribution ($\Delta z_{s_j}$) for any given $N_S$ is defined as:

$$\Delta z_{s_j} = z_s \frac{2^{\min(j-1, N_S - j)}}{2^{\lfloor \frac{N_S + 1}{2} \rfloor} + 2^{\lfloor \frac{N_S}{2} \rfloor} - 2}, \tag{S75}$$

$$z_s = \sum_{j=1}^{N_S} \Delta z_{s_j}, \tag{S76}$$

where $z_s$ is the total depth of the snow, and $\lfloor x \rfloor$ is the floor function (i.e. the nearest integer value to $x$ that is not greater than $x$). The layer distribution described by Eq. (S75) ensures that the layers near the ground and near the canopy air space are thinner than the intermediate layers, to improve the representation of exchanges between the snowpack and the canopy air space, soils, and incoming irradiance (Walko et al., 2000).

**S8   Canopy-Air-Space Pressure**

Canopy-air-space pressure $p_c$ is assumed to remain constant throughout the integration time step ($\Delta t_{\mathrm{Thermo}}$). At the end of the time step, the air pressure above canopy $p_a$ is updated using the meteorological forcing, at which time $p_c$ and $h_c$ are also updated. To determine $p_c$, we combine three assumptions:

1. Both canopy air space and the air above are a mix of two perfect gases, dry air and water

vapor (Dufour and van Mieghem, 1975):

$$p = \rho \mathcal{R} \left[ \frac{1}{\mathcal{M}_d}(1-w) + \frac{1}{\mathcal{M}_w}w \right] T = \rho \frac{\mathcal{R}}{\mathcal{M}_d} T_{\mathcal{V}}, \tag{S77}$$

$$T_{\mathcal{V}} = T \left[ 1 - \left( 1 - \frac{\mathcal{M}_d}{\mathcal{M}_w}w \right) \right], \tag{S78}$$

where $\mathcal{R}$ is the universal gas constant, and $\mathcal{M}_d$ and $\mathcal{M}_w$ are the molar masses of dry air and water (Table S3); and $T_{\mathcal{V}}$ is the virtual temperature, which is the temperature that pure dry air would be at if pressure and density were the same as the observed air:

2. $p_c$ instantaneously changes when $p_a$ is updated, and this update does not involve any exchange of mass or energy. This is equivalent to assuming that potential temperature of the canopy air space $\theta_c$ and air aloft $\theta_a$ do not change when pressure is updated, even if enthalpy and temperature change. Potential temperature, approximated to the potential temperature of dry air, is defined as:

$$\theta = T \left( \frac{p_0}{p} \right)^{\frac{\mathcal{R}}{\mathcal{M}_d q_{pd}}}, \tag{S79}$$

where $p_0$ is the reference pressure level and $q_{pd}$ is the specific heat of dry air at constant pressure (Table S3).

3. The layer between canopy air space depth $\bar{z}_c$ and reference height of the free air $z_a$ is in hydrostatic equilibrium:

$$\frac{\partial p}{\partial z} = -\rho g, \tag{S80}$$

where $g$ is the gravity acceleration (Table S3).

Combining these three assumptions and defining $\theta_{\mathcal{V}} \equiv \theta(T_{\mathcal{V}})$ yields:

$$p_c = \left[ p_a^{\frac{\mathcal{R}}{\mathcal{M}_d q_{pd}}} + \frac{g\,(z_a - \bar{z}_c)}{q_{pd}\overline{\theta_{\mathcal{V}}}} p_0^{\frac{\mathcal{R}}{\mathcal{M}_d q_{pd}}} \right]^{\frac{\mathcal{M}_d q_{pd}}{\mathcal{R}}}, \tag{S81}$$

where $\overline{\theta_{\mathcal{V}}}$ is the virtual potential temperature averaged between $z_a$ and $\bar{z}_c$. Once pressure is updated at the biophysics time step, temperature and enthalpy are also updated using Eq. (S79)

and Eq. (S50), respectively. Because canopy air pressure is known at all times, canopy air density $\rho_c$ can be determined diagnostically using Eq. (S77).

**S9    Soil thermal and hydraulic properties**

Most of the soil hydraulic properties in ED-2.2 are derived from LEAF-3 (Walko et al., 2000) and use the soil classification based on the United States Department of Agriculture (e.g. Cosby et al., 1984). Soils in tropical forests often fall under the *Clay* class of the USDA classification, even though their sand, silt, and clay fractions often vary significantly from the average values of this class. To avoid large deviations from observations, we further split the original *Clay* class into four categories, named as *Clayey sand*, *Clayey silt*, *Clay*, and *Heavy Clay*, as shown in Fig. S6; the default fractions of each component for the default soil texture types in ED-2.2 are listed in Table S8. In addition to the standard classes, the model can derive site-specific properties based on the actual clay, silt, and sand fractions, which can be provided directly by the user.

The main hydraulic properties follow the parameterization by Cosby et al. (1984), shown here for reference:

$$\vartheta_{\text{Po}} = 0.0505 - 0.0142\, f_{\mathcal{V}_{\text{Sand}}} - 0.0037\, f_{\mathcal{V}_{\text{Clay}}}, \tag{S82}$$

$$\Psi_{\text{Po}} = -0.01 \cdot 10^{2.17 - 1.58\, f_{\mathcal{V}_{\text{Sand}}} - 0.63\, f_{\mathcal{V}_{\text{Clay}}}}, \tag{S83}$$

$$b = 3.10 - 0.3 \cdot f_{\mathcal{V}_{\text{Sand}}} + 15.7 \cdot f_{\mathcal{V}_{\text{Clay}}}, \tag{S84}$$

$$\Upsilon_{\Psi_{\text{Po}}} = 6.817 \times 10^{-6} \cdot 10^{-0.60 + 1.26\, f_{\mathcal{V}_{\text{Sand}}} - 0.64\, f_{\mathcal{V}_{\text{Clay}}}}, \tag{S85}$$

where $f_{\mathcal{V}_{\text{Sand}}}$ and $f_{\mathcal{V}_{\text{Clay}}}$ are the volumetric fraction of sand and clay, respectively; $\vartheta_{\text{Po}}$ ( $m_W^3\, m^{-3}$) is the volumetric soil porosity (maximum soil moisture possible), $\Psi_{\text{Po}} \Upsilon_{\Psi_{\text{Po}}}$ ( m) is the soil matric potential at porosity, $b$ is the slope of the logarithmic water retention curve (Clapp and Hornberger, 1978), and $\Upsilon_{\Psi}^{(\text{Po})}$ ( $kg_W\, m^{-2}\, s^{-1}$) is the soil hydraulic conductivity at bubbling pressure, assumed to occur when soil moisture $\vartheta = \vartheta_{\text{Po}}$.

The equation that describes soil matric potential as a function of soil moisture is taken from Clapp and Hornberger (1978); soil hydraulic conductivity is defined after Brooks and Corey (1964), with an additional correction term applied to hydraulic conductivity to reduce conductivity in case the soil is partially or completely frozen:

$$\Psi = \Psi_{\text{Po}} \left( \frac{\vartheta_{\text{Po}}}{\vartheta} \right)^b, \tag{S86}$$

$$\Upsilon_\Psi = \left[10^{-7(1-\ell)}\right] \Upsilon_{\Psi_{Po}} \left(\frac{\vartheta}{\vartheta_{Po}}\right)^{2b+3}, \tag{S87}$$

where   $\ell$ is the fraction of liquid water of soil moisture.

[revised manuscript text omitted]

$$\varsigma_{R_{mg}} = \min\left[\varsigma_{R_m}^{\mathrm{Po}} + 0.11 - 0.40\,\vartheta_{g_{N_G}}, \varsigma_{R_m}^{\mathrm{Re}}\right], \tag{S107}$$

where $\varsigma_{R_m}^{\mathrm{Re}}$ and $\varsigma_{R_m}^{\mathrm{Po}}$ are the soil color-dependent reflectance for dry and saturated soils, respectively.

The temporary surface water reflectance $\varsigma_{R_{ms}}$ depends on the liquid fraction, snow grain size and age, impurities, and the direction of incoming radiation, but here we simply assume a linear interpolation of soil reflectance at saturation and pure snow reflectance ($\varsigma_{R_{ms}}^{\circledast}$; Table S4), assumed constant for each band:

$$\varsigma_{R_{ms}} = \varsigma_{R_{ms}}^{\circledast} + \ell_{s_{N_S}}\left(\varsigma_{R_m}^{\mathrm{Po}} - \varsigma_{R_{ms}}^{\circledast}\right). \tag{S108}$$

Following Verseghy (1991) and Walko et al. (2000), the transmissivity of intercepted irradiance for PAR and NIR is solved following Beer's law, with a direction-independent extinction coefficient:

$$\varsigma_{T_{ms}} = \begin{cases} \exp\left(-\dfrac{\sum_{j=1}^{N_S}\Delta\bar{z}_{s_j}}{f_{\mathrm{TSW}}\bar{\mu}_s}\right) & , \text{if } m \in (1,2) \\ 0 & , \text{if } m = 3 \end{cases}, \tag{S109}$$

where $\bar{\mu}_s = 0.05\,\mathrm{m}$ is the inverse of the optical depth per unit of temporary surface water depth, defined here to be the same coefficient used by Verseghy (1991) and Walko et al. (2000), and the additional $f_{\mathrm{TSW}}^{-1}$ term accounts for the clumping of the temporary surface water, when the water does not cover all ground. Temporary surface water is assumed to be opaque for the TIR band ($m = 3$), following Walko et al. (2000).

**S12 Solving the two-stream linear system of canopy radiation in ED-2.2.**

Because we assume that the optical properties are constant within each layer, it is possible to find an analytical solution for the full profile of direct and diffuse radiation. First, let $\dot{Q}_{mk}^{\odot}$, $\dot{Q}_{mk}^{\Downarrow}$, and

$\dot{Q}_{mk}^{\Uparrow}$ be the solution for band $m$ and interface $k$ immediately beneath the cohort (i.e. at $\tilde{\Phi} = \tilde{\Phi}_k$), and $\dot{Q}_{0_{mk}}^{\odot}$, $\dot{Q}_{0_{mk}}^{\Downarrow}$, and $\dot{Q}_{0_{mk}}^{\Uparrow}$ be the solution for band $m$ and interface $k$ immediately above the cohort (i.e. at $\tilde{\Phi} = 0$), as shown in Fig. 4. The direct radiation profile within each layer is simply given by:

$$\dot{Q}_{mk}^{\odot} = \dot{Q}_{0_{mk}}^{\odot} \exp\left(-\frac{\tilde{\Phi}_k}{\mu_k^{\odot}}\right), \tag{S110}$$

$$\dot{Q}_{0_{mk}}^{\odot} = \dot{Q}_{m(k+1)}^{\odot}, \tag{S111}$$

$$\dot{Q}_{m(N_T+1)}^{\odot} = \dot{Q}_{m(\infty,a)}^{\odot}, \tag{S112}$$

where $\dot{Q}_{m(\infty,a)}^{\odot}$ is the above-canopy, incoming direct radiation for band $m$ and serves as the top boundary condition. Because the value at interface $N_T + 1$ is known, it is possible to determine all levels by integrating the layers from top to bottom.

For the diffuse components, an analytic solution can be found by defining two auxiliary variables $\dot{Q}_{mk}^{+} \equiv \dot{Q}_{mk}^{\Downarrow} + \dot{Q}_{mk}^{\Uparrow}$ and $\dot{Q}_{mk}^{-} = \dot{Q}_{mk}^{\Downarrow} - \dot{Q}_{mk}^{\Uparrow}$. By subtracting (adding) Eq. (46) from (to) Eq. (47), and using Eq. (S110)-(S112) we obtain:

$$\frac{\mathrm{d}\dot{Q}_{mk}^{+}}{\mathrm{d}\tilde{\Phi}} = -\frac{1-(1-2\beta_{mk})\,\varsigma_{mk}}{\overline{\mu}_k}\dot{Q}_{mk}^{-} + \frac{(1-2\beta_{mk}^{\odot})\,\varsigma_{mk}}{\mu_{mk}^{\odot}}\dot{Q}_{m(k+1)}^{\odot}, \tag{S113}$$

$$\frac{\mathrm{d}\dot{Q}_{mk}^{-}}{\mathrm{d}\tilde{\Phi}} = -\frac{1-\varsigma_{mk}}{\overline{\mu}_k}\dot{Q}_{mk}^{+} + \frac{\varsigma_{mk}}{\mu_k^{\odot}}\dot{Q}_{m(k+1)}^{\odot} + \frac{2(1-\varsigma_{mk})}{\overline{\mu}_k}\dot{Q}_{mk}^{\blacklozenge}. \tag{S114}$$

By differentiating Eq. (S113) and Eq. (S114) and substituting the first derivatives by Eq. (S114) and Eq. (S113), we obtain two independent, second-order ordinary differential equations:

$$\frac{\mathrm{d}^2\dot{Q}_{mk}^{+}}{\mathrm{d}\tilde{\Phi}^2} = \varkappa_{mk}^2\,\dot{Q}_{mk}^{+} + \kappa_{mk}^{+}\exp\left(-\frac{\tilde{\Phi}}{\mu_k^{\odot}}\right) - 2\varkappa_{mk}^2\,\dot{Q}_{ik}^{\blacklozenge}, \tag{S115}$$

$$\frac{\mathrm{d}^2\dot{Q}_{mk}^{-}}{\mathrm{d}\tilde{\Phi}^2} = -\varkappa_{mk}^2\,\dot{Q}_{mk}^{-} + \kappa_{mk}^{-}\exp\left(-\frac{\tilde{\Phi}}{\mu_k^{\odot}}\right), \tag{S116}$$

where

$$\varkappa_{mk}^2 = \frac{[1-(1-2\beta_{mk})\,\varsigma_{mk}]\,(1-\varsigma_{mk})}{\overline{\mu}_k^2}, \tag{S117}$$

$$\kappa_{mk}^{+} = -\left[\frac{1-(1-2\beta_{mk})\,\varsigma_{mk}}{\overline{\mu}_k} + \frac{1-2\beta_{mk}^{\odot}}{\mu_k^{\odot}}\right]\frac{\varsigma_{mk}\,\dot{Q}_{m(k+1)}^{\odot}}{\mu_k^{\odot}}, \tag{S118}$$

$$\kappa_{mk}^{-} = -\left[ \frac{(1 - \varsigma_{mk})\left(1 - 2\beta_{mk}^{\odot}\right)}{\overline{\mu}_k} + \frac{1}{\mu_k^{\odot}} \right] \frac{\varsigma_{mk}\,\dot{Q}_{m(k+1)}^{\odot}}{\mu_k^{\odot}}. \tag{S119}$$

The solution of Eq. (S115)-(S116) is the combination of the homogeneous and the particular solution, and can be determined analytically:

$$\dot{Q}_{mk}^{+}\left(\tilde{\Phi}\right) = x_{mk}^{+-}\exp\left(-\varkappa_{mk}\,\tilde{\Phi}\right) + x_{mk}^{++}\exp\left(+\varkappa_{mk}\,\tilde{\Phi}\right) + \frac{\kappa^{+}\mu_k^{\odot 2}}{1 - \varkappa_{mk}^2\,\mu_k^{\odot 2}}\exp\left(-\frac{\tilde{\Phi}}{\mu_k^{\odot}}\right) + 2\dot{Q}_{mk}^{\blacklozenge} \tag{S120}$$

$$\dot{Q}_{mk}^{-}\left(\tilde{\Phi}\right) = x_{mk}^{--}\exp\left(-\varkappa_{mk}\,\tilde{\Phi}\right) + x_{mk}^{-+}\exp\left(+\varkappa_{mk}\,\tilde{\Phi}\right) + \frac{\kappa^{-}\mu_k^{\odot 2}}{1 - \varkappa_{mk}^2\,\mu_k^{\odot 2}}\exp\left(-\frac{\tilde{\Phi}}{\mu_k^{\odot}}\right) \tag{S121}$$

where $x_{mk}^{+-}$, $x_{mk}^{++}$, $x_{mk}^{--}$, and $x_{mk}^{-+}$ are coefficients to be determined. We can reduce the number of coefficients to two by differentiating Eq. (S120)-(S121) and comparing them to Eq. (S113)-(S114), and using the fact that they must be equal for any $\tilde{\Phi}$, $\mu_k^{\odot}$, $\varkappa_{mk}$, and $\dot{Q}_{mk}^{\blacklozenge}$. We call these parameters $x_{m(2k-1)}$ and $x_{m(2k)}$, $k \in \{1, 2, \ldots, N_T\}$. By further recalling the definition of $\dot{Q}_{mk}^{+}$ and $\dot{Q}_{mk}^{-}$, we obtain the profile of downward and upward diffuse irradiances:

$$\dot{Q}_{mk}^{\Downarrow}\left(\tilde{\Phi}\right) = x_{m(2k-1)}\,\mathcal{D}_{mk}^{+}\exp\left(-\varkappa_{mk}\,\tilde{\Phi}\right) + x_{m(2k)}\,\mathcal{D}_{mk}^{-}\exp\left(+\varkappa_{mk}\,\tilde{\Phi}\right) + P_{mk}^{+}\exp\left(-\frac{\tilde{\Phi}}{\mu_k^{\odot}}\right) + \dot{Q}_{mk}^{\blacklozenge}, \tag{S122}$$

$$\dot{Q}_{mk}^{\Uparrow}\left(\tilde{\Phi}\right) = x_{m(2k-1)}\,\mathcal{D}_{mk}^{-}\exp\left(-\varkappa_{mk}\,\tilde{\Phi}\right) + x_{m(2k)}\,\mathcal{D}_{mk}^{+}\exp\left(+\varkappa_{mk}\,\tilde{\Phi}\right) + P_{mk}^{-}\exp\left(-\frac{\tilde{\Phi}}{\mu_k^{\odot}}\right) + \dot{Q}_{mk}^{\blacklozenge}, \tag{S123}$$

where

$$\mathcal{D}_{mk}^{\pm} = \frac{1}{2}\left[ 1 \pm \sqrt{\frac{1 - \varsigma_{mk}}{1 - (1 - 2\beta_{mk})\,\varsigma_{mk}}} \right], \tag{S124}$$

$$P_{mk}^{\pm} = \frac{\left(\kappa_{mk}^{+} \pm \kappa_{mk}^{-}\right)\mu_k^{\odot 2}}{2\left(1 - \varkappa_{mk}^2\,\mu_k^{\odot 2}\right)}. \tag{S125}$$

To determine all vector elements $\left(x_{m(2k-1)}, x_{m(2k)}\right); k \in \{1, 2, \ldots, N_T, N_T + 1\}$ we need three independent systems of $2N_T + 2$ equations (one system of equations for each spectral band). For $k \in \{1, 2, \ldots, N_T\}$, the solution must meet the boundary conditions for all middle interfaces (Fig. 4), with one additional boundary condition for upward radiation coming out of the ground

(Line 1), and another for incoming downward radiation from above the canopy (Line $2N_T + 2$):

$$
\begin{aligned}
&\text{Line 1:} && \dot{Q}^{\Uparrow}_{m1} - \varsigma_{m0}\left(\dot{Q}^{\Downarrow}_{mk} + \dot{Q}^{\odot}_{mk}\right) - (1 - \varsigma_{m0})\dot{Q}^{\blacklozenge}_{m0} = 0 && \\
&\text{Line 2k:} && \dot{Q}^{\Downarrow}_{0_{mk}} - \dot{Q}^{\Downarrow}_{m(k+1)} = 0 && , 
[revised manuscript text omitted]

The ED-2.2 model uses the empirical parameterization of the originally developed by Beljaars and Holtslag (1991). For the unstable cases, Beljaars and Holtslag (1991) used the Businger-Dyer flux profile equations (Businger et al., 1971). For the stable cases, Beljaars and Holtslag (1991) implemented an empirical formulation that improved the vertical mixing between the canopy air space and the air above under stable conditions:

$$\psi_U(\zeta) = \begin{cases} 2\ln\left[\frac{1+Y(\zeta)}{2}\right] + \ln\left[\frac{1+Y^2(\zeta)}{2}\right] - 2\arctan[Y(\zeta)] + \frac{\pi}{2} & \text{, if } \mathrm{Ri}_B < 0 \\ y_1\zeta + y_2\left(\zeta - \frac{y_3}{y_4}\right)\exp(-y_4\zeta) + \frac{y_2 y_3}{y_4} & \text{, if } \mathrm{Ri}_B \geq 0 \end{cases}, \tag{S153}$$

$$\psi_\Theta(\zeta) = \begin{cases} 2\ln\left[\frac{1+Y^2(\zeta)}{2}\right] & \text{, if } \mathrm{Ri}_B < 0 \\ 1 - \left(1 - \frac{y_1}{y_5}\zeta\right)^{y_5} + x_2\left(\zeta - \frac{y_3}{y_4}\right)\exp(-y_4\zeta) + \frac{y_2 y_3}{y_4} & \text{, if } \mathrm{Ri}_B \geq 0 \end{cases}, \tag{S154}$$

$$Y(\zeta) = \sqrt[4]{1 - y_6\zeta}, \tag{S155}$$

where $\mathbf{y} = \left(-1; -\frac{2}{3}; 5; 0.35; \frac{3}{2}; 13\right)$ are empirical and adjustable parameters. Equation (S152)

cannot be solved analytically, therefore $\zeta^\star$ is calculated using a root-finding technique. Once $\zeta^\star$ is determined, we can find $u^\star$ using Eq. (S150), and define the canopy conductance $G_c$ ( m s$^{-1}$) using Eq. (S151) as the starting point, similarly to Oleson et al. (2013):

$$G_c = \frac{u^\star \theta_{\mathcal{V}}^\star}{\theta_{\mathcal{V}_a} - \theta_{\mathcal{V}_c}} = \frac{\kappa u^\star}{\Pr \left[ \ln \left( \frac{\zeta^\star}{\zeta_0} \right) - \psi_\Theta \left( \zeta^\star \right) + \psi_\Theta \left( \zeta_0 \right) \right]}. \tag{S156}$$

**S14.2   Derivation leaf and wood boundary layer conductances**

Following Monteith and Unsworth (2008), convection can be of two types: forced convection, which depends on mechanic mixing associated with the fluid velocity; and free convection, which is due to buoyancy of the boundary layer fluid. Although convection is often dominated by either forced or free convection, in ED-2.2 we always assume that the total conductance is a simple combination of forced and free convection conductances as if they were parallel:

$$G_{Qx_k} = G_{Qx_k}^{\text{Free}} + G_{Qx_k}^{\text{Forced}}, \tag{S157}$$

where $x_k$ can be either the leaf ($\lambda_k$) or the branch wood ($\beta_k$) boundary layer. For each convective regime, we define the conductance in terms of the Nusselt number Nu, a dimensionless number that corresponds to the ratio between heat exchange through convection and conduction:

$$G_{Qx_k} = \frac{\eta_c \, \text{Nu}}{x^\star}. \tag{S158}$$

where $\eta_c$ is the thermal diffusivity of canopy air space and $x^\star$ is the characteristic size of the obstacle. For leaves, the characteristic size $x_{\lambda_k}^\star$ is a PFT-dependent constant corresponding to the typical leaf width , whereas for branch wood the typical size $x_{\beta_k}^\star$ is assumed to be the typical diameter of twigs (Tables S5-S6).

Free convection is a result of the thermal gradient between the obstacle surface and the fluid, and this is normally expressed in terms of the Grashof number Gr, a dimensionless index that relates buoyancy and viscous forces. In ED-2.2 we use the same empirical functions as Monteith and Unsworth (2008), using flat plate geometry for leaves and horizontal cylinder geometry for branch wood:

$$
\mathrm{Nu}_{\lambda_k}^{(\mathrm{Free})} = \max \left[ \underbrace{0.50\,\mathrm{Gr}_{\lambda_k}^{\frac{1}{2}}}_{\text{Laminar}} , \underbrace{0.13\,\mathrm{Gr}_{\lambda_k}^{\frac{1}{3}}}_{\text{Turbulent}} \right] , \tag{S159}
$$

$$
\mathrm{Nu}_{\beta_k}^{(\mathrm{Free})} = \max \left[ \underbrace{0.48\,\mathrm{Gr}_{\beta_k}^{\frac{1}{2}}}_{\text{Laminar}} , \underbrace{0.09\,\mathrm{Gr}_{\beta_k}^{\frac{1}{3}}}_{\text{Turbulent}} \right] , \tag{S160}
$$

$$
\mathrm{Gr}_{x_k} = \frac{\varepsilon_c\, g\, \left(x_{x_k}^\star\right)^3}{v_c^2}\, \left| T_{x_k} - T_c \right| , \tag{S161}
$$

where $\varepsilon_c$ is the thermal dilatation coefficient for the canopy air space and $v_c$ is the kinematic viscosity of the canopy air space; $x_k$ represents either the leaf ($\lambda_k$) or wood ($\beta_k$) surface; and $g$ is the gravity acceleration. Like in Monteith and Unsworth (2008), thermal diffusivity and dynamic viscosity (both in $\mathrm{m^2\,s^{-1}}$) are assumed to be linear functions of the canopy air space temperature:

$$
\eta_c = 1.89 \cdot 10^{-5}\, [1 + 0.007\,(T_c - T_0)] , \tag{S162}
$$

$$
v_c = 1.33 \cdot 10^{-5}\, [1 + 0.007\,(T_c - T_0)] , \tag{S163}
$$

where the first term on the right hand side are the reference values at temperature $T_0 = 273.15\,\mathrm{K}$. Under the assumption that canopy air space is a perfect gas, thermal dilatation is $\varepsilon_c = T_c^{-1}$ (Dufour and van Mieghem, 1975).

For forced convection the flow of air through the object at different temperature causes the heat exchange, therefore Nusselt number is written as a function of the Reynolds number Re, a dimensionless index that relates inertial and viscous forces. Like in the free convection case, we use the same empirical functions as Monteith and Unsworth (2008) and the same shapes as the free convection case:

$$
\mathrm{Nu}_{\lambda_k}^{(\mathrm{Forced})} = \max \left[ \underbrace{0.60\,\mathrm{Re}_{\lambda_k}^{0.5}}_{\text{Laminar}} , \underbrace{0.032\,\mathrm{Re}_{\lambda_k}^{0.8}}_{\text{Turbulent}} \right] , \tag{S164}
$$

$$
\mathrm{Nu}_{\beta_k}^{(\mathrm{Forced})} = \max \left[ \underbrace{0.32 + 0.51\,\mathrm{Re}_{\beta_k}^{0.52}}_{\text{Laminar}} , \underbrace{0.24\,\mathrm{Re}_{\beta_k}^{0.60}}_{\text{Turbulent}} \right] , \tag{S165}
$$

$$
\mathrm{Re}_{x_k} = \frac{u_{t_k}\, x_{x_k}^\star}{\eta_c} , \tag{S166}
$$

where $u_{t_k}$ is the wind speed experienced by the cohort $k$, and $x_k$ represents either the leaf ($\lambda_k$) or wood ($\beta_k$) surface.

The wind profile within the canopy air space is determined in two steps. Above the tallest cohort, we assume that the wind can be determined from the similarity theory; from Eq. (S143) we define $\zeta_{c_j} = \zeta\left(z_{c_j}\right)$, and use wind profile function from the similarity theory (Eq. S150) to determine the wind speed at the top of the vegetated layer $u_{c_{N_C}} = u\left(\zeta_{c_{N_C}}\right)$. Within the canopy, we estimate the wind speed reduction using the wind profile as a function of cumulative drag ($\Xi_j$; Albini, 1981; Massman, 1997); the wind speed experienced by the cohort is the average wind between the layers where the bottom ($\hat{z}_{t_k}$) and top ($z_{t_k}$) of the crown are located:

$$u_{c_j} = u_{c_{N_C}} \exp\left(-\frac{\Xi_{c_{N_C}} - \Xi_{c_j}}{\xi_{\text{sfc}}}\right) \tag{S167}$$

$$u_{t_k} = \max\left[0.25\,\text{m\,s}^{-1}, \frac{u_{c_{N_C}}}{z_{c_j(k)} - z_{c_{\hat{j}}(k)}} \sum_{j'=\hat{j}(k)}^{j(k)} \left(u_{c_{j'}} \Delta z_{c_{j'}}\right)\right], \tag{S168}$$

where $c_{\hat{j}(k)}$ and $c_j(k)$ are the canopy air space layers corresponding to the bottom and top of the cohort's crown. The minimum wind speed of $0.25\,\text{m\,s}^{-1}$ is imposed to avoid conductance to become unrealistically low and to account for some mixing due to gusts when the mean wind is very weak. Once the heat conductance is determined, we use the same vapor to heat ratio as Leuning et al. (1995) to calculate the water vapor conductance:

$$G_{Wx_k} = 1.075\,G_{Qx_k}, \tag{S169}$$

where $x_k$ represents either the leaf ($\lambda_k$) or wood ($\beta_k$) surface. Similarly, we define the $CO_2$ boundary layer conductance for leaves using the ratio of diffusivities and convection between water and $CO_2$ ($f_{G\lambda}$, Table S4), following Cowan and Troughton (1971):

$$\hat{G}_{W\lambda_k} = f_{G\lambda}\,\hat{G}_{C\lambda_k}. \tag{S170}$$

**S14.3   Derivation of surface conductance**

The total resistance between the surface and the canopy air space is a combination of the air resistance if the surface were bare, and the resistance due to the presence of the vegetated canopy, assuming that these resistances are serial and thus additive (as mentioned by Walko et al., 2000); using that conductance is the inverse of resistance:

$$\frac{1}{G_{\text{Sfc}}} = \frac{1}{G_{\text{Bare}}} + \frac{1}{G_{\text{Veg}}}, \tag{S171}$$

[revised manuscript text omitted]

Both $p_{vi}$ and $p_{v\ell}$ are defined after the parameterization by ([Murphy and Koop, 2005]), which have high degree of accuracy ($< 0.05\%$) between 123 K and 332 K, and thus includes all the range of near-surface temperatures solved by ED-2.2:

$$p_{vi}(T) = \exp\left[9.550426 - \frac{5723.265}{T} + 3.53068\ln(T) - 0.00728332\,T\right], \tag{S186}$$

$$p_{v\ell}(T) = \exp\left\{Y_1(T) + Y_2(T)\tanh\left[0.0415\left(T - 218.8\right)\right]\right\}, \tag{S187}$$

$$Y_1(T) = 54.842763 - \frac{6763.22}{T} - 4.210\ln(T) + 0.000367\,T, \tag{S188}$$

$$Y_2(T) = 53.878 - \frac{1331.22}{T} - 9.44523\ln(T) + 0.014025\,T. \tag{S189}$$

Importantly, Eq. ([S186]) and Eq. ([S187]) yield the same value (within $4.1 \cdot 10^{-6}\%$ accuracy) at the water's triple point, which guarantees continuity of Eq. ([S185]).

The saturation specific humidity $w^{\equiv}$ is obtained using Eq. ([S185]) and the definition of specific humidity:

$$w_{\text{Sat}}(T, p) = \frac{\mathcal{M}_w\, p_{\text{Sat}}(T)}{\mathcal{M}_d\left[p - p_{\text{Sat}}(T)\right] + \mathcal{M}_w\, p_{\text{Sat}}(T)}, \tag{S190}$$

[revised manuscript text omitted]
}$ (wilting factor). The functional form of $f_{Wl_k}$ follows the previous versions of ED (Moorcroft et al., 2001; Medvigy et al., 2009). However, in ED-2.2 we define water availability ($W_{g_j}^\star$) in terms of soil matric potential, similarly to CLM (Oleson et al., 2013), which produces a more gradual transition from no-stress conditions to completely closed stomata as soil moisture approaches the wilting point (Fig. S9).

In ED-2.2, the wilting factor $f_{Wl_k}$ is defined as:

$$f_{Wl_k} = \cfrac{1}{1 + \cfrac{\text{Demand}}{\text{Supply}}} = \cfrac{1}{1 + \cfrac{\mathcal{M}_w \Lambda_k \dot{E}_k}{\hat{G}_{r_k} C_{r_k} W^{\star}_{g_{j0}}}}, \tag{S201}$$

$$W^{\star}_{g_j} = \sum_{j'=j}^{N_G} \left[ \rho_\ell \left( \vartheta_{\text{Fc}} - \vartheta_{\text{Wp}} \right) \Psi^{\star}_{g_{j'}} \Delta z_{g_{j'}} \right], \tag{S202}$$

$$\Psi^{\star}_{g_j} = \ell_{g_j} \cfrac{\max \left[ \min \left( \Psi_{g_j} + \frac{z_{g_j} + z_{g_{j+1}}}{2}, \Psi_{\text{Fc}} \right), \Psi_{\text{Wp}} \right] - \Psi_{\text{Wp}}}{\Psi_{\text{Fc}} - \Psi_{\text{Wp}}}, \tag{S203}$$

where $\hat{G}_{r_k}$ ($\text{m}^2 \text{kg}_C^{-1} \text{s}^{-1}$) is a PFT-dependent scaling parameter related to fine root conductance (Tables S5-S6); $\mathcal{M}_w$ is the molar mass of water (Table S3); $\dot{E}_k$ ($\text{mol}_W \text{m}_{\text{Leaf}}^{-2} \text{s}^{-1}$) is the leaf-level transpiration rate if soil moisture is not limiting; $C_{r_k}$ ($\text{kg}_C \text{m}^{-2}$) is the fine root biomass per individual; $\Lambda_k$ ($\text{m}^2 \text{m}^{-2}$) is the leaf area index of cohort $k$; $W^{\star}_{g_j}$ ($\text{kg}_W \text{m}^{-2}$) is the available water for photosynthesis integrated from soil layer $j$ to surface; $j0$ is the deepest soil layer that the cohort $k$ can access water; $z_{g_j}$ and $\Delta z_{g_j}$ are the depth and thickness of soil layer $j$ ($z_{g_j}$ is always negative, and $\Delta z_{g_j}$ is always positive); $\rho_\ell$ ($\text{kg}_W \text{m}^{-3}$) is the density of liquid water; $\vartheta_{\text{Fc}}$ and $\vartheta_{\text{Wp}}$ ($\text{m}^3 \text{m}^{-3}$) are the volumetric soil moistures at field capacity and at permanent wilting point, $\Psi_{g_j}$ ($m$) is the matric potential of layer $j$, $\Psi_{\text{Fc}}$ and $\Psi_{\text{Wp}}$ ($m$) are the matric potentials at field capacity and wilting point, $\Psi^{\star}_{g_j}$ (unitless) is a factor that represents the reduction of available water due to force needed to extract the water.

**S18   Allometric equations**

In ED-2.2, size is defined by a suite of dimensions, including tree height $z_{t_k}$ and rooting depth $z_{r_k}$ which directly affect the cohort access to light and water, and the carbon stocks in different tissues. Most allometric equations use the diameter at the breast height (DBH, cm) as the size-dependent explanatory variable. The only time DBH becomes the dependent variable is when the code calculates the growth of structural tissues ($\Delta t_{\text{CD}}$): structural carbon stocks are updated based on the cohort's net carbon balance, and DBH is calculated to be consistent with the updated structural carbon stocks. In this supplement, we present the allometric equations of ED-2.2 for tropical PFTs; the temperate counterparts have been previously described in Albani et al. (2006) and Medvigy et al. (2009).

The tree height of any cohort $k$ ($z_{t_k}$) is determined through a modified Weibull function:

$$z_{t_k} = \min \left\{ z_{t_{\max}}, \mathcal{Z}_0 + \mathcal{Z}_\infty \left[ 1 - \exp \left( -\mathcal{Z}_1 \cdot \text{DBH}_k^{\mathcal{Z}_2} \right) \right] \right\}, \tag{S204}$$

where $\mathcal{Z}_0$, $\mathcal{Z}_1$, $\mathcal{Z}_2$, and $\mathcal{Z}_\infty$ are PFT-dependent coefficients; and $z_{t_{\max}}$ is the maximum tree height, imposed to avoid excessive extrapolation of the allometric equations for carbon stocks. The coefficients are shown in Tables S5-S6; coefficients for tropical trees are provided by Poorter et al. (2006) allometric equation for moist forests in Bolivia; coefficients for temperate trees are from Albani et al. (2006).

The tree height at the bottom of the crown ($z_{t_k}^-$) is based on Poorter et al. (2006), and it is currently applied to tropical, subtropical, and temperate trees. For grasses, we fix the height to 1% of the total height, to avoid numeric singularities while assuming that most of the grass vertical profile has leaves:

$$
z_{t_k}^- = \begin{cases} \max\left(0.05, 0.01\, z_{t_k}\right) & \text{, if cohort } k \text{ is grass} \\ \max\left(0.05, z_{t_k} - 0.31\, z_{t_k}^{1.098}\right) & \text{, if cohort } k \text{ is tree} \end{cases}. \tag{S205}
$$

Maximum leaf biomass ($C_{l_k}^\bullet$, $\mathrm{kg\,m^{-2}}$), corresponding to the state when leaves are fully flushed, :

$$
C_{l_k}^\bullet = n_{t_k}\, \mathcal{C}_{0l_k} \mathrm{DBH}_k^{\mathcal{C}_{1l_k}}, \tag{S206}
$$

where $n_{t_k}$ ($\mathrm{plant\,m^{-2}}$) is the plant demographic density, and $\mathcal{C}_{0l}$ and $\mathcal{C}_{1l}$ are the PFT-dependent coefficients (Tables S5-S6). For tropical PFTs, the default parameters are derived from the allometric equations presented by Cole and Ewel (2006) and Calvo-Alvarado et al. (2008) for several commercial species in Costa Rica; for temperate PFTs, the default parameters are the same as in Albani et al. (2006) and Medvigy et al. (2009).

Maximum root biomass ($C_{r_k}^\bullet$, $\mathrm{kg\,m^{-2}}$) and maximum sapwood biomass ($C_{\sigma_k}^\bullet$, $\mathrm{kg\,m^{-2}}$) are determined from $C_{l_k}^\bullet$ using the same functional form as Moorcroft et al. (2001), whose formulation of sapwood biomass was was based on the pipe model by Shinozaki et al. (1964a,b):

$$
C_{r_k}^\bullet = f_{r_k} C_{l_k}^\bullet, \tag{S207}
$$

$$
C_{\sigma_k}^\bullet = \frac{\mathrm{SLA}_k}{f_{\sigma_k}} z_{t_k} C_{l_k}^\bullet, \tag{S208}
$$

where $f_{r_k}$ and $f_{\sigma_k}$ are PFT-dependent parameters, currently assumed to be the same as in the original ED-1 (Moorcroft et al., 2001, Tables S5-S6); SLA (Tables S5-S6) is the specific leaf area, determined from Kim et al. (2012) fit of specific leaf area as a function of leaf turnover rate, using the GLOPNET leaf economics dataset (Wright et al., 2004).

Total structural (heartwood) biomass ($C_{h_k}$, $\mathrm{kg_C\,m^{-2}}$)  functional form is the same functional form for all PFTs. For temperate PFTs, the parameters are the same as in Albani et al. (2006) and Medvigy et al. (2009). For tropical PFTs, the parameters are based on Baker et al. (2004) equation of above-ground biomass, which is in turn based on the allometric equation by Chave et al. (2001) for French Guiana. This allometric equation was used instead of the allometric equation based on Chambers et al. (2001) because in ED-2.2 the function relating $C_{h_k}$ and $\mathrm{DBH}_k$ must be bijective (i.e. given $n_{t_k}$, each $\mathrm{DBH}_k$ is associated with a single value of $C_{h_k}$ and vice versa), which cannot be attained with the polynomial fits of higher order. Structural biomass was assumed to be the difference between above-ground biomass and the biomass of leaves and 70% of the total sapwood, corresponding to the above-ground fraction. The estimate was fitted against DBH, yielding to:

$$C_{h_k} = \begin{cases} n_{t_k}\,\mathcal{C}_{0h_k}\,\mathrm{DBH}^{\mathcal{C}_{1h_k}} & \text{, if } \mathrm{DBH}_k \leq \mathrm{DBH_{Crit}} \\ n_{t_k}\,\mathcal{C}_{2h_k}\,\mathrm{DBH}^{\mathcal{C}_{3h_k}} & \text{, if } \mathrm{DBH}_k > \mathrm{DBH_{Crit}} \end{cases}, \tag{S209}$$

where $\mathrm{DBH_{Crit}}$ is the minimum DBH that results in $z_{t_k} = 35.0$ m, and the coefficients $\mathcal{C}_{0h}$, $\mathcal{C}_{1h}$, $\mathcal{C}_{2h}$, $\mathcal{C}_{3h}$ are defined for each PFT (Tables S5-S6).

The size-dependent rooting depth ($z_{r_k}$) is defined from an exponential function that allows tree depths to reach 5 m once trees reach canopy size ($z_{t_k} = 35$ m):

$$z_{r_k} = -1.114\,\mathrm{DBH}_k^{0.422}. \tag{S210}$$

The maximum rooting depth is shallow compared to Nepstad et al. (1994) results, however it produces a rooting profile similar to other dynamic global vegetation models, and reflects that little variation in soil moisture exists at very deep layers (Christoffersen, 2013).

Leaf area index ($\Lambda_k$, $\mathrm{m^2_{Leaf}\,m^{-2}}$) is determined from leaf biomass and specific leaf area:

$$\Lambda_k = \mathrm{SLA}_k\,C_{l_k}, \tag{S211}$$

where $n_k$ ( $\mathrm{plant\,m^{-2}}$) is the demographic density of cohort $k$.

No allometric equation was found for wood area index ($\Omega_k$, $\mathrm{m^2_{Wood}\,m^{-2}}$) for evergreen forests. We assumed the same allometric equation for temperate zone by Hörmann et al. (2003) for trees, and imposed maximum area at $\mathrm{DBH_{Crit}}$, similarly to $C_{l_k}$:

$$\Omega_k = \begin{cases} 0 & \text{if cohort } k \text{ is grass} \\ n_k \, 0.0096 \min \left(\text{DBH}, \text{DBH}_{\text{Crit}}\right)^{2.0947} & \text{if cohort } k \text{ is broadleaf tree} \\ n_k \, 0.02765 \min \left(\text{DBH}, \text{DBH}_{\text{Crit}}\right)^{1.9769} & \text{if cohort } k \text{ is conifer} \end{cases} . \qquad \text{(S212)}$$

Crown area index ($X_k$, $\text{m}^2_{\text{Crown}} \, \text{m}^{-2}$) is also based on Poorter et al. (2006), but re-written so it is a function of $\text{DBH}_k$. Like in the previous cases, crown area was capped at $\text{DBH}_{\text{Crit}}$, and local crown area was not allowed to exceed 1.0 or to be less than the leaf area index:

$$X_k = \begin{cases} \min \left[1.0, \max \left(\Lambda_k, n_k \, 1.126 \, \text{DBH}^{1.052}\right)\right] & \text{if cohort is tropical/subtropical} \\ \min \left[1.0, \max \left(\Lambda_k, n_k \, 2.490 \, \text{DBH}^{0.807}\right)\right] & \text{if cohort is temperate} \end{cases} . \qquad \text{(S213)}$$